

# A numerical model for solving the linearized gravity-wave equations by a multilayer method

Alexandru Doicu[1], Dmitry S. Efremenko[2], Thomas Trautmann[2], and Adrian Doicu[2]

[1]Independent researcher, 82110 Germering, Germany
[2]Deutsches Zentrum für Luft- und Raumfahrt (DLR), Institut für Methodik der Fernerkundung (IMF), 82234 Oberpfaffenhofen, Germany

**Correspondence:** Dmitry S. Efremenko (dmitry.efremenko@dlr.de)

**Abstract.** We developed a numerical model to solve the linearized gravity-wave equations by a multilayer approach. Specifically, the model handles the linearized equations including viscosity, thermal conduction, and ion drag. The solution methods are based on the matrix exponential formalism and encompass two main approaches: (i) global matrix methods and (ii) scattering matrix methods. Both methods are focused on determining either (i) the amplitudes of the characteristic solutions or (ii) the discrete values of the solution vector. Ascending and descending wave modes are distinguished based on the criterion that the real parts of the eigenvalues of the characteristic equation for ascending modes are smaller than those for descending modes. In global matrix methods, ascending and descending modes can be defined (i) at the upper and lower boundaries or (ii) in each layer. In contrast, scattering matrix methods necessitate explicitly determining the mode type within each layer. The model accommodates two types of lower boundary conditions and can handle both single-frequency waves and time wavepackets. Our simulations demonstrate that the solution methods are numerically stable and achieve comparable accuracies. Among them, the global matrix method for computing the amplitudes of the characteristic solutions is the most efficient.

## 1 Introduction

Time-step methods (Liu et al., 2013; Heale et al., 2014; Fritts et al., 2015) are used to solve fully nonlinear sets of governing equations, allowing for the modeling of wave breaking, secondary wave generation, and weakly nonlinear effects. However, as compared to linear methods for gravity waves (Midgley and Liemohn, 1966; Volland, 1969a, b; Francis, 1973; Yeh and Liu, 1974; Klostermeyer, 1972, 1980) they are computationally expensive. In (Pütz et al., 2019) it was found that a time-step model took several hours to run, while a linear method only took several seconds. In this regard, linear methods are more suitable for analyzing measured data.

The linearized equations can be transformed into a linear system of ordinary differential equations with variable coefficients that depend on the background atmospheric parameters and their height derivatives (the atmospheric parameters are assumed to be horizontally uniform but vertically varying). A common technique for integrating the linearized equations is the multilayer method first applied by Pfeffer and Zarichny (1962). In this method, the atmosphere is divided into a sequence of thin layers, and in each layer, a linear system of ordinary differential equations with constant coefficients is solved. The analytic wave



solutions in neighboring layers are matched by the continuity condition of the variables across the interface. There are two methods for deriving a linear system of ordinary differential equations with constant coefficients.

1. In the standard multilayer method, the atmospheric parameters, and in particular, the temperature and wind velocity, are assumed to be constant within each layer (Midgley and Liemohn, 1966; Volland, 1969a, b; Francis, 1973; Yeh and Liu, 1974). As a result of the piecewise constant approximation, the height derivatives of the atmospheric parameters are zero within each layer.

2. In the nonstandard multilayer method, the coefficients as a whole are approximated by their values in the middle of the layer (Klostermeyer, 1972). As a result, the height derivatives of the atmospheric parameters (approximated by their values in the middle of the layers) are also included in the resulting system of equations.

The criticism of standard multilayer methods by Hines (Hines, 1973) concerns whether the equations describing the state variables in a layer are physically realistic. He concluded that it is impossible to find the appropriate variables when either (i) the viscosity and the wind velocity are nonzero or (ii) the thermal conductivity and the temperature height derivative are nonzero. However, as mentioned by Knight et al. (2022), Hines' concern about the physical meaning of the state variables is not relevant for a nonstandard multilayer method. The reason is that in a purely mathematical context, it is sufficient to prove that the method converges to a correct solution in the infinitesimally thin layer limit. A justification of this result, based on a matrix–exponential representation for the solution, can be found in (Klostermeyer, 1980).

According to Volland (1969a), a layer is said to be isothermal if the background temperature is constant, and homogeneous if the kinematic viscosity, and so, the thermal diffusivity, is constant. In the case of an isothermal, homogeneous, and windless atmosphere, as in (Midgley and Liemohn, 1966; Volland, 1969a, b; Francis, 1973; Yeh and Liu, 1974), the dispersion relation, associated to the system of ordinary differential equations, separates into three pairs of ascending and descending gravity-wave, viscosity-wave, and thermal conduction-wave modes. The viscosity-wave and thermal conduction-wave modes are also referred to as dissipative modes. The main distinction between the two pairs of dissipative modes and the pair of gravity-wave modes is that the latter have smaller vertical wavenumber imaginary parts. This means that ascending gravity-wave modes do not decrease in amplitude as rapidly with increasing altitude as dissipative modes. On the other hand, if it is assumed that in each layer, the dynamic viscosity is constant, it is not possible to differentiate between ascending and descending modes for some wavenumbers, frequencies, and background parameters (Knight et al., 2022, 2019, 2021). In this context, Knight et al. (2022) explained that the problem of distinguishing ascending from descending modes is related to the problematic branch points of the root functions giving the vertical wavenumber as a function of complex frequency. Along this line, the authors proposed a technique called imaginary frequency shift to assist in achieving this separation.

The inclusion of dissipative modes in a linearized model produces a numerical swamping for floating-point arithmetic (Maeda, 1985) in which certain descending modes grow so rapidly in the upward direction that numerical overflow occurs when the system of differential equation is subject to lower and upper boundary conditions. Several methods have been proposed to reduce numerical swamping.



1. Midgley and Liemohn (1966) used an iterative method that can be regarded as a Gauss–Seidel group iteration. For the Gauss–Seidel iteration to converge, it is essential that the gravity solutions are only minimally coupled with the dissipative solutions. However, this condition is violated at critical layers where the eigenvalues of the gravity solutions can be nearly identical to those of one or more dissipative solutions. The condition's failure, resulting in significant coupling, manifests at altitudes of around 200 km or higher (Volland, 1969a, b). Klostermeyer (1980) avoids this problem by introducing the so called transfer matrix, which relates the amplitudes of the waves at different altitude levels. The atmosphere is divided into rough layers, such that the critical level is approximately in the middle of a rough layer. The transfer matrix equations for the rough layers are solved by the Gauss–Seidel iteration, while the fine structure inside a rough layer is determined by a direct solution of the transfer matrix equations specific to that layer.

2. Volland (1969b) applied the scattering matrix formalism to a three-layer atmosphere assuming (i) abrupt changes in variables at the interfaces between the different layers and (ii) that certain background parameters remain constant in the lower and upper layers. Knight et al. (2019) also formulated the problem in terms of scattering matrices which are closely related to the reflection and transmission matrices appearing in seismology (Pérez-Álvarez and García-Moliner, 2004). However, in contrast to Volland, the authors used a more rigorous approach, i.e., a sequence of composed scattering matrices instead of just a single stand-alone scattering matrix.

3. Maeda (1985) defined numerical swamping as the annihilation of linear independence among supposedly independent solutions. To address this challenge and to obtain a comprehensive set of special solutions that are linearly independent, he utilized a technique developed by Inoue and Horowitz (1966).

In radiative transfer, it is also necessary to solve a linear system of ordinary differential equations with constant coefficients. This arises by transforming the continuous dependence of radiance on direction into a dependence on a discrete set of direction. The standard methods for solving the linear system of ordinary differential equations are the discrete ordinate method (Stamnes, 1986; Wick, 1943; Chandrasekhar, 1950; Stamnes and Swanson, 1981) and the matrix operator method (Plass et al., 1973; Kattawar et al., 1973; van de Hulst, 1963; Nakajima and Tanaka, 1986). In the classical discrete ordinate method, the solution to these equations is expressed as a linear combination of characteristic solutions of the discretized problem. Conversely, the matrix operator method focuses on numerical computations of reflection and transmission matrices. Both methods can be formulated using the matrix exponential formalism. In the framework of the so called discrete ordinate method with matrix exponential, Doicu and Trautmann (2009a, b) designed stable numerical algorithms for computing the radiance field in a multi-layered atmosphere, while in the framework of the matrix operator method with matrix exponential, Budak et al. (2011, 2012) provided explicit and stable representations for the reflection and transmission matrices. A consistent overview of the matrix exponential description of radiative transfer can be found in (Efremenko et al., 2017; Efremenko and Kokhanovsky, 2021).

The main purpose of this article is to apply radiative transfer techniques to solve the linearized gravity-wave equations. As a prototype, we will consider the equations that describe gravity waves in the ionosphere, and that include viscosity, thermal conduction, and ion drag. In principle, a full wave model for the ionosphere comprises the hydrodynamic equations for the neutral atmosphere and the ionospheric equations. These two sets of equations are coupled through the ion drag, and should





be solved together. However, in order to simplify the analysis, we will decouple the two sets of equations by employing an approximation which is due to Klostermeyer (1972). This approximation, which states that the ion velocity can be estimated by the air-dragged ion velocity, allows us to solve the neutral-atmosphere equations separately.

Our paper is organized as follows. In Section 2 we review the linearized neutral-atmosphere and ionospheric equations and describe the method of Klostermeyer for decoupling these two sets of equations. In Section 3 we present the derivation of the matrix exponential solution of the linearized equations, while in Section 4 we describe stable numerical methods for computing this solution. Section 5, which is completely inspired by the works of Knight et al. (2022, 2019, 2021), addresses time-dependent boundary conditions (source functions) when the linearized equations are solved in the frequency domain. The concept of causality, rigorously addressed in (Knight et al., 2022, 2019, 2021), will also be briefly discussed.

## 2 Linearized equations

To design a full wave model for the ionosphere, we use the hydrodynamic equations for the neutral atmosphere and the ionospheric equations. In a linearized (perturbation) method, a quantity $f$ is expressed as

$$f = f_0 + f', \tag{1}$$

where $f_0$ and $f'$ are the unperturbed (background) and the perturbed quantity, respectively. The perturbations are assumed to be small so that it is justified to neglect all terms of higher than the first order. Concretely, we need to solve

1. the linearized hydrodynamic equations for the neutral atmosphere (e.g., (Midgley and Liemohn, 1966; Volland, 1969a))

$$\frac{\partial \rho'}{\partial t} = -\mathbf{u}' \cdot \nabla \rho_0 - \rho_0 \nabla \cdot \mathbf{u}' - \mathbf{u}_0 \cdot \nabla \rho' - \rho' \nabla \cdot \mathbf{u}_0, \tag{2}$$

$$\rho_0 \frac{\partial \mathbf{u}'}{\partial t} = -\nabla p' + \rho' \mathbf{g} + [\nabla \cdot \overline{\boldsymbol{\sigma}}]'$$
$$- \rho'(\mathbf{u}_0 \cdot \nabla)\mathbf{u}_0 - \rho_0(\mathbf{u}_0 \cdot \nabla)\mathbf{u}' - \rho_0(\mathbf{u}' \cdot \nabla)\mathbf{u}_0 - \mathbf{f}'_{\mathrm{ID}}, \tag{3}$$

$$\rho_0 c_{\mathrm{v}} \frac{\partial T'}{\partial t} = -p_0 \nabla \cdot \mathbf{u}' - p' \nabla \cdot \mathbf{u}_0 + [\overline{\boldsymbol{\sigma}} : \nabla \mathbf{u}]' + [\nabla \cdot (\lambda \nabla T)]'$$
$$- \rho' c_{\mathrm{v}}(\mathbf{u}_0 \cdot \nabla)T_0 - \rho_0 c_{\mathrm{v}}(\mathbf{u}_0 \cdot \nabla)T' - \rho_0 c_{\mathrm{v}}(\mathbf{u}' \cdot \nabla)T_0 - P'_{\mathrm{ID}}, \tag{4}$$

$$\frac{p'}{p_0} = \frac{T'}{T_0} + \frac{\rho'}{\rho_0}, \tag{5}$$

2. the linearized ion continuity and momentum equations (Huba et al., 2000)

$$\frac{\partial n'_i}{\partial t} + \nabla \cdot (n'_i \mathbf{u}_{0i}) + \nabla \cdot (n_{0i} \mathbf{u}'_i) = P'_i - n'_i \mathcal{L}_{0i} - n_{0i} \mathcal{L}'_i, \tag{6}$$





and

$$\frac{\partial \mathbf{u}_i'}{\partial t} + (\mathbf{u}_i' \cdot \nabla)\mathbf{u}_{0i} + (\mathbf{u}_{0i} \cdot \nabla)\mathbf{u}_i' = -\frac{1}{m_i}\left[\frac{K}{n_i}\nabla(n_i T_i) + \frac{K}{n_e}\nabla(n_e T_e)\right]'$$
$$+ \frac{e}{m_i c}[(\mathbf{u}_i - \mathbf{u}_e) \times \mathbf{B}]'$$
$$- \nu_{0in}(\mathbf{u}_i' - \mathbf{u}) - \nu_{in}'\mathbf{u}_{0i}$$
$$- \sum_j \nu_{0ij}(\mathbf{u}_i' - \mathbf{u}_j') - \sum_j \nu_{ij}'(\mathbf{u}_{0i} - \mathbf{u}_{0j}), \tag{7}$$

respectively, and

3. the linearized electrically neutral equation

$$n_e' = \sum_i n_i'. \tag{8}$$

In the neutral-atmosphere equations, $\rho$ is the density, $p$ the pressure, $T$ the temperature, $\mathbf{u}$ the velocity, $\mathbf{g}$ the gravitational acceleration, $c_\mathrm{v}$ the specific heat at constant volume, $\lambda$ the coefficient of thermal conductivity, $\overline{\sigma}$ the viscous stress tensor,

$\mathbf{f}_{\mathrm{ID}} = \rho\nu_{ni}(\mathbf{u} - \mathbf{u}_i)$ $\tag{9}$

the ion-drag force per unit volume,

$$P_{\mathrm{ID}} = \mathbf{f}_{\mathrm{ID}} \cdot (\mathbf{u} - \mathbf{u}_i) = \rho\nu_{ni}|\mathbf{u} - \mathbf{u}_i|^2 \tag{10}$$

the rate of work done by the ion drag force per unit volume, and $\nu_{ni}$ the neutral–ion collision frequency (the collision frequency between a neutral particle and all kind of ions). In the hydrodynamic equations, we neglected the Coriolis force, because we

are interested in gravity waves with an angular frequency $\omega > 2\Omega$, where $\Omega = 7.3 \times 10^{-5}$ s$^{-1}$ is the Earth's angular velocity. In the ionospheric equations, $n_i$, $\mathbf{u}_i$, $p_i = n_i K T_i$, $T_i$, and $m_i$ are the number density, velocity, partial pressure, temperature and mass of ion $i$, respectively, $n_e$, $\mathbf{u}_e$, $p_e = n_e K T_e$, $T_e$, and $m_e$ are the number density, velocity, partial pressure, temperature and mass of electrons, respectively, $\mathbf{B}$ is the magnetic induction, $e$ the electron charge, $c$ the speed of light, $K$ the Boltzmann constant, $P_i$ and $\mathcal{L}_i$ the ionization production rate and the loss rate due to chemical processes of ion $i$, respectively, $\nu_{in}$ the

collision frequency between the ion $i$ and the neutral $n$, $\nu_{ij}$ the collision frequency between the ions $i$ and $j$, and $\nu_{in} = \sum_n \nu_{i,n}$. Note that in the ionosphere, the neutral-ion collision frequency $\nu_{ni}$ is computed under the assumption that atomic oxygen O and O$^+$-ions are the main neutral and ionic constituents, that is, for $n = $ O and $i = $ O$^+$. Also note that Eq. (7) is in fact a linear combination of the ion and electron momentum equations (Huba et al., 2000).

The linearized neutral-atmosphere and ionospheric equations are coupled through the ion-drag perturbations $\mathbf{f}_{\mathrm{ID}}'$ and $P_{\mathrm{ID}}'$,

given respectively, by

$$\mathbf{f}_{\mathrm{ID}}' = \rho_0\nu_{0ni}(\mathbf{u}' - \mathbf{u}_i') + (\rho_0\nu_{ni}' + \rho'\nu_{0ni})(\mathbf{u}_0 - \mathbf{u}_{0i}), \tag{11}$$
$$P_{\mathrm{ID}}' = 2\rho_0\nu_{0ni}(\mathbf{u}_0 - \mathbf{u}_{0i}) \cdot (\mathbf{u}' - \mathbf{u}_i') + (\rho_0\nu_{ni}' + \rho'\nu_{0ni})|\mathbf{u}_0 - \mathbf{u}_{0i}|^2, \tag{12}$$





where the neutral-ion collision frequency $\nu_{ni}$ depends on the ion number density $n_i$ according to the relation $\nu_{ni} = (n_i/n_n)\nu_{in}$.

Klostermeyer (1972) proposed a simple method for solving the coupled systems of equations under the assumption that the

ion velocity can be approximated by the air-dragged ion velocity, i.e.,

$$\mathbf{u}_i \approx \mathbf{u}_{\mathrm{w}} = (\mathbf{u} \cdot \widehat{\mathbf{b}})\widehat{\mathbf{b}}, \tag{13}$$

where $\widehat{\mathbf{b}}$ is a unit vector in the direction of the magnetic induction. Assumption (13), which was employed, for example, before by Volland (1969a) and later by Francis (1973), implies $\mathbf{u}_{0i} = (\mathbf{u}_0 \cdot \widehat{\mathbf{b}})\widehat{\mathbf{b}}$ and $\mathbf{u}_i' = (\mathbf{u}' \cdot \widehat{\mathbf{b}})\widehat{\mathbf{b}}$ (a perturbation of $\widehat{\mathbf{b}}$ in the direction of the magnetic field is not considered), and eliminates the need of using the ion momentum equation (7). With $\mathbf{u}_i$ as in Eq.

(13), Eqs. (9) and (10) become

$$\mathbf{f}_{\mathrm{ID}} = \rho\nu_{ni}[\mathbf{u} - (\mathbf{u} \cdot \widehat{\mathbf{b}})\widehat{\mathbf{b}}], \tag{14}$$

$$P_{\mathrm{ID}} = \rho\nu_{ni}[|\mathbf{u}|^2 - (\mathbf{u} \cdot \widehat{\mathbf{b}})^2], \tag{15}$$

and their perturbed versions are,

$$\mathbf{f}_{\mathrm{ID}}' = \rho_0\nu_{0ni}[\mathbf{u}' - (\mathbf{u}' \cdot \widehat{\mathbf{b}})\widehat{\mathbf{b}}] + (\rho_0\nu_{ni}' + \rho'\nu_{0ni})[\mathbf{u}_0 - (\mathbf{u}_0 \cdot \widehat{\mathbf{b}})\widehat{\mathbf{b}}], \tag{16}$$

$$P_{\mathrm{ID}}' = 2\rho_0\nu_{0ni}[\mathbf{u}_0 \cdot \mathbf{u}' - (\mathbf{u}_0 \cdot \widehat{\mathbf{b}})(\mathbf{u}' \cdot \widehat{\mathbf{b}})] + (\rho_0\nu_{ni}' + \rho'\nu_{0ni})[|\mathbf{u}_0|^2 - (\mathbf{u}_0 \cdot \widehat{\mathbf{b}})^2]. \tag{17}$$

In fact, Klostermeyer solved the linearized neutral-atmosphere equations (2)–(5) together with the following ion continuity equation:

$$\frac{\partial n_i'}{\partial t} + n_{0i}\nabla \cdot \mathbf{u}_i' + \mathbf{u}_i' \cdot \nabla n_{0i} = 0, \ i = \mathrm{O}^+. \tag{18}$$

where, as already mentioned, $\mathbf{u}_i' = (\mathbf{u}' \cdot \widehat{\mathbf{b}})\widehat{\mathbf{b}}$. The ion continuity equation (18) is a simplified version of Eq. (6), in which

the term $\nabla \cdot (n_i'\mathbf{u}_{0i})$, as well as, the perturbed production and loss terms are neglected. Neglecting the term $\nabla \cdot (n_i'\mathbf{u}_{0i})$, where $\mathbf{u}_{0i} = (\mathbf{u}_0 \cdot \widehat{\mathbf{b}})\widehat{\mathbf{b}}$, means that in the continuity equation, the wind (background) velocity $\mathbf{u}_0$ is omitted. However, in Klostermeyer's formalism, this assumption is then relaxed, i.e., the ion-drag perturbations $\mathbf{f}_{\mathrm{ID}}'$ and $P_{\mathrm{ID}}'$ are computed by means of Eqs. (16)–(17), which still include the wind velocity. Adopting this solution method, we (i) use Eq. (18) to express $n_i'$ in terms of $\mathbf{u}'$, (ii) assume that in the ionosphere, the atomic oxygen O and $\mathrm{O}^+$-ions are the main neutral and ionic constituents,

and compute $\nu_{0ni}$ and $\nu_{ni}'$ by means of the relations (Stubbe, 1968; Shibata, 1983)

$$\nu_{0ni} = 7.22 \times 10^{-17}T_0^{0.37}n_{0i}, \ n = \mathrm{O}, i = \mathrm{O}^+, \ (\text{in MKS units}) \tag{19}$$

and

$$\frac{\nu_{ni}'}{\nu_{0ni}} = 0.37\frac{T'}{T_0} + \frac{n_i'}{n_{0i}}, \ n = \mathrm{O}, i = \mathrm{O}^+, \tag{20}$$

respectively, (iii) substitute these results into Eqs. (16) and (17), and (iv) solve the neutral-atmosphere equations (2)–(5) to-

gether with Eqs. (16)–(17) for $\mathbf{u}'$, $\rho^\jmath$, $p'$ and $T'$.

Some comments can be made here.





1. Shibata (1983) used a more precise representation for the ion velocity, that is, $\mathbf{u}_i = \mathbf{u}_{\mathrm{w}} + \mathbf{u}_{\mathrm{d}i} = (\mathbf{u} \cdot \widehat{\mathbf{b}})\widehat{\mathbf{b}} + \mathbf{u}_{\mathrm{d}i}$, where

$$\mathbf{u}_{\mathrm{d}i} = -D_{\mathrm{a}}\left(\frac{1}{n_i}\frac{\partial n_i}{\partial b} + \frac{1}{T_i}\frac{\partial T_i}{\partial b}\right)\widehat{\mathbf{b}} + \frac{1}{\nu_{in}}(\mathbf{g} \cdot \widehat{\mathbf{b}})\widehat{\mathbf{b}}, \; i = \mathrm{O}^+, n = \mathrm{O}, \tag{21}$$

is the plasma diffusion velocity, $D_{\mathrm{a}} = 2KT/(m_i \nu_{in})$ the ambipolar diffusion coefficient, and $b$ a coordinate along the magnetic induction. Under the assumption $T_i \approx T$, a perturbed version of Eq. (21) is solved together with (i) the ion continuity equation (6), in which the production and loss terms are neglected, and (ii) the linearized neutral-atmosphere equations (2)–(5).

2. In the model that we designed, we use the assumption $\mathbf{u}_i = (\mathbf{u} \cdot \widehat{\mathbf{b}})\widehat{\mathbf{b}}$ and solve the neutral-atmosphere equations (2)–(5) as in Klostermeyer's model. A next step could be the one in which we revise the assumption on the ion velocity and use the wave-induced perturbations (obtained by solving Eqs. (2)–(5)) to compute the perturbed ion density $n_i'$ and the perturbed ion velocity along the magnetic field line $u_{\mathrm{B}}' = \mathbf{u}_i' \cdot \widehat{\mathbf{b}}$ from the ionospheric equations (6)–(8). This topic will be discussed in more detail in the Conclusions.

## 3 General solution

The linearized neutral-atmosphere equations (2)–(5) are solved under the following assumptions:

**A1.** The geographic and geomagnetic coordinates are identical.

**A2.** The wave propagates in the meridional direction (the $x$-coordinate is positive southwards while the $z$-coordinate is positive upwards), i.e.,

$$f = f(x, z, t). \tag{22}$$

**A3.** All background (unperturbed) quantities vary only in the $z$-direction, i.e.,

$$f_0 = f_0(z), \tag{23}$$

while all perturbations vary harmonically in time and the $x$-direction, i.e.,

$$f' = f'(x, z, t) = \overline{f}(z)\mathrm{e}^{\mathrm{j}(\omega t - k_{\mathrm{x}} x)}, \tag{24}$$

where $\omega$ is the angular frequency and $k_{\mathrm{x}}$ the horizontal wavenumber.

**A4.** The atmosphere consists of a number of layers. In each layer, the dynamic (molecular) viscosity $\mu$ and the thermal conductivity $\lambda$ are computed as (Dalgarno and Smith, 1962)

$$\mu = 3.34 \times 10^{-7} T^{0.71}, \; \lambda = \frac{c_{\mathrm{p}}\mu}{\mathrm{Pr}} = 6.71 \times 10^{-4} T^{0.71} \text{ (all in MKS units)} \tag{25}$$





and their perturbed values as

$$\frac{\mu'}{\mu_0} = 0.71\frac{T'}{T_0}, \ \frac{\lambda'}{\lambda_0} = 0.71\frac{T'}{T_0}, \tag{26}$$

where $c_\mathrm{p}$ is the specific heat at constant pressure and Pr the Prandtl number. From Eq. (25) it is apparent that a constant background dynamic viscosity $\mu_0$ implies a constant background thermal conductivity $\lambda_0$ (Pr is constant for a given atmospheric composition), and obviously, a constant background kinematic viscosity $\mu_\mathrm{k} = \mu_0/\rho_0$ implies a constant "kinematic" thermal conductivity $\lambda_\mathrm{k} = \lambda_0/\rho_0 = \alpha_0 c_\mathrm{p}$, or a constant thermal diffusivity $\alpha_0$. In each layer, we assume non-zero height derivatives of the background temperature and wind velocity, as well as, a constant background dynamic viscosity, i.e.,

$$\frac{\mathrm{d}T_0}{\mathrm{d}z} \neq 0, \ \frac{\mathrm{d}u_0}{\mathrm{d}z} \neq 0, \ \mu_0 = \text{constant, and } \lambda_0 = \text{constant}, \tag{27}$$

where $\mathbf{u}_0 = (u_0, 0, 0)$ is the wind velocity. Along this line it should be pointed out that (i) in an isothermal atmosphere with constant wind velocity, $T_0$ and $u_0$ are supposed to be constant in each layer, i.e., $\mathrm{d}T_0/\mathrm{d}z = 0$ and $\mathrm{d}u_0/\mathrm{d}z = 0$, while (ii) in an homogeneous atmosphere, the background kinematic viscosity $\mu_\mathrm{k}$ is supposed to be constant in each layer.

In Appendix A it is shown that under the above assumptions, the linearized equations lead to a linear system of ordinary differential equations, written in matrix form as

$$\frac{1}{k_\mathrm{x}}\frac{\mathrm{d}\mathbf{e}}{\mathrm{d}z} = \mathrm{A}\mathbf{e}, \tag{28}$$

where

$$\mathbf{e} = [\widehat{u}, \widehat{w}, \widehat{p}, \widehat{T}, \widehat{\mathcal{U}}, \widehat{\mathcal{T}}]^T \tag{29}$$

is the unknown solution vector, and A is the propagation matrix with altitude independent elements (whose expressions follow from Eqs. (A28)–(A33) of A). In general, the unknowns (the hat quantities in Eq. (29)) are defined through the relation

$$\overline{f}(z) = C(z)\widehat{f}(z), \tag{30}$$

where $\overline{f}$ is defined by Eq. (24), and $C$ is a known quantity that ensures that $\widehat{f}$ is dimensionless and that may or may not depend on altitude (here, we indicate that $C$ depends on $z$). Specifically, for $\mathbf{u}_0 = (u_0, 0, 0)$ and $\mathbf{u}' = (u', 0, w')$, we have (cf. Eqs. (A22)–(A26) of A)

$$\overline{u}(z) = \frac{\omega_0}{k_\mathrm{x}}\widehat{u}(z), \tag{31}$$

$$\overline{w}(z) = \frac{\omega_0}{k_\mathrm{x}}\widehat{w}(z), \tag{32}$$

$$\overline{p}(z) = p_0(z)\widehat{p}(z), \tag{33}$$

$$\overline{T}(z) = T_0(z)\widehat{T}(z), \tag{34}$$





and

$$\widehat{\mathcal{U}} = 3\eta \frac{1}{k_x} \frac{\mathrm{d}\widehat{u}}{\mathrm{d}z}, \quad \widehat{\mathcal{T}} = \nu \frac{1}{k_x} \frac{\mathrm{d}\widehat{T}}{\mathrm{d}z}, \tag{35}$$

where $\omega_0$ is a reference frequency, and the dimensionless parameters $\eta$ and $\nu$ are given by Eq. (A27) of A.

If $(\lambda_n, \mathbf{v}_n)$ is an eigenpair of the matrix A, i.e., $A\mathbf{v}_n = \lambda_n \mathbf{v}_n$ for $n = 1, \dots, N$, where $N = \dim(\mathbf{e})$, the general solution of Eq. (28) is a linear combination of the characteristic solutions $\exp(k_x \lambda_n z)\mathbf{v}_n$, that is,

$$\mathbf{e}(z) = \sum_{n=1}^{N} a_n \mathrm{e}^{k_x \lambda_n z} \mathbf{v}_n$$

$$= [\mathbf{v}_1, \dots \mathbf{v}_N] \begin{bmatrix} \mathrm{e}^{k_x \lambda_1 z} & \cdots & 0 \\ \vdots & \ddots & \vdots \\ 0 & \cdots & \mathrm{e}^{k_x \lambda_N z} \end{bmatrix} \begin{bmatrix} a_1 \\ \vdots \\ a_N \end{bmatrix}$$

$$= V \mathrm{diag}[\mathrm{e}^{k_x \lambda_n z}]\mathbf{a}, \tag{36}$$

where

$$V = [\mathbf{v}_1, \dots, \mathbf{v}_N], \mathrm{diag}[\mathrm{e}^{k_x \lambda_n z}] = \begin{bmatrix} \mathrm{e}^{k_x \lambda_1 z} & \cdots & 0 \\ \vdots & \ddots & \vdots \\ 0 & \cdots & \mathrm{e}^{k_x \lambda_N z} \end{bmatrix}, \tag{37}$$

and $\mathbf{a} = [a_1, \dots, a_N]^T$. At $z = 0$, we have $\mathbf{e}(0) = V\mathbf{a}$; thus,

$$\mathbf{a} = V^{-1}\mathbf{e}(0), \tag{38}$$

implying (cf. Eq. (36)),

$$\mathbf{e}(z) = V\mathrm{diag}[\mathrm{e}^{k_x \lambda_n z}]V^{-1}\mathbf{e}(0) = \mathrm{e}^{k_x A z}\mathbf{e}(0), \tag{39}$$

and conversely,

$$\mathbf{e}(0) = V\mathrm{diag}[\mathrm{e}^{-k_x \lambda_n z}]V^{-1}\mathbf{e}(z) = \mathrm{e}^{-k_x A z}\mathbf{e}(z). \tag{40}$$

From the theory of gravity waves within an isothermal, nondissipative atmosphere, it is generally known that the amplitude of an ascending mode increases like $\exp[z/(2H_a)]$, where $H_a$ is the atmospheric scale height (Hines, 1960). This is necessary to keep the wave energy constant in an atmosphere where the pressure decreases exponentially with height. In this regard, we define the vertical wavenumber $k_{zn}$ through the relation

$$\mathrm{diag}[\mathrm{e}^{k_x \lambda_n z}] = \mathrm{diag}[\mathrm{e}^{z/(2H_a)}\mathrm{e}^{-\mathrm{j}k_{zn}z}], \tag{41}$$

yielding

$$\lambda_n = -\frac{\mathrm{j}}{k_x}k_{zn} + \frac{1}{2}\alpha, \tag{42}$$

 

and conversely,

$$k_{\mathrm{z}n} = \mathrm{j}k_{\mathrm{x}}\left(\lambda_n - \frac{1}{2}\alpha\right),\tag{43}$$

where $\alpha = 1/(k_{\mathrm{x}}H_{\mathrm{a}})$. The characteristic equation $\det(\mathrm{A} - \lambda\mathrm{I}_N) = 0$ has $N = 6$ solutions. As shown in A, in the case of an isothermal, homogeneous, and windless atmosphere, these solutions appear in pairs and correspond to (i) ascending and descending gravity-wave modes, (ii) ascending and descending viscosity-wave modes, and (iii) ascending and descending thermal conduction-wave modes. This classification is made according to the imaginary part of the vertical wavenumber $k_{\mathrm{z}n}$. In the more realistic case of a constant background dynamic viscosity, it is generally not possible to define ascending and descending modes as corresponding pairs. However, in our model we will use the same rule as in the case of an isothermal and homogeneous atmosphere, even though the traditional concept of classifying waves in pairs is no longer applicable. Specifically, we compute $k_{\mathrm{z}n}$ for $n = 1, \ldots, N$ by means of Eq. (25), and order the set $\{k_{\mathrm{z}n}\}_{n=1}^N$, and accordingly, $\{\lambda_n\}_{n=1}^N$, such that

$$\mathrm{Im}(k_{\mathrm{z}3}) < \mathrm{Im}(k_{\mathrm{z}2}) < \mathrm{Im}(k_{\mathrm{z}1}) < \mathrm{Im}(k_{\mathrm{z}4}) < \mathrm{Im}(k_{\mathrm{z}5}) < \mathrm{Im}(k_{\mathrm{z}6}).\tag{44}$$

By convention, (i) the pairs $(k_{\mathrm{z}1} = k_{\mathrm{z}1}^+, \lambda_1 = \lambda_1^+)$ and $(k_{\mathrm{z}4} = k_{\mathrm{z}1}^-, \lambda_4 = \lambda_1^-)$ will correspond to ascending and descending gravity-wave modes, respectively, (ii) the pairs $(k_{\mathrm{z}2} = k_{\mathbf{z}2}^+, \lambda_2 = \lambda_2^+)$ and $(k_{\mathrm{z}5} = k_{\mathbf{z}2}^-, \lambda_5 = \lambda_2^-)$ to ascending and descending viscosity-wave modes, respectively, and (iii) the pairs $(k_{\mathrm{z}3} = k_{\mathrm{z}3}^+, \lambda_3 = \lambda_3^+)$ and $(k_{\mathrm{z}6} = k_{\mathrm{z}3}^-, \lambda_6 = \lambda_3^-)$ to ascending and descending thermal conduction-wave modes, respectively. Thus, the vertical wavenumber is an auxiliary quantity that is used only to identify the different modes. According to the notation introduced above, $\{\lambda_m^+\}_{m=1}^M$, where $M = N/2$ is the number of modes, is the set of eigenvalues defining ascending modes, and $\{\lambda_m^-\}_{m=1}^M$ is the set of eigenvalues defining descending modes. Because $\mathrm{Re}(\lambda_n) = \mathrm{Im}(k_{\mathrm{z}n})/k_{\mathrm{x}} + \alpha/2$, it is obvious that we can renounce on the vertical wavenumber when identifying the different wave modes. We can simply order the set $\{\lambda_n\}_{n=1}^N$, such that

$$\mathrm{Re}(\lambda_3) < \mathrm{Re}(\lambda_2) < \mathrm{Re}(\lambda_1) < \mathrm{Re}(\lambda_4) < \mathrm{Re}(\lambda_5) < \mathrm{Re}(\lambda_6),\tag{45}$$

and use the same classification rule as above. Knight et al. (2019) employed this approach, providing a more intuitive explanation compared to the analogy with an isothermal and homogeneous atmosphere: For increasing $z$, the exponential term $\exp(k_{\mathrm{x}}\lambda_n z)$ will tend to be damped more for ascending modes than for descending modes; conversely, for decreasing $z$, the roles of ascending and descending modes are reversed.

To highlight the different wave modes, we organize the solution vector $\mathbf{e}(z)$ as

$$\begin{aligned}\mathbf{e}(z) &= \mathbf{e}_+(z) + \mathbf{e}_-(z)\\ &= \left(\sum_{m=1}^M a_m^+ \mathrm{e}^{k_{\mathrm{x}}\lambda_m^+ z}\mathbf{v}_m^+\right) + \left(\sum_{m=1}^M a_m^- \mathrm{e}^{k_{\mathrm{x}}\lambda_m^- z}\mathbf{v}_m^-\right)\\ &= [\mathrm{V}_+, \mathrm{V}_-]\begin{bmatrix}\mathrm{diag}[\mathrm{e}^{k_{\mathrm{x}}\lambda_m^+ z}] & \mathrm{O}_M\\ \mathrm{O}_M & \mathrm{diag}[\mathrm{e}^{k_{\mathrm{x}}\lambda_m^- z}]\end{bmatrix}\begin{bmatrix}\mathbf{a}_+\\ \mathbf{a}_-\end{bmatrix},\end{aligned}\tag{46}$$





where the eigenvector $\mathbf{v}_m^{\pm}$ corresponds to the eigenvalue $\lambda_m^{\pm}$,

$$V = [V_+, V_-], \ V_{\pm} = [\mathbf{v}_1^{\pm}, \ldots, \mathbf{v}_M^{\pm}], \tag{47}$$

$$\mathbf{a} = \begin{bmatrix} \mathbf{a}_+ \\ \mathbf{a}_- \end{bmatrix}, \ \mathbf{a}_{\pm} = \begin{bmatrix} a_1^{\pm} \\ \vdots \\ a_M^{\pm} \end{bmatrix}, \tag{48}$$

and $O_M$ is the zero matrix of dimension $M \times M$. Some useful relations are listed below

1. From Eq. (38), we find

$$\mathbf{a}_+ = [I_M, O_M]\mathbf{a} = [I_M, O_M]V^{-1}\mathbf{e}(0), \tag{49}$$

$$\mathbf{a}_- = [O_M, I_M]\mathbf{a} = [O_M, I_M]V^{-1}\mathbf{e}(0), \tag{50}$$

where $I_M$ is the identity matrix of dimension $M \times M$.

2. From Eq. (46), that is,

$$\mathbf{e}_{\pm}(z) = \sum_{m=1}^{M} a_m^{\pm} e^{k_x \lambda_m^{\pm} z} \mathbf{v}_m^{\pm} = V_{\pm} \mathrm{diag}[e^{k_x \lambda_m^{\pm} z}]\mathbf{a}_{\pm}, \tag{51}$$

we deduce that

$$\mathbf{e}_{\pm}(0) = V_{\pm}\mathbf{a}_{\pm}. \tag{52}$$

3. From Eq. (39), we obtain

$$\mathbf{e}_+(z) = T_+\mathbf{e}(0), \tag{53}$$

where

$$T_+ = V \begin{bmatrix} \mathrm{diag}[e^{k_x \lambda_m^+ z}] & O_M \\ O_M & O_M \end{bmatrix} V^{-1}, \tag{54}$$

while from Eq. (40), we find

$$\mathbf{e}_-(0) = T_-\mathbf{e}(z), \tag{55}$$

where

$$T_- = V \begin{bmatrix} O_M & O_M \\ O_M & \mathrm{diag}[e^{-k_x \lambda_m^- z}] \end{bmatrix} V^{-1}. \tag{56}$$





## 4 Solution for a stratified atmosphere

Consider an equidistant discretization of the atmosphere, i.e., $\widehat{z}_i = z_{\min} + (i-1)\Delta\widehat{z}$ for $i = 1, ..., 2L+1$. A layer $l$, where $l = 1, ..., L$ and $L$ is the number of layers, is bounded from below and from above by the grid points $z_l = \widehat{z}_{2l-1}$ and $z_{l+1} = \widehat{z}_{2l+1}$, respectively, and its center is located at the grid point $\overline{z}_l = \widehat{z}_{2l}$. The atmosphere extends from $z_{\min} = z_1 = \widehat{z}_1$ to $z_{\max} = z_{L+1} = \widehat{z}_{2L+1} = z_{\min} + L(2\Delta\widehat{z})$. We adopt a nonstandard multilayer method (Klostermeyer, 1972), and approximate the altitude dependent matrix A in each layer $l$ by its value at the layer center, i.e., $A_l = A(\overline{z}_l)$. The eigenpairs of the propagation matrix $A_l$ are denoted by $(\lambda_{nl}, \mathbf{v}_{nl})$ for $n = 1, ..., N$. The matrix differential equation (28) can be solved for (i) the amplitudes $\mathbf{a}_l$, $l = 1, ..., L$ of the characteristic solutions, or (ii) the discrete values $\mathbf{e}_l = \mathbf{e}(z_l)$, $l = 1, ..., L$ of the solution vector $\mathbf{e}(z)$.

### 4.1 Amplitudes of the characteristic solutions

In the layers $l$ and $l+1$, the solutions are given by (cf. Eq. (36))

$$\mathbf{e}_l(z) = V_l \text{diag}[e^{k_\mathrm{x}\lambda_{nl}(z-z_l)}]\mathbf{a}_l, \ z_l \leq z \leq z_{l+1}, \tag{57}$$

and

$$\mathbf{e}_{l+1}(z) = V_{l+1}\text{diag}[e^{k_\mathrm{x}\lambda_{n,l+1}(z-z_{l+1})}]\mathbf{a}_{l+1}, \ z_{l+1} \leq z \leq z_{l+2}, \tag{58}$$

respectively. The continuity condition at the interface $z = z_{l+1}$,

$$\mathbf{e}_l(z_{l+1}) = \mathbf{e}_{l+1}(z_{l+1}), \tag{59}$$

gives

$$V_l^{-1}V_{l+1}\mathbf{a}_{l+1} = \text{diag}[e^{k_\mathrm{x}\lambda_{nl}\Delta_l}]\mathbf{a}_l, \tag{60}$$

where $\Delta_l = z_{l+1} - z_l$. To obtain a stable system of equations, we define a scaling matrix $S_l^1$ with entries

$$[S_l^1]_{nn} = \begin{cases} e^{-k_\mathrm{x}\lambda_{nl}\Delta_l}, & \text{Re}(\lambda_{nl}) > 0 \\ \\ 1, & \text{Re}(\lambda_{nl}) \leq 0 \end{cases}, \tag{61}$$

and a scaling matrix $S_l^0$ by

$$S_l^0 = S_l^1\text{diag}[e^{k_\mathrm{x}\lambda_{nl}\Delta_l}], \text{ i.e., } [S_l^0]_{nn} = \begin{cases} 1, & \text{Re}(\lambda_{nl}) > 0 \\ \\ e^{k_\mathrm{x}\lambda_{nl}\Delta_l}, & \text{Re}(\lambda_{nl}) \leq 0 \end{cases}. \tag{62}$$

Multiplying Eq. (60) from the left with $S_l^1$ gives the continuity equation

$$\mathbb{A}_{l,l+1}^1\mathbf{a}_{l+1} - \mathbb{A}_{l,l+1}^0\mathbf{a}_l = \mathbf{0}_{2M}, \ l = 1, ..., L-1, \tag{63}$$




where $\mathbf{0}_{2M}$ is the $2M$-dimensional zero vector, and

$$\mathbb{A}^1_{l,l+1} = \mathrm{S}^1_l(\mathrm{V}^{-1}_l\mathrm{V}_{l+1}), \tag{64}$$

$$\mathbb{A}^0_{l,l+1} = \mathrm{S}^0_l. \tag{65}$$

Actually, we have $L-1$ continuity equations imposed at the levels $z_2, \ldots, z_L$ for the $L$ unknowns $\mathbf{a}_1, \ldots, \mathbf{a}_L$. The two missing equations are obtained from the lower and upper boundary conditions.

1. At the lower boundary, i.e., at $z = z_1 (= z_{\min})$, we assume that only the ascending wave modes transport energy upward. In this regard, we impose that in the layer $l = 1$, we have $a^+_{1,l=1} = s = $ finite, and that the rest of $a^+_{m,l=1}$ are zero, that is, $a^+_{m,l=1} = 0$ for $m \neq 1$ (Klostermeyer, 1972). Note that $a^+_{1,l=1}$ is the amplitude of the ascending gravity-wave modes, while the condition $a^+_{m,l=1} = 0$ for $m \neq 1$ means that the amplitudes of the ascending viscosity-wave and thermal conduction-wave modes are assumed to be zero. In this case, the boundary condition for ascending modes is

$$\mathbf{e}^+_{l=1}(z_1) = \begin{bmatrix} \widehat{u}^+_{l=1}(z_1) \\ \widehat{w}^+_{l=1}(z_1) \\ \widehat{p}^+_{l=1}(z_1) \\ \widehat{T}^+_{l=1}(z_1) \\ \widehat{\mathcal{U}}^+_{l=1}(z_1) \\ \widehat{\mathcal{T}}^+_{l=1}(z_1) \end{bmatrix} = \sum_{m=1}^{M} a^+_{m,l=1}\mathbf{v}^+_{m,l=1} = s\mathbf{v}^+_{1,l=1}. \tag{66}$$

Excluding for the moment the scale factor $s$, we express the boundary condition for amplitudes,

$$\mathbf{a}^+_{l=1} = \begin{bmatrix} a^+_{1,l=1} \\ a^+_{2,l=1} \\ \vdots \\ a^+_{M,l=1} \end{bmatrix} = \mathbf{i}_1 \text{ with } \mathbf{i}_1 = \begin{bmatrix} 1 \\ 0 \\ \vdots \\ 0 \end{bmatrix}, \tag{67}$$

in matrix form as

$$[\mathrm{I}_M, \mathrm{O}_M]\mathbf{a}_1 = [\mathrm{I}_M, \mathrm{O}_M]\begin{bmatrix} \mathbf{a}^+_1 \\ \mathbf{a}^-_1 \end{bmatrix} = \mathbf{i}_1, \tag{68}$$

where in general, $\mathbf{a}^{\pm}_{l_0} = \mathbf{a}^{\pm}_{l=l_0}$, for $l_0 = 1, \ldots, L$.

2. A reasonable upper boundary condition is that there is no downgoing energy at great altitudes, so that the amplitudes of all descending wave modes must be zero at the upper boundary (Klostermeyer, 1972). In this regard, we impose $a^-_{m,l=L} = 0$ for all $m = 1, \ldots, M$, in which case, in the layer $L$, the boundary condition for descending modes is

$$\mathbf{e}^-_{l=L}(z) = \sum_{m=1}^{M} a^-_{m,l=L} e^{k_{\mathrm{x}}\lambda^-_{m,l=L}z}\mathbf{v}^-_{m,l=L} = \mathbf{0}_{2M} \tag{69}$$





for all $z_L \leq z \leq z_{L+1}$. In matrix form, the boundary condition for amplitudes

$$\mathbf{a}_{l=L}^- = \begin{bmatrix} a_{1,l=L}^- \\ a_{2,l=L}^- \\ \vdots \\ a_{M,l=L}^- \end{bmatrix} = \mathbf{0}_M \tag{70}$$

is written as

$$[\mathrm{O}_M, \mathrm{I}_M]\mathbf{a}_L = [\mathrm{O}_M, \mathrm{I}_M] \begin{bmatrix} \mathbf{a}_L^+ \\ \mathbf{a}_L^- \end{bmatrix} = \mathbf{0}_M. \tag{71}$$

Comments.

1. The scaling matrices defined by Eqs. (61) and (62) do not take into account a classification of the wave modes as
ascending and descending (as defined by Eq. (45)). Consequently, the continuity equations (63) do not account for this
   classification, and the only equations in which it is necessary to distinguish between ascending and descending modes
   are the boundary condition equations (68) and (71). From this point of view, the method is similar to finite-difference
   methods (Lindzen and Kuo, 1969; Hickey et al., 1998, 2009). For the continuity equations and the boundary condition
   equations to be consistent (for both to rely on a categorization of wave modes as ascending or descending), we may
define the scaling matrices as

$$[\mathrm{S}_l^1]_{nn} = \begin{cases} 1, & n = 1, \dots, M \\ \\ \mathrm{e}^{-k_{\mathrm{x}} \lambda_{nl} \Delta_l}, & n = M+1, \dots, N \end{cases}, \tag{72}$$

and

$$\mathrm{S}_l^0 = \mathrm{S}_l^1 \mathrm{diag}[\mathrm{e}^{k_{\mathrm{x}} \lambda_{nl} \Delta_l}], \text{ i.e., } [\mathrm{S}_l^0]_{nn} = \begin{cases} \mathrm{e}^{k_{\mathrm{x}} \lambda_{nl} \Delta_l}, & n = 1, \dots, M \\ \\ 1, & n = M+1, \dots, N \end{cases}. \tag{73}$$

2. Knight et al. (2022, 2019) specified the lower boundary condition for ascending modes in terms of $M$ values $b_{1,k}$,
$k = 1, \dots, M$, as (compare with Eq. (66))

$$\left[ \frac{\mathrm{d}^{k-1} \mathbf{e}_{l=1}^+}{\mathrm{d}z^{k-1}}(z_1) \right]_q = s b_{1,k}, \ k = 1, \dots, M, \tag{74}$$

where as before, $s$ is a scale factor, and the notation $[\mathbf{x}]_q$ stands for the $q$th component of the vector $\mathbf{x}$. Using the relations

$$\frac{\mathrm{d}^{k-1} \mathbf{e}_{l=1}^+}{\mathrm{d}z^{k-1}}(z_1) = \sum_{m=1}^{M} a_{m,l=1}^+ (k_{\mathrm{x}} \lambda_{m,l=1}^+)^{k-1} \mathbf{v}_{m,l=1}^+, \ k = 1, \dots, M, \tag{75}$$





and

$$\left[\frac{\mathrm{d}^{k-1}\mathbf{e}_{l=1}^+}{\mathrm{d}z^{k-1}}(z_1)\right]_q = \widehat{\mathbf{i}}_q^T \frac{\mathrm{d}^{k-1}\mathbf{e}_{l=1}^+}{\mathrm{d}z^{k-1}}(z_1) = \sum_{m=1}^{M} a_{m,l=1}^+ (k_\mathrm{x}\lambda_{m,l=1}^+)^{k-1}\widehat{\mathbf{i}}_q^T\mathbf{v}_{m,l=1}^+, \tag{76}$$

where $\widehat{\mathbf{i}}_q$ is a $2M$-dimensional vector with components (compare with Eq. (67))

$$[\widehat{\mathbf{i}}_q]_k = \begin{cases} 1, & k=q \\ 0, & k \neq q \end{cases}, \; k = 1,\ldots,2M, \tag{77}$$

we find

$$\sum_{m=1}^{M}[\mathrm{A}]_{mk}a_{m,l=1}^+ = sb_{1,k}, \; k = 1,\ldots M, \tag{78}$$

where A is a matrix with entries

$$[\mathrm{A}]_{mk} = (k_\mathrm{x}\lambda_{m,l=1}^+)^{k-1}\widehat{\mathbf{i}}_q^T\mathbf{v}_{m,l=1}^+, \; m,k = 1,\ldots,M. \tag{79}$$

Setting $\mathbf{b}_1 = [b_{1,1},\ldots,b_{1,M}]^T$, and omitting the scale factor $s$, we consider the boundary condition for amplitudes

$$\mathbf{a}_1^+ = \mathrm{A}^{-1}\mathbf{b}_1, \tag{80}$$

that is (compare with Eq. (68))

$$[\mathrm{I}_M, \mathrm{O}_M]\mathbf{a}_1 = \mathrm{A}^{-1}\mathbf{b}_1. \tag{81}$$

Starting from the continuity equation (63), we will determine the amplitudes $\mathbf{a}_l$ by using two solution methods, namely, (i) the so-called global matrix method with matrix exponential and (ii) the scattering matrix method.

### 4.1.1 Global matrix method with matrix exponential

The continuity equations (63), and the boundary conditions (68) and (71) are assembled into a system of equations for the stratified atmosphere, i.e.,

$$\mathbb{A}\mathbf{a}_s = s\mathbf{b}, \tag{82}$$





where

$$
\mathbb{A} = \begin{bmatrix}
[O_M, I_M] & 0 & \dots & 0 & 0 \\
\mathbb{A}^1_{L-1,L} & -\mathbb{A}^0_{L-1,L} & \dots & 0 & 0 \\
\vdots & \vdots & \ddots & \vdots & \vdots \\
0 & 0 & \dots & \mathbb{A}^1_{12} & -\mathbb{A}^0_{12} \\
0 & 0 & \dots & 0 & [I_M, O_M]
\end{bmatrix}, \tag{83}
$$

$$
\mathbf{a} = \begin{bmatrix}
\mathbf{a}_L \\
\mathbf{a}_{L-1} \\
\vdots \\
\mathbf{a}_2 \\
\mathbf{a}_1
\end{bmatrix}, \text{ and } \mathbf{b} = \begin{bmatrix}
\mathbf{0}_M \\
\mathbf{0}_{2M} \\
\vdots \\
\mathbf{0}_{2M} \\
\mathbf{i}_1
\end{bmatrix}. \tag{84}
$$

For the lower boundary condition (81), $\mathbf{i}_1$ in Eq. (84) should be replaced by $A^{-1}\mathbf{b}_1$. The matrix $\mathbb{A}$ has $3M-1$ sub- and super-diagonals and it can be compressed into band-storage and inverted using standard methods. From Eq. (83), we see that the scale factor $s$ determines the solution, and can be interpreted as a source term. For this reason, in Eq. (82) we were more precise and showed that the vector of amplitudes depends on the source term $s$. In what follows, when we want to show that a quantity depends on the source term we will use the index $s$; when the index $s$ is omitted, it will be understood that $s = 1$ (according to this convention, $\mathbf{e}^+_{l=1}$ in Eqs. (66) and (74) should also depend on the index $s$). After solving Eq. (82), we compute the solution vector as

$$
\mathbf{e}_l = \mathbf{e}(z_l) = \begin{bmatrix}
\widehat{u}(z_l) \\
\widehat{w}(z_l) \\
\widehat{p}(z_l) \\
\widehat{T}(z_l) \\
\widehat{\mathcal{U}}(z_l) \\
\widehat{\mathcal{T}}(z_l)
\end{bmatrix} = V_l \mathbf{a}_l, \ l = 1, \dots, L, \tag{85}
$$

and the wave amplitudes by means of the relation

$$
\overline{f}_s(z) = sC(z)\widehat{f}(z), \tag{86}
$$

where $f$ stands for $u$, $w$, $p$, and $T$. The ascending and descending solution modes are computed by using Eq. (52), that is,

$$
\mathbf{e}^\pm_l = V^\pm_l \mathbf{a}^\pm_l, l = 1, \dots, L. \tag{87}
$$

According to the basic requirement of a linearization method, we can compute $s$ by imposing that the perturbed quantities are a small fraction of their unperturbed values. For example, we can impose,

$$
\max_z \left| \frac{T'(x,z,t)}{T_0(z)} \right| = \max_z \left| \widehat{T}(z) \right| = \delta_\mathrm{T}, \tag{88}
$$




or

$$\max_z \left| \frac{u'(x,z,t)}{u_0(z)} \right| = \frac{\omega_0}{k_\mathrm{x}} \max_z \left| \frac{\widehat{u}(z)}{u_0(z)} \right| = \delta_\mathrm{u}, \tag{89}$$

where $\delta_\mathrm{T}$ and $\delta_\mathrm{u}$ are prescribed tolerances. Alternatively, we can impose a value for the vertical energy flux at the lowest level $z_1 = z_\mathrm{min}$ as in (Hickey and Cole, 1988). In Section 5 $s$ will be assumed to be some time dependent source function as in (Knight et al., 2022, 2019, 2021).

#### 4.1.2 Scattering matrix method

We consider the continuity equation (63) and partition the matrices $\mathbb{A}_{l,l+1}^{0(1)}$ as

$$\mathbb{A}_{l,l+1}^{0(1)} = \begin{bmatrix} [\mathbb{A}_{l,l+1}^{0(1)}]_{11} & [\mathbb{A}_{l,l+1}^{0(1)}]_{12} \\ [\mathbb{A}_{l,l+1}^{0(1)}]_{21} & [\mathbb{A}_{l,l+1}^{0(1)}]_{22} \end{bmatrix}. \tag{90}$$

Further, we define the scattering matrix at the interface between the layers $l$ and $l+1$ (in fact, at the layer grid point $z_{l+1}$), $\mathrm{S}_{l,l+1}$ through the relation

$$\begin{bmatrix} \mathbf{a}_l^- \\ \mathbf{a}_{l+1}^+ \end{bmatrix} = \mathrm{S}_{l,l+1} \begin{bmatrix} \mathbf{a}_l^+ \\ \mathbf{a}_{l+1}^- \end{bmatrix}, \tag{91}$$

where

$$\mathrm{S}_{l,l+1} = \begin{bmatrix} \mathrm{R}_{l,l+1}^+ & \mathrm{T}_{l,l+1}^- \\ \mathrm{T}_{l,l+1}^+ & \mathrm{R}_{l,l+1}^- \end{bmatrix}, \tag{92}$$

and $\mathrm{R}_{l,l+1}^\pm$ and $\mathrm{T}_{l,l+1}^\pm$ with $\dim(\mathrm{R}_{l,l+1}^\pm) = \dim(\mathrm{T}_{l,l+1}^\pm) = M \times M$, are the reflection and transmission matrices, respectively. Eq. (91) is the so-called interaction principle equation at the interface $(l,l+1)$, and it is apparent that the scattering matrix $\mathrm{S}_{l,l+1}$ relates the amplitudes $\mathbf{a}_l^-$ and $\mathbf{a}_{l+1}^+$ of the waves leaving the interface with the amplitudes $\mathbf{a}_l^+$ and $\mathbf{a}_{l+1}^-$ of the waves entering the interface. Starting with Eq. (63), inserting the partitioning of the matrices $\mathbb{A}_{l,l+1}^{0(1)}$ as given by Eq. (90), using $\mathbf{a}_{l(l+1)} = [\mathbf{a}_{l(l+1)}^+, \mathbf{a}_{l(l+1)}^-]^T$, and rearranging the resulting equation in a fashion as given by Eq. (91), we find

$$\begin{bmatrix} \mathrm{R}_{l,l+1}^+ & \mathrm{T}_{l,l+1}^- \\ \mathrm{T}_{l,l+1}^+ & \mathrm{R}_{l,l+1}^- \end{bmatrix} = \begin{bmatrix} [\mathbb{A}_{l,l+1}^0]_{12} & -[\mathbb{A}_{l,l+1}^1]_{11} \\ [\mathbb{A}_{l,l+1}^0]_{22} & -[\mathbb{A}_{l,l+1}^1]_{21} \end{bmatrix}^{-1} \begin{bmatrix} -[\mathbb{A}_{l,l+1}^0]_{11} & [\mathbb{A}_{l,l+1}^1]_{12} \\ -[\mathbb{A}_{l,l+1}^0]_{21} & [\mathbb{A}_{l,l+1}^1]_{22} \end{bmatrix}. \tag{93}$$

We organize the computational process as an upward recurrence using the concept of a "stack". The stack $\mathcal{S}_{l_0 l}$ with $l_0 < l$, is a group of interfaces characterized by the interaction principle equation

$$\begin{bmatrix} \mathbf{a}_{l_0}^- \\ \mathbf{a}_l^+ \end{bmatrix} = \begin{bmatrix} \mathcal{R}_{l_0 l}^+ & \mathcal{T}_{l_0 l}^- \\ \mathcal{T}_{l_0 l}^+ & \mathcal{R}_{l_0 l}^- \end{bmatrix} \begin{bmatrix} \mathbf{a}_{l_0}^+ \\ \mathbf{a}_l^- \end{bmatrix}, \tag{94}$$

where the matrices $\mathcal{R}_{l_0 l}^\pm$ and $\mathcal{T}_{l_0 l}^\pm$ are obtained through a successive application of the interaction principle equation at the interfaces $(l_0, l_0+1)$, $(l_0+1, l_0+2)$,...,$(l-1, l)$. Adding a new layer $l+1$, and taking into account that at the interface $(l,l+1)$,





the reflection and transmission matrices are $\mathrm{R}_{l,l+1}^{\pm}$ and $\mathrm{T}_{l,l+1}^{\pm}$, respectively, we find that the interaction principle equation for the stack $\mathcal{S}_{l_0,l+1}$, is

$$
\begin{bmatrix} \mathbf{a}_{l_0}^- \\ \mathbf{a}_{l+1}^+ \end{bmatrix} = \begin{bmatrix} \mathcal{R}_{l_0,l+1}^+ & \mathcal{T}_{l_0,l+1}^- \\ \mathcal{T}_{l_0,l+1}^+ & \mathcal{R}_{l_0,l+1}^- \end{bmatrix} \begin{bmatrix} \mathbf{a}_{l_0}^+ \\ \mathbf{a}_{l+1}^- \end{bmatrix},
\tag{95}
$$

where $\mathcal{R}_{l_0,l+1}^{\pm}$ and $\mathcal{T}_{l_0,l+1}^{\pm}$ are computed recursively by using of the "adding formulas"

$$
\mathcal{R}_{l_0,l+1}^+ = \mathcal{R}_{l_0 l}^+ + \mathcal{T}_{l_0 l}^- (\mathrm{I}_M - \mathrm{R}_{l,l+1}^+ \mathcal{R}_{l_0 l}^-)^{-1} \mathrm{R}_{l,l+1}^+ \mathcal{T}_{l_0 l}^+,
\tag{96}
$$

$$
\mathcal{T}_{l_0,l+1}^- = \mathcal{T}_{l_0 l}^- (\mathrm{I}_M - \mathrm{R}_{l,l+1}^+ \mathcal{R}_{l_0 l}^-)^{-1} \mathrm{T}_{l,l+1}^-,
\tag{97}
$$

$$
\mathcal{T}_{l_0,l+1}^+ = \mathrm{T}_{l,l+1}^+ (\mathrm{I}_M - \mathcal{R}_{l_0 l}^- \mathrm{R}_{l,l+1}^+)^{-1} \mathcal{T}_{l_0 l}^+,
\tag{98}
$$

$$
\mathcal{R}_{l_0,l+1}^- = \mathrm{R}_{l,l+1}^- + \mathrm{T}_{l,l+1}^+ (\mathrm{I}_M - \mathcal{R}_{l_0 l}^- \mathrm{R}_{l,l+1}^+)^{-1} \mathcal{R}_{l_0 l}^- \mathrm{T}_{l,l+1}^-,
\tag{99}
$$

for $l = l_0 + 1, ..., L - 1$. The procedure is initialized with $\mathcal{R}_{l_0,l_0+1}^{\pm} = \mathrm{R}_{l_0,l_0+1}^{\pm}$ and $\mathcal{T}_{l_0,l_0+1}^{\pm} = \mathrm{T}_{l_0,l_0+1}^{\pm}$, and is repeated until the last interface is added to the stack. For the stack $\mathcal{S}_{1L}$, the interaction principle equation is

$$
\begin{bmatrix} \mathbf{a}_1^- \\ \mathbf{a}_L^+ \end{bmatrix} = \begin{bmatrix} \mathcal{R}_{1L}^+ & \mathcal{T}_{1L}^- \\ \mathcal{T}_{1L}^+ & \mathcal{R}_{1L}^- \end{bmatrix} \begin{bmatrix} \mathbf{a}_1^+ \\ \mathbf{a}_L^- \end{bmatrix},
\tag{100}
$$

and from the boundary conditions for amplitudes (67) and (70), that is, from the relations $\mathbf{a}_1^+ = \mathbf{i}_1$ and $\mathbf{a}_L^- = \mathbf{0}_M$, respectively, we find

$$
\mathbf{a}_1^- = \mathcal{R}_{1L}^+ \mathbf{a}_1^+ \text{ and } \mathbf{a}_L^+ = \mathcal{T}_{1L}^+ \mathbf{a}_1^+.
\tag{101}
$$

To restore the entire set of amplitude vectors $\mathbf{a}_l$, we consider the interaction principle equations for the stacks $\mathcal{S}_{1l}$ and $\mathcal{S}_{lL}$, yielding

$$
\mathbf{a}_l^+ = (\mathrm{I}_M - \mathcal{R}_{1l}^- \mathcal{R}_{lL}^+)^{-1} \mathcal{T}_{1l}^+ \mathbf{a}_1^+,
\tag{102}
$$

$$
\mathbf{a}_l^- = \mathcal{R}_{lL}^+ \mathbf{a}_l^+,
\tag{103}
$$

for $l = L - 1, ..., 1$. The solution vector and the wave amplitudes are then computed by using Eqs. (85) and (86), respectively. In contrast to the previous method, this approach requires a clear differentiation between ascending and descending modes as defined by Eq. (45).

## 4.2 Discrete values of the solution vector

In this section, the global matrix method with matrix exponential and the scattering matrix method will be formulated for the discrete values of the solution vector.




### 4.2.1 Global matrix method with matrix exponential

In the layer $l$, with boundaries $z_l$ and $z_{l+1}$, the discrete values $\mathbf{e}_{l+1} = \mathbf{e}(z_{l+1})$ and $\mathbf{e}_l = \mathbf{e}(z_l)$ are related through the relation
(cf. Eq. (39))

$$\mathbf{e}_{l+1} = V_l \text{diag}[e^{k_x \lambda_{nl} \Delta_l}] V_l^{-1} \mathbf{e}_l, \tag{104}$$

or equivalently,

$$V_l^{-1} \mathbf{e}_{l+1} = \text{diag}[e^{k_x \lambda_{nl} \Delta_l}] V_l^{-1} \mathbf{e}_l. \tag{105}$$

Taking into account that by Eq. (39), we have $\mathbf{e}_l = V_l \mathbf{a}_l$ and $\mathbf{e}_{l+1} = V_{l+1} \mathbf{a}_{l+1}$, we see that Eq. (60) and (105) are completely
equivalent. Multiplying Eq. (105) with the scaling matrix $S_l^1$, we obtain the layer equation

$$\mathbb{A}_l^1 \mathbf{e}_{l+1} - \mathbb{A}_l^0 \mathbf{e}_l = \mathbf{0}_{2M}, \ l = 1, ..., L-1, \tag{106}$$

where

$$\mathbb{A}_l^1 = S_l^1 V_l^{-1}, \tag{107}$$
$$\mathbb{A}_l^0 = S_l^0 V_l^{-1}, \tag{108}$$

and $S_l^1$ and $S_l^0$, are given by Eqs. (61) and (62), respectively. Essentially, we have $L-1$ equations imposed on layers $1, \ldots, L-1$
for the $L$ unknowns $\mathbf{e}_1, \ldots, \mathbf{e}_L$. On the layer $l = 1$, the boundary condition (cf. Eq. (67)) $\mathbf{a}_1^+ = \mathbf{i}_1$, translates into (cf. Eq. (49))

$$[I_M, O_M] V_1^{-1} \mathbf{e}_1 = \mathbf{i}_1, \tag{109}$$

while on the layer $l = L$, the boundary condition (cf. Eq. (70)) $\mathbf{a}_L^- = \mathbf{0}_M$ translates into (cf. Eq. (50))

$$[O_M, I_M] V_L^{-1} \mathbf{e}_L = \mathbf{0}_M. \tag{110}$$

As before, the layer equations (106) together with the boundary conditions (109) and (110) are assembled into a system of
equations for the stratified atmosphere, i.e.,

$$\mathbb{A} \mathbf{e}_s = s \mathbf{b}, \tag{111}$$





where

$$\mathbb{A} = \begin{bmatrix} [O_M, I_M]V_L^{-1} & 0 & \dots & 0 & 0 \\ \mathbb{A}_{L-1}^1 & -\mathbb{A}_{L-1}^0 & \dots & 0 & 0 \\ \vdots & \vdots & \ddots & \vdots & \vdots \\ 0 & 0 & \dots & \mathbb{A}_1^1 & -\mathbb{A}_1^0 \\ 0 & 0 & \dots & 0 & [I_M, O_M]V_1^{-1} \end{bmatrix}, \tag{112}$$

$$\mathbf{e} = \begin{bmatrix} \mathbf{e}_L \\ \mathbf{e}_{L-1} \\ \vdots \\ \mathbf{e}_2 \\ \mathbf{e}_1 \end{bmatrix} \text{ and } \mathbf{b} = \begin{bmatrix} \mathbf{0}_M \\ \mathbf{0}_{2M} \\ \vdots \\ \mathbf{0}_{2M} \\ \mathbf{i}_1 \end{bmatrix}. \tag{113}$$

After solving Eq. (111) for $s = 1$, we compute the wave amplitudes by using Eq. (86).

Comments.

1. The ascending and descending solution modes can be derived by using the upward and downward recurrence relations
(cf. Eqs. (53)–(56), (66) and (69))

$$\mathbf{e}_{l+1}^+ = T_l^+ \mathbf{e}_l, \text{ for } l = 1, \dots, L-1, \text{ with } \mathbf{e}_1^+ = \mathbf{v}_1^+, \text{ and} \tag{114}$$

$$\mathbf{e}_l^- = T_l^- \mathbf{e}_{l+1}, \text{ for } l = L-1, \dots, 1, \text{ with } \mathbf{e}_L^- = \mathbf{0}_{2M}, \tag{115}$$

respectively, where

$$T_l^+ = V_l \begin{bmatrix} \text{diag}[e^{k_x \lambda_{ml}^+ \Delta_l}] & O_M \\ O_M & O_M \end{bmatrix} V_l^{-1}, \tag{116}$$

$$T_l^- = V_l \begin{bmatrix} O_M & O_M \\ O_M & \text{diag}[e^{-k_x \lambda_{ml}^- \Delta_l}] \end{bmatrix} V_l^{-1}. \tag{117}$$

Obviously, the relation $\mathbf{e}_l = \mathbf{e}_l^+ + \mathbf{e}_l^-$, $l = 1, \dots, L$, can be used to verify the numerical algorithm.

2. If we assume that the ascending modes are the dominant modes, i.e., $\mathbf{e}_l \approx \mathbf{e}_l^+$ for $l = 1, \dots, L$, we may compute the solution vector by means of the upward recurrence relation (cf. Eq. (114))

$$\mathbf{e}_{l+1} = T_l^+ \mathbf{e}_l, \text{ for } l = 1, \dots, L-1, \text{ with } \mathbf{e}_1 = \mathbf{v}_1^+. \tag{118}$$

**4.2.2 Global matrix method with the Padé approximation to the matrix exponential**

The layer equation (106) was derived from the solution representation (104). In fact, this equation is simply the matrix-exponential representation of the solution, i.e.,

$$\mathbf{e}_{l+1} = e^{k_x \Delta_l A_l} \mathbf{e}_l, \tag{119}$$





where the matrix exponential is calculated using an eigendecomposition of the propagation matrix $\mathrm{A}_l$, i.e., $\mathrm{A}_l = \mathrm{V}_l \mathrm{diag}[\lambda_{nl}]\mathrm{V}_l^{-1}$.

However, instead of an eigendecomposition method, we can use the Padé approximation to compute the matrix exponential (Doicu and Trautmann, 2009a, b). This method is presumably more efficient than the eigendecomposition method.

The $n$th diagonal Padé approximation to the matrix exponential is

$$\mathrm{e}^{x\mathrm{A}} = [\mathrm{D}(x\mathrm{A})]^{-1}\mathrm{N}(x\mathrm{A}), \tag{120}$$

where $\mathrm{D}(x\mathrm{A})$ and $\mathrm{N}(x\mathrm{A})$ are polynomials in $\mathrm{A}x$ of degree $n$ given respectively, by

$$\mathrm{D}(x\mathrm{A}) = \sum_{k=0}^{n}(-1)^k c_k x^k \mathrm{A}^k, \tag{121}$$

$$\mathrm{N}(x\mathrm{A}) = \mathrm{D}(-x\mathrm{A}) = \sum_{k=0}^{n} c_k x^k \mathrm{A}^k. \tag{122}$$

The coefficients $c_k$ are defined by

$$c_k = \frac{(2n-k)!n!}{(2n)!k!(n-k)!}, \tag{123}$$

and can be computed recursively by means of the relation

$$c_k = \frac{n-k+1}{k(2n-k+1)}c_{k-1}, \ k \geq 1 \tag{124}$$

with the initial value $c_0 = 1$. The layer equation (119) then becomes

$$\mathrm{D}(k_{\mathrm{x}}\Delta_l \mathrm{A}_l)\mathbf{e}_{l+1} = \mathrm{N}(k_{\mathrm{x}}\Delta_l \mathrm{A}_l)\mathbf{e}_l, \tag{125}$$

that is

$$\mathbb{A}_l^1 \mathbf{e}_{l+1} - \mathbb{A}_l^0 \mathbf{e}_l = \mathbf{0}_{2M}, \ l = 1, ..., L-1, \tag{126}$$

where

$$\mathbb{A}_l^1 = \mathrm{D}k_{\mathrm{x}}\Delta_l(\mathrm{A}_l), \tag{127}$$

$$\mathbb{A}_l^0 = \mathrm{N}(k_{\mathrm{x}}\Delta_l \mathrm{A}_l). \tag{128}$$

Now, the layer equations (126) together with the boundary conditions (109) and (110) are assembled into a system of equations for the stratified atmosphere, which is then solved by standard methods for band matrices. Taking into account that the boundary conditions (109) and (110) are expressed in terms of the eigenvector matrix, we see that in this approach, the eigendecomposition method must only be used in the lower and upper layers, i.e., for $l = 1$ and $l = L$, while in the rest of the layers, the Padé approximation is used.

Comments.





1. The first-order Padé approximation is equivalent with the finite-difference scheme

$$\frac{\mathbf{e}_{l+1} - \mathbf{e}_l}{\Delta_l} = k_{\mathrm{x}} \mathrm{A}_l \frac{\mathbf{e}_{l+1} + \mathbf{e}_l}{2}. \tag{129}$$

2. Considering the Taylor series approximation to the matrix exponential,

$$\mathrm{e}^{x\mathrm{A}} = \mathrm{I} + \sum_{k=1}^{n} \frac{1}{k!} x^k \mathrm{A}^k, \tag{130}$$

we deduce that for $k = 1$, we have

$$\mathrm{e}^{x\mathrm{A}} = \mathrm{I} + x\mathrm{A}, \tag{131}$$

which, when used in Eq. (119), is equivalent with the forward finite-difference scheme

$$\frac{\mathbf{e}_{l+1} - \mathbf{e}_l}{\Delta_l} = k_{\mathrm{x}} \mathrm{A}_l \mathbf{e}_l. \tag{132}$$

### 4.2.3 Scattering matrix method

The scattering matrix method can also be formulated in terms of the discrete values of the solution vector. Starting from the interaction principle equation (74), using Eq. (52), i.e., $\mathbf{e}_l^{\pm} = \mathrm{V}_l^{\pm} \mathbf{a}_l^{\pm}$, and Eqs. (49)–(50), i.e., (here we use the more precise notation $(\mathrm{V}_l)^{-1}$ instead of $\mathrm{V}_l^{-1}$)

$$\mathbf{a}_l^+ = [\mathrm{I}_M, \mathrm{O}_M]\mathbf{a}_l = [\mathrm{I}_M, \mathrm{O}_M](\mathrm{V}_l)^{-1}\mathbf{e}_l, \tag{133}$$

$$\mathbf{a}_l^- = [\mathrm{O}_M, \mathrm{I}_M]\mathbf{a}_l = [\mathrm{O}_M, \mathrm{I}_M](\mathrm{V}_l)^{-1}\mathbf{e}_l, \tag{134}$$

we find that for the stack $\mathcal{S}_{l_0 l}$, the interaction principle equation involving the discrete values of the solution vector is

$$\begin{bmatrix} \mathbf{e}_{l_0}^- \\ \mathbf{e}_{l+1}^+ \end{bmatrix} = \begin{bmatrix} \mathcal{R}_{l_0 l}^+ & \mathcal{T}_{l_0 l}^- \\ \mathcal{T}_{l_0 l}^+ & \mathcal{R}_{l_0 l}^- \end{bmatrix} \begin{bmatrix} \mathbf{e}_{l_0}^+ \\ \mathbf{e}_{l+1}^- \end{bmatrix}, \tag{135}$$

where $\dim(\mathcal{R}_{l_0 l}^{\pm}) = \dim(\mathcal{T}_{l_0 l}^{\pm}) = 2M \times 2M$, and

$$\begin{bmatrix} \mathcal{R}_{l_0 l}^+ & \mathcal{T}_{l_0 l}^- \\ \mathcal{T}_{l_0 l}^+ & \mathcal{R}_{l_0 l}^- \end{bmatrix} = (\mathrm{I}_{4M} - \mathrm{A})^{-1}\mathrm{A}, \tag{136}$$

with

$$\mathrm{A} = \begin{bmatrix} \mathrm{V}_{l_0}^- \mathcal{R}_{l_0 l}^+ [(\mathrm{V}_{l_0})^{-1}]_1 & \mathrm{V}_{l_0}^- \mathcal{T}_{l_0 l}^- [(\mathrm{V}_l)^{-1}]_2 \\ \mathrm{V}_l^+ \mathcal{T}_{l_0 l}^+ [(\mathrm{V}_{l_0})^{-1}]_1 & \mathrm{V}_l^+ \mathcal{R}_{l_0 l}^- [(\mathrm{V}_l)^{-1}]_2 \end{bmatrix}, \tag{137}$$

$$\mathrm{V}_l = [\mathrm{V}_l^+, \mathrm{V}_l^-], \text{ and } (\mathrm{V}_l)^{-1} = \begin{bmatrix} [(\mathrm{V}_l)^{-1}]_1 \\ [(\mathrm{V}_l)^{-1}]_2 \end{bmatrix}. \tag{138}$$





The boundary values of the solution vector $\mathbf{e}_1^-$ and $\mathbf{e}_L^+$ are computed from Eqs. (101) with $\mathbf{e}_1^+ = \mathbf{v}_1^+$ and $\mathbf{e}_L^- = \mathbf{0}_{2M}$, while the rest of the discrete values $\mathbf{e}_l^+$ and $\mathbf{e}_l^-$ are obtained from Eqs. (102) and (103) with $\mathbf{e}$ replacing $\mathbf{a}$, respectively. Because the dimensions of the matrices $\mathcal{R}_{l_0 l}^\pm$ and $\mathcal{T}_{l_0 l}^\pm$ are twice as large as those of the matrices $R_{l_0 l}^\pm$ and $T_{l_0 l}^\pm$, the computation time will be higher. The computation time can be somewhat reduced, if the Neumann series representation $(\mathrm{I}_{4M} - \mathrm{A})^{-1}\mathrm{A} = \sum_{n=1}^{\infty} \mathrm{A}^n$ is used (this expansion is valid for $\|\mathrm{A}\| < 1$, where $\|\cdot\|$ is some matrix norm).

## 535  5  Time dependent source function

If the source term is time-dependent (or more precisely, if it lacks a temporal dependence in the form of $\exp(\mathrm{j}\omega t)$), the perturbed quantity $f'(x,z,t)$ is not a wave with a specified angular frequency $\omega$ (a single-frequency wave). In this case, the equations should be treated in the frequency domain by considering the Fourier transform in time (Knight et al., 2022, 2019, 2021). This is defined by

$$\widetilde{f'}(x,z,\omega) = \int_{-\infty}^{\infty} f'(x,z,t)\mathrm{e}^{-\mathrm{j}\omega t}\mathrm{d}t = \mathcal{F}[f'](x,z,\omega) \tag{139}$$

and its inverse by

$$f'(x,z,t) = \frac{1}{2\pi} \int_{-\infty}^{\infty} \widetilde{f'}(x,z,\omega)\mathrm{e}^{\mathrm{j}\omega t}\mathrm{d}\omega = \mathcal{F}^{-1}[\widetilde{f'}](x,z,t). \tag{140}$$

Applying the Fourier transform to the linearized equations (A12)–(A16) of Appendix A, using the result

$$\mathcal{F}\left[\frac{\partial f'}{\partial t}\right](x,z,\omega) = \mathrm{j}\omega \widetilde{f'}(x,z,\omega), \tag{141}$$

and setting

$$\widetilde{f'}(x,z,\omega) = \overline{\widetilde{f}}(z,\omega)\mathrm{e}^{-\mathrm{j}k_x x} \tag{142}$$

as the counterpart of Eq. (24) (in which the exponential term $\exp(\mathrm{j}\omega t)$ is absorbed into the expression of $\overline{f}(z)$), and

$$\overline{\widetilde{f}}(z,\omega) = C(z)\widehat{\widetilde{f}}(z,\omega) \tag{143}$$

as the counterpart of Eq. (30), we are led to the system of differential equations (A28)–(A33) of Appendix A (or equivalently, to
the matrix differential equation (28)), but with $\widehat{\widetilde{f}}(z,\omega)$ replacing $\widehat{f}(z)$. In the frequency domain, the lower boundary conditions (66) and (74) for a unit source function become





$$\mathbf{e}_{l=1}^{+}(z_1,\omega) = \begin{bmatrix} \widehat{u}_{l=1}^{+}(z_1,\omega) \\ \widehat{w}_{l=1}^{+}(z_1,\omega) \\ \widehat{p}_{l=1}^{+}(z_1,\omega) \\ \widehat{T}_{l=1}^{+}(z_1,\omega) \\ \widehat{\mathcal{U}}_{l=1}^{+}(z_1,\omega) \\ \widehat{\mathcal{T}}_{l=1}^{+}(z_1,\omega) \end{bmatrix} = \sum_{m=1}^{M} a_{m,l=1}^{+}(\omega)\mathbf{v}_{m,l=1}^{+}(\omega) = \mathbf{v}_{1,l=1}^{+}(\omega), \tag{144}$$

for $a_{m,l=1}^{+}(\omega) = \delta_{m1}$, where $\delta_{m1}$ is the Kronecker delta, and

$$\left[\frac{\mathrm{d}^{k-1}\mathbf{e}_{l=1}^{+}}{\mathrm{d}z^{k-1}}(z_1,\omega)\right]_q = b_{1,k}, \ k=1,...,M, \tag{145}$$

respectively. As in Section 4.1.1, with $\widetilde{\widehat{f}}(z,\omega)$ being the solution of the differential equation (28) for a unit source function (in the frequency domain), we compute the perturbed quantity $f_s'(x,z,t)$ by taking the inverse transform (140) with

$$\widetilde{f_s'}(x,z,\omega) = C(z)\widetilde{s}(\omega)\widetilde{\widehat{f}}(z,\omega)\mathrm{e}^{-\mathrm{j}k_{\mathrm{x}}x}, \tag{146}$$

that is,

$$f_s'(x,z,t) = \frac{1}{2\pi}\int\limits_{-\infty}^{\infty}\widetilde{f_s'}(x,z,\omega)\mathrm{e}^{\mathrm{j}\omega t}\mathrm{d}\omega = \frac{C(z)}{2\pi}\mathrm{e}^{-\mathrm{j}k_{\mathrm{x}}x}\int\limits_{-\infty}^{\infty}\widetilde{s}(\omega)\widetilde{\widehat{f}}(z,\omega)\mathrm{e}^{\mathrm{j}\omega t}\mathrm{d}\omega, \tag{147}$$

where $\widetilde{s}(\omega)$ is the Fourier transform of some source function $s(t)$, i.e., $\widetilde{s} = \mathcal{F}[s]$. The computation of $\widetilde{\widehat{f}}(z,\omega)$ can be performed without any limitations using any of the methods presented in Section 4. Note that the most general situation is when the source function depends on time and the horizontal coordinate, that is, $s = s(x,t)$. In this case, we have to consider the two-dimensional Fourier transform in time and space. However, a simplification occurs in case that the variables are separable, i.e., $s = s_t(t)s_x(x)$; in this case, the term $\widetilde{s}(\omega)\exp(-\mathrm{j}k_{\mathrm{x}}x)$ in Eq. (146) should be replaced by the product of one-dimensional

Fourier transforms $\widetilde{s}_t(\omega)\widetilde{s}_x(k_{\mathrm{x}})$.

We conclude this section with a comment related to causality as discussed in (Knight et al., 2019). Causality means that a wave field in response to any source function can never be nonzero prior to the earliest time at which the source function is nonzero. According to the classification rule (45), we have

$$\mathrm{Re}[\lambda_{1l}^{+}(\omega)] < \mathrm{Re}[\lambda_{1l}^{-}(\omega)], \tag{148}$$

for any $l = 1,\ldots,L$ and any real frequency $\omega$. However, as shown by Knight et al. (2019), to preserve causality in solutions of two-point boundary value problems, a much stronger condition should be met. This condition is

$$\max_{l=1,...,L}\mathrm{Re}[\lambda_{1l}^{+}(\varpi)] < \min_{l=1,...,L}\mathrm{Re}[\lambda_{1l}^{-}(\varpi)]. \tag{149}$$





for all $\varpi = \mathrm{Re}(\varpi) + \mathrm{j}\,\mathrm{Im}(\varpi) \in \overline{U}_\delta$, where for any $\delta \geq 0$, $\overline{U}_\delta$ is the closed lower half-plane $\mathrm{Im}(\varpi) \leq -\delta$. Eq. (149) states that the upper altitude bound of the real part of the eigenvalues for ascending modes should be smaller than the lower altitude bound of the same values for descending modes. In some situation, condition (149) is not satisfied for $\delta = 0$, but it is satisfied for some $\delta > 0$. If so, an imaginary frequency shifting, i.e., $\omega \to \omega - \mathrm{j}\delta$ is required to preserve causality (Knight et al., 2019). To summarize this approach, we consider the shifted source function in the frequency domain

$$\widetilde{s}_\delta(\omega) = \int_{-\infty}^{\infty} s(t)\mathrm{e}^{-\mathrm{j}(\omega - \mathrm{j}\delta)t}\mathrm{d}t = \widetilde{s}(\omega - \mathrm{j}\delta), \tag{150}$$

and let

$$\widetilde{f'_{s\delta}}(x,z,\omega) = C(z)\widetilde{s}(\omega - \mathrm{j}\delta)\widehat{\widetilde{f}}(z,\omega - \mathrm{j}\delta)\mathrm{e}^{-\mathrm{j}k_{\mathrm{x}}x} \tag{151}$$

be the Fourier transform of the perturbed quantity with frequency shifting $f'_{s\delta}(x,z,t)$, i.e., $\widetilde{f'_{s\delta}} = \mathcal{F}[f'_{s\delta}]$, where as usual, $\widehat{\widetilde{f}}(z,\omega - \mathrm{j}\delta)$ is solution of the differential equation (28) for a unit source function. By Cauchy theorem it can be shown that the perturbed quantity with frequency shifting computes as

$$f'_{s\delta}(x,z,t) = \frac{1}{2\pi}\int_{-\infty}^{\infty} \widetilde{f'_{s\delta}}(x,z,\omega)\mathrm{e}^{\mathrm{j}\omega t}\mathrm{d}\omega = \mathrm{e}^{-\delta t}f'_s(x,z,t), \tag{152}$$

where $f'_s$ is the perturbed quantity without frequency shifting given by Eq. (147). As a result, we obtain the so called shift-invariance property of the solution, i.e.,

$$f'_s(x,z,t) = \mathrm{e}^{\delta t}f'_{s\delta}(x,z,t). \tag{153}$$

Summarizing, in the frequency shifting approach we (i) compute $\widehat{\widetilde{f}}(z,\omega - \mathrm{j}\delta)$ as the solution of the differential equation (28) for a unit source function, (ii) calculate $\widetilde{f'_{s\delta}}$ by means of Eq. (151), (iii) determine $f'_{s\delta}$ as $f'_{s\delta} = \mathcal{F}[\widetilde{f'_{s\delta}}]$, and (iv) compute $f'_s$ from the shift-invariance property (153). An issue that may arise with this approach is that the interaction between a large frequency $\delta$ (which ensures the causality condition (149)) and the rounding errors in computing a small $f'_{s\delta}$, through the exponential term $\exp(\delta t)$, can cause the left-hand side of Eq. (153) to explode for large $t$. Since this problem appears frequently in our simulations, we decided to renounce on the causality condition (149) (in other words, on the frequency shifting), and to compute $f'_s$ directly by using Eq. (147). Parenthetically we note that instead of condition (149) we can give another condition that preserves causality. From Eqs. (151)–(153), we find

$$f'_s(x,z,t) = \mathrm{e}^{\delta t}f'_{s\delta}(x,z,t) = C(z)\mathrm{e}^{-\mathrm{j}k_{\mathrm{x}}x}\int_{-\infty}^{\infty} s(t_1)\widehat{f}_\delta(z,t-t_1)\mathrm{d}t_1, \tag{154}$$

where

$$\widehat{f}_\delta(z,t-t_1) = \frac{1}{2\pi}\int_{-\infty}^{\infty} \widehat{\widetilde{f}}(z,\omega - \mathrm{j}\delta)\mathrm{e}^{\mathrm{j}(\omega - \mathrm{j}\delta)(t-t_1)}\mathrm{d}\omega. \tag{155}$$





Assume that the source function $s(t)$ is applied at $t = 0$, and in view of Eq. (154), consider the convolution integral

$$I(z,t) = \int\limits_0^\infty s(t_1)\widehat{f}_\delta(z, t - t_1)\mathrm{d}t_1. \tag{156}$$

Further, assume that

at any altitude level $z$, $\widehat{f}_\delta(z, \tau) = 0$ for all $\tau \leq T$ with some $T \geq 0$. $\tag{157}$

In this case, for $t \leq T$ and $t_1 \geq 0$, we have $\tau = t - t_1 \leq T$, implying $\widehat{f}_\delta(z, \tau = t - t_1) = 0$. Consequently, for $t \leq T$, we have $I(z,t) = 0$, and we conclude that the effect (reaction) $I(z,t)$ appears at $t > T$, while the cause (action) $s(t)$ appears at $t = 0$. Thus, in the new formulation, condition (157) is the analogue of condition (149). The problem arising here is that in practice, condition (157) should be verified for all $z$ within the considered altitude range. However, if we decide not to be so strict in verifying causality, we can determine the pair $(z_0, t_0) = \arg\max_{z,t} \widehat{f}_\delta(z,t)$ and then compare $s(t)$ with $\widehat{f}_\delta(z_0, t)$.

## 6 Numerical simulations

We designed a numerical model for solving the ionospheric linearized gravity-wave equations. The solution methods included in the model are the following.

**Method 1:** Global matrix method with matrix exponential for the amplitudes of the characteristic solutions;

**Method 2:** Scattering matrix method for the amplitudes of the characteristic solutions;

**Method 3:** Global matrix method with matrix exponential for the discrete values of the solution vector;

**Method 4:** Global matrix method with the Padé approximation to the matrix exponential for the discrete values of the solution vector;

**Method 5:** Scattering matrix method for the discrete values of the solution vector.

The main difference between global and scattering matrix methods is that the former with the scaling matrices (61) and (62), require defining ascending and descending modes only at the upper and lower boundaries, while the latter require an explicit determination of the mode type at every altitude.

The input parameters of the numerical model are the background quantities $\rho_0$, $T_0$, $u_0$ and $n_{0i}$. These are delivered by the SAMI2 model of the Naval Research Lab (Huba et al., 2000), while the derivatives $\mathrm{d}T_0/\mathrm{d}z$, $\mathrm{d}u_0/\mathrm{d}z$, and $\mathrm{d}n_{0i}/\mathrm{d}z$ are computed by finite-differences. Note that in SAMI2, the neutral atmosphere is specified using the Mass Spectrometer Incoherent Scatter model (MSIS) (Hedin, 1987), and the Horizontal Wind Model (HWM) (Hedin et al., 1991). The background quantities $T_0$, $u_0$ and $n_{0i}$, as well as their derivatives are illustrated in Fig. 1.

In our numerical analysis, we consider two types of source functions, namely

$$s(t) = s_0 \mathrm{e}^{\mathrm{j}\omega_0 t}, \quad \widetilde{s}(\omega) = 2\pi s_0 \delta(\omega - \omega_0), \tag{158}$$





**Figure 1.** Background quantities $T_0$, $u_0$ and $n_{0i}$ ($i = O^+$) delivered by SAMI2 (upper panels), and their height derivatives (lower panels).




and

$$s(t) = s_0 e^{-(t-t_0)^2/(2\sigma_t^2)} e^{j\omega_0(t-t_0)}, \quad \widetilde{s}(\omega) = \frac{\sqrt{2\pi}}{\sigma_\omega} s_0 e^{-(\omega-\omega_0)^2/(2\sigma_\omega^2)} e^{-j\omega t_0},$$ (159)

where $\omega_0$ is the reference frequency (the central frequency in the Fourier spectrum), $t_0$ is the time at which the source function
is maximum, $\sigma_t$ and $\sigma_\omega = 1/\sigma_t$ are the standard deviations in the time and frequency domains, respectively, and $s_0$ is the
amplitude of the source function (computed from Eq. (88) with $\delta_T = 0.1$). The first type of source function leads to a single-
frequency solution with frequency $\omega_0$, that is,

$$f'(x,z,t) = C(z)\widetilde{\widehat{f}}(z,\omega_0)e^{j(\omega_0 t - k_x x)},$$ (160)

while the second type corresponds to a time wavepacket (with a Gaussian pulse function as envelope). The reference frequency
used in the definitions (31) and (32) of $\widehat{u}$ and $\widehat{w}$, respectively, is $\omega_0$. The quantities of interest (which will be calculated and
analyzed) are the perturbations

$$\overline{f}_s(z) = sC(z)\widetilde{\widehat{f}}(z,\omega_0)$$ (161)

in the first case, and

$$\overline{f}_s(z,t) = \frac{C(z)}{2\pi} \int_{-\infty}^{\infty} \widetilde{s}(\omega)\widetilde{\widehat{f}}(z,\omega)e^{j\omega t}d\omega$$ (162)

in the second case.

The numerical analysis is performed under the following assumptions.

1. The atmosphere extends from $z_{\min} = 50\,\mathrm{km}$ to $z_{\max} = 500\,\mathrm{km}$, the number of layers is $L = 400$, the number of grid
points is $2L+1 = 801$, the altitude discretization step is $\Delta\widehat{z} = 0.56\,\mathrm{km}$, and the layer thickness is $2\Delta\widehat{z} = 1.12\,\mathrm{km}$.

2. Unless stated otherwise, we assume non-zero height derivatives for background temperature and wind velocity, as well
as a constant background dynamic viscosity in each layer (in other words, we assume the layer conditions (27)).

3. We choose $\omega_0 = \kappa_\omega \max_z \nu_{0ni}(z)$ with $0.6 \leq \kappa_\omega \leq 1$, where $\nu_{0ni}$ is the background neutral-ion collision frequency given
by Eq. (19). For $\kappa_\omega$ ranging from 0.6 to 1.0, $\omega_0$ varies between $1.25\times10^{-3}\,\mathrm{s}^{-1}$ and $2.10\times10^{-3}\,\mathrm{s}^{-1}$, and correspondingly,
the time period $T_0 = 2\pi/\omega_0$ varies between 50 min and 83 min.

4. For the first type of source function, we analyze $\overline{f}_s(z)$ in the spatial frequency domain by applying a nonuniform Fourier
Transform (FT) with $N_k = 256$ points. The sampling wavenumber is $\Delta k_z = k_{z0}/20$, where $k_{z0} = 2\pi/\lambda_{z0}$ and $\lambda_{z0} = 50$
km. The Fourier spectrum is smoothed using cubic smoothing spline, while for a better comparison, the FT-amplitudes
are normalized by their maximum values.

5. For the second type of source function, we choose $\sigma_\omega = \omega_0/30$, and $t_0 = \kappa_t\sigma_t$ with $\kappa_t = 4$ and $\sigma_t = 1/\sigma_\omega$, so that with
a good approximation, $s(t) \approx 0$ for $t \leq 0$ and $t \geq 2\kappa_t\sigma_t$. Thus, the time domain is $t_{\min} = 0$ and $t_{\max} = 2\kappa_t\sigma_t$, while





the frequency domain is $\omega_{\min} = \omega_0 - \kappa_t \sigma_\omega$ and $\omega_{\max} = \omega_0 + \kappa_t \sigma_\omega$. We perform a nonuniform Fourier transform with $N_t = N_\omega = 512$ points; accordingly, the sampling time is $\Delta t = (t_{\max} - t_{\min})/(N_t - 1)$ and the sampling frequency is $\Delta \omega = (\omega_{\max} - \omega_{\min})/(N_\omega - 1)$.

6. If not stated otherwise, we use Method 1 as solution method and set the frequency parameter $\kappa_\omega$ to 0.8.

## 6.1    Single-frequency waves

Our numerical analysis is organized as follows. In a first step, we test the numerical model, and in a second step, we evaluate the accuracy and efficiency of the proposed solution methods. Further, we discuss the significance of the vertical wavenumber, and analyze the pairwise classification of ascending and descending modes, as well as, the influence of the ion drag, lower boundary condition, and lower altitude level on the results. Finally, we calculate the ascending and descending wave modes by using Methods 1 and 3.

*Model testing.* To test the numerical model we compare the results obtained by solving (see Appendix A)

1. the linearized equations (A28)–(A33) for an isothermal, homogeneous, and windless atmosphere, i.e., for the layer conditions

$$\frac{\mathrm{d}T_0}{\mathrm{d}z} = 0, \ u_0 = 0, \ \mu_{\mathrm{k}} = \text{constant, and } \lambda_{\mathrm{k}} = \text{constant,} \tag{163}$$

and

2. the linearized equations (A57)–(A62).

In this simulation, the altitude range is from $z_{\min} = 125\,\mathrm{km}$ to $z_{\max} = 450\,\mathrm{km}$. The results in Fig. 2 show that there are no visible differences in the altitude profiles of temperature $\overline{T}_s$, vertical velocity $\overline{w}_s$, and horizontal velocity $\overline{u}_s$.

*Accuracy and efficiency of the solution methods.* To test the accuracy of the solution methods, we choose Method 1 as a reference, and calculate the relative root-mean square error of a method as

$$\varepsilon_f^{\text{Method}} = \sqrt{\frac{\sum_{z_i} [\overline{f}_s^{\text{Method}}(z_l) - \overline{f}_s^{\text{Method 1}}(z_l)]^2}{\sum_{z_i} [\overline{f}_s^{\text{Method 1}}(z_l)]^2}}.$$

In Fig. 3 we plot the relative root-mean square error in the perturbed temperature, vertical velocity, and horizontal velocity versus the frequency parameters $\kappa_\omega$. The results were obtained using Method 4 with the second- and third-order Padé approximations, and Method 5 with a third-order Neumann series approximation for matrix inversion. We present only these errors, because the errors for Methods 2 and 3 are smaller than $10^{-6}$. The elapsed time (wall time) of all solution methods are shown

in Fig. 4. The following conclusions can be drawn.

1. The relative root-mean square errors of the third-order Padé approximation are generally smaller than $5 \times 10^{-3}$. In contrast, the errors of the second-order Padé approximation can reach values of $5 \times 10^{-2}$.



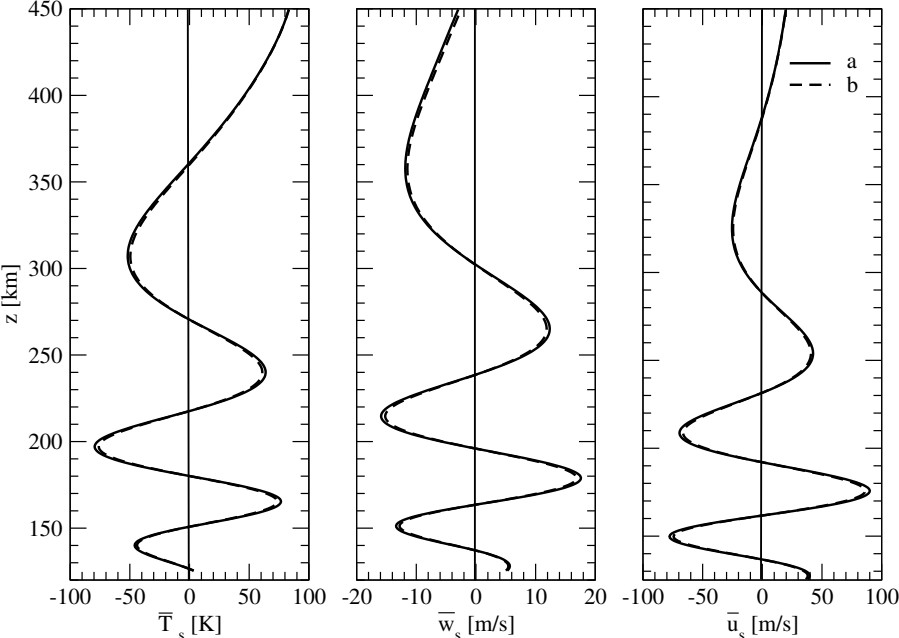

**Figure 2.** Altitude profiles of the perturbed temperature $\overline{T}_s$, vertical velocity $\overline{w}_s$, and horizontal velocity $\overline{u}_s$ computed by solving the linearized equations (A28)–(A33) (a) and (A57)–(A62) (b) of Appendix A.

2. Excluding an outlier, the relative root-mean square errors of Method 5 are of about $3 \times 10^{-2}$. These large errors may be due to the fact that the rounding errors may affect the calculation of the reflection and transmission matrices of a stack

(these matrices have larger dimensions and require more matrix inversion operations than in Method 2).

3. The scattering matrix methods (Methods 2 and 5) are more time consuming than the global matrix methods (Methods 1, 3, and 4). This is because scattering matrix methods necessitate many matrix operations per layer, while solving a system of equations compressed into band-storage is not so time expensive.

The conclusion of this simulation is that Method 1 is not only the most accurate but also the most efficient.

*Vertical wavelength.* In Fig. 5, we illustrate the altitude profiles of the perturbed temperature $\overline{T}_s$, vertical velocity $\overline{w}_s$, and horizontal velocity $\overline{u}_s$, as well as, their Fourier spectra in the spatial frequency domain. The calculation is done for three value of the frequency parameter $\kappa_\omega$, namely, $1.0$, $0.8$, and $0.6$. The variation of the vertical wavelength $\lambda_{z0}$, corresponding to the maximum FT-amplitudes of the perturbed temperature, vertical velocity, and horizontal velocity, with respect to the frequency parameter $\kappa_\omega$ is depicted in Fig. 6. The results reveal that when the contribution of the ion drag increases (i.e., the frequency

parameters $\kappa_\omega$ decreases)

1. the widths of the Fourier spectra increases,

2. the vertical wavelength $\lambda_{z0}$, corresponding to the maximum FT-amplitude, decreases, and



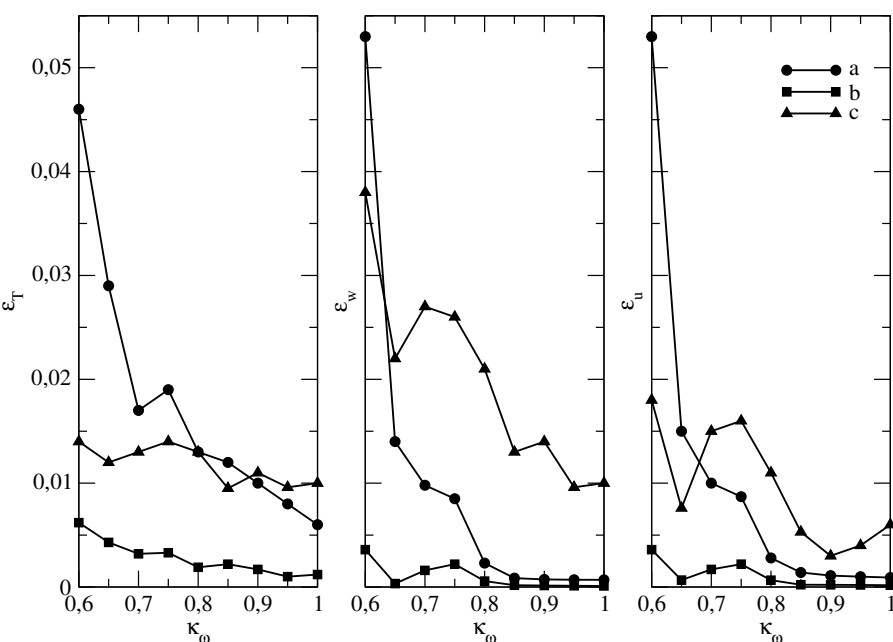

**Figure 3.** Relative root-mean square error in the perturbed temperature, vertical velocity, and horizontal velocity. The results are computed by Method 4 with the second-order Padé approximation (a) and the third-order Padé approximation (b), and by Method 5 (c).

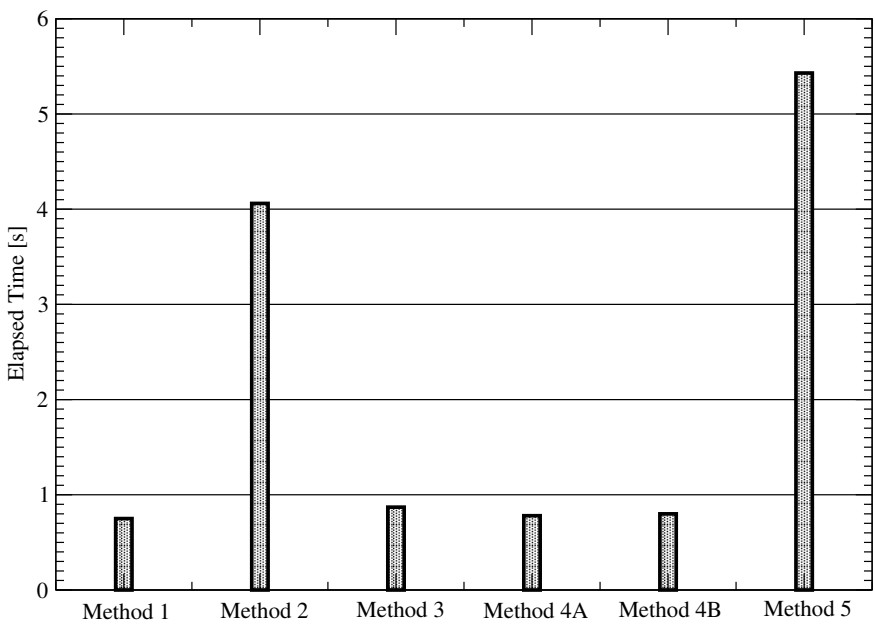

**Figure 4.** Elapsed time of all solution methods. Method 4A corresponds to the second-order Padé approximation, while Method 4B corresponds to the third-order Padé approximation.





3. the wave is more absorbed (a wave with a larger vertical wavelength penetrates deeper into the atmosphere).

What appears to be problematic for our further analysis is the fact that for a fixed $\kappa_\omega$, we cannot associate a unique vertical
wavelength to the wave (the $\lambda_{z0}$ determined by analyzing the temperature, vertical velocity, and horizontal velocity Fourier
spectra are different). Solving this problem requires more effort in the future.

*Pairwise classification of ascending and descending* modes. In the upper panels of Fig. 7, we plot the imaginary part of the
vertical wavenumber $k_{zn}$, $n = 1, \dots, 6$ for the layer conditions (27) and (163). As expected, the plots demonstrate that only in
the second case, the vertical wavenumbers appear in pairs. In the first case, the problematic altitude range for gravity waves
seems to be from $50\,\mathrm{km}$ to $120\,\mathrm{km}$, where the imaginary parts of the wavenumber for ascending and descending modes are
nearly identical, either positive or negative. Although the vertical wavenumbers do not appear in pairs, the results obtained using
both (i) global matrix methods with the scaling matrices (61) and (62), and (ii) scattering matrix methods with classification
rule (44) exhibit very close agreement. Furthermore, the use of global matrix methods with the scaling matrices (72) and (73)
(which take into account the classification of modes as ascending and descending) does not produce a significant change in
the results (although not shown here, the relative errors are less than $10^{-5}$). On the other hand, because in the altitude range
from 50 to 120 km, the amplitudes of the waves are small, categorizing wave modes into pairs seems not to be essential.
Parenthetically, we note that the wave modes almost appear in pairs for

1. the layer conditions

$$\frac{\mathrm{d}T_0}{\mathrm{d}z} \neq 0, \ \frac{\mathrm{d}u_0}{\mathrm{d}z} = 0, \ \mu_0 = \text{constant, and } \lambda_0 = \text{constant,} \tag{164}$$

that is, for a zero height derivative of the wind velocity, and

2. the layer conditions (27) but with the altitude $z$ ranging from $z_{\min} = 125\,\mathrm{km}$ to $z_{\max} = 450\,\mathrm{km}$.

This can be seen in the lower panels of Fig. 7.

*Ion drag*. The ion-drag is important for time frequencies $\omega_0$ that are smaller than the neutral-ion collision frequency $\nu_{0ni}$. To
verify this result, we analyze the influence of the ion-drag on the perturbed temperature $\overline{T}_s$, vertical velocity $\overline{w}_s$, and horizontal
velocity $\overline{u}_s$. The simulations are performed when the ion drag is excluded in the linearized equations, and when it is included;
in the second case, the frequency parameter $\kappa_\omega$ is chosen as $1.2$ and $0.8$. The results in Fig. 8, show a complete agreement
between the cases (i) ion-drag excluded and (ii) ion-drag included with $\kappa_\omega = 1.2$.

*Lower Boundary Condition*. In Fig. 9 we show the results for the lower boundary conditions (68) and (81) with $\mathbf{b}_1 = [1, 0, \dots, 0]^T$. The boundary condition (81) is imposed on the horizontal velocity $\widehat{u}_1^+$ ($q = 1$) and the vertical velocity $\widehat{w}_1^+$
($q = 2$). In all cases, the amplitude $s_0$ in Eq. (158) is computed from Eq. (88) with $\delta_{\mathrm{T}} = 0.1$. Small differences are visible in
the case (81) with $q = 2$, compared to the other two cases.

*Lower altitude level*. To model waves in the ionosphere, the lower altitude level was chosen by Volland (1969b), Kloster-
meyer (1972), and Shibata (1983) at $150\,\mathrm{km}$, by Hickey and Cole (1988) at $120\,\mathrm{km}$, and by Maeda (1985) at 100 km. Our
choice $z_{\min} = 50\,\mathrm{km}$ corresponds to that of Knight et al. (2019). It is interesting to see what is the effect of the lower altitude



**Figure 5.** Altitude profiles of the perturbed temperature $\overline{T}_s$, vertical velocity $\overline{w}_s$, and horizontal velocity $\overline{u}_s$ (upper panels), and their normalized FT-amplitudes in the spatial frequency domain (lower panels). The results correspond to $\kappa_\omega = 1$ (a), $\kappa_\omega = 0.8$ (b), and $\kappa_\omega = 0.6$ (c).




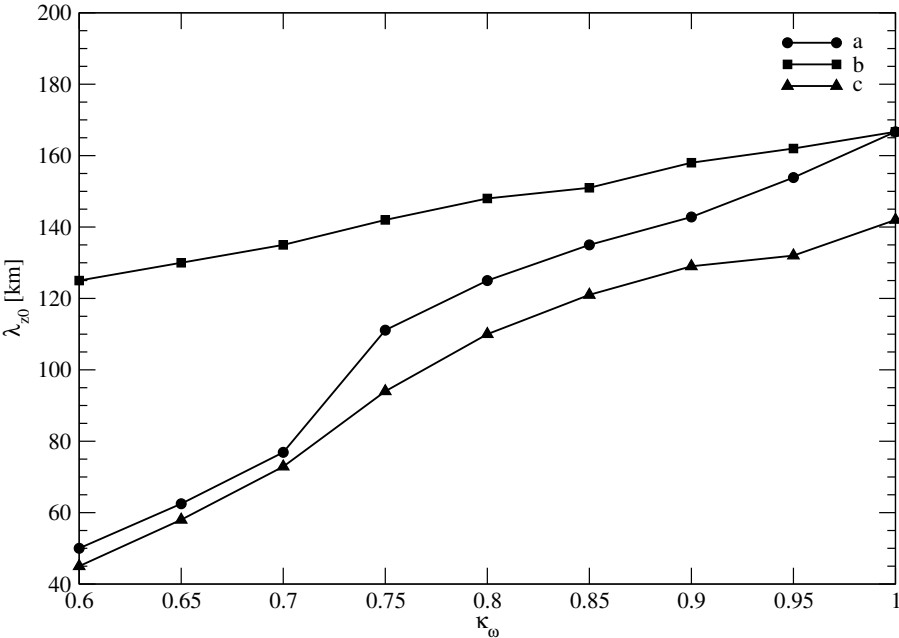

**Figure 6.** Vertical wavelengths, corresponding to the maximum FT-amplitudes, versus the frequency parameter $\kappa_\omega$. The results are obtained by analyzing the Fourier spectra of the perturbed temperature (a), vertical velocity (b), and horizontal velocity (c).

level on the results. In Fig. 10, we plot the perturbed temperature altitude profile and its Fourier spectrum for $z_{\min} = 50\,\text{km}$ and $z_{\min} = 70\,\text{km}$. In the two cases, the Fourier spectrum is practically unchanged, which makes us conclude that modifying the lower altitude level roughly causes a shift in the wave phase.

     *Computing ascending and descending wave modes.* The ascending and descending modes can be computed with Method 1 by using Eq. (87), or with Method 3 by using the recurrence relations (114) and (115). The results in Fig. 11 demonstrate that (i)
both methods yield almost the same results, and (ii) the dominant wave mode is the ascending mode. Encouraged by this result, we calculated the solution vector by using the recurrence relation (118). The plots in Fig. 12 suggest that the approximation $\mathbf{e}_l \approx \mathbf{e}_l^+$ for $l = 1, \ldots, L$, does look quite reasonable; visible differences appear at high altitudes, say, at $z \geq 350\,\text{km}$. Because this method is very efficient, it can be used to obtain quickly a primary information about the gravity wave.

## 6.2   Time wavepacket

For the source function (159), we plot in Fig. 13, the time dependent perturbed temperature and vertical velocity profiles $\overline{T}_s(z,t)$ and $\overline{w}_s(z,t)$, respectively. For each perturbed temperature profile, we determine the pair $(z_0, t_0) = \arg\max_{z,t} \overline{T}_s(z,t)$, and depict in Fig. 14, the Gaussian envelope of the source function, still denoted by $s(t)$, together with the perturbation $\overline{T}_s(z_0, t)$. The plots reveal

     1. a stronger attenuation of the waves in the case $\kappa_\omega = 0.6$;



**Figure 7.** Upper panels: The imaginary part of the vertical wavenumber $k_{zn}$, $n = 1,\ldots,6$ for the layer conditions (27) (left), and (163) (right). Lower panels: The imaginary part of the vertical wavenumber $k_{zn}$, $n = 1,\ldots,6$ for the layer conditions (164) (left), and (27) with $z_{\min} = 125\,\mathrm{km}$ and $z_{\max} = 450\,\mathrm{km}$ (right). The labels 1 and 4 correspond to ascending and descending gravity-wave modes, respectively, 2 and 5 to ascending and descending viscosity-wave modes, respectively, and 3 and 6 to ascending and descending thermal conduction-wave modes, respectively.





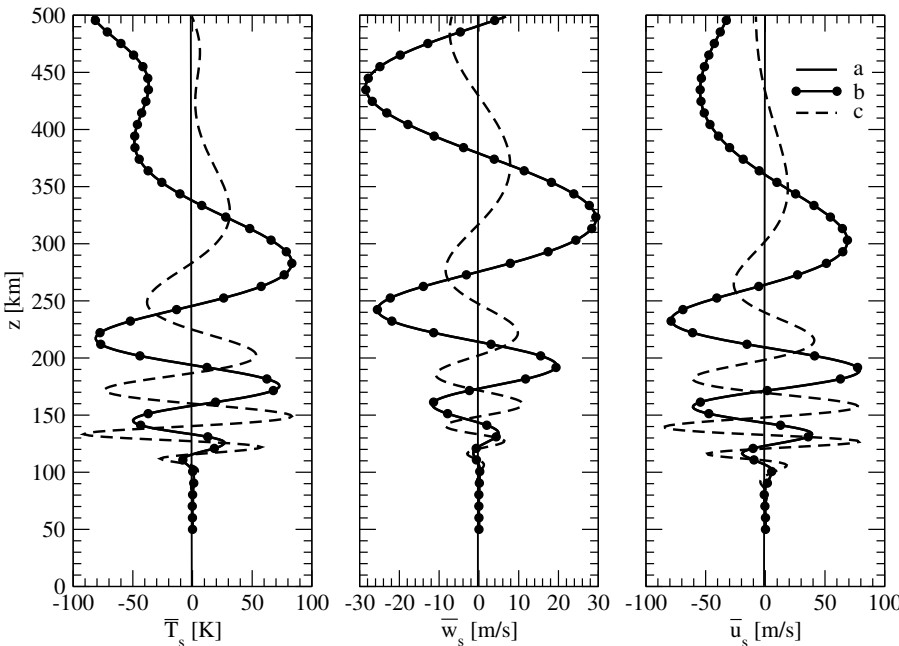

**Figure 8.** Altitude profiles of the perturbed temperature $\overline{T}_s$, vertical velocity $\overline{w}_s$, and horizontal velocity $\overline{u}_s$ when the ion drag is excluded in the linearized equations (a), and when it is included; in the latter case, the frequency parameter $\kappa_\omega$ is 1.2 (b) and 0.8 (c).

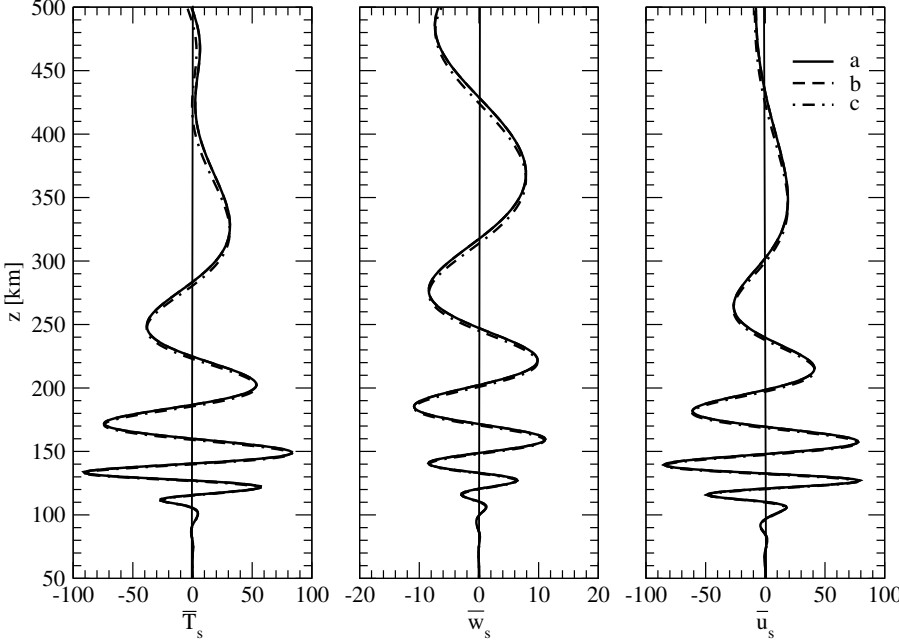

**Figure 9.** Altitude profiles of the perturbed temperature $\overline{T}_s$, vertical velocity $\overline{w}_s$, and horizontal velocity $\overline{u}_s$ for the lower boundary conditions (68) (a), and (81) with $q = 1$ (b), and $q = 2$ (c).



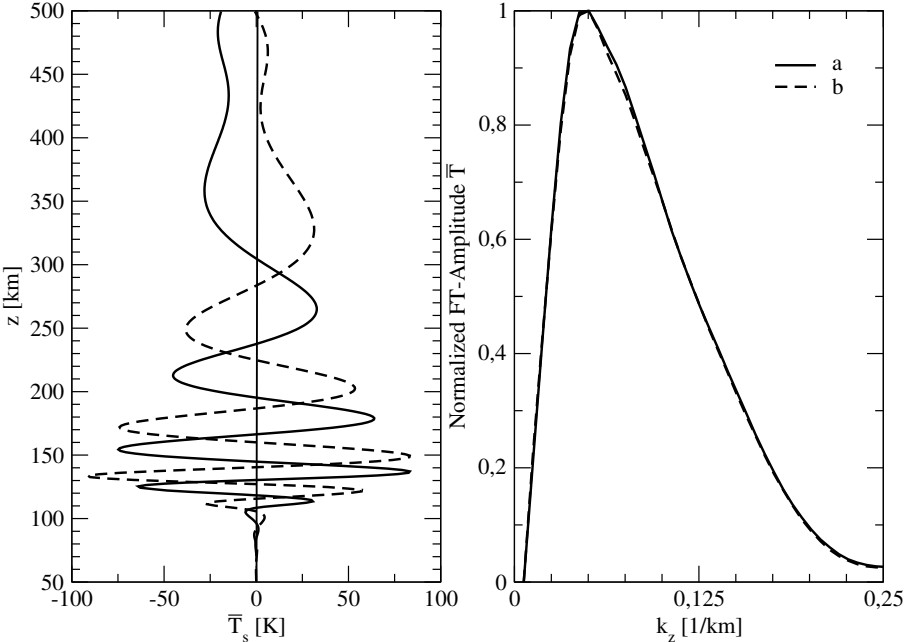

**Figure 10.** Perturbed temperature altitude profile and its Fourier spectrum for $z_{\min} = 50\,\text{km}$ (a) and $z_{\min} = 70\,\text{km}$ (b).

2. a positive time shift between the maxima of the wave and source function envelops; the time shift is (i) $168\,\text{mins}$ for
       $\kappa_\omega = 1$, (ii) $219\,\text{mins}$ for $\kappa_\omega = 0.8$, and (iii) $376\,\text{mins}$ for $\kappa_\omega = 0.6$.

Note that the computation time was $25\,\text{s}$ for Method 1 and $1530\,\text{s}$ for Method 2. If in the case of a pulse source function, we interpret causality in terms of the occurrence of maximum values, we can conclude that this is preserved.

   To understand the link between causality and imaginary frequency shifting, we plot in the upper panels of Fig. 15 the real part

of the eigenvalue $\lambda_n$, $n = 1,\ldots,6$ without and with an imaginary frequency shifting $\delta$, that is, for $\delta = 0$ and $\delta = 5 \times 10^{-3}$. The results show that the causality condition (14) is not satisfied in the case $\delta = 0$, but is almost satisfied in the case $\delta = 5 \times 10^{-3}$. Coming to the causality condition (157), we illustrate in the lower panels of Fig. 15, the Gaussian envelope of the source function $s(t)$ together with the perturbations $\widehat{T}(z_0, t) = \widehat{T}_{\delta=0}(z_0, t)$ and $\widehat{T}_\delta(z_0, t)$ with $\delta = 5 \times 10^{-3}$. The plots highlight the fact that $\widehat{T}(z_0, t)$ has small oscillations around 0 for $t \leq 0$, and that these oscillations become smaller, in the case of $\widehat{T}_\delta(z_0, t)$.

However, if we are not so strict in defining causality, we can accept the solution without imaginary frequency shifting as realistic.

## 7  Conclusions

We designed a numerical model for solving the linearized gravity-wave equations by a multilayer method. The numerical model employs the following solution methods: (i) global matrix methods using matrix exponentials, and (ii) scattering matrix





**Figure 11.** Upper panels: Altitude profiles of the perturbed temperature $\overline{T}_s$ corresponding to ascending and descending modes and being computed with Method 1 (a) and Method 3 (b). Lower panels: Altitude profiles of the perturbed temperature $\overline{T}_s$, vertical velocity $\overline{w}_s$, and horizontal velocity $\overline{u}_s$ corresponding to the total wave mode (a), ascending mode (b), and descending mode (c). The results are computed with Method 1.



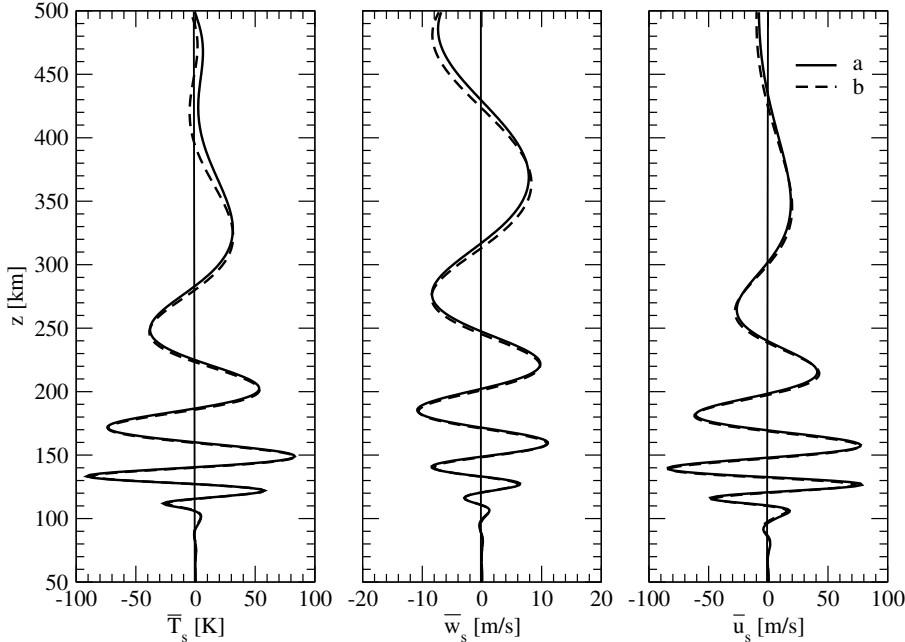

**Figure 12.** Altitude profiles of the perturbed temperature $\overline{T}_s$, vertical velocity $\overline{w}_s$, and horizontal velocity $\overline{u}_s$ computed with Method 1 (a), and by using the recurrence relation (118) (b).

methods for determining either (i) the amplitudes of the characteristic solutions or (ii) the discrete values of the solution vector. Ascending and descending wave modes are identified according to the rule that the real parts of the eigenvalues of the characteristic equation for ascending modes are smaller than those for descending modes (or equivalently, the imaginary parts of the vertical wavenumbers are smaller). Global matrix methods with the scaling matrices (61) and (62), require defining ascending and descending modes only at the upper and lower boundaries, while the use of the scaling matrices (72) and

(73) requires a classification of modes as ascending and descending in each layer. Scattering matrix methods also require an explicit determination of the mode type at every altitude. The model includes two types of lower boundary condition, namely, (68) and (81). Depending on the type of the source function, single-frequency waves or time wavepackets can be analyzed. The amplitude of the source function can be computed by imposing an upper bound for the perturbed temperature (88), or the horizontal wind velocity (89). The model is devoted to the solution of the linearized equations with viscosity, thermal

conduction, and ion drag included. According to Eqs. (A53)–(A55), it can be simplified to an isothermal atmosphere with constant wind velocity, a homogeneous atmosphere, and an atmosphere without ion drag.

     Numerical simulations demonstrate that both global matrix and scattering matrix methods achieve comparable accuracies. However, the former exhibit significantly greater efficiency than the latter, especially noticeable in simulations involving time wavepackets. Within global matrix methods, the approach that solves for the amplitudes of the characteristic solutions appears

to offer the highest efficiency and accuracy.




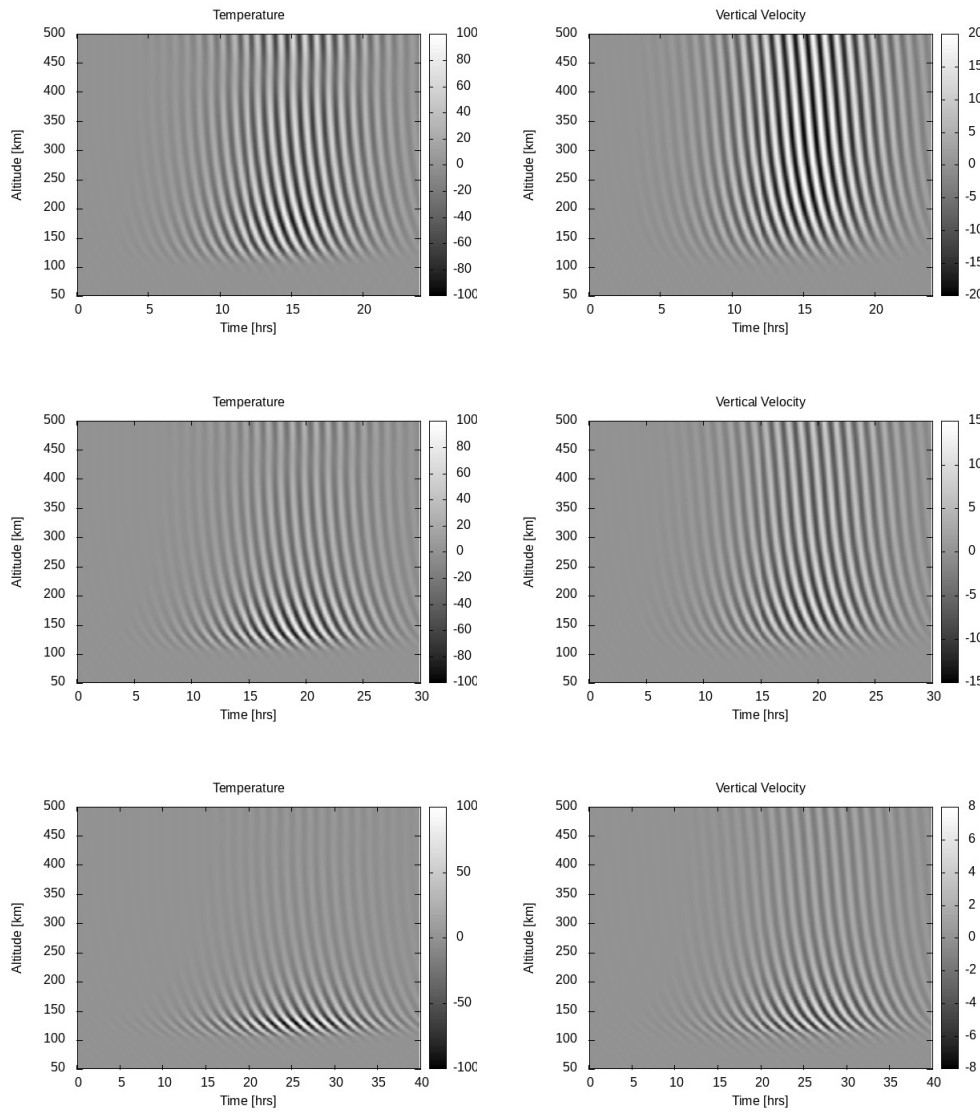

**Figure 13.** Perturbed temperature (left panels) and vertical velocity (right panels) as functions of time and altitude. The upper panels correspond to the frequency parameter $\kappa_\omega = 1$, the middle panels to $\kappa_\omega = 0.8$, and the lower panels to $\kappa_\omega = 0.6$.





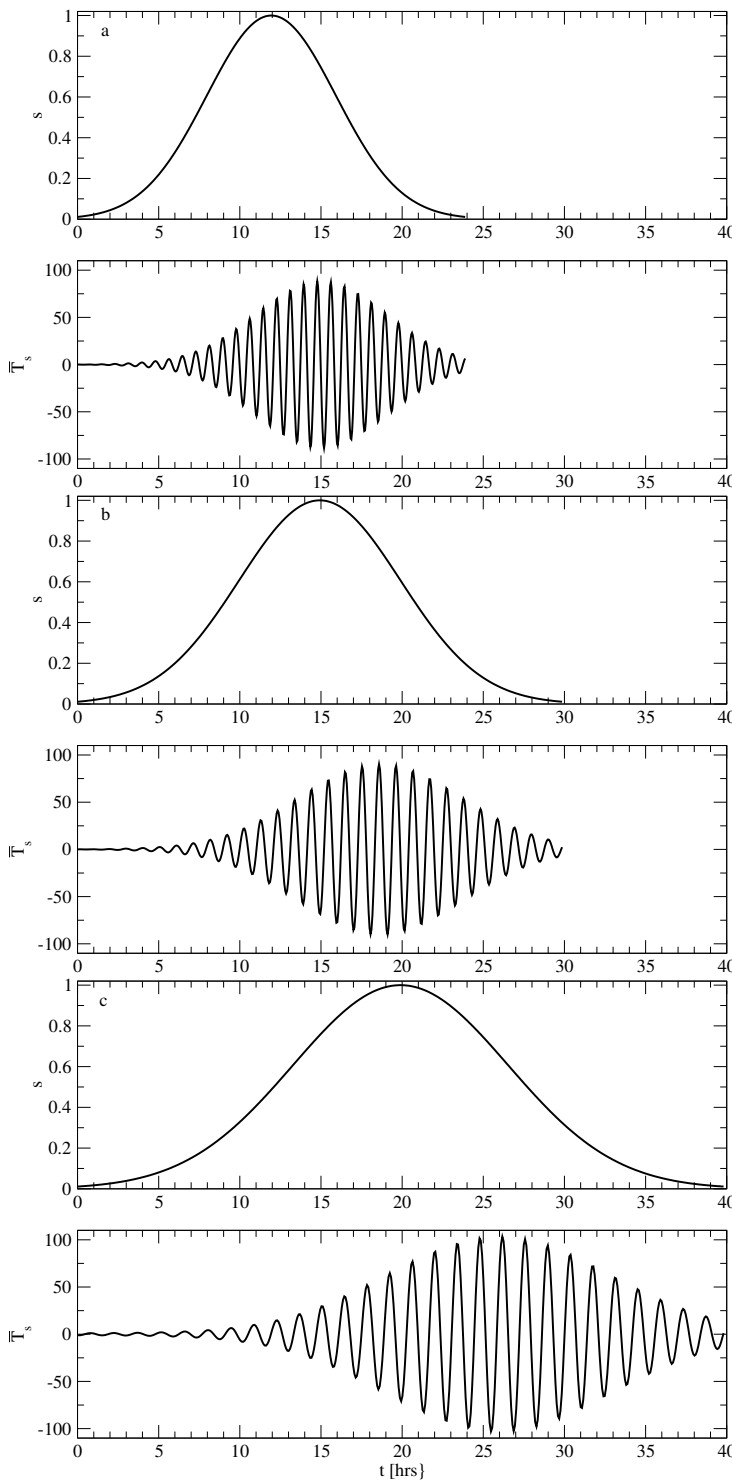

**Figure 14.** The Gaussian envelope of the source function $s$ together with the perturbation $\overline{T}_s(z_0, \cdot)$. The results correspond to the frequency parameter $\kappa_\omega = 1$ (a), $\kappa_\omega = 0.8$ (b), and $\kappa_\omega = 0.6$ (c).



**Figure 15.** Upper panels: The real part of the eigenvalue $\lambda_n$, $n = 1, \ldots, 6$ for $\delta = 0$ (left) and $\delta = 5 \times 10^{-3}$ (right). The upper bound for ascending modes (AM) is indicated by a solid line, while the lower bound for descending modes (DM) is indicated by a dashed line. Lower panels: The Gaussian envelope of the source function $s$ together with the perturbation $\widehat{T}(z_0, \cdot) = \widehat{T}_{\delta=0}(z_0, \cdot)$ and $\widehat{T}_\delta(z_0, \cdot)$ with $\delta = 5 \times 10^{-3}$.





The linearized equations on which the solution methods were tested correspond to the ionosphere. In fact, the final goal of our research is to design a complete model for analyzing ionospheric gravity waves from satellite limb measurements. The approach presented in this paper represents only the first component of this complete model. Our strategy involves a two-step process: first, to solve the neutral-atmosphere equations using Klostermeyer's approximation, and then, in the second step, to
use the wave-induced perturbations from the initial step to solve the ionospheric equations for the perturbed density of $O^+$-ions and to calculate the volume emission rate of a line with a wavelength of 630.0 nm. The ionospheric equations will be solved using the SAMI2 model (Huba et al., 2000). Thus, our strategy is to prioritize the ionospheric equations of the SAMI2 model, and to manage the wave-induced perturbations using the present approximate approach. The final goal we proposed also explains some of the approximations we made. First, we use a linearized model for faster processing of the measurements
made by the satellite instrument. Second, we consider a two-dimensional geometry because for limb measurements, it is much easier in this case to account for the curvature of the atmosphere through a series of approximations, such as the shallow-atmosphere approximation and an order-of-magnitude approximation. A comprehensive description of this complete model will be addressed in a future paper.

## Appendix A:  The linearized hydrodynamic equations for the neutral atmosphere

In this appendix we present a general model for the hydrodynamic equations including viscosity, thermal conduction, and ion drag, together with some simplified models, which are frequently found in the literature.

### A1. General model

To solve the linearized hydrodynamic equations for the neutral atmosphere, we use assumptions A1–A4, and let

$$\mathbf{u}_0 = (u_0, 0, 0), \tag{A1}$$

$$\mathbf{u}' = (u', 0, w'), \tag{A2}$$

$$\mathbf{g} = (0, 0, -g), \tag{A3}$$

$$\widehat{\mathbf{b}} = (-\cos I, 0, -\sin I), \tag{A4}$$

where $I$ the geomagnetic inclination (being positive in the northern hemisphere). For

$$f(x, z, t) = f_0(z) + f'(x, z, t), \tag{A5}$$





that is,

$$u(x,z,t) = u_0(z) + u'(x,z,t), \tag{A6}$$

$$w(x,z,t) = w'(x,z,t), \tag{A7}$$

$$p(x,z,t) = p_0(z) + p'(x,z,t), \tag{A8}$$

$$T(x,z,t) = T_0(z) + T'(x,z,t), \tag{A9}$$

$$\rho(x,z,t) = \rho_0(z) + \rho'(x,z,t), \tag{A10}$$

and

$$n_i'(x,z,t) = n_{0i}(z) + n_i'(x,z,t), \tag{A11}$$

the linearized equations (2)–(5) can be written in component form as

$$\frac{\partial \rho'}{\partial t} = -w' \frac{\partial \rho_0}{\partial z} - \rho_0 \left( \frac{\partial u'}{\partial x} + \frac{\partial w'}{\partial z} \right) - u_0 \frac{\partial \rho'}{\partial x}, \tag{A12}$$

$$\begin{aligned}
\rho_0 \frac{\partial u'}{\partial t} =& -\frac{\partial p'}{\partial x} + \frac{4}{3}\mu_0 \frac{\partial^2 u'}{\partial x^2} + \mu_0 \frac{\partial^2 u'}{\partial z^2} + \frac{1}{3}\mu_0 \frac{\partial^2 w'}{\partial x \partial z} \\
&+ \frac{\mathrm{d}\mu_0}{\mathrm{d}z} \left( \frac{\partial u'}{\partial z} + \frac{\partial w'}{\partial x} \right) + \mu_0 V_\mathrm{T} \left( \frac{T'}{T_0} \right) + \mu_0 V_\mathrm{dT} \frac{\partial}{\partial z} \left( \frac{T'}{T_0} \right), \\
&- \rho_0 \left( u_0 \frac{\partial u'}{\partial x} + w' \frac{\partial u_0}{\partial z} \right) - f'_{\mathrm{ID}x},
\end{aligned} \tag{A13}$$

$$\begin{aligned}
\rho_0 \frac{\partial w'}{\partial t} =& -\frac{\partial p'}{\partial z} - \rho'g + \frac{4}{3}\mu_0 \frac{\partial^2 w'}{\partial z^2} + \mu_0 \frac{\partial^2 w'}{\partial x^2} + \frac{1}{3}\mu_0 \frac{\partial^2 u'}{\partial x \partial z} \\
&+ \frac{\mathrm{d}\mu_0}{\mathrm{d}z} \left( \frac{4}{3} \frac{\partial w'}{\partial z} - \frac{2}{3} \frac{\partial u'}{\partial x} \right) + \mu_0 V_\mathrm{dT} \left( \frac{1}{T_0} \frac{\partial T'}{\partial x} \right) \\
&- \rho_0 u_0 \frac{\partial w'}{\partial x} - f'_{\mathrm{ID}z},
\end{aligned} \tag{A14}$$

$$\begin{aligned}
\rho_0 c_\mathrm{v} \frac{\partial T'}{\partial t} =& -p_0 \left( \frac{\partial u'}{\partial x} + \frac{\partial w'}{\partial z} \right) + \lambda_0 \frac{\partial^2 T'}{\partial x^2} + \lambda_0 \frac{\partial^2 T'}{\partial z^2} + \frac{\mathrm{d}\lambda_0}{\mathrm{d}z} \frac{\partial T'}{\partial z} \\
\quad &+ \lambda_0 C_\mathrm{T} \left( \frac{T'}{T_0} \right) + \lambda_0 C_\mathrm{dT} \frac{\partial}{\partial z} \left( \frac{T'}{T_0} \right) \\
&+ \mu_0 W_\mathrm{u} \frac{\partial u'}{\partial z} + \mu_0 W_\mathrm{u} \frac{\partial w'}{\partial x} + \mu_0 W_\mathrm{T} \frac{T'}{T_0} \\
&- \rho_0 c_\mathrm{v} \left( u_0 \frac{\partial T'}{\partial x} + w' \frac{\partial T_0}{\partial z} \right) - P'_{\mathrm{ID}},
\end{aligned} \tag{A15}$$

and

$$\frac{p'}{p_0} = \frac{T'}{T_0} + \frac{\rho'}{\rho_0}. \tag{A16}$$





In the above equations, $\mathbf{f}'_{\text{ID}} = (f'_{\text{IDx}}, 0, f'_{\text{IDz}})$, $\mu_0$ and $\lambda_0$ are computed by using the relations (cf. Eq. (25))

$$\mu_0 = 3.34 \times 10^{-7} T_0^{0.71} \text{ and } \lambda_0 = 6.71 \times 10^{-4} T_0^{0.71}, \tag{A17}$$

respectively, while the quantities $V_{\text{T}}$, $V_{\text{dT}}$, $C_{\text{T}}$, $C_{\text{dT}}$, $W_{\text{u}}$, and $W_{\text{T}}$ are given respectively, by

$$V_{\text{T}} = 0.71 \frac{\mathrm{d}^2 u_0}{\mathrm{d}z^2} + 0.71 \frac{1}{\mu_0} \frac{\mathrm{d}\mu_0}{\mathrm{d}z} \frac{\mathrm{d}u_0}{\mathrm{d}z}, \ V_{\text{dT}} = 0.71 \frac{\mathrm{d}u_0}{\mathrm{d}z}, \tag{A18}$$

$$C_{\text{T}} = 0.71 \frac{\mathrm{d}^2 T_0}{\mathrm{d}z^2} + 0.71 \frac{1}{\lambda_0} \frac{\mathrm{d}\lambda_0}{\mathrm{d}z} \frac{\mathrm{d}T_0}{\mathrm{d}z}, \ C_{\text{dT}} = 0.71 \frac{\mathrm{d}T_0}{\mathrm{d}z}, \tag{A19}$$

$$W_{\text{u}} = 2 \frac{\mathrm{d}u_0}{\mathrm{d}z}, \ W_{\text{T}} = 0.71 \left( \frac{\mathrm{d}u_0}{\mathrm{d}z} \right)^2. \tag{A20}$$

In a second step, we use Eq. (A16) to eliminate the mass density perturbation $\rho'$ in Eqs. (A12)–(A15), and in a third step, we assume the plane wave solutions

$$f'(x, z, t) = \overline{f}(z) \mathrm{e}^{\mathrm{j}(\omega t - k_{\text{x}} x)}, \tag{A21}$$

where $f$ stands for $u$, $w$, $p$, $T$, and $n_i$. Under assumption (A21), the linearized equations (A12)–(A15) transform into a linear
system of ordinary differential equations. This system of ordinary differential equations will be expressed in terms of the dimensionless quantities $\widehat{u}$, $\widehat{w}$, $\widehat{p}$, and $\widehat{T}$, defined through the relations

$$\overline{u}(z) = \frac{\omega_0}{k_{\text{x}}} \widehat{u}(z), \tag{A22}$$

$$\overline{w}(z) = \frac{\omega_0}{k_{\text{x}}} \widehat{w}(z), \tag{A23}$$

$$\overline{p}(z) = p_0(z) \widehat{p}(z), \tag{A24}$$

$$\overline{T}(z) = T_0(z) \widehat{T}(z), \tag{A25}$$

and of the derivatives $\widehat{\mathcal{U}}$ and $\widehat{\mathcal{T}}$, defined through the relations

$$\widehat{\mathcal{U}} = 3\eta \frac{1}{k_{\text{x}}} \frac{\mathrm{d}\widehat{u}}{\mathrm{d}z}, \ \widehat{\mathcal{T}} = \nu \frac{1}{k_{\text{x}}} \frac{\mathrm{d}\widehat{T}}{\mathrm{d}z}, \tag{A26}$$

where $\omega_0$ is a reference frequency, and

$$\eta = \mathrm{j} \frac{\omega_0 \mu_0}{3 p_0} \text{ and } \nu = \mathrm{j} \frac{k_{\text{x}}^2 \lambda_0 T_0}{\omega_0 p_0} \tag{A27}$$

are dimensionless parameters. The linear system of ordinary differential equations consist of six equations. These are:

$$\frac{1}{k_{\text{x}}} \frac{\mathrm{d}\widehat{u}}{\mathrm{d}z} = \frac{1}{3\eta} \widehat{\mathcal{U}}, \tag{A28}$$





$$\frac{1}{k_{\mathrm{x}}}\frac{\mathrm{d}\widehat{w}}{\mathrm{d}z} = \frac{1}{k_{\mathrm{x}}}A_{\mathrm{u}}\widehat{u} + \frac{1}{k_{\mathrm{x}}}A_{\mathrm{w}}\widehat{w} + \frac{1}{\omega_0}A_{\mathrm{p}}\widehat{p} - \frac{1}{\omega_0}A_{\mathrm{p}}\widehat{T},$$ (A29)

$$\frac{k_{\mathrm{x}}^2}{\omega_0}\left(\frac{4}{3}\mu_0 A_{\mathrm{p}} - p_0\right)\left(\frac{1}{k_x}\frac{\mathrm{d}\widehat{p}}{\mathrm{d}z}\right)$$

$$= -\left[\frac{2}{3}\mathrm{j}k_{\mathrm{x}}\frac{\mathrm{d}\mu_0}{\mathrm{d}z} + \left(\frac{4}{3}\frac{\mathrm{d}\mu_0}{\mathrm{d}z} + \frac{4}{3}\mu_0 A_{\mathrm{w}}\right)A_{\mathrm{u}}\right]\widehat{u}$$
$$+ \left[\mathrm{j}\widehat{\omega}\rho_0 + \mu_0 k_{\mathrm{x}}^2 - \frac{4}{3}\mu_0\frac{\mathrm{d}A_{\mathrm{w}}}{\mathrm{d}z} - \left(\frac{4}{3}\frac{\mathrm{d}\mu_0}{\mathrm{d}z} + \frac{4}{3}\mu_0 A_{\mathrm{w}}\right)A_{\mathrm{w}}\right]\widehat{w}$$
$$+ \frac{k_{\mathrm{x}}}{\omega_0}\left[\rho_0 g + \frac{\mathrm{d}p_0}{\mathrm{d}z} - \frac{4}{3}\mu_0\frac{\mathrm{d}A_{\mathrm{p}}}{\mathrm{d}z} - \left(\frac{4}{3}\frac{\mathrm{d}\mu_0}{\mathrm{d}z} + \frac{4}{3}\mu_0 A_{\mathrm{w}}\right)A_P\right]\widehat{p}$$
$$+ \frac{k_{\mathrm{x}}}{\omega_0}\left[\mathrm{j}k_{\mathrm{x}}\mu_0 V_{\mathrm{dT}} - \rho_0 g + \frac{4}{3}\mu_0\frac{\mathrm{d}A_{\mathrm{p}}}{\mathrm{d}z} + \left(\frac{4}{3}\frac{\mathrm{d}\mu_0}{\mathrm{d}z} + \frac{4}{3}\mu_0 A_{\mathrm{w}}\right)A_P\right]\widehat{T}$$
$$+ \frac{k_{\mathrm{x}}}{3\eta}\left(\frac{1}{3}\mathrm{j}k_{\mathrm{x}}\mu_0 - \frac{4}{3}\mu_0 A_{\mathrm{u}}\right)\widehat{\mathcal{U}} + \frac{4}{3}\frac{k_{\mathrm{x}}^2\mu_0 A_{\mathrm{p}}}{\omega_0\nu}\widehat{\mathcal{T}} + \rho_0\nu_{0ni}\widehat{f}_{\mathrm{ID}z},$$ (A30)

$$\frac{1}{k_{\mathrm{x}}}\frac{\mathrm{d}\widehat{T}}{\mathrm{d}z} = \frac{1}{\nu}\widehat{\mathcal{T}},$$ (A31)

$$\frac{k_{\mathrm{x}}^2\mu_0}{3\eta}\left(\frac{1}{k_{\mathrm{x}}}\frac{\mathrm{d}\widehat{\mathcal{U}}}{\mathrm{d}z}\right)$$

$$= \left[\mathrm{j}\widehat{\omega}\rho_0 + \frac{4}{3}k_{\mathrm{x}}^2\mu_0 + \frac{1}{3}\mathrm{j}k_{\mathrm{x}}\mu_0 A_{\mathrm{u}}\right]\widehat{u}$$
$$+ \left(\mathrm{j}k_{\mathrm{x}}\frac{\mathrm{d}\mu_0}{\mathrm{d}z} + \rho_0\frac{\mathrm{d}u_0}{\mathrm{d}z} + \frac{1}{3}\mathrm{j}k_{\mathrm{x}}\mu_0 A_{\mathrm{w}}\right)\widehat{w}$$
$$+ \frac{k_{\mathrm{x}}}{\omega_0}\left(\frac{1}{3}\mathrm{j}k_{\mathrm{x}}\mu_0 A_{\mathrm{p}} - \mathrm{j}k_{\mathrm{x}}p_0\right)\widehat{p}$$
$$- \frac{k_{\mathrm{x}}}{\omega_0}\left(\frac{1}{3}\mathrm{j}k_{\mathrm{x}}\mu_0 A_{\mathrm{p}} + \mu_0 V_{\mathrm{T}}\right)\widehat{T}$$

$$- \frac{k_{\mathrm{x}}}{3\eta}\left(\frac{\mathrm{d}\mu_0}{\mathrm{d}z} - \mu_0\frac{1}{\eta}\frac{\mathrm{d}\eta}{\mathrm{d}z}\right)\widehat{\mathcal{U}} - \frac{k_{\mathrm{x}}^2\mu_0 V_{\mathrm{dT}}}{\omega_0\nu}\widehat{\mathcal{T}} + \rho_0\nu_{0ni}\widehat{f}_{\mathrm{ID}x},$$ (A32)

and





$$\frac{k_{\mathrm{x}}^3 \lambda_0 T_0}{\omega_0 \nu}\left(\frac{1}{k_{\mathrm{x}}} \frac{\mathrm{d}\widehat{\mathcal{T}}}{\mathrm{d}z}\right)$$

$$= (-\mathrm{j}k_{\mathrm{x}}p_0 + p_0 A_{\mathrm{u}})\widehat{u}$$

$$+ \left(\rho_0 c_{\mathrm{v}} \frac{\mathrm{d}T_0}{\mathrm{d}z} + \mathrm{j}k_{\mathrm{x}}\mu_0 W_{\mathrm{u}} + p_0 A_{\mathrm{w}}\right)\widehat{w}$$

$$+ \frac{k_{\mathrm{x}}}{\omega_0} p_0 A_{\mathrm{p}}\widehat{p}$$

$$+ \frac{k_{\mathrm{x}}}{\omega_0}\left[\mathrm{j}c_{\mathrm{v}}\widehat{\omega}\rho_0 T_0 + k_{\mathrm{x}}^2 \lambda_0 T_0 - \lambda_0 C_{\mathrm{T}} - \mu_0 W_{\mathrm{T}} - \frac{\mathrm{d}\lambda_0}{\mathrm{d}z}\frac{\mathrm{d}T_0}{\mathrm{d}z} - p_0 A_{\mathrm{p}} - \lambda_0 \frac{\mathrm{d}^2 T_0}{\mathrm{d}z^2}\right]\widehat{T}$$

$$- \frac{k_{\mathrm{x}}\mu_0 W_{\mathrm{u}}}{3\eta}\widehat{\mathcal{U}} - \frac{k_{\mathrm{x}}^2}{\omega_0 \nu}\left(\frac{\mathrm{d}\lambda_0}{\mathrm{d}z}T_0 + 2\lambda_0\frac{\mathrm{d}T_0}{\mathrm{d}z} + \lambda_0 C_{\mathrm{dT}} - \lambda_0\frac{1}{\nu}\frac{\mathrm{d}\nu}{\mathrm{d}z}T_0\right)\widehat{\mathcal{T}} + \rho_0 \nu_{0ni}\widehat{P}_{\mathrm{ID}}. \tag{A33}$$

In Eqs. (A28)–(A33),

1. $A_{\mathrm{u}}$, $A_{\mathrm{w}}$, and $A_{\mathrm{p}}$ are given respectively, by

$$A_{\mathrm{u}} = \mathrm{j}k_{\mathrm{x}}, \; A_{\mathrm{w}} = -\frac{1}{\rho_0}\frac{\mathrm{d}\rho_0}{\mathrm{d}z}, \; A_{\mathrm{p}} = -\mathrm{j}\widehat{\omega}, \tag{A34}$$

where $\widehat{\omega} = \omega - k_{\mathrm{x}}u_0$ is the intrinsic frequency,

2. the specific heat at constant volume is computed as $c_{\mathrm{v}} = R_{\mathrm{M}}/(\gamma - 1)$, where $\gamma = 1.4$ is the ratio of specific heats and $R_{\mathrm{M}} = 287\,\mathrm{J/kg\,K}$ the specific gas constant,

3. the background neutral-ion collision frequency $\nu_{0ni}$ is calculated from Eq. (19),

4. the ion-drag terms in Eqs. (A30) and (A32) are given by

$$\widehat{f}_{\mathrm{IDx}} = \left(f_{\mathrm{xu}} + \frac{1}{\omega_0}f_{\mathrm{xn}}N_{\mathrm{u}} + \frac{1}{\omega_0}f_{\mathrm{xn}}N_{\mathrm{dw}}A_{\mathrm{u}}\right)\widehat{u}$$

$$+ \left(f_{\mathrm{xw}} + \frac{1}{\omega_0}f_{\mathrm{xn}}N_{\mathrm{w}} + \frac{1}{\omega_0}f_{\mathrm{xn}}N_{\mathrm{dw}}A_{\mathrm{w}}\right)\widehat{w}$$

$$+ \frac{k_{\mathrm{x}}}{\omega_0}\left(f_{\mathrm{xn}} + \frac{1}{\omega_0}f_{\mathrm{xn}}N_{\mathrm{dw}}A_{\mathrm{p}}\right)\widehat{p}$$

$$+ \frac{k_{\mathrm{x}}}{\omega_0}\left(f_{\mathrm{xT}} - f_{\mathrm{xn}} - \frac{1}{\omega_0}f_{\mathrm{xn}}N_{\mathrm{dw}}A_{\mathrm{p}}\right)\widehat{T}$$

$$+ \frac{k_{\mathrm{x}}}{\omega_0}f_{\mathrm{xn}}N_{\mathrm{du}}\frac{1}{3\eta}\widehat{\mathcal{U}} \tag{A35}$$




and

$$\widehat{f}_{\mathrm{IDz}} = \left( f_{\mathrm{zu}} + \frac{1}{\omega_0} f_{\mathrm{zn}} N_{\mathrm{u}} + \frac{1}{\omega_0} f_{\mathrm{zn}} N_{\mathrm{dw}} A_{\mathrm{u}} \right) \widehat{u}$$
$$+ \left( f_{\mathrm{zw}} + \frac{1}{\omega_0} f_{\mathrm{zn}} N_{\mathrm{w}} + \frac{1}{\omega_0} f_{\mathrm{zn}} N_{\mathrm{dw}} A_{\mathrm{w}} \right) \widehat{w}$$
$$+ \frac{k_{\mathrm{x}}}{\omega_0} \left( f_{\mathrm{zn}} + \frac{1}{\omega_0} f_{\mathrm{zn}} N_{\mathrm{dw}} A_{\mathrm{p}} \right) \widehat{p}$$
$$+ \frac{k_{\mathrm{x}}}{\omega_0} \left( f_{\mathrm{zT}} - f_{\mathrm{zn}} - \frac{1}{\omega_0} f_{\mathrm{zn}} N_{\mathrm{dw}} A_{\mathrm{p}} \right) \widehat{T}$$
$$+ \frac{k_{\mathrm{x}}}{\omega_0} f_{\mathrm{zn}} N_{\mathrm{du}} \frac{1}{3\eta} \widehat{\mathcal{U}}, \tag{A36}$$

respectively, where

$$f_{\mathrm{xu}} = \sin^2 I, \ f_{\mathrm{xw}} = -\cos I \sin I, \tag{A37}$$
$$f_{\mathrm{xT}} = 0.37 \sin^2 I u_0, \ f_{\mathrm{xn}} = \sin^2 I u_0, \tag{A38}$$
$$f_{\mathrm{zu}} = -\cos I \sin I, \ f_{\mathrm{zw}} = \cos^2 I, \tag{A39}$$
$$f_{\mathrm{zT}} = -0.37 \cos I \sin I u_0, \ f_{\mathrm{zn}} = -\cos I \sin I u_0, \tag{A40}$$

and

$$N_{\mathrm{u}} = \frac{\omega_0}{\omega} \left( k_{\mathrm{x}} \cos^2 I + \mathrm{j} \frac{1}{n_{0i}} \frac{\mathrm{d} n_{0i}}{\mathrm{d} z} \cos I \sin I \right), \tag{A41}$$
$$N_{\mathrm{w}} = \frac{\omega_0}{\omega} \left( k_{\mathrm{x}} \cos I \sin I + \mathrm{j} \frac{1}{n_{0i}} \frac{\mathrm{d} n_{0i}}{\mathrm{d} z} \sin^2 I \right), \tag{A42}$$
$$N_{\mathrm{du}} = \mathrm{j} \frac{\omega_0}{\omega} \cos I \sin I, \ N_{\mathrm{dw}} = \mathrm{j} \frac{\omega_0}{\omega} \sin^2 I, \tag{A43}$$

and finally,

5. the ion-drag term in Eq. (A33) is given by

$$\widehat{P}_{\mathrm{ID}} = \left( P_{\mathrm{u}} + \frac{1}{\omega_0} P_{\mathrm{n}} N_{\mathrm{u}} + \frac{1}{\omega_0} P_{\mathrm{n}} N_{\mathrm{dw}} A_{\mathrm{u}} \right) \widehat{u}$$
$$+ \left( P_{\mathrm{w}} + \frac{1}{\omega_0} P_{\mathrm{n}} N_{\mathrm{w}} + \frac{1}{\omega_0} P_{\mathrm{n}} N_{\mathrm{dw}} A_{\mathrm{w}} \right) \widehat{w}$$
$$+ \frac{k_{\mathrm{x}}}{\omega_0} \left( P_{\mathrm{n}} + \frac{1}{\omega_0} P_{\mathrm{n}} N_{\mathrm{dw}} A_{\mathrm{p}} \right) \widehat{p}$$
$$+ \frac{k_{\mathrm{x}}}{\omega_0} \left( P_{\mathrm{T}} - P_{\mathrm{n}} - \frac{1}{\omega_0} P_{\mathrm{n}} N_{\mathrm{dw}} A_{\mathrm{p}} \right) \widehat{T}$$
$$+ \frac{k_{\mathrm{x}}}{\omega_0} P_{\mathrm{n}} N_{\mathrm{du}} \frac{1}{3\eta} \widehat{\mathcal{U}}, \tag{A44}$$





where

$$P_{\mathrm{u}} = 2\sin^2 I u_0, \ \ P_{\mathrm{w}} = -2\cos I \sin I u_0, \tag{A45}$$

$$P_{\mathrm{T}} = 0.37\sin^2 I u_0^2, \ \ P_{\mathrm{n}} = \sin^2 I u_0^2. \tag{A46}$$

Note that the ion drag terms $\widehat{f}_{\mathrm{ID}x}$, $\widehat{f}_{\mathrm{ID}z}$, and $\widehat{P}_{\mathrm{ID}}$ are computed as described in the main text. In short, in a first step, we assume the representations $n_i'(x,z,t) = \overline{n}_i(z)\exp[\mathrm{j}(\omega t - k_{\mathrm{x}}x)]$ and $\overline{n}_i(z) = n_{0i}(z)\widehat{n}_i(z)$, and use the continuity equation (18) to obtain

$$k_{\mathrm{x}}\widehat{n}_i = \frac{1}{k_{\mathrm{x}}}\left(N_{\mathrm{u}}\widehat{u} + N_{\mathrm{w}}\widehat{w} + N_{\mathrm{du}}\frac{\partial\widehat{u}}{\partial z} + N_{\mathrm{dw}}\frac{\partial\widehat{w}}{\partial z}\right), \tag{A47}$$

In a second step, we use Eq. (A47) together with Eqs. (19) and (20) to derive a representation for $\widehat{\nu}_{ni}$, which is then substituted in Eqs. (16) and (17) to compute the desired quantities.

The linear system of ordinary differential equations (A28)–(A33) can be written in matrix form as

$$\frac{1}{k_{\mathrm{x}}}\frac{\mathrm{d}\mathbf{e}}{\mathrm{d}z} = \mathrm{A}\mathbf{e}, \tag{A48}$$

where

$$\mathbf{e} = [\widehat{u}, \widehat{w}, \widehat{p}, \widehat{T}, \widehat{\mathcal{U}}, \widehat{\mathcal{T}}]^T. \tag{A49}$$

For a numerical solution, the atmosphere is divided into thin layers, and the altitude dependent matrix A is approximated by its value in the middle of each layer.

Some comments can be made here.

1. The background quantities $\rho_0$, $T_0$, $p_0 = \rho_0 R_{\mathrm{M}}T_0$, $u_0$ and $n_{0i}$ are assumed to be input parameters of the model, in which case, the derivatives

$$\frac{\mathrm{d}T_0}{\mathrm{d}z}, \ \frac{\mathrm{d}u_0}{\mathrm{d}z}, \ \text{and} \ \frac{1}{n_{0i}}\frac{\mathrm{d}n_{0i}}{\mathrm{d}z}$$

are computed by finite-differences, and the derivatives $\mathrm{d}p_0/\mathrm{d}z$ and $\mathrm{d}\rho_0/\mathrm{d}z$ with the relations

$$\frac{\mathrm{d}p_0}{\mathrm{d}z} = -\rho_0 g, \tag{A50}$$

$$\frac{1}{\rho_0}\frac{\mathrm{d}\rho_0}{\mathrm{d}z} = \frac{1}{p_0}\frac{\mathrm{d}p_0}{\mathrm{d}z} - \frac{1}{T_0}\frac{\mathrm{d}T_0}{\mathrm{d}z} = -\frac{1}{H_{\mathrm{a}}} - \frac{1}{T_0}\frac{\mathrm{d}T_0}{\mathrm{d}z}, \tag{A51}$$

where

$$H_{\mathrm{a}} = \frac{p_0}{\rho_0 g} = \frac{R_{\mathrm{M}}T_0}{g} \tag{A52}$$

is the atmospheric scale height.

2. The model can be particularized as follows:





(a) for an isothermal atmosphere with constant wind velocity (characterized by $T_0 = $ constant and $u_0 = $ constant), we
set

$$\frac{\partial T_0}{\partial z} = 0, \ \frac{\partial u_0}{\partial z} = 0, \ \frac{1}{\rho_0}\frac{\partial \rho_0}{\partial z} = -\frac{1}{H_\mathrm{a}}, \ \text{and} \ \frac{\partial A_\mathrm{w}}{\partial z} = 0. \tag{A53}$$

(b) for a homogeneous atmosphere (characterized by $\mu_\mathrm{k} = \mu_0/\rho_0 = $ constant and $\lambda_\mathrm{k} = \lambda_0/\rho_0 = $ constant), we use the
computational formulas

$$\frac{\mathrm{d}\mu_0}{\mathrm{d}z} = \mu_k \frac{\mathrm{d}\rho_0}{\mathrm{d}z} \ \text{and} \ \frac{\mathrm{d}\lambda_0}{\mathrm{d}z} = \lambda_k \frac{\mathrm{d}\rho_0}{\mathrm{d}z}, \tag{A54}$$

and

(c) for an atmosphere without ion drag, we set

$$\widehat{f}_\mathrm{IDx} = 0, \ \widehat{f}_\mathrm{IDz} = 0, \ \text{and} \ \widehat{P}_\mathrm{ID} = 0. \tag{A55}$$

## A2. Simplified models

Consider an isothermal, homogeneous, and windless atmosphere, that is,

$$\frac{\mathrm{d}T_0}{\mathrm{d}z} = 0, \ u_0 = 0, \ \mu_\mathrm{k} = \text{constant}, \ \text{and} \ \lambda_\mathrm{k} = \text{constant}. \tag{A56}$$

Assuming $\mathbf{f}'_\mathrm{ID} \approx \rho_0 \nu_{0ni}[\mathbf{u}' - (\mathbf{u}' \cdot \widehat{\mathbf{b}})\widehat{\mathbf{b}}]$ and $P'_\mathrm{ID} \approx 0$ (compare with Eqs. (16) and (17)), we are led to the following linear
system of ordinary differential equations ($\omega_0 = \omega$):

$$\frac{1}{k_\mathrm{x}}\frac{\mathrm{d}\widehat{u}}{\mathrm{d}z} = \frac{1}{3\eta}\widehat{\mathcal{U}}, \tag{A57}$$

$$\frac{1}{k_\mathrm{x}}\frac{\mathrm{d}\widehat{w}}{\mathrm{d}z} = \mathrm{j}\widehat{u} + \alpha\widehat{w} - \mathrm{j}\widehat{p} + \mathrm{j}\widehat{T}, \tag{A58}$$

$$(1+4\eta)\frac{1}{k_\mathrm{x}}\frac{\mathrm{d}\widehat{p}}{\mathrm{d}z} = -(2\eta\alpha - \beta\sigma\cos I \sin I)\widehat{u} + \mathrm{j}[3\eta - \beta + \mathrm{j}\sigma\beta\cos^2 I]\widehat{w}$$

$$+ \alpha\widehat{T} + \widehat{\mathcal{U}} + 4\frac{\eta}{\nu}\widehat{\mathcal{T}}, \tag{A59}$$

$$\frac{1}{k_\mathrm{x}}\frac{\mathrm{d}\widehat{T}}{\mathrm{d}z} = \frac{1}{\nu}\widehat{\mathcal{T}}, \tag{A60}$$

$$\frac{1}{k_\mathrm{x}}\frac{\mathrm{d}\widehat{\mathcal{U}}}{\mathrm{d}z} = [3\eta - \beta + \mathrm{j}\beta\sigma\sin^2 I]\widehat{u} - \mathrm{j}(2\eta\alpha + \beta\sigma\cos I \sin I)\widehat{w}$$

$$+ (\eta+1)\widehat{p} - \eta\widehat{T} + \alpha\widehat{\mathcal{U}}, \tag{A61}$$

$$\frac{1}{k_\mathrm{x}}\frac{\mathrm{d}\widehat{\mathcal{T}}}{\mathrm{d}z} = \mathrm{j}\alpha\widehat{w} + \widehat{p} + \left(\nu - \frac{\gamma}{\gamma-1}\right)\widehat{T} + \alpha\widehat{\mathcal{T}}, \tag{A62}$$





where the dimensionless parameters $\eta$ and $\nu$ are given by Eq. (A27), while the parameters $\alpha$, $\beta$, and $\sigma$ are given respectively, by

$$\alpha = \frac{1}{k_{\mathrm{x}} H_{\mathrm{a}}}, \tag{A63}$$

$$\beta = \frac{\omega^2}{k_{\mathrm{x}}^2 g H_{\mathrm{a}}}, \tag{A64}$$

$$\sigma = \frac{\nu_{0ni}}{\omega}. \tag{A65}$$

For $\mathbf{e} = [\widehat{u}, \widehat{w}, \widehat{p}, \widehat{T}, \widehat{\mathcal{U}}, \widehat{\mathcal{T}}]^T$, the matrix differential equation associated to the system of differential equations (A57)–(A62) is given by Eq. (A48). By further assuming that the geomagnetic field is either in the horizontal or the vertical direction, that is, by neglecting the terms containing the product $\cos I \sin I$ in Eqs. (A59) and (A61), we are led to the model developed by Francis (1973). In this case, the characteristic equation, also known as the dispersion equation, is the cubic equation

$$C_3 R^3 + C_2 R^2 + C_1 R + C_0 = 0, \tag{A66}$$

with the coefficients

$$C_3 = -3\eta\nu(1 + 4\eta), \tag{A67}$$

$$\begin{aligned} C_2 = {} & \frac{3\eta(1 + 4\eta)}{\gamma - 1} + \nu\beta(1 + 7\eta) + 3\eta \\ & - \mathrm{j}\sigma\beta\nu[(1 + 4\eta)\sin^2 I + 3\eta\cos^2 I], \end{aligned} \tag{A68}$$

$$\begin{aligned} C_1 = {} & -[\beta^2 - 2\eta\alpha^2(1 + 3\eta)]\nu - \frac{\beta(1 + 7\eta)}{\gamma - 1} - \beta \\ & + \mathrm{j}\sigma\beta\sin^2 I \left[ \frac{\gamma + 4\eta}{\gamma - 1} + \nu(1 + \eta + \beta) \right] \\ & + \mathrm{j}\sigma\beta\cos^2 I \left[ \frac{3\eta}{\gamma - 1} - \nu(1 + \eta - \beta) \right], \end{aligned} \tag{A69}$$

$$\begin{aligned} C_0 = {} & \frac{\beta^2 - 2\eta\alpha^2(1 + 3\eta)}{\gamma - 1} + \alpha^2(1 + 3\eta) \\ & + \mathrm{j}\frac{\sigma\beta}{\gamma - 1}[\cos^2 I(\gamma + \eta - \beta) - \sin^2 I(\gamma + \eta + \beta)]. \end{aligned} \tag{A70}$$

If $R_m$, $m = 1, \ldots, 3$ are the solutions of the dispersion equation, the corresponding vertical wavenumbers $k_{\mathrm{z}m}^{\pm}$ are computed as

$$k_{\mathrm{z}m}^{\pm} = \mp k_{\mathrm{x}} \sqrt{R_m - 1 - \frac{\alpha^2}{4}}. \tag{A71}$$

The wavenumbers $k_{\mathrm{z}m}^{+}$ with $\mathrm{Im}(k_{\mathrm{z}m}^{+}) < 0$ are associated with ascending modes, and the wavenumbers $k_{\mathrm{z}m}^{-}$ with $\mathrm{Im}(k_{\mathrm{z}m}^{-}) > 0$ with descending modes. If we organize the wavenumbers $k_{\mathrm{z}m}^{+}$ as

$$\mathrm{Im}(k_{\mathrm{z}3}^{+}) < \mathrm{Im}(k_{\mathrm{z}2}^{+}) < \mathrm{Im}(k_{\mathrm{z}1}^{+}) < 0,$$





then (i) $k_{z1}^+$ and $k_{z1}^-$ will correspond to ascending and descending gravity-wave modes, respectively, (ii) $k_{z2}^+$ and $k_{z2}^-$ to ascending and descending viscosity-wave modes, respectively, and (iii) $k_{z3}^+$ and $k_{z3}^-$ to ascending and descending thermal conduction-wave modes, respectively. Note that when the ion drag is completely neglected, the model simplifies to that of Midgley and Liemohn (1966).

Consider an isothermal, homogeneous, and windless atmosphere, and neglect viscosity and ion drag. In this case, the linear system of ordinary differential equations simplify to (Volland, 1969b)

$$\frac{1}{k_\mathrm{x}}\frac{\partial \widehat{w}}{\partial z} = \alpha\widehat{w} + \mathrm{j}\left(\frac{1}{\beta}-1\right)\widehat{p} + \mathrm{j}\widehat{T}, \tag{A72}$$

$$\frac{1}{k_\mathrm{x}}\frac{\partial \widehat{p}}{\partial z} = -\mathrm{j}\beta\widehat{w} + \alpha\widehat{T}, \tag{A73}$$

$$\frac{1}{k_\mathrm{x}}\frac{\partial \widehat{T}}{\partial z} = \frac{1}{\nu}\widehat{\mathcal{T}}, \tag{A74}$$

$$\frac{1}{k_\mathrm{x}}\frac{\partial \widehat{\mathcal{T}}}{\partial z} = \mathrm{j}\alpha\widehat{w} + \widehat{p} + \left(\nu - \frac{\gamma}{\gamma-1}\right)\widehat{T} + \alpha\widehat{\mathcal{T}}, \tag{A75}$$

and the additional equation

$$\widehat{u} = \frac{1}{\beta}\widehat{p}. \tag{A76}$$

Thus, the matrix differential equation (A48) is solved for $\mathbf{e} = [\widehat{w}, \widehat{p}, \widehat{T}, \widehat{\mathcal{T}}]^T$, and the dispersion equation is the quadratic equation

$$C_2 R^2 + C_1 R + C_0 = 0, \tag{A77}$$

with

$$C_2 = \nu\beta, \tag{A78}$$

$$C_1 = -\beta^2\nu - \beta\frac{\gamma}{\gamma-1}, \tag{A79}$$

$$C_0 = \frac{\beta^2}{\gamma-1} + \alpha^2. \tag{A80}$$

For $\mathrm{Im}(k_{z2}^+) < \mathrm{Im}(k_{z1}^+) < 0$, the permissible modes are (i) the ascending and descending gravity-wave modes associated to the pair $(k_{z1}^+, k_{z1}^-)$, and (ii) the ascending and descending thermal conduction-wave modes associated to the pair $(k_{z2}^+, k_{z2}^-)$.

    We conclude this appendix by presenting Hines' criticism of standard multilayer methods (Hines, 1973). Although this criticism does not pertain to our adopted method (a nonstandard multilayer method), we find it valuable to discuss. This is especially pertinent since the solution methods we proposed can be applied to any linear system of ordinary differential

equations with constant coefficients. Hines mentioned that in a standard multilayer method, the choice of the unknowns (the components of the vector $\mathbf{e}$) should depend on the physical meaning of the continuity condition at the layer interfaces. The choice (A49) is the same as the one used by Volland (1969b) and implies the continuity of the pressure $\widehat{p}$. Midgley and Liemohn





(1966) used instead the continuity of $\widehat{p} + \mathrm{j}\widehat{w}/(k_{\mathrm{x}}H_{\mathrm{a}})$, which reflects the continuity of the pressure perturbation $p' + \mathrm{j}wp_0/(\omega H_{\mathrm{a}})$ evaluated at the interface between two layers; this interface being displaced by a distance $\mathrm{j}w/\omega$ from its ambient position because of the perturbing effect of the wave. To guarantee this interfacial condition, the components of the vector $\mathbf{e}$ are chosen as

$$e_1 = \widehat{u}, \, e_2 = \widehat{w}, \, e_3 = \widehat{p} + \mathrm{j}\alpha\widehat{w}, \, e_4 = \widehat{T}, \, e_5 = \widehat{\mathcal{U}}, \, e_6 = \widehat{\mathcal{T}}. \tag{A81}$$

Inserting Eq. (A49) into Eqs. (A57)–(A62), we are led to the system of equations

$$\frac{1}{k_{\mathrm{x}}}\frac{\mathrm{d}e_1}{\mathrm{d}z} = \frac{1}{3\eta}e_5, \tag{A82}$$

$$\frac{1}{k_{\mathrm{x}}}\frac{\mathrm{d}e_2}{\mathrm{d}z} = \mathrm{j}(e_1 - e_3 + e_6), \tag{A83}$$

$$\frac{1}{k_{\mathrm{x}}}\frac{\mathrm{d}e_3}{\mathrm{d}z} = -[\eta_1(2\eta\alpha - \beta\sigma b_x b_z) + \alpha]e_1 + \mathrm{j}\eta_1[3\eta - \beta + \mathrm{j}\sigma\beta(1 - b_z^2)]e_2$$
$$+ \alpha e_3 + \alpha\eta_1 e_4 + \eta_1 e_5 + \left(4\frac{\eta\eta_1}{\nu} - \alpha\right)e_6, \tag{A84}$$

$$\frac{1}{k_{\mathrm{x}}}\frac{\mathrm{d}e_4}{\mathrm{d}z} = \frac{1}{\nu}e_6, \tag{A85}$$

$$\frac{1}{k_{\mathrm{x}}}\frac{\mathrm{d}e_5}{\mathrm{d}z} = [3\eta - \beta + \mathrm{j}\beta\sigma(1 - b_x^2)]e_1 - \mathrm{j}(3\eta\alpha + \beta\sigma b_x b_z + \alpha)e_2$$
$$+ (\eta + 1)e_3 - \eta e_4 + \alpha e_5, \tag{A86}$$

$$\frac{1}{k_{\mathrm{x}}}\frac{\mathrm{d}e_6}{\mathrm{d}z} = e_3 + \left(\nu - \frac{\gamma}{\gamma - 1}\right)e_4 + \alpha e_6, \tag{A87}$$

where $\eta_1 = 1/(1 + 4\eta)$. On the other hand, Hines (1973) and Francis (1973) showed that the continuity of

$$\widehat{p} + \frac{\mathrm{j}}{k_{\mathrm{x}}H_{\mathrm{a}}}\widehat{w} + 2\eta\widehat{u} + \mathrm{j}\frac{4\eta}{k_{\mathrm{x}}}\frac{\mathrm{d}\widehat{w}}{\mathrm{d}z},$$

or equivalently, of the pressure perturbation

$$p' + \mathrm{j}\frac{wp_0}{\omega H_{\mathrm{a}}} - \frac{2\mu}{3}\left(2\frac{\partial w}{\partial z} - \mathrm{j}k_{\mathrm{x}}u\right),$$

is more realistic. In this case, the components of the vector $\mathbf{e}$ are chosen as

$$e_1 = \widehat{u}, \, e_2 = \widehat{w}, \, e_4 = \widehat{T}, \, e_6 = \widehat{\mathcal{T}}, \tag{A88}$$

$$e_3 = \widehat{p} + \mathrm{j}\alpha\widehat{w} + 2\eta\widehat{u} + \mathrm{j}\frac{4\eta}{k_{\mathrm{x}}}\frac{\mathrm{d}\widehat{w}}{\mathrm{d}z}, \tag{A89}$$

$$e_5 = \widehat{\mathcal{U}} + \mathrm{j}4\eta\widehat{w}, \tag{A90}$$



and the underlying system of equations is

$$\frac{1}{k_{\mathrm{x}}}\frac{\mathrm{d}e_1}{\mathrm{d}z} = -\mathrm{j}\frac{4}{3}e_2 + \frac{1}{3\eta}e_5, \tag{A91}$$

$$\frac{1}{k_{\mathrm{x}}}\frac{\mathrm{d}e_2}{\mathrm{d}z} = \mathrm{j}\eta_1[(1+2\eta)e_1 - e_3 + e_4], \tag{A92}$$

$$(1-4\eta\eta_1)\frac{1}{k_{\mathrm{x}}}\frac{\mathrm{d}e_3}{\mathrm{d}z} = -\eta_1[(4\eta+1)\alpha - \beta\sigma b_x b_z]e_1$$

$$+\mathrm{j}\left\{\frac{16}{3}\eta\eta_1(1+2\eta) - \eta\left(\eta_1 - \frac{8}{3}\right) - \eta_1[\beta - \mathrm{j}\sigma\beta(1-b_z^2)]\right\}e_2$$

$$+\alpha\eta_1 e_3 + \left[\eta_1 - \frac{4}{3}\eta_1(1+2\eta) + \frac{2}{3}\right]e_5, \tag{A93}$$

$$\frac{1}{k_{\mathrm{x}}}\frac{\mathrm{d}e_4}{\mathrm{d}z} = \frac{1}{\nu}e_6, \tag{A94}$$

$$\frac{1}{k_{\mathrm{x}}}\frac{\mathrm{d}e_5}{\mathrm{d}z} = \{-\beta + \mathrm{j}\beta\sigma(1-b_x^2) + \eta[4\eta\eta_1(1+2\eta) - 2\eta + 1]\}e_1$$

$$-\mathrm{j}[\alpha(7\eta+1) + \beta\sigma b_x b_z]e_2 + [1 - \eta(4\eta\eta_1 - 1)]e_3$$

$$+\eta(4\eta\eta_1 - 1)e_4 + \alpha e_5, \tag{A95}$$

$$\frac{1}{k_{\mathrm{x}}}\frac{\mathrm{d}e_6}{\mathrm{d}z} = 2\eta[2\eta_1(1+2\eta) - 1]e_1 + (1 - 4\eta\eta_1)e_3$$

$$+\left(4\eta\eta_1 + \nu - \frac{\gamma}{\gamma-1}\right)e_4 + \alpha e_6. \tag{A96}$$

Both system of equations (A82)–(A87) and (A91)–(A96) can be solved using the methods proposed in Section 4.

*Author contributions.* AlD: writing (original draft preparation), and conceptualization. DSE: writing (original draft preparation), and validation. TT: writing (original draft preparation), and formal analysis. AdD: writing (original draft preparation), and methodology

*Competing interests.* The authors declare that they have no conflict of interest.



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
