# Peer review of "A numerical model for solving the linearized gravity-wave equations by a multilayer method"

_EGUsphere, 2025_

## Referee Comment (RC1)

**Review of Doicu et al., "A numerical model for solving the linearized gravity-wave equations by a multilayer method"**
Harold Knight
Computational Physics, Inc.
September 5, 2025

**Overview**

This paper describes a new modeling technique for atmospheric gravity waves (GWs) and examines the effect of ion drag on GW propagation and dissipation in the thermosphere. The authors find that a banded-matrix technique can be applied to thermospheric GW modeling problems with the same effectiveness as scattering-matrix techniques but with greater computational efficiency. While the aforementioned results are of scientific interest, some of the material in the paper is unnecessary and should be deleted, especially given the length of the paper. Also, further elaboration is needed in some instances identified below.

I am happy to see that the authors include a discussion of results from Knight et al. (2019,2021,2022), which are relevant to their work, but their discussion of causality reflects some misunderstandings and should be substantially revised, as indicated below.

Here are some general issues affecting clarity that occur through the paper:
1. Most citations just give the article reference without a specific section and/or equation number.
2. The same symbols are used in multiple contexts in some instances identified below.
3. The figure labels are not sufficiently descriptive, making it difficult for the reader to understand the figures. In almost all figures (e.g., Figure 5), lines and symbols are labeled a, b, c, etc., and the reader has to look at the caption to know what is meant. This makes it unnecessarily difficult to interpret the figures. Descriptive information should be given in the line/symbol labels instead of a, b, c, etc.
4. Often, equations involve terms that are not introduced until many lines after the equation. It is better to introduce terms before they appear in equations.

**Specific Comments**

Lines 1-11. The abstract should be revised to reflect changes suggested below. The main results of scientific interest are the effectiveness and efficiency of a banded-matrix technique and the examination of the effects of ion drag, so these should be the focus of the abstract.

Line 14. This sentence should mention that it is talking about upper-atmospheric gravity waves.

Lines 15-16. Please add Knight et al. (2024) to this list.

Lines 26-34. Knight et al. (2022, Section 1) introduced the term "numerical multilayer method" to replace the term "nonstandard" suggested by Hines (1973). The term "physical multilayer" can be used in place of "standard". The terms "nonstandard" and "standard" are anachronistic and nondescriptive, considering that the supposed "standard" approach is not in use anymore. I

encourage the authors to use the terms "numerical multilayer" and "physical multilayer" in place of "nonstandard" and "standard", respectively.

Lines 47-52. It should be mentioned here that the assumption of locally constant kinematic molecular viscosity is unrealistic, as discussed in Knight et al. (2019, Section 1). Also, Knight et al. (2024, Section 6.5) describes the effect of relaxing the assumption that $\mu_z = 0$, where $\mu$ is dynamic molecular viscosity. The effects are small, generally. Knight et al. (2021,2024) allow $\mu$ to vary according to the standard Dalgarno and Smith (1962) formula, but terms involving $\mu_z$ are omitted from the vertical structure equations associated with the main results.

Lines 57-65. This discussion of Midgley and Liemohn (1966) seems too detailed. I have not seen this problem involving coupling and critical layers in any other context. Perhaps it is specific to their assumption of locally constant kinematic viscosity.

Line 76. Change direction to directions.

Line 110. Define $\bar{\sigma}$ or give a specific reference for it. Also, what does the prime symbol mean in $[\nabla \cdot \bar{\sigma}]'$?

Lines 113-123. It is best to leave out the equations of motion for ions, since this is not being modeled here.

Lines 145-182. This discussion should be simplified, given that the ion continuity equation is not modeled in this paper and ionospheric effects are left for future work. I suggest using words rather than equations. Note that Knight et al. (2025) modeled the effects of atmospheric gravity waves on the ionosphere by driving an ionospheric model with the numerical multilayer method of Knight et al. (2024). It was seen in Knight et al. (2025, Section 4.3) that the downward flow of ions from the plasmasphere is needed in order to properly describe plasma fluctuations resulting from gravity waves.

Line 192. The opposite sign convention for frequency and horizontal wavenumbers is typical for gravity-wave studies. See, e.g., Knight et al. (2025, eq. 1). This should be mentioned here to prevent confusion.

Line 196. Give the value for $Pr$. Also, the numerical formula for thermal conductivity, $\lambda = 6.71 \times 10^{-4} T^{0.71}$, does not look right. Is constant $c_p$ assumed? If so, what value? The specific heat at constant pressure varies with composition.

Line 220. Something to consider for future work is that there are disadvantages in converting to nondimensional quantities. It makes it impossible to check equations in terms of units, and it leads to more complicated formulas as in the Appendix.

Line 226. Say Appendix or Appendix A instead of just A.

Line 227. The notation $\lambda_n$ for eigenvalues conflicts with the notation $\lambda$ for thermal conductivity. This should be fixed.

Line 249. Say "Appendix A.2" instead of just "A". Say that it is to be expected that the vertical wavenumbers will separate into pairs when kinematic viscosity is assumed to be constant given earlier work, e.g., Volland (1969b).

Lines 260. Is it really possible to distinguish between viscosity-wave modes and thermal-conduction-wave modes? What is the theoretical basis for this distinction? Presumably, such a distinction would change based on the Prandtl number.

Line 264. "we can renounce on the vertical wavenumber" does not sound right in English. Perhaps instead say "we can put aside the concept of vertical wavenumber."

Lines 267-270. You say "providing a more intuitive explanation compared with the analogy with an isothermal and homogeneous atmosphere." This should be reworded. The classification of upgoing and downgoing roots done earlier by Volland, etc. had no theoretical justification other than the wish for dissipation to increase rather than decrease magnitudes. The Knight et al. approach, which will be discussed in Section 5, requires other conditions beyond (45) having to do with causality, and was motivated by theoretical concerns rather than intuition.

Line 299. It would make sense to say "a numerical multilayer method (Knight et al., 2022)" here instead of "nonstandard." Note that the term "nonstandard" does not appear in Klostermeyer (1972).

Lines 313-321. Why is it necessary to apply (61) and (62)? It appears that the method would work setting $S^1$ to the identity matrix and $S^0$ to the diagonal matrix of exponential terms or by setting $S^1$ to the diagonal matrix of inverse exponential terms and $S^0$ to the identity matrix. What specific problem does applying the condition $\text{Re}(\lambda_{nl}) > 0$, etc., prevent? Line 313 says "To obtain a stable system of equations." Does the banded-matrix method not work without dividing up the diagonal terms this way? This step of dividing up the diagonal terms is not needed with the scattering-matrix approach. It is not done in the Knight et al. references.

Lines 348-353. These lines do not need to be included. It says "For the continuity equations and the boundary conditions to be consistent …" There does not appear to be any need for such consistency. If there is no numerical or mathematical reason for (72) and (73), then there is no need to include them.

Line 354. I suggest beginning by saying something like "Here define an alternative type of lower boundary condition …"

Line 356. For clarity, it should be stated here that eq. (74) is a generalization of Knight et al. (2022, eq. 2.19), which refers to the first state variable rather than an arbitrary state variable.

Line 356. The following is for the authors' information. The motivation for this form of the lower boundary condition comes from Knight et al. (2019, eq. 3.5), which is used in the statement of Knight et al. (2019, Theorem 1). For causality results, it is necessary to formulate boundary conditions in terms of state variables rather than modes, since modes are in the frequency domain.

Note also that Knight et al. (2022, eq. 2.19) should be used with caution in cases where conditions are nearly inviscid at the lower boundary. See Knight et al. (2024, eq. 5.5).

Line 364. The symbol $A$ is overused in this paper. See lines 267 and 300, for instance. Also, a similar symbol is used for the continuity equation and the global matrix. A different symbol should be used here.

Line 377. Move the explanation at lines 383-386 to before eq. (83). Otherwise, the term $a_s$ is confusing.

Line 381. "The matrix A has 3M – 1 sub- and super-diagonals." Give the numbers of sub- and super- diagonals separately. Are these distinct from the diagonal? 3M – 1 does not seem right. It looks like there should be 4M diagonal bands total.

Line 382. Give more specifics on the method used to solve the banded-matrix equation, including the software package and/or a book or article reference. Do you actually invert the banded matrix or do you solve a linear equation of the form $A \vec{v} = \vec{b}$. This will make a difference in numerical efficiency. Your global matrix is extremely ill-conditioned due to the very large and very small dissipative wavenumbers towards the inviscid end of the altitude range. One benefit of the scattering-matrix approach is that it avoids this problem, at least as formulated in Knight et al. (2019). It would be interesting to see some discussion of how your banded-matrix method avoids the problem of ill conditioned matrices, especially with a lower boundary at or below 50 km altitude, where kinematic viscosity is very small.

Lines 393-395. It is not clear why the authors feel the need to state this condition. Generally, in discussion of linear methods, it goes without saying that the approximation is valid in the asymptotic limit of small perturbations. It does not appear that the authors have an actual estimate of the range of validity of their linear equations, so why bother stating eq. (88)? Knight et al. (2024, eq. 6.2) give a condition for wave breaking, but weakly nonlinear effects can occur at smaller amplitudes. Unless the authors can give a good reason for including eq. (88) and the related discussion, it should be deleted.

Lines 399-400. This statement is problematic because $s$ is in the frequency domain up to here, while in Section 5 it becomes a time-dependent function. I suggest rewording so that the term $s$ is not explicitly mentioned.

Line 403. Do 1 and 2 correspond to + and -? State this earlier.

Lines 406-408. There is a notational conflict with eqs. (61-62), since this reuses the symbol $S$.

Line 410. Where does the term "interaction principle" come from?

Line 414. Give a reference for this formulation of the scattering matrix.

Lines 424-427. Give a reference for these equations. They are similar to Knight et al. (2019, eqs. 4.30-33), for instance.

Line 102. Give a reference for this equation. It is similar to Knight et al. (2019, eq. 4.34), for instance.

Line 441. I find Section 4.2 problematic and think it should be deleted. In Section 6, it is seen that there is no advantage in using the formulations in Section 4.2, so why add all of this unnecessary detail? If you think it is important, then summarize this alternative in a few sentences, including the finding in Section 6 that it offers no advantage. That aside, the use of the word "discrete" in the title of this section is unclear. In what sense is it discrete? How is it any more discrete than the approach of Section 4.1? Discrete ordinates are mentioned in the introduction, but that does not appear to be related.

Line 554. Shouldn't $b_{1,k}$ depend on $\omega$?

Lines 563-565. This sentence about the separable case appears to be unnecessary.

Line 572. It should be mentioned that in Knight et al. (2024, eq. 2.29) a relaxed condition is given, in which the bounds are allowed to vary with attitude.

Lines 577-594. Since imaginary frequency shifting is not used in this work, much of this summary should be deleted. Regardless, the explanation given here is incomplete, in that there is no indication of how $\delta$ was selected. Methods for determining the minimum sufficient $\delta$ are described in Knight et al. (2019,2021,2022). Instead of giving this summary, the paper should say that the imaginary frequency shifting technique was not applied and that further study is needed to determine the effect. Also, below I will suggest a new figure for Section 6 that will give a good indication of whether problematic branch points are present. If there are problematic branch points, they will primarily affect nearby frequencies.

Lines 590-594. The problem of the numerical blowup associated with the exponential growth term is discussed at length in Knight et al. (2021, App. B) and should be cited here. Numerical blowup is especially a problem for large time domains. If, in future work, you are unable to obtain results without the blowup, then I suggest reducing the size of the time domain and considering narrower time wave packets. Care is needed in selecting $\delta$. It should be large enough to prevent the crossing seen in Knight et al. (2019, Fig 2b), but not much greater than that. Rigorous methods are discussed in Knight et al. (2019,2021,2022), but it would suffice to look for the curve-crossing issue.

I recommend replacing the discussion from line 577 to 594 with brief references to Knight et al. (2019, Section 3.4) and Knight et al. (2021, Section 2.4). You can mention that you encountered the numerical blowup problem mentioned in Knight et al. (2021, App. B) and that further study is needed to resolve this issue.

It is good that you introduce imaginary frequency shifting in lines 570-675, however, since that allows you to explore the effect of $\delta$ on the root-crossing issue illustrated in Knight et al. (2019, Fig. 2b).

Lines 594-607. This alternative approach should not be included in the paper. In practice, it is impossible to verify (157) rigorously without Titchmarsh's theorem (Knight et al., 2019, Section 2). A concise statement of the causality condition is given in the short paragraph following the proof of Lemma 1 in Knight et al. (2019, Section 2). Using the notation given there, the condition $\beta(t) = 0$ for $t < 0$ can be required for the lower boundary condition $\beta$, but Titchmarsh's theorem is needed to establish it for $w$ (Knight et al., 2019, eq. 2.4).

Line 608. It does not appear that the horizontal wavenumber or wavelength is ever given in Section 6. It is important to specify this.

Lines 613-616. Methods 3, 4, and 5 should not be included here, given that they offer no advantages in accuracy or efficiency. The derivations in Section 4.2 do not seem scientifically interesting, given that they mostly rearrange terms from Section 4.1. (The Pade approximation would be of interest if it actually provided advantages, but it does not, so there is no apparent need to mention it except perhaps very briefly.) Method 2 is of interest because a related method is currently in use for atmospheric gravity waves (referring to the Knight et al. work).

Line 620. This does not look like a complete list of background parameters. It might be complete with $p_0$, $c_v$, and $Pr$ added.

Lines 520-525. What are the input parameters for SAMI2 and HWM?

Line 627. $p_0$, $c_v$, and $\rho_0$ should also be shown in a figure. The density scale height, $H = -\rho_0/(\partial\rho_0/\partial z)$, should also be shown.

Lines 649-652. I suggest omitting the Fourier transform in the altitude dimension. There are several reasons for this:
1. It is confusing, since it means that there are two types of vertical wavenumbers, one obtained from the vertical structure equations and one obtained directly from the Fourier transform.
2. It creates notational ambiguity, since the same notation is used for both types of vertical wavenumbers.
3. Taking the Fourier transform in the vertical dimension does not make sense given that the vertical wavenumbers coming from the vertical structure equations include both real and imaginary parts. See Knight et al. (2025, Section 4.2, first paragraph). Vertical wavelengths are not defined, strictly speaking, when significant dissipation is occurring.
4. The Fourier transform makes the most sense with periodic or unbounded domains, neither of which applies to the altitude dimension.
Aside from that, the values given here are difficult to interpret. If the vertical Fourier transform is left in the paper (which I recommend against), then the actual value for $\Delta k_z$ should be given, and $N_k \Delta k_z$ should approximately equal 450 km.

Line 655. Give a reference for the nonuniform Fourier transform.

Line 658. It would be more helpful to state that $\kappa_\omega = 0.8$ when such results are discussed, both in the text and in the figures and/or figure captions.

Lines 665-672. There is no need to include this discussion, and it should be deleted, along with Figure 2. Figure 2 merely confirms that the derivations in Appendix A.2 are correct, and it should go without saying that they are correct.

Lines 673-689. Again, methods 3, 4, and 5 should be omitted, making Figure 3 unnecessary.

Continuing with lines 673-689, Figure 4 probably becomes unnecessary if it is just a comparison of the first two methods. Given $M = 3$, I would expect about factor of three ratio of processing times between methods 2 and 1, assuming that the banded-matrix method solves the linear equation $A \vec{v} = \vec{b}$ directly rather than inverting $A$. This is because the scattering-matrix approach effectively solves for a general three-dimensional lower-boundary condition, meaning that it does more computations than would be needed for a specific lower-boundary condition, in principle. The ratio in Figure 2 is more like a factor of five. This makes me wonder whether your scattering-matrix computations are done as efficiently as they could be. Rigorous analysis of the computational steps involved with methods 1 and 2 would be needed to clarify this. I am not suggesting that such analysis be included in the current paper, but I would like for your paper to mention that more rigorous analysis is needed to make the result definite.

Lines 690-701. It is not clear what is gained by merely comparing results for different values of $\kappa_\omega$. This is because there is no way of knowing to what extent differences in neutral-atmospheric dynamics are contributing to the differences. To clarify this, I recommend combining the results in the upper panels of Figure 5 with the results shown in Figure 8 in the same figure (maybe a different figure for each state variable) and discussing these results together. I would give results without ion damping for each of the three $\kappa_\omega$ values so they can be compared in each case.

Why are the apparent vertical wavelengths in the upper panels of Figure 5 so similar for the three $\kappa_\omega$? As mentioned above, I could not see where you specified the horizontal wavelength. The vertical wavelength coming from the vertical structure equations should change with $\kappa_\omega$, assuming that the horizontal wavelength is kept fixed. These issues need to be clarified in Section 6.

As indicated above, I recommend omitting the type of analysis shown in the lower panels of Figure 5 and in Figure 6. You can replace it with a comparison of results for the three $\kappa_\omega$ values, with and without ion damping, as described above. If you want to talk about vertical wavenumbers, I recommend looking at vertical profiles of the upgoing gravity-wave roots and interpreting differences in model results in terms of those. It would also be good to include discussion of previous analysis of the effects of ion damping on gravity waves and relate it to your present work.

Lines 702-722. Pairwise classification of vertical wavenumbers is less important than being able to divide the roots into separated upgoing and downgoing sets. Figure 7 should include more descriptive titles and labels giving the meaning of the panels. The figure caption is difficult to interpret because it merely refers to equation numbers without reminding the reader of the meaning.

While Figure 7 illustrates the differences between two governing-equation assumptions (i.e., locally varying and constant kinematic viscosity) in their effect on vertical wavenumbers, which is of some value, it does not say much about whether the roots can be separated into upgoing and

downgoing sets. To do this, one would need to look at how the roots vary with frequency. This applies even for fixed-frequency cases. I recommend giving a figure like Knight et al. (2019, Fig. 2b) for several different altitudes, e.g., 150, 250, 350, and 450 km. If any of the roots cross like in Knight et al. (2019, Fig. 2b), it means that there is a problematic branch point nearby.

Generally, there is no problem for single-frequency results provided that the frequency is far from the problematic branch point. Even though the global method does not explicitly require upgoing and downgoing modes to be defined at intermediate altitudes, the solution still may not be valid without appropriate imaginary frequency shifting for problematic branch points occurring over the entire altitude range. I hope to write a paper clarifying these issues in the future.

Also, Figure 7 shows altitude in the x-axis, but it is standard to put altitude in the y-axis.

Line 718. "The ion-drag is important for time frequencies …" Give a specific reference for this.

Lines 718-722. As mentioned above, this discussion should be combined with the discussion in lines 690-701. Also, the results discussed here are puzzling. It says there is complete agreement between results with and without ion drag for $\kappa_\omega = 1.2$. This does not seem possible. Surely, ion drag would have some effect. The authors should double-check this and provide further explanation if there really is no effect. In particular, they should look at the vertical wavenumbers (obtained from the vertical structure equations) and see whether there is any difference.

Aside from this, the caption of Figure 8 is puzzling. Case (a) is with ion drag excluded. What is $\kappa_\omega$ for (a)? If $\kappa_\omega = 0.8$ for (a), then the similarity between results for (a) and (b) makes even less sense, given that $\kappa_\omega = 1.2$ for (b). The wording here and in the text should be made clearer, and errors, if any, should be corrected.

Lines 723-726. These lines should be deleted. Figure 9 gives a comparison of nearly identical results, and if the results are identical there is most likely a trivial reason for it, so the discussion, along with Figure 9, does not need to be included.

Line 729. Say Knight (2019, Section 6.1).

Lines 729-732. I do not see the scientific interest of this. One would expect the results to be similar.

Lines 733-738. This is similar to some previous work, which should be cited. Knight et al. (2019) defines the "transmission-only" approximation, which is similar to your eq. (118), and Knight et al. (2021) discusses a single-mode approximation, which is related to the transmission-only approximation. Additionally, Knight et al. (2019, Section 6) shows the upgoing and downgoing contributions to a wavefield. Although (118) is introduced in Section 4.2, which I recommend deleting, it should be possible to give very similar definition in Section 4.1.2.

Figure 14 is unnecessary, since the information is already conveyed by Figure 13.

Lines 747-748. This reflects a naïve view of causality. Causality is really about whether upgoing and downgoing modes are defined and valid. For frequencies near problematic branch points, a

single-frequency solution will be incorrect, regardless of whether the peak in amplitude seems to be moving with altitude.

Lines 749-756. This discussion is problematic. Firstly, Figure 15 is the wrong type of plot for analyzing issues with causality, i.e., whether upgoing and downgoing roots are valid. What is needed is a figure like I described for lines 702-722 above, showing the imaginary parts vs. frequency. There is no indication of how the $\delta$ value was selected. Note how in Fig 2b of Knight et al. (2019), two roots cross, while in Fig. 2d they do not cross. This indicates that the $\delta$ value used in Fig. 2d was sufficient. If $\delta$ is not large enough to prevent the roots from crossing, then it will not work. The bottom three panels of Figure 15 should be omitted. To really assess the effect of problematic branch points, you need a solution that is known to be correct. Just observing that the solution is small before $t = 0$ is not sufficient.

Regarding Figure 15, are the eigenvalues specific to $\omega_0$? This should be stated.

Line 747. The extreme difference in computation time between methods 1 and 2 is very puzzling given that only a factor of five difference was seen for the single-frequency case. What possible reason could there be for this? It seems like this must be a mistake.

Line 751. Should (14) be (149)?

Line 751. Units should be given for $\delta$.

Lines 768-769. "The amplitude of the source function can be computed …" This is unclear. Why would one want to compute the amplitude of the source function? Generally, one starts with the source and computes the wavefield from that.

Lines 773-774. As discussed above for line 747, there is no apparent reason why there should be a difference in relative efficiency between single-frequency and time-varying cases.

Appendix A. Converting to non-dimensional form makes the equations more complicated than they would be otherwise, and it also makes it impossible to check equations via units.

Line 793. "A1-A4" is unclear. Does this mean eqs. (A1-4)? Maybe say "eqs. A1-A4 below".

Lines 916-917. This statement is redundant with discussion in the main text.

Line 927. Say whether this is density or pressure scale height.

Line 940. Say "$\mu_0 = \mu_k = $ constant", etc., here.

Lines 997-1037. These lines would belong in a separate section, but I do not think they should be included in the paper at all. If you have fresh insights into Hines' criticism, I suggest describing them briefly in the main text without any additional equations.

Final comment: It would be advantageous for the authors to show that they can reproduce a previous result. To this end, they could apply their method 1 to the case illustrated by Figure 2a in Knight et al. (2022). It would be interesting to hear whether they get similar results, although it would not be necessary to add a figure for this.

**New References**

Knight, H., Broutman, D., & Eckermann, S. (2024). Compressible and anelastic governing-equation solution methods for thermospheric gravity waves with realistic background parameters. Theoretical and Computational Fluid Dynamics, 38(4), 479–509. https://doi.org/10.1007/s00162-024-00709-x

Knight, H. K., Richards, P. G., Martinis, C. R., & Goncharenko, L. P. (2025). Modeling MSTIDs produced by gravity waves with parameters obtained from all-sky imager observations and comparisons to incoherent scatter radar observations. Journal of Geophysical Research: Space Physics, 130, e2025JA033906. https://doi.org/10.1029/2025JA033906

---

## Author Comment (AC1)

**Response to reviewer comments**

Based on the comments provided by the reviewers, we undertook a major revision and completely reformulated the manuscript. The major changes to the manuscript are listed below.

1. The revised manuscript is organized as follows. In Section 2, we present the derivation of the matrix exponential solution of the linearized equations, while Section 3 describes stable numerical methods for computing the amplitudes of the characteristic solution in a stratified atmosphere. Here, we focus only on the global matrix method based on matrix exponentials and the scattering matrix method for computing the amplitudes of the characteristic solutions. Section 4 addresses the computation of the perturbed quantities for both harmonic and non-harmonic source functions, that is, for single-frequency waves and time-dependent wave packets. The concepts of causality and the imaginary frequency shift are also briefly discussed. Aspects of the numerical implementation are addressed in Section 5, and representative simulation results are presented in Section 6. Additional theoretical issues are discussed in the appendices. Some of the theoretical aspects discussed, especially in Appendices A and B, may perhaps be unnecessary; however, our intention was to provide a self-consistent and complete description of the models.

2. The linearization model is described in Appendix A. We reformulate the linearized equations for the state vector $\mathbf{e} = [\widehat{u}, \widehat{w}, \widehat{T}, \widehat{\mathcal{U}}, \widehat{\mathcal{W}}, \widehat{\mathcal{T}}]^{\mathrm{T}}$, following the formulations of Vadas and Nicolls (2012) and Knight et al. (2024), instead of using $\mathbf{e} = [\widehat{u}, \widehat{w}, \widehat{p}, \widehat{T}, \widehat{\mathcal{U}}, \widehat{\mathcal{T}}]^{T}$, as adopted in earlier studies by Midgley and Liemohn (1966), Volland (1969), Francis (1973), and Yeh and Liu (1974). Appendix A provides a general model that accounts for the altitude derivatives of the background velocity $u_0$, temperature $T_0$, density scale height $H_\rho$, and dynamic viscosity $\mu_0$. In addition, it includes a simplified model for an isothermal ($T_0$ = constant), homogeneous ($\mu_{\mathrm{k}0} = \mu_0/\rho_0$ = constant), and windless atmosphere without ion drag. Appendix A is organized into the following sections: *Hydrodynamic equations, Linearized equations, Plane wave solution*, and *Dispersion equation*.

3. The computation of the ion-drag force and ion-drag heating is presented in Appendix B. In the revised version, ion-drag effects are incorporated in an approximate manner with the explicit aim of decoupling the hydrodynamic and ion equation systems, while explicitly accounting for the plasma diffusion velocity. This is achieved by neglecting perturbed production and loss terms in the ion continuity equation, reducing the ion momentum equation to ambipolar diffusion by neglecting ion inertia and ion–ion collisions and retaining only field-aligned transport, and assuming fast field-aligned diffusion so that ion perturbations and the plasma diffusion velocity remain nearly constant along magnetic field lines. Appendix

B is organized into the following sections: *Ion equations, Linearized equations, Decoupled system of equations,* and *Plane wave solutions.*

4. The application the global matrix method based on matrix exponentials and the scattering matrix method to compute the grid-point values of the state vector is described in Appendix C. This appendix is organized into the following sections: *Global matrix method with matrix exponential* and *Scattering matrix method.*

5. In Appendix D, we discuss several implementation issues related to the computation of lower and upper bounds for the wave period, the choice of frequency and time discretization for the Fourier transform, and the determination of the imaginary frequency shift using a practical, albeit heuristic, approach.

6. We plan to provide a freely available open-source code for solving the linearized gravity-wave equations on GitHub. Accordingly, only representative simulation results are presented in Section 6. The simulation results are new for two reasons: (i) we employ a new linearization model, and (ii) the previous implementation contained an error in the coefficient of thermal conductivity, which was set to $\lambda_0 = 6.71 \times 10^{-7} T_0^{0.71}$ instead of the correct value $\lambda_0 = 6.71 \times 10^{-4} T_0^{0.71}$, thereby substantially reducing the effect of thermal conduction.

For a better understanding of the revised manuscript, we have included the new version in our response (not in the final form required by the journal). Please find below our detailed replies (in black font) to the reviewer comments (in blue font).

**Reviewer 1**

We are sincerely grateful to Harold Knight for his helpful and insightful comments. His detailed explanations greatly clarified several results from his work that we had used in this paper without a full understanding. We also warmly acknowledge his recommendation of his two most recent papers, which were previously unknown to us and have now become an important foundation for the present study.

Here are some general issues affecting clarity that occur through the paper:

1. Most citations just give the article reference without a specific section and/or equation number. Throughout the revised manuscript, we have supplemented citations with explicit section and equation numbers wherever necessary.

2. The same symbols are used in multiple contexts in some instances identified below. We have reviewed the notation and eliminated ambiguous or overlapping symbol usage.

3. The figure labels are not sufficiently descriptive, making it difficult for the reader to understand the figures. In almost all figures (e.g., Figure 5), lines and symbols are labeled a, b, c, etc., and the reader has to look at the caption to know what is meant. This makes it unnecessarily difficult to interpret the figures. Descriptive information should be given in the line/symbol labels instead of a, b, c, etc. All figures have been revised to include descriptive labels directly on the lines and symbols, rather than generic labels such as a, b, c.

4. Often, equations involve terms that are not introduced until many lines after the equation. It is better to introduce terms before they appear in equations. We have reorganized the presentation of equations so that all variables, parameters, and coefficients are introduced and defined before they appear in the equations.

Specific Comments

Lines 1-11. The abstract should be revised to reflect changes suggested below. The main results of scientific interest are the effectiveness and efficiency of a banded-matrix technique and the examination of the effects of ion drag, so these should be the focus of the abstract.

In the abstract we state: *"Particular emphasis is placed on the global matrix method, which exploits the structured form of the multilayer system to achieve high computational efficiency while maintaining numerical accuracy.“ and "The impact of ion drag on wave characteristics is quantified within this framework."*

Line 14. This sentence should mention that it is talking about upper-atmospheric gravity waves.

We agree and have revised the sentence to explicitly refer to upper-atmospheric gravity waves: *"Time-step methods [1,2,3] are commonly used to solve fully non-linear sets of governing equations for upper-atmospheric gravity waves, thereby allowing the modeling of wave breaking, secondary wave generation, and weakly nonlinear effects."*

Lines 15-16. Please add Knight et al. (2024) to this list.

We have added Knight et al. (2024) to this list.

Lines 26-34. Knight et al. (2022, Section 1) introduced the term "numerical multilayer method" to replace the term "nonstandard" suggested by Hines (1973). The term "physical multilayer" can be used in place of "standard". The terms "nonstandard" and "standard" are anachronistic and nondescriptive, considering that the supposed "standard" approach is not in use anymore. I encourage the authors to use the terms "numerical multilayer" and "physical multilayer" in place of "nonstandard" and "standard", respectively.

Following the reviewer's suggestion, we have replaced the terms "nonstandard" and "standard" with "numerical multilayer" and "physical multilayer," respectively, as introduced by Knight et al. (2022, Section 1).

Lines 47-52. It should be mentioned here that the assumption of locally constant kinematic molecular viscosity is unrealistic, as discussed in Knight et al. (2019, Section 1). Also, Knight et al. (2024, Section 6.5) describes the

effect of relaxing the assumption that $\mu z = 0$, where $\mu$ is dynamic molecular viscosity. The effects are small, generally. Knight et al. (2021,2024) allow $\mu$ to vary according to the standard Dalgarno and Smith (1962) formula, but terms involving $\mu z$ are omitted from the vertical structure equations associated with the main results.

We have clarified this point by stating in the main text that *"On the other hand, the assumption of locally constant kinematic viscosity is unrealistic as discussed in Knight et al. [16]."*, and by explaining in Appendix A that *"Essentially, the difference between the model of Vadas and Nicolls [35] and that of Knight et al. [11] is that, in the latter, the derivative of the dynamic viscosity $d\mu_0/dz$ is omitted. In the code, for testing purposes, we included a hard-coded logical flag that selects the linearized model to be used. Our numerical simulations show that there are no significant differences between the general model and that of Vadas and Nicolls [35], and that the effect of the assumption $d\mu_0/dz = 0$ is relatively small. This latter assumption was discussed in detail in Knight et al. [11]."*

Lines 57-65. This discussion of Midgley and Liemohn (1966) seems too detailed. I have not seen this problem involving coupling and critical layers in any other context. Perhaps it is specific to their assumption of locally constant kinematic viscosity.

We agree with the reviewer that the original discussion of Midgley and Liemohn (1966) was overly detailed. Accordingly, we have reduced and streamlined this section. It now reads: *"Midgley and Liemohn [4] employed an iterative method that can be regarded as a Gauss–Seidel group iteration. However, the Gauss–Seidel iteration may fail to converge in certain situations, in particular when gravity and dissipative modes become strongly coupled, as discussed in Refs. [5,6]. Klostermeyer [10] avoids this difficulty by introducing the concept of a transfer matrix, which relates the wave amplitudes at different altitude levels and provides a more robust framework for treating such coupling."*

Line 76. Change direction to directions.

We have changed.

Line 110. Define $\overline{\boldsymbol{\sigma}}$ or give a specific reference for it. Also, what does the prime symbol mean in $(\nabla \cdot \overline{\sigma})'$ ?

The stress tensor $\overline{\boldsymbol{\sigma}}$ is defined in Appendix A. In an earlier version, the prime symbol in $(\nabla \cdot \overline{\sigma})'$ denoted the perturbation of the stress-divergence term. This notation has been removed in the revised manuscript.

Lines 113-123. It is best to leave out the equations of motion for ions, since this is not being modeled here.

The modeling of the ion equations is described in detail in Appendix B.

Lines 145-182. This discussion should be simplified, given that the ion continuity equation is not modeled in this paper and ionospheric effects are left for future work. I suggest using words rather than equations. Note that Knight et al. (2025) modeled the effects of atmospheric gravity waves on the ionosphere by driving an ionospheric model with the numerical multilayer method of Knight et al. (2024). It was seen in Knight et al. (2025, Section 4.3) that the downward flow of ions from the plasmasphere is needed in order to properly describe

plasma fluctuations resulting from gravity waves.

In the revised manuscript, this discussion has been removed from the main text. As noted above, the modeling of the ion equations is described in detail in Appendix B. In addition, the following statement has been added to the conclusions: *"The linearized equations on which the solution methods were tested correspond to ionospheric conditions. The ultimate goal of our research is to develop a comprehensive model for analyzing ionospheric gravity waves using satellite measurements. The approach presented in this paper represents only the first component of such a model. Two options are envisaged for extending it to a more complete formulation.*

1. *Fully coupled neutral–ion model. The linearized hydrodynamic equations would be solved together with the ion equations. In this case, the ion continuity equation would include perturbed production and loss terms, whereas ion inertia and ion–ion collisions would continue to be neglected in the ion momentum equation, and only transport parallel to the magnetic field lines would be retained. The state vector would then be augmented by two additional components, namely the perturbed ion number density and the ion diffusion velocity.*

2. *Two-step coupling strategy. In the first step, the neutral-atmosphere equations are solved using the fast field-aligned diffusion approximation. In the second step, the wave-induced perturbations obtained from the neutral solution are used as input to solve the ionospheric equations for the perturbed $O^+$ ion density. The ionospheric equations may be solved using the SAMI2 model for low latitudes, where the $\mathbf{E} \times \mathbf{B}$ drift is neglected, or the SAMI3 model at higher altitudes, where the $\mathbf{E} \times \mathbf{B}$ drift is included. In this strategy, priority is given to the ionospheric equations of the SAMI framework, while wave-induced perturbations are handled using the approximate approach developed in the present study. Along similar lines, Knight et al. [34] solved the neutral-atmosphere equations without ion drag in a first step, and subsequently addressed the ionospheric response using the Field-Line Interhemispheric Plasma (FLIP) model [44]."*

Line 192. The opposite sign convention for frequency and horizontal wavenumbers is typical for gravity-wave studies. See, e.g., Knight et al. (2025, eq. 1). This should be mentioned here to prevent confusion.

In the main text, we have added the following remark: *"Note that in some gravity-wave studies, the opposite sign convention for frequency and horizontal wavenumber is used (e.g. [34])."*

Line 196. Give the value for $Pr$. Also, the numerical formula for thermal conductivity, $\lambda = 6.71 \times 10{-4}\, T\, 0.71$, does not look right. Is constant $cp$ assumed? If so, what value? The specific heat at constant pressure varies with composition. The value of the Prandtl number is given in the Numerical simulations section ( Pr=0.66).

In Appendix A, we define

$$c_{\mathrm{p}} = c_{\mathrm{v}} + R_{\mathrm{M}}, \ \ \gamma = \frac{c_{\mathrm{p}}}{c_{\mathrm{v}}}, \ \ \Lambda = \frac{c_{\mathrm{p}}\mu}{\mathrm{Pr}},$$

where $c_{\mathrm{p}}$ is the specific heat at constant pressure, $\gamma$ the ratio of specific heats, Pr the Prandtl number, $\Lambda$ the coefficient of thermal conductivity, and $R_{\mathrm{M}}$ the specific gas constant. Moreover, to derive the heat equation, we assume that $c_{\mathrm{p}}$ and Pr are constant, which yields

$$\frac{\partial T}{\partial t} = -\left(\gamma - 1\right) T \nabla \cdot \mathbf{u} - \mathbf{u} \cdot \nabla T + \frac{1}{\rho c_{\mathrm{v}}} \overline{\boldsymbol{\sigma}} : \nabla \mathbf{u} + \frac{\gamma}{\rho \mathrm{Pr}} \nabla \cdot (\mu \nabla T) - \frac{1}{\rho c_{\mathrm{v}}} q_{\mathrm{ID}}.$$

The empirical power-law expression used for $\Lambda$ (as a function of $T$) is consistent with this closure through $\Lambda = c_{\mathrm{p}}\mu/\mathrm{Pr}$, under the assumption of constant $c_{\mathrm{p}}$ and Pr.

Line 226. Say Appendix or Appendix A instead of just A.

"A" has been replaced by "Appendix A."

Line 227. The notation $\lambda n$ for eigenvalues conflicts with the notation $\lambda$ for thermal conductivity. This should be fixed.

This has been corrected by denoting the coefficient of thermal conductivity by $\Lambda$.

Line 249. Say "Appendix A.2" instead of just "A". Say that it is to be expected that the vertical wavenumbers will separate into pairs when kinematic viscosity is assumed to be constant given earlier work, e.g., Volland (1969b).

We reformulate the text as follows: *"As shown in Appendix A, for a constant kinematic viscosity the solutions occur in pairs and correspond to (i) ascending and descending gravity-wave modes, (ii) ascending and descending viscosity-wave modes, and (iii) ascending and descending thermal-conduction wave modes [6,7]. In that appendix, this pairing is explicitly demonstrated by deriving the dispersion relation for the special case of an isothermal (constant background temperature), homogeneous (constant kinematic viscosity), and windless atmosphere without ion drag."*

Lines 260. Is it really possible to distinguish between viscosity-wave modes and thermal- conduction-wave modes? What is the theoretical basis for this distinction? Presumably, such a distinction would change based on the Prandtl number.

We agree with the reviewer that there is no significant distinction between viscosity and thermal-conduction waves. In the section *Simulation results*, we plot the imaginary part of the vertical wavenumber for all wave types and state: *"The plots reveal a clear distinction between gravity waves and viscosity- and thermal-conduction waves. However, the viscosity and thermal-conduction waves are very close to each other."*

Line 264. "we can renounce on the vertical wavenumber" does not sound right in English. Perhaps instead say "we can put aside the concept of vertical wavenumber."

We replaced the phrase with *"we can put aside the concept of the vertical wavenumber."*

Lines 267-270. You say "providing a more intuitive explanation compared with the analogy with an isothermal and homogeneous atmosphere." This should be reworded. The classification of upgoing and downgoing roots done earlier by Volland, etc. had no theoretical justification other than the wish for dissipation to increase rather than decrease magnitudes. The Knight et al. approach, which will be discussed in Section 5, requires other conditions beyond (45) having to do with causality, and was motivated by theoretical concerns rather than intuition.

We reformulate the text as follows: *"A commonly cited interpretation of condition (20) is that, for increasing $z$, the exponential term $exp(k_x\lambda_n z)$ will tend to be damped more for ascending modes than for descending modes; conversely, for decreasing $z$, the roles of ascending and descending modes are reversed. However, such a classification of upgoing and downgoing roots (e.g., Volland [6] and related works) was primarily heuristic and lacked a rigorous theoretical justification. By contrast, the approach of Knight et al. [16], which is discussed in Section 4, introduces additional constraints beyond condition (20) that are explicitly related to causality and is therefore grounded in theoretical considerations rather than heuristic arguments."*

Line 299. It would make sense to say "a numerical multilayer method (Knight et al., 2022)" here instead of "nonstandard." Note that the term "nonstandard" does not appear in Klostermeyer (1972).

We replaced "nonstandard" with "a numerical multilayer method [9,15,16]" as suggested.

Lines 313-321. Why is it necessary to apply (61) and (62)? It appears that the method would work setting $S$ 1 to the identity matrix and $S$ 0 to the diagonal matrix of exponential terms or by setting $S$ 1 to the diagonal matrix of inverse exponential terms and $S$ 0 to the identity matrix. What specific problem does applying the condition $\mathrm{Re}(\lambda_{nl}) > 0$ etc., prevent?

In the revised manuscript the scaling matrices are denoted by $\mathrm{K}_l^1$ and $\mathrm{K}_l^0$, and the layer matrices are written as $\mathbb{A}_{l,l+1}^1 = \mathrm{K}_l^1(\mathrm{V}_l^{-1}\mathrm{V}_{l+1})$ and $\mathbb{A}_{l,l+1}^0 = \mathrm{K}_l^0$. To explain the role of the scaling matrices, we included the following text: *"The scaling matrices $\mathrm{K}_l^1$ and $\mathrm{K}_l^0$ prevent a possible blow-up of the exponential terms for $Re(\lambda_{nl}) > 0$ and $Re(\lambda_{nl}) \leq 0$, respectively. Such scaling techniques are standard in radiative transfer theory and are commonly used to obtain stable numerical algorithms for computing the radiance field in multilayered atmospheres [29,30]."* To verify the robustness of the approach, we performed a series of simulations in which the number of altitude levels was reduced from 800 to 400, 200, 100, and 10, corresponding to an increase in the grid spacing $\Delta_l$ from 0.5 km to 42 km. In all cases, the results remained stable and reliable.

Line 313 says "To obtain a stable system of equations." Does the banded-matrix method not work without dividing up the diagonal terms this way? This step of dividing up the diagonal terms is not needed with the scattering-matrix approach. It is not done in the Knight et al. references.

In general, the computation of the matrix exponential $\exp(\mathsf{A}x)$ becomes numerically problematic when the argument of the exponential term $\exp(\lambda x)$ is large, that is, when $\lambda x \gg 1$. For small values of $\lambda x$, no serious numerical

difficulties arise. In radiative transfer theory, it is well known that the global (banded) matrix method requires such scaling to ensure numerical stability, whereas the scattering-matrix approach does not. The reason is that, in the latter case, the interaction principle equation is inherently numerically stable.

Lines 348-353. These lines do not need to be included. It says "For the continuity equations and the boundary conditions to be consistent . . . " There does not appear to be any need for such consistency. If there is no numerical or mathematical reason for (72) and (73), then there is no need to include them.

These lines have been removed and Eqs. (72) and (73) have been deleted.

Line 354. I suggest beginning by saying something like "Here define an alternative type of lower boundary condition . . .".

We revised the text to begin with the sentence: *"An alternative type of lower boundary condition was proposed by Knight et al. [15,16]."*

Line 356. For clarity, it should be stated here that eq. (74) is a generalization of Knight et al. (2022, eq. 2.19), which refers to the first state variable rather than an arbitrary state variable.

For clarity, we now state: "Note that Eq. (47) generalizes Eq. (2.19) in Ref. [15], which is formulated for the first state variable rather than for an arbitrary state variable."

Line 356. The following is for the authors' information. The motivation for this form of the lower boundary condition comes from Knight et al. (2019, eq. 3.5), which is used in the statement of Knight et al. (2019, Theorem 1). For causality results, it is necessary to formulate boundary conditions in terms of state variables rather than modes, since modes are in the frequency domain.Note also that Knight et al. (2022, eq. 2.19) should be used with caution in cases where conditions are nearly inviscid at the lower boundary. See Knight et al. (2024, eq. 5.5).

We added the following explanation to the manuscript: *"The boundary condition (48) is a localized condition that prescribes the value of a single state variable while enforcing vanishing slope and curvature at the boundary. It effectively acts as an external driver applied to one variable and is appropriate for non-harmonic source functions. Note that this form of the lower boundary condition is used in the statement of Theorem 1 in Ref. 16. For causality considerations, boundary conditions must be expressed in terms of state variables rather than modal amplitudes, since modes are defined in the frequency domain."*

Line 364. The symbol $A$ is overused in this paper. See lines 267 and 300, for instance. Also, a similar symbol is used for the continuity equation and the global matrix. A different symbol should be used here.

We replaced the symbol A with B to avoid overuse.

Line 377. Move the explanation at lines 383-386 to before eq. (83). Otherwise, the term a$s$ is confusing.

In the revised manuscript the index s, which indicates the dependency of a perturbed quantity on the source factor, has been removed in Section 4 (Source function).

Line 381. "The matrix A has $3M - 1$ sub- and super-diagonals." Give the numbers of sub- and super- diagonals separately. Are these distinct from the

diagonal? 3M – 1 does not seem right. It looks like there should be 4M diagonal bands total.

We revised the text to read: *"The matrix $\mathbb{A}$ has $3M - 1$ sub- and superdiagonals (excluding the main diagonal)..."*

Line 382. Give more specifics on the method used to solve the banded-matrix equation, including the software package and/or a book or article reference. Do you actually invert the banded matrix or do you solve a linear equation of the form $A\boldsymbol{v} = \boldsymbol{b}$ This will make a difference in numerical efficiency. Your global matrix is extremely ill-conditioned due to the very large and very small dissipative wavenumbers towards the inviscid end of the altitude range. One benefit of the scattering-matrix approach is that it avoids this problem, at least as formulated in Knight et al. (2019). It would be interesting to see some discussion of how your banded-matrix method avoids the problem of ill conditioned matrices, especially with a lower boundary at or below 50 km altitude, where kinematic viscosity is very small.

We added the following explanation to the manuscript: *"The matrix $\mathbb{A}$ has $3M - 1$ sub- and superdiagonals (excluding the main diagonal) and can therefore be stored in banded form and treated using standard band-matrix techniques. To solve the resulting banded system of linear equations, we employed the LAPACK routines ZGBTRF and ZGBTRS. The routine ZGBTRF performs an LU factorization with partial pivoting of the complex band matrix, and ZGBTRS subsequently uses this factorization to solve the linear system for the prescribed right-hand side. In this approach, the inverse of the full system matrix is not computed explicitly, which improves the computational efficiency."* Moreover, for lower boundary conditions imposed at 40, 45, 50, 60, 70, 75, and 80 km, we found that all solution methods yield nearly identical results. Accordingly, the maximum values of the perturbed vertical velocity are 50.86, 20.35, 14.57, 14.19, 16.16, 8.39, and 3.44 m/s. In these simulations: the horizontal wavelength is $\lambda_x = 400$ km, the wave period is $\lambda_t = 40$ min, the imaginary frequency shift for a single-frequency wave is $10^{-6}$ s$^{-1}$, and the lower boundary conditions are imposed on the vertical velocity with $f_{b2} = 1 \times 10^{-2}$ ms$^{-1}$.

Lines 393-395. It is not clear why the authors feel the need to state this condition. Generally, in discussion of linear methods, it goes without saying that the approximation is valid in the asymptotic limit of small perturbations. It does not appear that the authors have an actual estimate of the range of validity of their linear equations, so why bother stating eq. (88)? Knight et al. (2024, eq. 6.2) give a condition for wave breaking, but weakly nonlinear effects can occur at smaller amplitudes. Unless the authors can give a good reason for including eq. (88) and the related discussion, it should be deleted.

In the revised manuscript, we clarified that the amplitude of the source function is specified through a normalization condition rather than as a strict criterion for linear validity. Specifically, we determine the amplitude of the source function by imposing a lower-boundary value on the amplitude of the perturbed temperature, horizontal velocity, or vertical velocity. We added the following text in Section 4: *"The amplitude of the source function A is specified by imposing the normalization condition:* $|Re\{f'_{sq_0}(x = 0, z_1, t_0)\}| = f_{bq_0}$, *for*

*some prescribed boundary value $f_{bq_0} > 0$. For example, in the case $q_0 = 1$, $f_{bq_0}$ may be chosen as a fraction of the maximum horizontal velocity of neutrals in the south direction over the altitude range, whereas in the case $q_0 = 3$, $f_{bq_0}$ may be chosen as a fraction of the maximum temperature of neutrals over the altitude range."*

 This statement is problematic because $s$ is in the frequency domain up to here, while in Section 5 it becomes a time-dependent function. I suggest rewording so that the term $s$ is not explicitly mentioned.

As noted above, in the revised manuscript the source factor s appears only in Section 4. For time-dependent wave packets, we now state: *"Applying the Fourier transform to Eq. (81) and using Eqs. (79) and (80), we obtain the following boundary conditions in the frequency domain (note that $AS(\omega)$ is the Fourier transform of $As(t)$):*

$$\widehat{F}_{sq_0}(z_1, \omega) = AS(\omega), \;\; \frac{\partial \widehat{F}_{sq_0}}{\partial z}(z_1, \omega) = 0, \;\; \frac{\partial^2 \widehat{F}_{sq_0}}{\partial z^2}(z_1, \omega) = 0.$$

*Comparing Eqs. (86) and (48), we see that the latter corresponds to the choice $b_{1,2} = b_{1,3} = 0$. In this case, the source factor $s = b_{1,1}$ can be identified with $AS(\omega)$."* Similarly, for a single-frequency wave, we state: *"Accordingly, the localized boundary conditions are imposed directly on the harmonic amplitudes at $\omega_0$, namely*

$$\widehat{F}_{sq_0}(z_1, \omega_0) = A, \;\; \frac{\partial \widehat{F}_{sq_0}}{\partial z}(z_1, \omega_0) = 0, \;\; \frac{\partial^2 \widehat{F}_{sq_0}}{\partial z^2}(z_1, \omega_0) = 0.$$

*Again, by comparing Eqs. (98) and (48), we see that the source factor $s = b_{1,1}$ can be identified with $A$."*

 Do 1 and 2 correspond to + and -? State this earlier.

We clarified this point by adding the following explanation: *"We consider the continuity equation (38) and partition the matrices $\mathbb{A}^i_{l,l+1}$, with $i = 0, 1$, as..."*

 There is a notational conflict with eqs. (61-62), since this reuses the symbol $S$.

The scaling matrices are now denoted by $K^1_l$ and $K^0_l$

 Where does the term "interaction principle" come from? Line 414. Give a reference for this formulation of the scattering matrix. Lines 424-427. Give a reference for these equations. They are similar to Knight et al. (2019, eqs. 4.30-33), for instance. Line 102. Give a reference for this equation. It is similar to Knight et al. (2019, eq. 4.34), for instance.

The term "interaction principle equation," the concept of a "stack," as well as the term "adding formulas" are standard in radiative transfer theory. As examples, we indicated Refs. [31,32]. We also note: *"Note that Eqs. (68)–(71) are mathematically equivalent to Eqs. (4.30)–(4.33) in Ref. [16]".*

 I find Section 4.2 problematic and think it should be deleted. In Section 6, it is seen that there is no advantage in using the formulations in Section 4.2, so why add all of this unnecessary detail? If you think it is

important, then summarize this alternative in a few sentences, including the finding in Section 6 that it offers no advantage. That aside, the use of the word "discrete" in the title of this section is unclear. In what sense is it discrete? How is it any more discrete than the approach of Section 4.1? Discrete ordinates are mentioned in the introduction, but that does not appear to be related.

We agree with the reviewer that Section 4.2 does not offer a practical advantage. Accordingly, Section 4.2 has been deleted from the main text. The solution methods described there, which we consider to be of purely theoretical interest, have been moved to Appendix C, entitled Solution methods for the grid-point values of the state vector. In addition, the term "discrete values" has been replaced by "grid-point values" to avoid ambiguity.

Line 554. Shouldn't $b1,k$ depend on $\omega$?

Yes, $b_{1,1}$ depend on $\omega$; this is clarified in our response to the comment referring to Lines 399–400.

Lines 563-565. This sentence about the separable case appears to be unnecessary.

The sentence has been deleted.

Line 572. It should be mentioned that in Knight et al. (2024, eq. 2.29) a relaxed condition is given, in which the bounds are allowed to vary with attitude.

In Section 4, we added the following text: *"Knight et al. [11] subsequently relaxed the requirement that (106) hold for all layers $l$ for a fixed $\sigma$. The new condition, which we refer to as the Layerwise Causality (LC) condition, requires that at each layer $l$ there is a $\sigma_l$ such that*

$$Re[\lambda_{1l}^+(\omega - j\delta)] < \sigma_l < Re[\lambda_{1l}^-(\omega - j\delta)], \tag{1}$$

*for all $\omega \in \mathbb{R}$. Equivalently, if at each layer $l$,*

$$d_l(\omega) = Re[\lambda_{1l}^-(\omega - j\delta)] - Re[\lambda_{1l}^+(\omega - j\delta)] > 0, \tag{2}$$

*for all $\omega \in \mathbb{R}$, then the multilayer algorithm will still preserve causality. The LC condition requires a strict separation between the two eigenvalue families within each layer, but it does not require that the same separator works for all layers. Thus, each layer may have its own separating value $\sigma_l$. Equivalently, $d_l(\omega) > 0$ means that, in layer $l$, the real parts of the eigenvalues associated with the ascending and descending gravity waves remain separated (and therefore do not cross) as functions of the real frequency $\omega$ after the shift."*

Lines 577-594. Since imaginary frequency shifting is not used in this work, much of this summary should be deleted. Regardless, the explanation given here is incomplete, in that there is no indication of how $\delta$ was selected. Methods for determining the minimum sufficient $\delta$ are described in Knight et al. (2019,2021,2022). Instead of giving this summary, the paper should say that the imaginary frequency shifting technique was not applied and that further study is needed to determine the effect. Also, below I will suggest a new figure for Section 6 that will give a good indication of whether problematic branch points are present. If there are problematic branch points, they will primarily affect nearby frequencies.

Lines 590-594. The problem of the numerical blowup associated with the exponential growth term is discussed at length in Knight et al. (2021, App. B) and should be cited here. Numerical blowup is especially a problem for large time domains. If, in future work, you are unable to obtain results without the blowup, then I suggest reducing the size of the time domain and considering narrower time wave packets. Care is needed in selecting $\delta$. It should be large enough to prevent the crossing seen in Knight et al. (2019, Fig 2b), but not much greater than that. Rigorous methods are discussed in Knight et al. (2019,2021,2022), but it would suffice to look for the curve-crossing issue. I recommend replacing the discussion from line 577 to 594 with brief references to Knight et al. (2019, Section 3.4) and Knight et al. (2021, Section 2.4). You can mention that you encountered the numerical blowup problem mentioned in Knight et al. (2021, App. B) and that further study is needed to resolve this issue. It is good that you introduce imaginary frequency shifting in lines 570-675, however, since that allows you to explore the effect of $\delta$ on the root-crossing issue illustrated in Knight et al. (2019, Fig. 2b).

In the new algorithm implementation, we employ an imaginary frequency shift. In Subsection 4.3, we summarize the imaginary frequency shift approach and conclude the discussion with the following text: *"A potential numerical issue with this approach is that a large frequency shift $\delta$, while ensuring the causality condition, may amplify rounding errors when recovering $f'_{sq}$ from $f'_{\delta sq}$ through the exponential term $exp[\delta(t-t_0)]$. For large t, this may lead to an uncontrolled growth of the right-hand side of Eq. (115). Therefore, care is required in selecting $\delta$: it must be large enough to ensure the layerwise causality condition , but not significantly larger than that. Rigorous methods for determining the minimum sufficient $\delta$ were described by Knight et al. in Refs. [15,16,17], while the numerical blow-up associated with the exponential growth term was discussed in Appendix B of Ref. [17]. In our implementation we employ a heuristic approach that combines (i) the Layerwise Causality (LC) condition applied at selected altitude levels, and (ii) a Source-Function Reconstruction (SFR) test. First, an admissible interval $[\delta_{\min}, \delta_{\max}]$ is constructed by enforcing the SFR criterion, and within this interval, the LC condition is applied at selected altitude levels to obtain a refined lower bound $\delta_{\min}$. In the final selection step, a discrete set of candidate shifts is evaluated, and for each candidate the LC condition is checked over the entire altitude range. Among all shifts that satisfy causality at all altitudes, the algorithm selects the one whose maximum-amplitude vector is closest to the center of mass of the admissible solutions. A detailed description of this approach is provided in Appendix D."* Thus, the imaginary frequency shift approach is discussed in Appendix D. In Section 6 (Numerical Simulations), we illustrate how this algorithm operates in practice.

Lines 594-607. This alternative approach should not be included in the paper. In practice, it is impossible to verify (157) rigorously without Titchmarsh's theorem (Knight et al., 2019, Section 2). A concise statement of the causality condition is given in the short paragraph following the proof of Lemma 1 in Knight et al. (2019, Section 2). Using the notation given there, the condition $\beta(t) = 0$ for $t < 0$ can be required for the lower boundary condition $\beta$, but

Titchmarsh's theorem is needed to establish it for $w$ (Knight et al., 2019, eq. 2.4).

We agree with the reviewer and have therefore removed this alternative approach from the manuscript.

Line 608. It does not appear that the horizontal wavenumber or wavelength is ever given in Section 6. It is important to specify this.

In Section 6, we have now specified the values of the horizontal wavelength and the wave period.

Lines 613-616. Methods 3, 4, and 5 should not be included here, given that they offer no advantages in accuracy or efficiency. The derivations in Section 4.2 do not seem scientifically interesting, given that they mostly rearrange terms from Section 4.1. (The Pade approximation would be of interest if it actually provided advantages, but it does not, so there is no apparent need to mention it except perhaps very briefly.) Method 2 is of interest because a related method is currently in use for atmospheric gravity waves (referring to the Knight et al. work).

In the code, we implemented the following methods: the Global Matrix Method for the Amplitudes (GMMA) of the characteristic solutions; the Scattering Matrix Method for the Amplitudes (SMMA) of the characteristic solutions; and the Global Matrix Method for the Nodal (grid-point) values (GMMN) of the state vector. The first two methods are described in the main text, whereas the third method is described in Appendix C. We do not wish to omit the third method, because it is the approach commonly used in radiative transfer theory, where the solution is formulated in terms of the grid-point values of the radiance.

Line 620. This does not look like a complete list of background parameters. It might be complete with $p0$ , $cv$ , and $Pr$ added.

In Section 6, we have now specified the value of the Prandtl number and added the following text: "*Background atmospheric parameters. The code provides altitude-dependent profiles of the input parameters used in the numerical model. These include the temperature, mass density, pressure, southward horizontal velocity, atmospheric scale height, density scale height, specific heat capacity, ratio of specific heats, sound speed, number density of $O^+$ ions, neutral–ion collision frequency, ion–neutral collision frequency, and the diffusion velocity. In addition, altitude derivatives of the temperature, horizontal velocity, mass density, pressure, density scale height, and ion number density are also provided. As an illustrative example, Fig. 1 shows the background temperature $T_0$, horizontal velocity $u_0$ and the ion number density $n_{i0}$, together with their corresponding altitude derivatives.*"

Lines 520-525. What are the input parameters for SAMI2 and HWM?

We have added a new section, namely Section 5 (Numerical Implementation), in which the input data and their use are described in detail. We now state: "*The code uses as input the data file produced by the International Reference Ionosphere (IRI) code available at https://ccmc.gsfc.nasa.gov/models/IRI˜2016/ is used. From these date, we read*

1. *the date (year, month, and day) and the time,*

2. *the geographic latitude and longitude,*

3. *the magnetic dip angle,*

4. *the solar radio flux f10.7 and its 81-day average,*

5. *the number density of $O^+$ ions as a function of altitude.*

*In its present implementation, the IRI data files correspond to the locations summarized in Table 1. The altitude grid extends from $z_{\min} = 80$ km to $z_{\max} = 500$ km with a step size of $dz = 1.0$ km. Users may generate custom data files by running the IRI code and specifying the corresponding file names in the input namelist.*

*The IRI data are subsequently used in a manner analogous to that in the SAMI2 model of the Naval Research Laboratory (https://github.com/NRL-Plasma-Physics-Division/SAMI2). In the present implementation, the ionospheric equations follow the SAMI2 framework originally developed by Huba et al. [40] and described in detail by Huba [41]. In SAMI2, the neutral atmospheric parameters– namely the neutral number density, total mass density, and temperature–are specified using the MSIS family of models. In this study, these parameters are based on the MSIS formulation of Hedin [42], while we note that more recent updates are provided by the NRLMSIS 2.0 model of Emmert et al. [43]. The meridional and zonal winds are specified using the Horizontal Wind Model. In the present implementation, we follow the formulation of Hedin et al. [44], while more recent updates are described by Drob et al. [45]."* In addition, at the beginning of Section 6 we now specify the numerical input used in the simulations: *"The simulations are performed using, as input, an IRI data file corresponding to the EISCAT Tromsø (auroral) location on 11 February 2012 at 10:00. The solar zenith angle is 84.2°, the magnetic inclination angle is 78.28°, the daily solar radio flux F10.7 is 109.4 sfu, and the 81-day averaged solar radio flux is 116.9 sfu, where $1\,sfu = 10^{-22}\ Wm^{-2}Hz^{-1}$. In the simulations, the Prandtl number is 0.66, the magnetic index is 7.0, and the horizontal wind model 14 implemented in SAMI2 is used. The altitude grid extends from 80 km to 500 km and contains 801 grid points. The lower boundary can, in principle, be set to smaller altitudes (e.g., 50 km), but we choose 80 km because this level is typically adopted as the lower boundary for ionospheric equations. Unless stated otherwise, the horizontal wavelength is $\lambda_x = 400$ km, the wave period is $\lambda_t = 40$ min, and the imaginary frequency shift for a single-frequency wave is $10^{-6}\ s^{-1}$. The lower boundary conditions are imposed on the vertical velocity with $f_{b2} = 5 \times 10^{-2}\,ms^{-1}$ in Eqs. (85) and (101)."*

Line 627. $p_0$ , $c_v$ , and $\rho_0$ should also be shown in a figure. The density scale height $H$, should also be shown.

The freely available code computes these quantities, among others, as altitude-dependent input parameters. In order to avoid an overabundance of figures showing background profiles, we decided to display only the background temperature, the horizontal velocity, and the ion number density, together with

their corresponding altitude derivatives, as representative examples of the input data.

Lines 649-652. I suggest omitting the Fourier transform in the altitude dimension. There are several reasons for this: 1. It is confusing, since it means that there are two types of vertical wavenumbers, one obtained from the vertical structure equations and one obtained directly from the Fourier transform. 2. It creates notational ambiguity, since the same notation is used for both types of vertical wavenumbers. 3. Taking the Fourier transform in the vertical dimension does not make sense given that the vertical wavenumbers coming from the vertical structure equations include both real and imaginary parts. See Knight et al. (2025, Section 4.2, first paragraph). Vertical wavelengths are not defined, strictly speaking, when significant dissipation is occurring. 4. The Fourier transform makes the most sense with periodic or unbounded domains, neither of which applies to the altitude dimension. Aside from that, the values given here are difficult to interpret. If the vertical Fourier transform is left in the paper (which I recommend against), then the actual value for $\Delta k_z$ should be given, and $N_k \Delta k_z$ should approximately equal 450 km. Line 655. Give a reference for the nonuniform Fourier transform.

We agree with the reviewer's assessment and have removed the consideration of the Fourier transform in the altitude dimension from the revised manuscript.

Lines 665-672. There is no need to include this discussion, and it should be deleted, along with Figure 2. Figure 2 merely confirms that the derivations in Appendix A.2 are correct, and it should go without saying that they are correct.

Lines 673-689. Again, methods 3, 4, and 5 should be omitted, making Figure 3 unnecessary. Continuing with lines 673-689, Figure 4 probably becomes unnecessary if it is just a comparison of the first two methods. Given $M = 3$, I would expect about factor of three ratio of processing times between methods 2 and 1, assuming that the banded-matrix method solves the linear equation $A\boldsymbol{v} = \boldsymbol{b}$ directly rather than inverting $A$. This is because the scattering-matrix approach effectively solves for a general three-dimensional lower-boundary condition, meaning that it does more computations than would be needed for a specific lower-boundary condition, in principle. The ratio in Figure 2 is more like a factor of five. This makes me wonder whether your scattering-matrix computations are done as efficiently as they could be. Rigorous analysis of the computational steps involved with methods 1 and 2 would be needed to clarify this. I am not suggesting that such analysis be included in the current paper, but I would like for your paper to mention that more rigorous analysis is needed to make the result definite.

We agree with the reviewer's assessment. In the revised manuscript, Figures 2, 3, and 4 have been removed, and the corresponding discussion has been deleted. Instead, we now include the following concise summary: "*Accuracy and efficiency of the solution methods. Taking the global matrix method for amplitudes as a reference, we find that the relative root-mean-square errors in the perturbed temperature, vertical velocity, and horizontal velocity obtained with the other two solution methods are smaller than $10^{-6}$. Thus, all methods ex-*

hibit comparable accuracy. On the other hand, we find that the scattering matrix method is more time-consuming than the global matrix methods, particularly for time-dependent wave packets. This is because the scattering matrix approach requires numerous matrix operations in each layer, whereas solving a system of equations compressed into band storage is computationally less expensive." We deliberately leave open the possibility for users to further improve the implementation of the scattering-matrix method and, eventually, its computational efficiency. Instead of these figures, we have included Fig. 3, which shows the altitude profiles of the perturbed temperature, vertical velocity, and horizontal velocity computed using the general and simplified models.

Lines 690-701. It is not clear what is gained by merely comparing results for different values of $\kappa\omega$. This is because there is no way of knowing to what extent differences in neutral-atmospheric dynamics are contributing to the differences. To clarify this, I recommend combining the results in the upper panels of Figure 5 with the results shown in Figure 8 in the same figure (maybe a different figure for each state variable) and discussing these results together. I would give results without ion damping for each of the three $\kappa\omega$ values so they can be compared in each case. Why are the apparent vertical wavelengths in the upper panels of Figure 5 so similar for the three $\kappa\omega$? As mentioned above, I could not see where you specified the horizontal wavelength. The vertical wavelength coming from the vertical structure equations should change with $\kappa\omega$, assuming that the horizontal wavelength is kept fixed. These issues need to be clarified in Section 6. As indicated above, I recommend omitting the type of analysis shown in the lower panels of Figure 5 and in Figure 6. You can replace it with a comparison of results for the three $\kappa\omega$ values, with and without ion damping, as described above. If you want to talk about vertical wavenumbers, I recommend looking at vertical profiles of the upgoing gravity-wave roots and interpreting differences in model results in terms of those. It would also be good to include discussion of previous analysis of the effects of ion damping on gravity waves and relate it to your present work.

In the previous implementation, an error was present in the coefficient of thermal conductivity, which was set to $\lambda_0 = 6.71 \times 10^{-7}T_0^{0.71}$ instead of the correct value $\lambda_0 = 6.71 \times 10^{-4}T_0^{0.71}$, thereby substantially reducing the effect of thermal conduction in favor of ion-drag effects. After correcting this error, we have revised the numerical analysis accordingly. In the revised manuscript, Fig. 8 of the previous version has been replaced by a new Fig. 4, which focuses explicitly on the effect of ion drag. The new Fig. 4 includes the following explanation: "*Ion Drag. The effect of ion drag on the perturbed temperature, vertical velocity, and horizontal velocity is illustrated in Fig. 4. A moderate attenuation is observed in the altitude range from 180 to 350 km, where the ion number density is relatively high. When $\mathbf{E} \times \mathbf{B}$ drifts are not included, ion drag does not exhibit the classical regime in which auroral convection strongly drives the neutral atmosphere. Instead, ion drag mainly arises from diffusion- and pressure-gradient-driven ion motion along the magnetic field, as well as from any relative ion–neutral motion induced by neutral winds. Consequently, ion drag does not constitute the dominant forcing mechanism for the neutral*

*perturbations in this configuration."*

After Fig. 4, we included a new figure, namely Fig. 5, illustrating the influence of the horizontal wavelength $\lambda_x$ and the wave period $\lambda_t$ on the perturbed quantities. We found these effects to be of interest and summarize them as follows:

1. *"Horizontal wavelength. As the horizontal wavelength increases, horizontal pressure gradients become weaker, which reduces the driving of the wave motion. This also weakens the coupling between horizontal and vertical motions, resulting in smaller gravity-wave amplitudes."*

2. *"Wave period. As the wave period increases, the buoyancy restoring force acts more slowly, leading to weaker oscillations for a given forcing. In addition, dissipative processes such as viscosity and thermal diffusion act more effectively on low-frequency waves, further reducing their amplitudes."*

Lines 702-722. Pairwise classification of vertical wavenumbers is less important than being able to divide the roots into separated upgoing and downgoing sets. Figure 7 should include more descriptive titles and labels giving the meaning of the panels. The figure caption is difficult to interpret because it merely refers to equation numbers without reminding the reader of the meaning. While Figure 7 illustrates the differences between two governing-equation assumptions (i.e., locally varying and constant kinematic viscosity) in their effect on vertical wavenumbers, which is of some value, it does not say much about whether the roots can be separated into upgoing and downgoing sets. To do this, one would need to look at how the roots vary with frequency. This applies even for fixed-frequency cases. I recommend giving a figure like Knight et al. (2019, Fig. 2b) for several different altitudes, e.g., 150, 250, 350, and 450 km. If any of the roots cross like in Knight et al. (2019, Fig. 2b), it means that there is a problematic branch point nearby. Generally, there is no problem for single-frequency results provided that the frequency is far from the problematic branch point. Even though the global method does not explicitly require upgoing and downgoing modes to be defined at intermediate altitudes, the solution still may not be valid without appropriate imaginary frequency shifting for problematic branch points occurring over the entire altitude range. I hope to write a paper clarifying these issues in the future.

In the revised manuscript, Fig. 2 is now the counterpart of Fig. 7 from the previous version and has been substantially revised, including clearer titles, labels, and an expanded caption. The new Fig. 2 includes the following explanation: *"In the upper and middle panels of Fig. 2, we plot the imaginary part of the vertical wavenumber for ascending $(k_{z1})$ and descending $(k_{z4})$ gravity waves, computed using the general and the simplified models, respectively. The plots demonstrate that only in the latter case do the vertical wavenumbers appear in pairs. In the former case, a problematic altitude range for gravity waves is observed between 80 km and 120 km, where the imaginary parts of the vertical wavenumbers for the ascending and descending modes are nearly identical, being either both positive or both negative. In the lower panel of Fig. 2,*

we show the imaginary part of the vertical wavenumber $k_z$ for all wave types $(k_{zn}, n = 1, \ldots, 6)$, computed using the general model. The plots reveal a clear distinction between gravity waves and viscosity- and thermal-conduction waves. However, the viscosity and thermal-conduction waves are very close to each other. In the upper panel of Fig. 2, we also compare the imaginary part of the vertical wavenumber computed with and without ion drag. No pronounced effect of ion drag on the vertical wavenumber is observed. A small effect appears in the altitude range from 180 to 300 km, where the ion number density is relatively high. This finding is consistent with the results of Shibata [43], who showed that, for gravity waves, plasma diffusion is of minor importance with respect to the vertical wavenumber, which is mainly controlled by dissipation due to viscosity and thermal conduction in the neutral gas." We emphasize that, in Fig. 2, the pairwise classification of vertical wavenumbers for the simplified model is included primarily to support the statements in Section 2 and the dispersion equation in Appendix A. The crossing of the curves corresponding to the real parts of the eigenvalues associated with ascending and descending gravity waves as functions of frequency is encapsulated in the Layerwise Causality (LC) condition. In the code, this condition is checked first at selected altitude levels and, in a second step, over the entire altitude range. If the LC condition is not satisfied, an error message is issued and, in the second case, the corresponding solution is considered invalid. Finally, we note that a routine for plotting the eigenvalue curves as functions of frequency could, in principle, be incorporated into the code without difficulty. However, we do not regard this as necessary for the purposes of the present study (code description), since any user with access to the freely available source code can readily generate such plots and carry out this analysis independently.

Line 718. "The ion-drag is important for time frequencies . . ." Give a specific reference for this.

This sentence has been removed from the revised manuscript.

Lines 718-722. As mentioned above, this discussion should be combined with the discussion in lines 690-701. Also, the results discussed here are puzzling. It says there is complete agreement between results with and without ion drag for $\kappa\omega = 1.2$. This does not seem possible. Surely, ion drag would have some effect. The authors should double-check this and provide further explanation if there really is no effect. In particular, they should look at the vertical wavenumbers (obtained from the vertical structure equations) and see whether there is any difference. Aside from this, the caption of Figure 8 is puzzling. Case (a) is with ion drag excluded. What is $\kappa\omega$ for (a)? If $\kappa\omega = 0.8$ for (a), then the similarity between results for (a) and (b) makes even less sense, given that $\kappa\omega = 1.2$ for (b). The wording here and in the text should be made clearer, and errors, if any, should be corrected.

The numerical analysis related to the effects of ion drag has been revised. We refer to our response to the comments on Lines 690–701 for a detailed description of the corrections and the revised interpretation.

Lines 723-726. These lines should be deleted. Figure 9 gives a comparison of nearly identical results, and if the results are identical there is most likely a

trivial reason for it, so the discussion, along with Figure 9, does not need to be included.

Lines 723–726 and Fig. 9 have been deleted from the revised manuscript.

Lines 733-738. This is similar to some previous work, which should be cited. Knight et al. (2019) defines the "transmission-only" approximation, which is similar to your eq. (118), and Knight et al. (2021) discusses a single-mode approximation, which is related to the transmission-only approximation. Additionally, Knight et al. (2019, Section 6) shows the upgoing and downgoing contributions to a wavefield. Although (118) is introduced in Section 4.2, which I recommend deleting, it should be possible to give very similar definition in Section 4.1.2.

We have revised this part of the manuscript and now explicitly relate our formulation to the work of Knight et al. In particular, the new Fig. 6 addresses the computation of ascending and descending wave modes and is explained as follows: *"The ascending and descending solution modes in layer $l$, denoted by $\mathbf{e}_l^+$ and $\mathbf{e}_l^-$, respectively, can be computed using the GMMA through Eq. (61) or using the GMMN via the recurrence relations (269) and (270). The total solution mode $\mathbf{e}_l$, obtained from Eq. (59) in the GMMA formulation and by solving Eq. (266) in the GMMN formulation, should satisfy the relation $\mathbf{e}_l = \mathbf{e}_l^+ + \mathbf{e}_l^-$. In all our simulations, this identity is satisfied. Furthermore, the results shown in Fig. 6 indicate that the ascending mode is dominant, except in the altitude range between 120 km and 180 km in the case of the general model. This finding, which is consistent with the results presented by Knight et al. [16] (see their Fig. 6), suggests that in a simplified model one may assume the ascending modes to be dominant at altitudes above 200 km, that is, $\mathbf{e}_l \approx \mathbf{e}_l^+$ for $l = 1, \ldots, L$. Under this assumption, the state vector can be computed using the upward recurrence relation (269). In Ref. [17], this approach was referred to as the transmission-only approximation, whereas in Ref. [16] a related single-mode approximation was introduced."*

Lines 747-748. This reflects a naïve view of causality. Causality is really about whether upgoing and downgoing modes are defined and valid. For frequencies near problematic branch points, asingle-frequency solution will be incorrect, regardless of whether the peak in amplitude seems to be moving with altitude.

Lines 749-756. This discussion is problematic. Firstly, Figure 15 is the wrong type of plot for analyzing issues with causality, i.e., whether upgoing and downgoing roots are valid. What is needed is a figure like I described for lines 702-722 above, showing the imaginary parts vs. frequency. There is no indication of how the $\delta$ value was selected. Note how in Fig 2b of Knight et al. (2019), two roots cross, while in Fig. 2d they do not cross. This indicates that the $\delta$ value used in Fig. 2d was sufficient. If $\delta$ is not large enough to prevent the roots from crossing, then it will not work. The bottom three panels of Figure 15 should be omitted. To really assess the effect of problematic branch points, you need a solution that is known to be correct. Just observing that the solution is small before $t = 0$ is not sufficient. Regarding Figure 15, are the eigenvalues specific to $\omega 0$ ? This should be stated.

We are grateful to the reviewer for these comments, which clarified our understanding of the concept of causality and the role of problematic branch points. In the revised manuscript, the previous Fig. 15 has been removed, and the corresponding discussion has been deleted. The new Fig. 7 is now the counterpart of Fig. 13 from the previous version.

Line 747. The extreme difference in computation time between methods 1 and 2 is very puzzling given that only a factor of five difference was seen for the single-frequency case. What possible reason could there be for this? It seems like this must be a mistake.

Lines 773-774. As discussed above for line 747, there is no apparent reason why there should be a difference in relative efficiency between single-frequency and time-varying cases.

In the revised manuscript, we no longer report the computation times of the solution methods. Instead, we leave it to the user to assess the efficiency of the implemented methods and, if desired, to further optimize their performance.

Lines 768-769. "The amplitude of the source function can be computed ..." This is unclear. Why would one want to compute the amplitude of the source function? Generally, one starts with the source and computes the wavefield from that.

We agree with the reviewer's comment. However, as explained in our response to the comments on Lines 393–395, in our approach the amplitude of the source function is determined indirectly by imposing a lower-boundary value on the amplitude of the perturbed temperature, horizontal velocity, or vertical velocity. In other words, we prescribe the effect at the lower boundary and infer the corresponding source amplitude, rather than prescribing the source amplitude a priori, thereby prioritizing the effect over the cause.

Appendix A. Converting to non-dimensional form makes the equations more complicated than they would be otherwise, and it also makes it impossible to check equations via units.

Line 793. "A1-A4" is unclear. Does this mean eqs. (A1-4)? Maybe say "eqs. A1-A4 below".

Lines 916-917. This statement is redundant with discussion in the main text.

Line 927. Say whether this is density or pressure scale height.

Line 940. Say "$\mu0 = \mu k =$ constant", etc., here.

Lines 997-1037. These lines would belong in a separate section, but I do not think they should be included in the paper at all. If you have fresh insights into Hines' criticism, I suggest describing them briefly in the main text without any additional equations.

Appendix A has been completely reformulated, and all of the reviewer's comments have been taken into account. In particular, the notation has been clarified, redundant statements have been removed, ambiguous definitions have been specified, and the extended discussion in Lines 997–1037 has been deleted.

Final comment: It would be advantageous for the authors to show that they can reproduce a previous result. To this end, they could apply their method 1 to the case illustrated by Figure 2a in Knight et al. (2022). It would be interesting

to hear whether they get similar results, although it would not be necessary to add a figure for this.

Our code is designed to operate with realistic, altitude-dependent background parameters, and for this reason it is not straightforward to reproduce the results presented in Knight et al. (2022), which are based on a more idealized configuration. However, we are able to reproduce the results reported in Knight et al. (2024), which are formulated within a framework more consistent with the present model.

New References Knight, H., Broutman, D., & Eckermann, S. (2024). Compressible and anelastic governing- equation solution methods for thermospheric gravity waves with realistic background parameters. Theoretical and Computational Fluid Dynamics, 38(4), 479–509. https://doi.org/10.1007/s00162-024-00709-x Knight, H. K., Richards, P. G., Martinis, C. R., & Goncharenko, L. P. (2025). Modeling MSTIDs produced by gravity waves with parameters obtained from all-sky imager observations and comparisons to incoherent scatter radar observations. Journal of Geophysical Research: Space Physics, 130, e2025JA033906. https://doi.org/10.1029/2025JA033906.

We have included these references in the revised list of references.

**A numerical model for solving the linearized gravity-wave equations by a multilayer method**

Alexandru Doicu[a], Dmitry S. Efremenko[b], Thomas Trautmann[b], Adrian Doicu[b]

January 22, 2026

[a]Independent Researcher, 82110 Germering, Germany

[b]Deutsches Zentrum für Luft- und Raumfahrt (DLR), Institut für Methodik der Fernerkundung (IMF), 82234 Oberpfaffenhofen, Germany

**Abstract**

We developed a numerical model for solving the linearized gravity-wave equations using a multilayer approach that explicitly accounts for viscosity, thermal conduction, and ion drag. The solution strategy is based on a matrix-exponential formalism and comprises two classes of methods: global matrix methods and scattering matrix methods. The model supports both single-frequency waves and time-dependent wave packets. Particular emphasis is placed on the global matrix method, which exploits the structured form of the multilayer system to achieve high computational efficiency while maintaining numerical accuracy. Numerical experiments demonstrate that all methods yield identical accuracy, although the global matrix method is significantly more efficient than the scattering matrix method, especially for time-dependent wave packets. The impact of ion drag on wave characteristics is quantified within this framework. The implementation is freely available as open-source code on GitHub.

**1   Introduction**

Time-step methods [1, 2, 3] are commonly used to solve fully nonlinear sets of governing equations for upper-atmospheric gravity waves, thereby allowing the modeling of wave breaking, secondary wave generation, and weakly nonlinear effects. However, as compared to linear methods for gravity waves [4, 5, 6, 7, 8, 9, 10, 11] they are computationally expensive. In Ref. [12] it was found that a time-step model took several hours to run, while a linear method only took several seconds. In this regard, linear methods are more suitable for analyzing measured data.

[revised manuscript text omitted]

The main purpose of this article is to apply radiative transfer techniques to solve the linearized gravity-wave equations. As a prototype, we will consider the equations that describe gravity waves in the ionosphere, and that include viscosity, thermal conduction, and ion drag. In principle, a full wave model for the ionosphere comprises the hydrodynamic equations for the neutral atmosphere and the ionospheric equations. These two sets of equations are coupled through the ion drag, and should be solved together. However, to simplify the analysis, we decouple the two sets of equations by adopting a fast field-aligned diffusion approximation, which may be viewed as a generalization of an approximation originally proposed by Klostermeyer [9].

Our paper is organized as follows. In Section 2, we present the derivation of the matrix exponential solution of the linearized equations, while Section 3 describes stable numerical methods for computing the amplitudes of the characteristic solution in a stratified atmosphere. Section 4, which is largely inspired by the works of Knight et al. [11, 15, 16, 17], addresses the computation of the perturbed quantities for both harmonic and non-harmonic source functions, that is, for single-frequency waves and time-dependent wave packets. The concepts of causality and the imaginary frequency shift, which are rigorously treated in Refs. [15–17], are also briefly discussed. Aspects of the numerical implementation are addressed in Section 5, and representative simulation results are presented in Section 6. Additional theoretical issues are discussed in the appendices. Appendix A contains the linearized hydrodynamic equations for the neutral atmosphere and the derivation of the underlying system of differential equations. Appendix B outlines the linearized ionospheric equations and discusses the assumptions employed to decouple the hydrodynamic and ionospheric systems. Appendix C describes methods for computing grid-point values of the state vector in a stratified atmosphere. Appendix D addresses several implementation issues, including a practical, albeit heuristic, approach for determining the imaginary frequency shift.

**2 Matrix exponential solution of the linearized equations**

To design a full wave model for the ionosphere, we use the hydrodynamic equations for the neutral atmosphere and the ionospheric equations. In a linearized (perturbation) method, a quantity $f$ is expressed as

$$f = f_0 + f', \tag{1}$$

where $f_0$ and $f'$ are the unperturbed (background) and the perturbed quantity, respectively. The perturbations are assumed to be small so that it is justified to neglect all terms of higher than the first order.

Concretely, we solve the linearized hydrodynamic equations for the neutral atmosphere together with the linearized ion continuity and momentum equations. The linearized neutral-atmosphere equations are solved under the following assumptions:

**A1.** The geographic and geomagnetic coordinates are identical.

**A2.** The wave propagates in the meridional plane (the $x$-coordinate is positive southwards while the $z$-coordinate is positive upwards), i.e.,

$$f = f(x, z, t). \tag{2}$$

**A3.** All background (unperturbed) quantities vary only in the $z$-direction, i.e.,

$$f_0 = f_0(z), \tag{3}$$

while all perturbations vary harmonically in time and the $x$-direction, i.e.,

$$f' = f'(x, z, t) = \overline{f}(z)e^{j(\omega t - k_x x)}, \tag{4}$$

where $\omega$ is the angular frequency and $k_x$ the horizontal wavenumber. Note that in some gravity-wave studies, the opposite sign convention for frequency and horizontal wavenumber is used (e.g. Ref. [34]).

The linearization model is described in Appendix A. It provides a general framework that accounts for the altitude derivatives of the background velocity $u_0$, temperature $T_0$, density scale Height $H_\rho$, and dynamic viscosity $\mu_0$. Apart from the ion-drag terms, the formulation follows a structure similar to those employed by Vadas and Nicolls [35] and Knight et al. [11].

The computation of the ion-drag force and ion-drag heating is presented in Appendix B. The ion-drag terms are introduced in an approximate manner, with the explicit aim of decoupling the hydrodynamic and ion equation systems. To this end, we adopt the following assumptions:

**B1.** In the ion continuity equation, the perturbed production and loss terms are neglected.

**B2.** In the ion momentum equation, ion inertia and ion–ion collisions are neglected, and only transport parallel to the magnetic field lines is retained. Under these assumptions, the ion momentum equation reduces to the ambipolar diffusion equation.

**B3.** To decouple the ion continuity equation from the diffusion equation, fast field-aligned diffusion is assumed, meaning that the field-aligned diffusion is sufficiently strong for the relative ion perturbation and the perturbed diffusion velocity to remain nearly constant along a magnetic field line.

The linearized equations lead to a linear system of ordinary differential equations, written in matrix form as

$$\frac{1}{k_{\mathrm{x}}}\frac{\mathrm{d}\mathbf{e}}{\mathrm{d}z} = \mathbf{A}\mathbf{e}, \tag{5}$$

where

$$\mathbf{e} = [\widehat{u}, \widehat{w}, \widehat{T}, \widehat{\mathcal{U}}, \widehat{\mathcal{W}}, \widehat{\mathcal{T}}]^{\mathrm{T}} \tag{6}$$

is the state vector, and A is the propagation matrix with altitude independent elements (whose expressions follow from Eqs. (157)–(162) of Appendix A). In general, the unknowns (the hat quantities in Eq. (6)) are defined through the relation

$$\overline{f}(z) = C(z)\widehat{f}(z), \tag{7}$$

where $\overline{f}$ is defined by Eq. (4), and $C$ is a known quantity that ensures that $\widehat{f}$ is dimensionless and that may or may not depend on altitude (here, we indicate that $C$ depend on $z$). Specifically, for the background velocity $\mathbf{u}_0 = (u_0, 0, 0)$, and the perturbed velocity $\mathbf{u}' = (u', 0, w')$, we have (cf. Eqs. (155) and (156) of Appendix A)

$$\overline{u}(z) = \frac{\omega_0}{k_{\mathrm{x}}}\widehat{u}(z), \tag{8}$$

$$\overline{w}(z) = \frac{\omega_0}{k_{\mathrm{x}}}\widehat{w}(z), \tag{9}$$

$$\overline{T}(z) = T_0(z)\widehat{T}(z), \tag{10}$$

and

$$\widehat{\mathcal{U}} = \frac{\mathrm{d}\widehat{u}}{\mathrm{d}z}, \quad \widehat{\mathcal{W}} = \frac{\mathrm{d}\widehat{w}}{\mathrm{d}z}, \quad \widehat{\mathcal{T}} = \frac{\mathrm{d}\widehat{T}}{\mathrm{d}z}.$$

where $\omega_0$ is a reference frequency.

If $(\lambda_n, \mathbf{v}_n)$ is an eigenpair of the matrix A, i.e., $\mathbf{A}\mathbf{v}_n = \lambda_n\mathbf{v}_n$ for $n = 1, \ldots, N$, where $N = \dim(\mathbf{e})$, the general solution of Eq. (5) is a linear combination of the characteristic solutions $\exp(k_{\mathrm{x}}\lambda_n z)\mathbf{v}_n$, that is,

$$\mathbf{e}(z) = \sum_{n=1}^{N} a_n e^{k_{\mathrm{x}}\lambda_n z}\mathbf{v}_n$$

$$= [\mathbf{v}_1, \ldots \mathbf{v}_N] \begin{bmatrix} e^{k_{\mathrm{x}}\lambda_1 z} & \cdots & 0 \\ \vdots & \ddots & \vdots \\ 0 & \cdots & e^{k_{\mathrm{x}}\lambda_N z} \end{bmatrix} \begin{bmatrix} a_1 \\ \vdots \\ a_N \end{bmatrix}$$

$$= \mathrm{V}\mathrm{diag}[e^{k_{\mathrm{x}}\lambda_n z}]\mathbf{a}, \tag{11}$$

where

$$\mathrm{V} = [\mathbf{v}_1, \ldots, \mathbf{v}_N], \ \mathrm{diag}[e^{k_{\mathrm{x}}\lambda_n z}] = \begin{bmatrix} e^{k_{\mathrm{x}}\lambda_1 z} & \cdots & 0 \\ \vdots & \ddots & \vdots \\ 0 & \cdots & e^{k_{\mathrm{x}}\lambda_N z} \end{bmatrix}, \tag{12}$$

and $\mathbf{a} = [a_1, \ldots, a_N]^T$. At $z = 0$, we have $\mathbf{e}(0) = \mathrm{V}\mathbf{a}$; thus,

$$\mathbf{a} = \mathrm{V}^{-1}\mathbf{e}(0), \tag{13}$$

implying (cf. Eq. (11)),

$$\mathbf{e}(z) = \mathrm{V}\mathrm{diag}[\mathrm{e}^{k_\mathrm{x}\lambda_n z}]\mathrm{V}^{-1}\mathbf{e}(0) = \mathrm{e}^{k_\mathrm{x}\mathrm{A}z}\mathbf{e}(0), \tag{14}$$

and conversely,

$$\mathbf{e}(0) = \mathrm{V}\mathrm{diag}[\mathrm{e}^{-k_\mathrm{x}\lambda_n z}]\mathrm{V}^{-1}\mathbf{e}(z) = \mathrm{e}^{-k_\mathrm{x}\mathrm{A}z}\mathbf{e}(z). \tag{15}$$

From the theory of gravity waves within an isothermal, nondissipative atmosphere, it is generally known that the amplitude of an ascending modes increases like $\exp[z/(2H_\mathrm{a})]$, where $H_\mathrm{a}$ is the atmospheric scale height [36]. This is necessary to keep the wave energy constant in an atmosphere where the pressure decreases exponentially with height. In this regard, we define the vertical wavenumber $k_{\mathrm{z}n}$ through the relation

$$\mathrm{diag}[\mathrm{e}^{k_\mathrm{x}\lambda_n z}] = \mathrm{diag}[\mathrm{e}^{z/(2H_\mathrm{a})}\mathrm{e}^{-\mathrm{j}k_{\mathrm{z}n}z}], \tag{16}$$

yielding

$$\lambda_n = -\frac{\mathrm{j}}{k_\mathrm{x}}k_{\mathrm{z}n} + \frac{1}{2}\alpha, \tag{17}$$

and conversely,

$$k_{\mathrm{z}n} = \mathrm{j}k_\mathrm{x}\left(\lambda_n - \frac{1}{2}\alpha\right), \tag{18}$$

where $\alpha = 1/(k_\mathrm{x}H_\mathrm{a})$. The characteristic equation $\det(\mathrm{A} - \lambda\mathrm{I}_N) = 0$ has $N = 6$ solutions. As shown in Appendix A, for a constant kinematic viscosity the solutions occur in pairs and correspond to (i) ascending and descending gravity-wave modes, (ii) ascending and descending viscosity-wave modes, and (iii) ascending and descending thermal-conduction wave modes [6, 7]. In that appendix, this pairing is explicitly demonstrated by deriving the dispersion relation for the special case of an isothermal (constant background temperature), homogeneous (constant kinematic viscosity), and windless atmosphere without ion drag. This solution classification is made according to the imaginary part of the vertical wavenumber $k_{\mathrm{z}n}$. In the more realistic case of a constant background dynamic viscosity, it is generally not possible to define ascending and descending modes as corresponding pairs (see Eq. (168) in Appendix A). However, in our model we will use the same rule as in the case of a homogeneous atmosphere, even though the traditional concept of classifying waves in pairs is no longer applicable. Specifically, we compute $k_{\mathrm{z}n}$ for $n = 1, \ldots, N$ by means of Eq. (18), and order the set $\{k_{\mathrm{z}n}\}_{n=1}^N$, and accordingly, $\{\lambda_n\}_{n=1}^N$, such that

$$\mathrm{Im}(k_{\mathrm{z}3}) < \mathrm{Im}(k_{\mathrm{z}2}) < \mathrm{Im}(k_{\mathrm{z}1}) < \mathrm{Im}(k_{\mathrm{z}4}) < \mathrm{Im}(k_{\mathrm{z}5}) < \mathrm{Im}(k_{\mathrm{z}6}). \tag{19}$$

By convention, (i) the pairs $(k_{\mathrm{z}1} = k_{\mathrm{z}1}^+, \lambda_1 = \lambda_1^+)$ and $(k_{\mathrm{z}4} = k_{\mathrm{z}1}^-, \lambda_4 = \lambda_1^-)$ will correspond to ascending and descending gravity-wave modes, respectively,

(ii) the pairs $(k_{z2} = k_{z2}^+, \lambda_2 = \lambda_2^+)$ and $(k_{z5} = k_{z2}^-, \lambda_5 = \lambda_2^-)$ to ascending and descending viscosity-wave modes, respectively, and (iii) the pairs $(k_{z3} = k_{z3}^+, \lambda_3 = \lambda_3^+)$ and $(k_{z6} = k_{z3}^-, \lambda_6 = \lambda_3^-)$ to ascending and descending thermal conduction-wave modes, respectively. Thus, the vertical wavenumber is an auxiliary quantity that is used only to identify the different modes. According to the notation introduced above, $\{\lambda_m^+\}_{m=1}^M$, where $M = N/2$ is the number of modes, is the set of eigenvalues defining ascending modes, and $\{\lambda_m^-\}_{m=1}^M$ is the set of eigenvalues defining descending modes. Because $\mathrm{Re}(\lambda_n) = \mathrm{Im}(k_{zn})/k_x + \alpha/2$, it is obvious that we can put aside the concept of vertical wavenumber when identifying the different wave modes. We can simply order the set $\{\lambda_n\}_{n=1}^N$, such that

$$\mathrm{Re}(\lambda_3) < \mathrm{Re}(\lambda_2) < \mathrm{Re}(\lambda_1) < \mathrm{Re}(\lambda_4) < \mathrm{Re}(\lambda_5) < \mathrm{Re}(\lambda_6), \qquad (20)$$

and use the same classification rule as above. A commonly cited interpretation of condition (20) is that, for increasing $z$, the exponential term $\exp(k_x \lambda_n z)$ will tend to be damped more for ascending modes than for descending modes; conversely, for decreasing $z$, the roles of ascending and descending modes are reversed. However, such a classification of upgoing and downgoing roots (e.g., Ref. [6] and related works) was primarily heuristic and lacked a rigorous theoretical justification. By contrast, the approach of Knight et al. [16], which is discussed in Section 4, introduces additional constraints beyond condition (20) that are explicitly related to causality and is therefore grounded in theoretical considerations rather than heuristic arguments.

To highlight the different wave modes, we organize the state vector $\mathbf{e}(z)$ as

$$\begin{aligned}
\mathbf{e}(z) &= \mathbf{e}_+(z) + \mathbf{e}_-(z) \\
&= \left( \sum_{m=1}^M a_m^+ e^{k_x \lambda_m^+ z} \mathbf{v}_m^+ \right) + \left( \sum_{m=1}^M a_m^- e^{k_x \lambda_m^- z} \mathbf{v}_m^- \right) \\
&= [\mathrm{V}_+, \mathrm{V}_-] \begin{bmatrix} \mathrm{diag}[e^{k_x \lambda_m^+ z}] & 0_M \\ 0_M & \mathrm{diag}[e^{k_x \lambda_m^- z}] \end{bmatrix} \begin{bmatrix} \mathbf{a}_+ \\ \mathbf{a}_- \end{bmatrix}, \qquad (21)
\end{aligned}$$

where the eigenvector $\mathbf{v}_m^\pm$ corresponds to the eigenvalue $\lambda_m^\pm$,

$$\mathrm{V} = [\mathrm{V}_+, \mathrm{V}_-], \quad \mathrm{V}_\pm = [\mathbf{v}_1^\pm, \dots, \mathbf{v}_M^\pm], \qquad (22)$$

$$\mathbf{a} = \begin{bmatrix} \mathbf{a}_+ \\ \mathbf{a}_- \end{bmatrix}, \quad \mathbf{a}_\pm = \begin{bmatrix} a_1^\pm \\ \vdots \\ a_M^\pm \end{bmatrix}, \qquad (23)$$

and $0_M$ is the zero matrix of dimension $M \times M$. Some useful relations are listed below

1. From Eq. (13), we find

$$\mathbf{a}_+ = [\mathrm{I}_M, 0_M]\mathbf{a} = [\mathrm{I}_M, 0_M]\mathrm{V}^{-1}\mathbf{e}(0), \qquad (24)$$

$$\mathbf{a}_- = [0_M, \mathrm{I}_M]\mathbf{a} = [0_M, \mathrm{I}_M]\mathrm{V}^{-1}\mathbf{e}(0), \qquad (25)$$

where $\mathrm{I}_M$ is the identity matrix of dimension $M \times M$.

2. From Eq. (21), that is,

$$\mathbf{e}_{\pm}(z) = \sum_{m=1}^{M} a_m^{\pm} e^{k_x \lambda_m^{\pm} z} \mathbf{v}_m^{\pm} = \mathrm{V}_{\pm} \mathrm{diag}[e^{k_x \lambda_m^{\pm} z}] \mathbf{a}_{\pm}, \qquad (26)$$

we deduce that

$$\mathbf{e}_{\pm}(0) = \mathrm{V}_{\pm} \mathbf{a}_{\pm}. \qquad (27)$$

3. From Eq. (14), we obtain

$$\mathbf{e}_{+}(z) = \mathrm{T}_{+} \mathbf{e}(0), \qquad (28)$$

where

$$\mathrm{T}_{+} = \mathrm{V} \begin{bmatrix} \mathrm{diag}[e^{k_x \lambda_m^{+} z}] & 0_M \\ 0_M & 0_M \end{bmatrix} \mathrm{V}^{-1}, \qquad (29)$$

while from Eq. (15), we find

$$\mathbf{e}_{-}(0) = \mathrm{T}_{-} \mathbf{e}(z), \qquad (30)$$

where

$$\mathrm{T}_{-} = \mathrm{V} \begin{bmatrix} 0_M & 0_M \\ 0_M & \mathrm{diag}[e^{-k_x \lambda_m^{-} z}] \end{bmatrix} \mathrm{V}^{-1}. \qquad (31)$$

**3 Solution of the linearized equations for a stratified atmosphere**

Consider an equidistant discretization of the atmosphere, i.e., $\widehat{z}_i = z_{\min} + (i - 1)\Delta\widehat{z}$ for $i = 1, ..., 2L + 1$. A layer $l$, where $l = 1, \ldots, L$ and $L$ is the number of layers, is bounded from below and from above by the grid points $z_l = \widehat{z}_{2l-1}$ and $z_{l+1} = \widehat{z}_{2l+1}$, respectively, and its center is located at the grid point $\overline{z}_l = \widehat{z}_{2l}$. The atmosphere extends from $z_{\min} = z_1 = \widehat{z}_1$ to $z_{\max} = z_{L+1} = \widehat{z}_{2L+1} = z_{\min} + L(2\Delta\widehat{z})$. We adopt a numerical multilayer method [9, 15, 16], and approximate the altitude dependent matrix A in each layer $l$ by its value at the layer center, i.e., $\mathrm{A}_l = \mathrm{A}(\overline{z}_l)$. The eigenpairs of the propagation matrix $\mathrm{A}_l$ are denote by $(\lambda_{nl}, \mathbf{v}_{nl})$ for $n = 1, \ldots, N$. The matrix differential equation (5) can be solved either (i) in terms of the amplitudes $\mathbf{a}_l$, $l = 1, \ldots, L$ of the characteristic solutions, or (ii) in terms of the grid-point values $\mathbf{e}_l = \mathbf{e}(z_l)$, $l = 1, \ldots, L$ of the state vector $\mathbf{e}(z)$. In the following we present the method based on the amplitude of the characteristic solutions, whereas the second method is described in Appendix C.

In the layers $l$ and $l + 1$, the solutions are given by (cf. Eq. (11))

$$\mathbf{e}_l(z) = \mathrm{V}_l \mathrm{diag}[e^{k_x \lambda_{nl}(z - z_l)}] \mathbf{a}_l, \ \ z_l \leq z \leq z_{l+1}, \qquad (32)$$

and
$$\mathbf{e}_{l+1}(z) = V_{l+1}\text{diag}[e^{k_x\lambda_{n,l+1}(z-z_{l+1})}]\mathbf{a}_{l+1}, \ \ z_{l+1} \leq z \leq z_l, \tag{33}$$

respectively. The continuity condition at the interface $z = z_{l+1}$,

$$\mathbf{e}_l(z_{l+1}) = \mathbf{e}_{l+1}(z_{l+1}), \tag{34}$$

gives

$$V_l^{-1}V_{l+1}\mathbf{a}_{l+1} = \text{diag}[e^{k_x\lambda_{n,l}\Delta_l}]\mathbf{a}_l, \tag{35}$$

where $\Delta_l = z_{l+1} - z_l$. To obtain a stable system of equations, we define a scaling matrix $K_l^1$ with entries

$$[K_l^1]_{nn} = \begin{cases} e^{-k_x\lambda_{nl}\Delta_l}, & \text{Re}(\lambda_{nl}) > 0 \\ \\ 1, & \text{Re}(\lambda_{nl}) \leq 0 \end{cases}, \tag{36}$$

and a second scaling matrix $K_l^0$ by

$$K_l^0 = K_l^1\text{diag}[e^{k_x\lambda_{nl}\Delta_l}], \text{ i.e., } [K_l^0]_{nn} = \begin{cases} 1, & \text{Re}(\lambda_{nl}) > 0 \\ \\ e^{k_x\lambda_{nl}\Delta_l}, & \text{Re}(\lambda_{nl}) \leq 0 \end{cases}. \tag{37}$$

Multiplying Eq. (35) from the left with $K_l^1$ yields the continuity equation

$$\mathbb{A}_{l,l+1}^1\mathbf{a}_{l+1} - \mathbb{A}_{l,l+1}^0\mathbf{a}_l = \mathbf{0}_{2M}, \ \ l = 1, ..., L-1, \tag{38}$$

where $\mathbf{0}_{2M}$ is the $2M$-dimensional zero vector, and

$$\mathbb{A}_{l,l+1}^1 = K_l^1(V_l^{-1}V_{l+1}), \tag{39}$$
$$\mathbb{A}_{l,l+1}^0 = K_l^0. \tag{40}$$

The scaling matrices $K_l^1$ and $K_l^0$ prevent a possible blow-up of the exponential terms for $\text{Re}(\lambda_{nl}) > 0$ and $\text{Re}(\lambda_{nl}) \leq 0$, respectively. Such scaling techniques are standard in radiative transfer theory and are commonly used to obtain stable numerical algorithms for computing the radiance field in multilayered atmospheres [29, 30].

Actually, we have $L-1$ continuity equations imposed at the levels $z_2, \ldots, z_L$ for the $L$ unknowns $\mathbf{a}_1, \ldots, \mathbf{a}_L$. The two missing equations are obtained from the lower and upper boundary conditions.

1. At the lower boundary, i.e., at $z = z_1(= z_{\min})$, we assume that only the ascending wave modes transport energy upward. In this regard, we impose that in the layer $l = 1$, we have $a_{1,l=1}^+ = s = $ finite, and that the rest of $a_{m,l=1}^+$ are zero, that is, $a_{m,l=1}^+ = 0$ for $m \neq 1$ [9]. Note that $a_{1,l=1}^+$ is the amplitude of the ascending gravity-wave modes, while the condition

$a^+_{m,l=1} = 0$ for $m \neq 1$ means that the amplitudes of the ascending viscosity-wave and thermal conduction-wave modes are assumed to be zero. In this case, the boundary condition for ascending modes is

$$\mathbf{e}^+_{l=1}(z_1) = \begin{bmatrix} \widehat{u}^+_{l=1}(z_1) \\ \widehat{w}^+_{l=1}(z_1) \\ \widehat{T}^+_{l=1}(z_1) \\ \widehat{\mathcal{U}}^+_{l=1}(z_1) \\ \widehat{\mathcal{W}}^+_{l=1}(z_1) \\ \widehat{\mathcal{T}}^+_{l=1}(z_1) \end{bmatrix} = \sum_{m=1}^{M} a^+_{m,l=1}\mathbf{v}^+_{m,l=1} = s\mathbf{v}^+_{1,l=1}. \qquad (41)$$

Excluding for the moment the scale factor $s$, we express the boundary condition for amplitudes,

$$\mathbf{a}^+_{l=1} = \begin{bmatrix} a^+_{1,l=1} \\ a^+_{2,l=1} \\ \vdots \\ a^+_{M,l=1} \end{bmatrix} = \mathbf{i}_1 \text{ with } \mathbf{i}_1 = \begin{bmatrix} 1 \\ 0 \\ \vdots \\ 0 \end{bmatrix}, \qquad (42)$$

in matrix form as

$$[\mathbf{I}_M, \mathbf{0}_M]\mathbf{a}_1 = [\mathbf{I}_M, \mathbf{0}_M]\begin{bmatrix} \mathbf{a}^+_1 \\ \mathbf{a}^-_1 \end{bmatrix} = \mathbf{i}_1, \qquad (43)$$

where in general, $\mathbf{a}^{\pm}_{l_0} = \mathbf{a}^{\pm}_{l=l_0}$, for $l_0 = 1, \ldots, L$. The boundary condition (41) is a modal (eigenvector-based) boundary condition, which imposes that the state at $z_1$ is exactly aligned with a chosen eigenmode. In this way, a pure normal mode is injected into the system.

2. A reasonable upper boundary condition is that there is no downgoing energy at great altitudes, so that the amplitudes of all descending wave modes must be zero at the upper boundary [9]. In this regard, we impose $a^-_{m,l=L} = 0$ for all $m = 1, \ldots, M$, in which case, in the layer $L$, the boundary condition for descending modes is

$$\mathbf{e}^-_{l=L}(z) = \sum_{m=1}^{M} a^-_{m,l=L}\mathbf{e}^{k_\mathbf{x}\lambda^-_{m,l=L}z}\mathbf{v}^-_{m,l=L} = \mathbf{0}_{2M} \qquad (44)$$

for all $z_L \leq z \leq z_{L+1}$. In matrix form, the boundary condition for amplitudes

$$\mathbf{a}^-_{l=L} = \begin{bmatrix} a^-_{1,l=L} \\ a^-_{2,l=L} \\ \vdots \\ a^-_{M,l=L} \end{bmatrix} = \mathbf{0}_M \qquad (45)$$

is written as

$$[\mathbf{0}_M, \mathbf{I}_M]\mathbf{a}_L = [\mathbf{0}_M, \mathbf{I}_M]\begin{bmatrix} \mathbf{a}^+_L \\ \mathbf{a}^-_L \end{bmatrix} = \mathbf{0}_M. \qquad (46)$$

Comments.

1. The scaling matrices defined by Eqs. (36) and (37) do not take into account a classification of the wave modes as ascending and descending (as defined by Eq. (20)). Consequently, the continuity equations (38) do not account for this classification, and the only equations in which it is necessary to distinguish between ascending and descending modes are the boundary condition equations (43) and (46). From this point of view, the method is similar to finite-difference methods [37, 38, 39].

2. An alternative type of lower boundary condition was proposed by Knight et al. [15, 16]. In this approach, the lower boundary condition for ascending modes is prescribed in terms of $M$ values $b_{1,k}$, $k = 1, \ldots, M$, according to (compare with Eq. (41))

$$\left[ \frac{\mathrm{d}^{k-1} \mathbf{e}_{l=1}^{+}}{\mathrm{d}z^{k-1}}(z_1) \right]_q = b_{1,k}, \quad k = 1, ..., M, \tag{47}$$

where the notation $[\mathbf{x}]_q$ denotes the $q$th component of the vector $\mathbf{x}$. In the present context, this refers to the first $M$ components, corresponding to $\widehat{u}$ ($q = 1$), $\widehat{w}$ ($q = 2$), and $\widehat{T}$ ($q = 3$). In Eq. 47, $k$ denotes the derivative order, and in the case $M = 3$, we have explicitly,

$$\left[ \mathbf{e}_{l=1}^{+}(z_1) \right]_q = b_{1,1}, \quad \left[ \frac{\mathrm{d}\mathbf{e}_{l=1}^{+}}{\mathrm{d}z}(z_1) \right]_q = b_{1,2}, \quad \left[ \frac{\mathrm{d}^2 \mathbf{e}_{l=1}^{+}}{\mathrm{d}z^2}(z_1) \right]_q = b_{1,3}. \tag{48}$$

Note that Eq. 47 generalizes Eq. (2.19) in Ref. [15], which is formulated for the first state variable rather than for an arbitrary state variable. Using the relations

$$\frac{\mathrm{d}^{k-1} \mathbf{e}_{l=1}^{+}}{\mathrm{d}z^{k-1}}(z_1) = \sum_{m=1}^{M} a_{m,l=1}^{+} (k_{\mathrm{x}} \lambda_{m,l=1}^{+})^{k-1} \mathbf{v}_{m,l=1}^{+}, \quad k = 1, \ldots, M, \tag{49}$$

and

$$\left[ \frac{\mathrm{d}^{k-1} \mathbf{e}_{l=1}^{+}}{\mathrm{d}z^{k-1}}(z_1) \right]_q = \widehat{\mathbf{i}}_q^T \frac{\mathrm{d}^{k-1} \mathbf{e}_{l=1}^{+}}{\mathrm{d}z^{k-1}}(z_1) = \sum_{m=1}^{M} a_{m,l=1}^{+} (k_{\mathrm{x}} \lambda_{m,l=1}^{+})^{k-1} \widehat{\mathbf{i}}_q^T \mathbf{v}_{m,l=1}^{+},$$
$$\tag{50}$$

where $\widehat{\mathbf{i}}_q$ is a $2M$-dimensional vector with components (compare with Eq. (42))

$$[\widehat{\mathbf{i}}_q]_k = \begin{cases} 1, & k = q \\ 0, & k \neq q \end{cases}, \quad k = 1, \ldots, 2M, \tag{51}$$

we find

$$\sum_{m=1}^{M} [\mathsf{B}]_{mk} a_{m,l=1}^{+} = b_{1,k}, \quad k = 1, ...M, \tag{52}$$

where B is a matrix with entries

$$[\mathsf{B}]_{mk} = (k_{\mathrm{x}}\lambda^+_{m,l=1})^{k-1}\widehat{\mathbf{i}}^T_q \mathbf{v}^+_{m,l=1}, \ \ m,k = 1,\dots,M. \tag{53}$$

Setting $\mathbf{b}_1 = [b_{1,1},\dots,b_{1,M}]^{\mathrm{T}}$, we consider the boundary condition for amplitudes

$$\mathbf{a}^+_1 = \mathsf{B}^{-1}\mathbf{b}_1, \tag{54}$$

that is (compare with Eq. (43))

$$[\mathsf{I}_M,0_M]\mathbf{a}_1 = \mathsf{B}^{-1}\mathbf{b}_1. \tag{55}$$

For the choice $b_{1,k} = 0$ with $k \geq 2$, the first component of the boundary-value vector $\mathbf{b}_1$ can be identified with the scale factor $s$, that is, $s = b_{1,1}$. Consequently, for a unit scale factor and $M = 3$, we have $\mathbf{b}_1 = [1,0,0]^{\mathrm{T}}$. The boundary condition (48) is a localized condition that prescribes the value of a single state variable while enforcing vanishing slope and curvature at the boundary. It effectively acts as an external driver applied to one variable and is appropriate for non-harmonic source functions. Note that this form of the lower boundary condition is used in the statement of Theorem 1 in Ref. [16]. For causality considerations, boundary conditions must be expressed in terms of state variables rather than modal amplitudes, since modes are defined in the frequency domain.

3. The eigenvectors are not uniquely defined and may be scaled by an arbitrary nonzero complex factor. When the LAPACK routine ZGEEV is used, the eigenvectors are returned with a built-in normalization, namely unit Euclidean norm together with a fixed phase convention. In the present work, we follow Knight et al. [15, 16] and apply a component-wise normalization, i.e.,

$$[\mathbf{v}_n]_j = \frac{1}{|[\mathbf{v}_n]_q|}[\mathbf{v}_n]_j, \ \ j = 1,\dots,N,$$

in which each eigenvector is rescaled such that a selected reference component has unit magnitude ($|[\mathbf{v}_n]_q| = 1$). This reference component is chosen to correspond to a boundary value, thereby fixing the overall amplitude of the eigenmode in a manner consistent with the imposed boundary conditions.

Starting from the continuity equation (38), we will determine the amplitudes $\mathbf{a}_l$ by using two solution methods, namely, (i) the so-called global matrix method with matrix exponential and (ii) the scattering matrix method.

**3.1 Global matrix method with matrix exponential**

The continuity equations (38), and the boundary conditions (43) and (46) for a unit scale factor, are assembled into a system of equations for the stratified atmosphere, i.e.,

$$\mathbb{A}\mathbf{a} = \mathbf{b}, \tag{56}$$

where

$$
\mathbb{A} = \begin{bmatrix} [0_M, \mathrm{I}_M] & 0 & \dots & 0 & 0 \\ \mathbb{A}_{L-1,L}^1 & -\mathbb{A}_{L-1,L}^0 & \dots & 0 & 0 \\ \vdots & \vdots & \ddots & \vdots & \vdots \\ 0 & 0 & \dots & \mathbb{A}_{12}^1 & -\mathbb{A}_{12}^0 \\ 0 & 0 & \dots & 0 & [\mathrm{I}_M, 0_M] \end{bmatrix}, \tag{57}
$$

$$
\mathbf{a} = \begin{bmatrix} \mathbf{a}_L \\ \mathbf{a}_{L-1} \\ \vdots \\ \mathbf{a}_2 \\ \mathbf{a}_1 \end{bmatrix}, \text{ and } \mathbf{b} = \begin{bmatrix} \mathbf{0}_M \\ \mathbf{0}_{2M} \\ \vdots \\ \mathbf{0}_{2M} \\ \mathbf{i}_1 \end{bmatrix}. \tag{58}
$$

For the lower boundary condition (55), $\mathbf{i}_1$ in Eq. (58) should be replaced by $\mathrm{B}^{-1}\mathbf{b}_1$, where, for a unit scale factor, $\mathbf{b}_1 = [1, 0, 0]^{\mathrm{T}}$. The matrix $\mathbb{A}$ has $3M - 1$ sub- and superdiagonals (excluding the main diagonal) and can therefore be stored in banded form and treated using standard band-matrix techniques. To solve the resulting banded system of linear equations, we employed the LA-PACK routines ZGBTRF and ZGBTRS. The routine ZGBTRF performs an LU factorization with partial pivoting of the complex band matrix, and ZGBTRS subsequently uses this factorization to solve the linear system for the prescribed right-hand side. In this approach, the inverse of the full system matrix is not computed explicitly, which improves the computational efficiency.

After solving Eq. (56), we compute the state vector as

$$
\mathbf{e}_l = \mathbf{e}(z_l) = \begin{bmatrix} \widehat{u}(z_l) \\ \widehat{w}(z_l) \\ \widehat{T}(z_l) \\ \widehat{\mathcal{U}}(z_l) \\ \widehat{\mathcal{W}}(z_l) \\ \widehat{\mathcal{T}}(z_l) \end{bmatrix} = \mathrm{V}_l \mathbf{a}_l, \ \ l = 1, \dots, L, \tag{59}
$$

and the wave amplitudes by means of the relation

$$
\overline{f}(z) = C(z)\widehat{f}(z), \tag{60}
$$

where $f$ stands for $u$, $w$, and $T$. The ascending and descending solution modes are computed by using Eq. (27), that is,

$$
\mathbf{e}_l^{\pm} = \mathrm{V}_l^{\pm} \mathbf{a}_l^{\pm}, \ l = 1, \dots, L. \tag{61}
$$

**3.2 Scattering matrix method**

We consider the continuity equation (38) and partition the matrices $\mathbb{A}_{l,l+1}^i$, with $i = 0, 1$, as

$$
\mathbb{A}_{l,l+1}^i = \begin{bmatrix} [\mathbb{A}_{l,l+1}^i]_{11} & [\mathbb{A}_{l,l+1}^i]_{12} \\ [\mathbb{A}_{l,l+1}^i]_{21} & [\mathbb{A}_{l,l+1}^i]_{22} \end{bmatrix}. \tag{62}
$$

Further, we define the scattering matrix at the interface between the layers $l$ and $l+1$ (in fact, at the layer grid point $z_{l+1}$), $S_{l,l+1}$ through the relation

$$\begin{bmatrix} \mathbf{a}_l^- \\ \mathbf{a}_{l+1}^+ \end{bmatrix} = S_{l,l+1} \begin{bmatrix} \mathbf{a}_l^+ \\ \mathbf{a}_{l+1}^- \end{bmatrix}, \tag{63}$$

where

$$S_{l,l+1} = \begin{bmatrix} R_{l,l+1}^+ & T_{l,l+1}^- \\ T_{l,l+1}^+ & R_{l,l+1}^- \end{bmatrix}, \tag{64}$$

and $R_{l,l+1}^\pm$ and $T_{l,l+1}^\pm$ with $\dim(R_{l,l+1}^\pm) = \dim(T_{l,l+1}^\pm) = M \times M$, are the reflection and transmission matrices, respectively. In analogy with radiative transfer theory (e.g., Refs. [31, 32]), Eq. (63) is referred to as the interaction principle equation at the interface $(l, l+1)$. It shows that the scattering matrix $S_{l,l+1}$ relates the amplitudes $\mathbf{a}_l^-$ and $\mathbf{a}_{l+1}^+$ of the waves leaving the interface with the amplitudes $\mathbf{a}_l^+$ and $\mathbf{a}_{l+1}^-$ of the waves entering the interface. From Eqs. (38) and (63), we find

$$\begin{bmatrix} R_{l,l+1}^+ & T_{l,l+1}^- \\ T_{l,l+1}^+ & R_{l,l+1}^- \end{bmatrix} = \begin{bmatrix} [\mathbb{A}_{l,l+1}^0]_{12} & -[\mathbb{A}_{l,l+1}^1]_{11} \\ [\mathbb{A}_{l,l+1}^0]_{22} & -[\mathbb{A}_{l,l+1}^1]_{21} \end{bmatrix}^{-1} \begin{bmatrix} -[\mathbb{A}_{l,l+1}^0]_{11} & [\mathbb{A}_{l,l+1}^1]_{12} \\ -[\mathbb{A}_{l,l+1}^0]_{21} & [\mathbb{A}_{l,l+1}^1]_{22} \end{bmatrix}. \tag{65}$$

We organize the computational process as an upward recurrence using the concept of a "stack". The stack $\mathcal{S}_{l_0 l}$ with $l_0 < l$, is a group of interfaces characterized by the interaction principle equation

$$\begin{bmatrix} \mathbf{a}_{l_0}^- \\ \mathbf{a}_l^+ \end{bmatrix} = \begin{bmatrix} \mathcal{R}_{l_0 l}^+ & \mathcal{T}_{l_0 l}^- \\ \mathcal{T}_{l_0 l}^+ & \mathcal{R}_{l_0 l}^- \end{bmatrix} \begin{bmatrix} \mathbf{a}_{l_0}^+ \\ \mathbf{a}_l^- \end{bmatrix}, \tag{66}$$

where the matrices $\mathcal{R}_{l_0 l}^\pm$ and $\mathcal{T}_{l_0 l}^\pm$ are obtained through a successive application of the interaction principle equation at the interfaces $(l_0, l_0+1)$, $(l_0+1, l_0+2)$,...,$(l-1, l)$. Adding a new layer $l+1$, and taking into account that at the interface $(l, l+1)$, the reflection and transmission matrices are $R_{l,l+1}^\pm$ and $T_{l,l+1}^\pm$, respectively, we find that the interaction principle equation for the stack $\mathcal{S}_{l_0,l+1}$, is

$$\begin{bmatrix} \mathbf{a}_{l_0}^- \\ \mathbf{a}_{l+1}^+ \end{bmatrix} = \begin{bmatrix} \mathcal{R}_{l_0,l+1}^+ & \mathcal{T}_{l_0,l+1}^- \\ \mathcal{T}_{l_0,l+1}^+ & \mathcal{R}_{l_0,l+1}^- \end{bmatrix} \begin{bmatrix} \mathbf{a}_{l_0}^+ \\ \mathbf{a}_{l+1}^- \end{bmatrix}, \tag{67}$$

where $\mathcal{R}_{l_0,l+1}^\pm$ and $\mathcal{T}_{l_0,l+1}^\pm$ are computed recursively by using of the "adding formulas"

$$\mathcal{R}_{l_0,l+1}^+ = \mathcal{R}_{l_0 l}^+ + \mathcal{T}_{l_0 l}^- (I - R_{l,l+1}^+ \mathcal{R}_{l_0 l}^-)^{-1} R_{l,l+1}^+ \mathcal{T}_{l_0 l}^+, \tag{68}$$

$$\mathcal{T}_{l_0,l+1}^- = \mathcal{T}_{l_0 l}^- (I - R_{l,l+1}^+ \mathcal{R}_{l_0 l}^-)^{-1} T_{l,l+1}^-, \tag{69}$$

$$\mathcal{T}_{l_0,l+1}^+ = T_{l,l+1}^+ (I - \mathcal{R}_{l_0 l}^- R_{l,l+1}^+)^{-1} \mathcal{T}_{l_0 l}^+, \tag{70}$$

$$\mathcal{R}_{l_0,l+1}^- = R_{l,l+1}^- + T_{l,l+1}^+ (I - \mathcal{R}_{l_0 l}^- R_{l,l+1}^+)^{-1} \mathcal{R}_{l_0 l}^- T_{l,l+1}^-, \tag{71}$$

for $l = l_0+1, ..., L-1$. Note that Eqs. (68)–(71) are mathematically equivalent to Eqs. (4.30)–(4.33) in Ref. [16]. The procedure is initialized with $\mathcal{R}_{l_0,l_0+1}^\pm =$

$\mathrm{R}^{\pm}_{l_0,l_0+1}$ and $\mathcal{T}^{\pm}_{l_0,l_0+1} = \mathrm{T}^{\pm}_{l_0,l_0+1}$, and is repeated until the last interface is added to the stack. For the stack $\mathcal{S}_{1L}$, the interaction principle equation is

$$\begin{bmatrix} \mathbf{a}^-_1 \\ \mathbf{a}^+_L \end{bmatrix} = \begin{bmatrix} \mathcal{R}^+_{1L} & \mathcal{T}^-_{1L} \\ \mathcal{T}^+_{1L} & \mathcal{R}^-_{1L} \end{bmatrix} \begin{bmatrix} \mathbf{a}^+_1 \\ \mathbf{a}^-_L \end{bmatrix}, \tag{72}$$

and from the boundary conditions for amplitudes (42) and (45), that is, from the relations $\mathbf{a}^+_1 = \mathbf{i}_1$ and $\mathbf{a}^-_L = \mathbf{0}_M$, respectively, we find

$$\mathbf{a}^-_1 = \mathcal{R}^+_{1L}\mathbf{a}^+_1 \text{ and } \mathbf{a}^+_L = \mathcal{T}^+_{1L}\mathbf{a}^+_1. \tag{73}$$

For the lower boundary condition (55), $\mathbf{a}^+_1$ in Eq. (73) is given by $\mathbf{a}^+_1 = \mathrm{B}^{-1}\mathbf{b}_1$, where, for a unit scale factor, $\mathbf{b}_1 = [1,0,0]^{\mathrm{T}}$. To restore the entire set of amplitude vectors $\mathbf{a}_l$, we consider the interaction principle equations for the stacks $\mathcal{S}_{1l}$ and $\mathcal{S}_{lL}$, yielding

$$\mathbf{a}^+_l = (I - \mathcal{R}^-_{1l}\mathcal{R}^+_{lL})^{-1}\mathcal{T}^+_{1l}\mathbf{a}^+_1, \tag{74}$$

$$\mathbf{a}^-_l = \mathcal{R}^+_{lL}\mathbf{a}^+_l, \tag{75}$$

for $l = L-1, ..., 1$. The state vector and the wave amplitudes are then computed by using Eqs. (59) and (60), respectively. In contrast to the previous method, this approach requires a clear differentiation between ascending and descending modes as defined by Eq. (20).

**4   Source function**

In the derivation so far, the amplitude vector is uniquely defined up to a multiplicative factor, namely the scale factor $s$. Accordingly, the general solution can be written as $\mathbf{a}_{\mathrm{s}} = s\mathbf{a}$, where, here and it what follows, the subscript s indicates the dependence on $s$. Since $\mathbf{a}_{\mathrm{s}}$ satisfies the equation $\mathbb{A}\mathbf{a}_{\mathrm{s}} = s\mathbf{b}$ (cf. Eq. (56)), the scale factor can be interpreted as a source factor. The source factor is constant in the case of a harmonic (monochromatic) source function, corresponding to a single-frequency wave, but is time dependent for a non-harmonic source function, corresponding to a time-dependent wave packet. In this section, we describe the computation of the perturbed quantities for both harmonic and non-harmonic source functions. We also present a brief overview of the causality condition and the imaginary frequency shift introduced by Knight et al. [16], and latter extended and applied in Refs. [11, 15, 17, 34]. Although it would be sufficient to simply refer to these works, we include a short discussion here because the underlying mathematical structure provides valuable insight into the method.

**4.1   Non-harmonic source (time-dependent wave packet)**

If the source term is not purely harmonic in time (i.e., it cannot be written as a single factor $\exp(\mathrm{j}\omega t)$), the perturbed quantity $f'(x, z, t)$ is not a single-frequency wave with a specified angular frequency $\omega$. In this case, the equations

are treated in the frequency domain by considering the Fourier transform in time [15, 16, 17]. This is defined by

$$F'(x, z, \omega) = \int_{-\infty}^{\infty} f'(x, z, t) e^{-j\omega t} dt = \mathcal{F}[f'(x, z, t)](x, z, \omega) \qquad (76)$$

and its inverse by

$$f'(x, z, t) = \frac{1}{2\pi} \int_{-\infty}^{\infty} F'(x, z, \omega) e^{j\omega t} d\omega = \mathcal{F}^{-1}[F'(x, z, \omega)](x, z, t). \qquad (77)$$

Applying the Fourier transform to the linearized equations (145)–(147) of Appendix A, using the result

$$\mathcal{F}\left[\frac{\partial f'}{\partial t}(x, z, t)\right](x, z, \omega) = j\omega F'(x, z, \omega), \qquad (78)$$

and setting
$$F'(x, z, \omega) = \overline{F}(z, \omega) e^{-jk_x x} \qquad (79)$$

as the counterpart of Eq. (148) (in which the exponential term $\exp(j\omega t)$ is absorbed into $\overline{f}(z)$), together with

$$\overline{F}(z, \omega) = C(z)\widehat{F}(z, \omega) \qquad (80)$$

as the counterpart of Eq. (7), we are led to the system of differential equations (157)–(162) of Appendix A (or equivalently, to the matrix differential equation (5)), but with $\widehat{F}(z, \omega)$ replacing $\widehat{f}(z)$.

At the lower boundary $z_1$, we consider the localized boundary conditions

$$f'_{sq_0}(x, z_1, t) = C_{q_0}(z_1)s(x, t), \quad \frac{\partial f'_{sq_0}}{\partial z}(x, z_1, t) = 0, \quad \frac{\partial^2 f'_{sq_0}}{\partial z^2}(x, z_1, t) = 0, \quad (81)$$

where $q_0$ takes the values 1, 2, and 3 for the horizontal velocity, vertical velocity, and temperature, respectively. In Eq. (81), the source function is given by

$$s(x, t) = As(t) e^{-jk_x x}, \qquad (82)$$

with $A$ denoting the scalar source amplitude and $s(t)$ its prescribed time dependence. Here, and in what follows, the index s s is used to indicate that a quantity depends on the source function. In our implementation, the time-dependent part of the source function is chosen as

$$s(t) = e^{j\omega_0(t-t_0)} e^{-\dfrac{(t-t_0)^2}{2\sigma_t^2}} \qquad (83)$$

with the Fourier transform

$$S(\omega) = \frac{\sqrt{2\pi}}{\sigma_\omega} e^{-j\omega t_0} e^{-\dfrac{(\omega-\omega_0)^2}{2\sigma_\omega^2}}, \qquad (84)$$

where $\omega_0$ is the reference frequency (the central frequency in the Fourier spectrum), $t_0$ is the time at which the source function is maximum, and $\sigma_{\mathrm{t}}$ and $\sigma_\omega = 1/\sigma_{\mathrm{t}}$ are the standard deviations in the time and frequency domains, respectively. The amplitude of the source function $A$ is specified by imposing the normalization condition:

$$|\mathrm{Re}\{f'_{sq_0}(x=0, z_1, t_0)\}| = f_{\mathrm{b}q_0}, \tag{85}$$

for some prescribed boundary value $f_{\mathrm{b}q_0} > 0$. For example, in the case $q_0 = 1$, $f_{\mathrm{b}q_0}$ may be chosen as a fraction of the maximum horizontal velocity of neutrals in the south direction over the altitude range, whereas in the case $q_0 = 3$, $f_{\mathrm{b}q_0}$ may be chosen as a fraction of the maximum temperature of neutrals over the altitude range.

Applying the Fourier transform to Eq. (81) and using Eqs. (79) and (80), we obtain the following boundary conditions in the frequency domain (note that $AS(\omega)$ is the Fourier transform of $As(t)$):

$$\widehat{F}_{sq_0}(z_1, \omega) = AS(\omega), \quad \frac{\partial \widehat{F}_{sq_0}}{\partial z}(z_1, \omega) = 0, \quad \frac{\partial^2 \widehat{F}_{sq_0}}{\partial z^2}(z_1, \omega) = 0. \tag{86}$$

Comparing Eqs. (86) and (48), we see that the latter corresponds to the choice $b_{1,2} = b_{1,3} = 0$. In this case, the source factor $s = b_{1,1}$ can be identified with $AS(\omega)$. Therefore, as in Section 3, we define $\widehat{F}_q(z, \omega) = [\mathbf{e}(z, \omega)]_q$, $q = 1, 2, 3$, where $\mathbf{e}(z, \omega)$ denotes the solution of the differential equation (5) for a unit source factor in the frequency domain (i.e., for $\mathbf{b}_1 = [1, 0, 0]^{\mathrm{T}}$). The perturbed quantity $f'_{sq}(x, z, t)$ is then obtained by applying the inverse transform (77) to

$$F'_{sq}(x, z, \omega) = AS(\omega)\mathrm{e}^{-\mathrm{j}k_{\mathrm{x}}x}C_q(z)\widehat{F}_q(z, \omega), \tag{87}$$

that is,

$$f'_{sq}(x, z, t) = \frac{1}{2\pi}\int_{-\infty}^{\infty} F'_{sq}(x, z, \omega)\mathrm{e}^{\mathrm{j}\omega t}\mathrm{d}\omega = A\overline{f}_q(z, t)\mathrm{e}^{-\mathrm{j}k_{\mathrm{x}}x}, \tag{88}$$

where

$$\overline{f}_q(z, t) = \frac{C_q(z)}{2\pi}\int_{-\infty}^{\infty} S(\omega)\widehat{F}_q(z, \omega)\mathrm{e}^{\mathrm{j}\omega t}\mathrm{d}\omega. \tag{89}$$

For $S(\omega)$ as above, $\overline{f}_q(z, t)$ can be written as

$$\overline{f}_q(z, t) = \frac{C_q(z)}{2\pi}\int_{-\infty}^{\infty} \mathscr{S}(\omega)\widehat{F}_q(z, \omega)\mathrm{e}^{\mathrm{j}\omega(t - t_0)}\mathrm{d}\omega, \tag{90}$$

with

$$\mathscr{S}(\omega) = \frac{\sqrt{2\pi}}{\sigma_\omega}\mathrm{e}^{-\frac{(\omega - \omega_0)^2}{2\sigma_\omega^2}}. \tag{91}$$

The computation of $\widehat{F}_q(z,\omega)$ can be performed using any of the methods presented in Section 3. The computed quantity is $\overline{f}_q(z,t)$, the amplitude $A > 0$, is determined from the normalization condition (85) as

$$A = \frac{f_{\mathrm{b}q_0}}{|\mathrm{Re}\{\overline{f}_{q_0}(z_1, t_0)\}|}, \tag{92}$$

and the perturbed quantity $f'_{\mathrm{s}q}(x, z, t)$ is computed from Eq. (88).

**4.2 Monochromatic source (single-frequency wave)**

The case of a monochromatic source is obtained as a special case of the above approach by choosing

$$s(t) = \mathrm{e}^{\mathrm{j}\omega_0 t}, \tag{93}$$

whose Fourier transform is

$$S(\omega) = \int_{-\infty}^{\infty} s(t)\,\mathrm{e}^{-\mathrm{j}\omega t}\,\mathrm{d}t = \int_{-\infty}^{\infty} \mathrm{e}^{-\mathrm{j}(\omega-\omega_0)t}\,\mathrm{d}t = 2\pi\delta(\omega - \omega_0), \tag{94}$$

where the equality is understood in the sense of distributions.

The lower boundary conditions in the time domain are specified as

$$f'_{\mathrm{s}q_0}(x, z_1, t) = \frac{1}{2\pi} C_{q_0}(z_1)\, s(x, t), \quad \frac{\partial f'_{\mathrm{s}q_0}}{\partial z}(x, z_1, t) = 0, \quad \frac{\partial^2 f'_{\mathrm{s}q_0}}{\partial z^2}(x, z_1, t) = 0, \tag{95}$$

with

$$s(x, t) = As(t)\mathrm{e}^{-\mathrm{j}k_{\mathrm{x}}x} = A\mathrm{e}^{\mathrm{j}\omega_0 t}\,\mathrm{e}^{-\mathrm{j}k_{\mathrm{x}}x}. \tag{96}$$

Applying the Fourier transform with respect to time yields

$$\widehat{F}_{\mathrm{s}q_0}(z_1, \omega) = A\,\delta(\omega - \omega_0). \tag{97}$$

Since the forcing is monochromatic, the frequency-domain problem is solved only at the excitation frequency $\omega = \omega_0$. Accordingly, the localized boundary conditions are imposed directly on the harmonic amplitudes at $\omega_0$, namely

$$\widehat{F}_{\mathrm{s}q_0}(z_1, \omega_0) = A, \quad \frac{\partial \widehat{F}_{\mathrm{s}q_0}}{\partial z}(z_1, \omega_0) = 0, \quad \frac{\partial^2 \widehat{F}_{\mathrm{s}q_0}}{\partial z^2}(z_1, \omega_0) = 0. \tag{98}$$

Again, by comparing Eqs. (98) and (48), we see that the source factor $s = b_{1,1}$ can be identified with $A$. In this regard, let $\widehat{F}_q(z, \omega_0) = \big[\mathbf{e}(z, \omega_0)\big]_q$, $q = 1, 2, 3$, where $\mathbf{e}(z, \omega_0)$ denotes the solution of the differential equation (5) for a unit source factor in the frequency domain. The perturbed quantity $f'_{\mathrm{s}q}(x, z, t)$ is then obtained by the inverse Fourier transform (77) of $F'_{\mathrm{s}q}(x, z, \omega)$ (cf. Eq. (87)), and the result is

$$f'_{\mathrm{s}q}(x, z, t) = A\,\overline{f}_{\mathrm{s}q}(z)\mathrm{e}^{\mathrm{j}(\omega_0 t - k_{\mathrm{x}}x)}, \tag{99}$$

where

$$\overline{f}_{sq}(z) = C_q(z)\,\widehat{F}_q(z,\omega_0).\tag{100}$$

Thus, the computed quantity is $\overline{f}_{sq}(z)$, and the amplitude $A$ is determined from the normalization condition

$$|\mathrm{Re}\{f'_{sq_0}(x=0,z_1,t=0)\}| = f_{bq_0},\tag{101}$$

which yields

$$A = \frac{f_{bq_0}}{|\mathrm{Re}\{\overline{f}_{sq}(z_1)\}|}.\tag{102}$$

Note that for a monochromatic source, the modal boundary condition (41) can be used instead of the localized boundary condition (98).

**4.3 Causality and imaginary frequency shifting**

Causality means that the wave field in response to any source function cannot be nonzero prior to the earliest time at which the source function is nonzero. According to the classification rule (20), we have

$$\mathrm{Re}[\lambda_{1l}^+(\omega)] < \mathrm{Re}[\lambda_{1l}^-(\omega)],\tag{103}$$

for any layer $l = 1,\dots,L$ and any real frequency $\omega$. To preserve causality in solutions of two-point boundary value problems, a stronger condition is required, namely

$$\max_{l=1,\dots,L} \mathrm{Re}[\lambda_{1l}^+(\omega)] < \min_{l=1,\dots,L} \mathrm{Re}[\lambda_{1l}^-(\omega)]\tag{104}$$

for all $\omega \in \mathbb{R}$. Equivalently, this condition requires that there exists a single real constant $\sigma$, such that

$$\mathrm{Re}[\lambda_{1l}^+(\omega)] < \sigma < \mathrm{Re}[\lambda_{1l}^-(\omega)],\tag{105}$$

for all $l$ and all $\omega \in \mathbb{R}$.

In some situations, condition (105) is not satisfied on the real frequency axis but can be enforced by introducing an imaginary frequency shift $\omega \to \omega - \mathrm{j}\delta$. Following Ref. [16], we impose a causality requirement, which we refer to as the Global Causality (GC) condition. This condition demands that there exists a single real constant $\sigma$, such that

$$\mathrm{Re}[\lambda_{1l}^+(\omega - \mathrm{j}\delta)] < \sigma < \mathrm{Re}[\lambda_{1l}^-(\omega - \mathrm{j}\delta)],\tag{106}$$

for all $l$ and all $\omega \in \mathbb{R}$.

Knight et al. [11] subsequently relaxed the requirement that (106) hold for all layers $l$ for a fixed $\sigma$. The new condition, which we refer to as the Layerwise Causality (LC) condition, requires that at each layer $l$ there is a $\sigma_l$ such that

$$\mathrm{Re}[\lambda_{1l}^+(\omega - \mathrm{j}\delta)] < \sigma_l < \mathrm{Re}[\lambda_{1l}^-(\omega - \mathrm{j}\delta)],\tag{107}$$

for all $\omega \in \mathbb{R}$. Equivalently, if at each layer $l$,

$$d_l(\omega) = \mathrm{Re}[\lambda_{1l}^-(\omega - \mathrm{j}\delta)] - \mathrm{Re}[\lambda_{1l}^+(\omega - \mathrm{j}\delta)] > 0, \qquad (108)$$

for all $\omega \in \mathbb{R}$, then the multilayer algorithm will still preserve causality. The LC condition requires a strict separation between the two eigenvalue families within each layer, but it does not require that the same separator works for all layers. Thus, each layer may have its own separating value $\sigma_l$. Equivalently, $d_l(\omega) > 0$ means that, in layer $l$, the real parts of the eigenvalues associated with the ascending and descending gravity waves remain separated (and therefore do not cross) as functions of the real frequency $\omega$ after the shift.

To summarize the approach for computing $f'_{Sq}(x, z, t)$ in the case of imaginary frequency shifting, we introduce the shifted spectrum

$$S_\delta(\omega) = S(\omega - \mathrm{j}\delta) = \int_{-\infty}^{\infty} s(t)\mathrm{e}^{-\mathrm{j}(\omega - \mathrm{j}\delta)t}\mathrm{d}t = \int_{-\infty}^{\infty} \left[ s(t)\mathrm{e}^{-\delta t} \right] \mathrm{e}^{-\mathrm{j}\omega t}\mathrm{d}t, \qquad (109)$$

which can be viewed as the analytic continuation of $S(\omega)$ to complex frequencies. Note that the shift $\omega \to \omega - \mathrm{j}\delta$ corresponds in the time domain to multiplication by $\exp(-\delta t)$. Let

$$F'_{\delta sq}(x, z, \omega) = AS(\omega - \mathrm{j}\delta)C_q(z)\widehat{F}_q(z, \omega - \mathrm{j}\delta)\mathrm{e}^{-\mathrm{j}k_\mathrm{x}x} \qquad (110)$$

be the Fourier transform (in time) of the perturbed quantity with frequency shifting $f'_{\delta sq}(x, z, t)$, where as usual, $\widehat{F}_q(z, \omega - \mathrm{j}\delta) = [\mathbf{e}(z, \omega - \mathrm{j}\delta)]_q$ is solution of the differential equation (5) for a unit source factor in the frequency domain. Under the usual analyticity and decay assumptions (so that contour shifting is permitted), Cauchy's theorem yields the shift relation

$$f'_{\delta sq}(x, z, t) = \frac{1}{2\pi} \int_{-\infty}^{\infty} F'_{\delta sq}(x, z, \omega)\mathrm{e}^{\mathrm{j}\omega t}\mathrm{d}\omega = \mathrm{e}^{-\delta t}f'_{sq}(x, z, t), \qquad (111)$$

where $f'_{sq}$ is the perturbed quantity without frequency shifting given by Eq. (88). Equivalently, this implies the shift-invariance property

$$f'_{sq}(x, z, t) = \mathrm{e}^{\delta t}f'_{\delta sq}(x, z, t). \qquad (112)$$

Summarizing, the computational steps for the frequency-shifting approach are as follows:

1. Compute $\mathbf{e}(z, \omega - \mathrm{j}\delta)$ as the solution of the differential equation (5) for a unit source factor, and set $\widehat{F}_q(z, \omega - \mathrm{j}\delta) = [\mathbf{e}(z, \omega - \mathrm{j}\delta)]_q$.

2. Calculate $F'_{\delta sq}$ by means of Eq. (110) with $S(\omega)$ replaced by $S(\omega - \mathrm{j}\delta)$ and $\widehat{F}_q(z, \omega)$ replaced by $\widehat{F}_q(z, \omega - \mathrm{j}\delta)$.

3. Compute $f'_{\delta sq}$ by inverse Fourier transform

$$f'_{\delta sq}(x, z, t) = A\mathrm{e}^{-\mathrm{j}k_x x}\frac{C_q(z)}{2\pi} \int_{-\infty}^{\infty} S(\omega - \mathrm{j}\delta)\widehat{F}_q(\omega - \mathrm{j}\delta)e^{\mathrm{j}\omega t}\,\mathrm{d}\omega. \qquad (113)$$

4. Recover $f'_{sq}$ from the shift-invariance property (112).

For $S(\omega)$ as in Eq. (84), it is convenient to write the recovered solution in the form

$$f'_{sq}(x,z,t) = A\overline{f}_{\delta q}(z,t)e^{-jk_x x}, \qquad (114)$$

where (compare with Eq. (90))

$$\overline{f}_{\delta q}(z,t) = e^{\delta(t-t_0)}\frac{C_q(z)}{2\pi}\int_{-\infty}^{\infty}\mathscr{S}(\omega - j\delta)\widehat{F}_q(z,\omega - j\delta)e^{j\omega(t-t_0)}\,d\omega, \qquad (115)$$

and $\mathscr{S}$ is given by Eq. (91). The computed quantity is $\overline{f}_{\delta q}(z,t)$, the amplitude $A > 0$ is determined from the normalization condition (85) as

$$A = \frac{f_{bq_0}}{|\text{Re}\{\overline{f}_{\delta q}(z_1,t_0)\}|}, \qquad (116)$$

and the perturbed quantity $f'_{sq}(x,z,t)$ is computed from Eq. (114). The Fourier integral in Eq. (115) is evaluated using a direct discrete Fourier transform (FT) rather than a fast Fourier transform (FFT). The frequency and time discretization used in the Fourier transform are discussed in Appendix D.

A potential numerical issue with this approach is that a large frequency shift $\delta$, while ensuring the causality condition, may amplify rounding errors when recovering $f'_{sq}$ from $f'_{\delta sq}$ through the exponential term $\exp[\delta(t-t_0)]$. For large $t$, this may lead to an uncontrolled growth of the right-hand side of Eq. (115). Therefore, care is required in selecting $\delta$: it must be large enough to ensure the layerwise causality condition , but not significantly larger than that. Rigorous methods for determining the minimum sufficient $\delta$ were described by Knight et al. in Refs. [15, 16, 17], while the numerical blow-up associated with the exponential growth term was discussed in Appendix B of Ref. [17]. In our implementation we employ a heuristic approach that combines (i) the Layerwise Causality (LC) condition applied at selected altitude levels, and (ii) a Source-Function Reconstruction (SFR) test. First, an admissible interval $[\delta_{\min}, \delta_{\max}]$ is constructed by enforcing the SFR criterion, and within this interval, the LC condition is applied at selected altitude levels to obtain a refined lower bound $\delta_{\min}$. In the final selection step, a discrete set of candidate shifts is evaluated, and for each candidate the LC condition is checked over the entire altitude range. Among all shifts that satisfy causality at all altitudes, the algorithm selects the one whose maximum-amplitude vector is closest to the center of mass of the admissible solutions. A detailed description of this approach is provided in Appendix D.

**5 Numerical implementation**

An implementation of the method is freely available as an open-source code on GitHub. The code uses as input the data file produced by the International Reference Ionosphere (IRI) code available at https://ccmc.gsfc.nasa.gov/models/IRI~2016/ is used. From these date, we read

| Location | Latitude [deg] | Longitude [deg] | Height [km] |
|---|---|---|---|
| Jicamarca (Peru) | -12 | 283 | 300 |
| Arecibo (Puerto Rico) | +18 | 293 | 300 |
| Millstone Hill (USA) | +42 | 288 | 300 |
| Saint-Santin (France) | +44 | 2 | 300 |
| EISCAT Tromsø (Auroral) | +70 | 19 | 300 |
| Svalbard archipelago (Norway) | +80 | 15 | 300 |

Table 1: Geographic locations and heights of the IRI data sets

1. the date (year, month, and day) and the time,

2. the geographic latitude and longitude,

3. the magnetic dip angle,

4. the solar radio flux f10.7 and its 81-day average,

5. the number density of $O^+$ ions as a function of altitude.

In its present implementation, the IRI data files correspond to the locations summarized in Table 1. The altitude grid extends from $z_{\min} = 80$ km to $z_{\max} = 500$ km with a step size of $dz = 1.0$ km. Users may generate custom data files by running the IRI code and specifying the corresponding file names in the input namelist.

The IRI data are subsequently used in a manner analogous to that in the SAMI2 model of the Naval Research Laboratory (https://github.com/NRL-Plasma-Physics-Division/SAMI2). In the present implementation, the ionospheric equations follow the SAMI2 framework originally developed by Huba et al. [40] and described in detail by Huba [41]. In SAMI2, the neutral atmospheric parameters–namely the neutral number density, total mass density, and temperature–are specified using the MSIS family of models. In this study, these parameters are based on the MSIS formulation of Hedin [42], while we note that more recent updates are provided by the NRLMSIS 2.0 model of Emmert et al. [43]. The meridional and zonal winds are specified using the Horizontal Wind Model. In the present implementation, we follow the formulation of Hedin et al. [44], while more recent updates are described by Drob et al. [45].

The derivatives of the background parameters are computed using central finite differences. Prior to applying the finite-difference calculations, the background parameters are smoothed by means of cubic spline interpolation with regularization.

Other features of the model are summarized as follows:

1. Two linearization models are included in the code:

   (a) a general model that accounts for the altitude derivatives of the background velocity, temperature, density scale height, and dynamic viscosity; and

(b) a simplified model for an isothermal, homogeneous, and windless atmosphere without ion drag.

2. The methods for solving the linearized equations based on the matrix exponential formalism comprise:

    (a) the Global Matrix Method for the Amplitudes (GMMA) of the characteristic solutions;

    (b) the Scattering Matrix Method for the Amplitudes (SMMA) of the characteristic solutions; and

    (c) the Global Matrix Method for the Nodal (grid-point) values (GMMN) of the state vector.

3. At the lower boundary, we impose that a selected component of the state vector is finite and that its first and second derivatives with respect to height vanish. At the upper boundary, we assume that there is no downward energy propagation, i.e., the amplitudes of all descending wave modes are set to zero.

4. The code first computes the wave parameters for a single-frequency wave and then for a time-dependent wave packet.

5. Typical values of the horizontal wavelength lie in the range 300–700 km.

6. The algorithm computes lower and upper bounds for the wave period by solving the inviscid dispersion equation for two prescribed minimum and maximum values of the vertical wavelength. The computational procedure is described in Appendix D. The user then selects an appropriate value within this range.

7. For a single-frequency wave, the output quantity of interest is $A\overline{f}_{sq}(z)$, where $\overline{f}_{sq}(z)$ and $A$ are given by Eqs. (100) and (102), respectively, whereas for a time-dependent wave packet the corresponding output quantity is $A\overline{f}_{\delta q}(z,t)$, where $\overline{f}_{\delta q}(z,t)$ and $A$ are given by Eqs. (115) and (116), respectively.

**6  Numerical simulations**

The simulations are performed using, as input, an IRI data file corresponding to the EISCAT Tromsø (auroral) location on 11 February 2012 at 10:00. The solar zenith angle is 84.2°, the magnetic inclination angle is 78.28°, the daily solar radio flux F10.7 is 109.4 sfu, and the 81-day averaged solar radio flux is 116.9 sfu, where $1\,\text{sfu} = 10^{-22}\ \text{Wm}^{-2}\text{Hz}^{-1}$. In the simulations, the Prandtl number is 0.66, the magnetic index is 7.0, and the horizontal wind model 14 implemented in SAMI2 is used. The altitude grid extends from 80 km to 500 km and contains 801 grid points. The lower boundary can, in principle, be set to smaller altitudes (e.g., 50 km), but we choose 80 km because this level

is typically adopted as the lower boundary for ionospheric equations. Unless stated otherwise, the horizontal wavelength is $\lambda_x = 400$ km, the wave period is $\lambda_t = 40$ min, and the imaginary frequency shift for a single-frequency wave is $10^{-6}$ s$^{-1}$. The lower boundary conditions are imposed on the vertical velocity with $f_{b2} = 5 \times 10^{-2}$ ms$^{-1}$ in Eqs. (85) and (101).

*Accuracy and efficiency of the solution methods.* Taking the global matrix method for amplitudes as a reference, we find that the relative root-mean-square errors in the perturbed temperature, vertical velocity, and horizontal velocity obtained with the other two solution methods are smaller than $10^{-6}$. Thus, all methods exhibit comparable accuracy. On the other hand, we find that the scattering matrix method is more time-consuming than the global matrix methods, particularly for time-dependent wave packets. This is because the scattering matrix approach requires numerous matrix operations in each layer, whereas solving a system of equations compressed into band storage is computationally less expensive.

*Background atmospheric parameters.* The code provides altitude-dependent profiles of the input parameters used in the numerical model. These include the temperature, mass density, pressure, southward horizontal velocity, atmospheric scale height, density scale height, specific heat capacity, ratio of specific heats, sound speed, number density of O$^+$ ions, neutral–ion collision frequency, ion–neutral collision frequency, and the diffusion velocity. In addition, altitude derivatives of the temperature, horizontal velocity, mass density, pressure, density scale height, and ion number density are also provided. As an illustrative example, Fig. 1 shows the background temperature $T_0$, horizontal velocity $u_0$ and the ion number density $n_{i0}$, together with their corresponding altitude derivatives.

*Pairwise classification of ascending and descending* modes. In the upper and middle panels of Fig. 2, we plot the imaginary part of the vertical wavenumber for ascending ($k_{z1}$) and descending ($k_{z4}$) gravity waves, computed using the general and the simplified models, respectively. The plots demonstrate that only in the latter case do the vertical wavenumbers appear in pairs. In the former case, a problematic altitude range for gravity waves is observed between 80 km and 120 km, where the imaginary parts of the vertical wavenumbers for the ascending and descending modes are nearly identical, being either both positive or both negative. In the lower panel of Fig. 2, we show the imaginary part of the vertical wavenumber $k_z$ for all wave types ($k_{zn}, n = 1, \ldots, 6$), computed using the general model. The plots reveal a clear distinction between gravity waves and viscosity- and thermal-conduction waves. However, the viscosity and thermal-conduction waves are very close to each other. In the upper panel of Fig. 2, we also compare the imaginary part of the vertical wavenumber computed with and without ion drag. No pronounced effect of ion drag on the vertical wavenumber is observed. A small effect appears in the altitude range from 180 to 300 km, where the ion number density is relatively high. This finding is consistent with the results of Shibata [46], who showed that, for gravity waves, plasma diffusion is of minor importance with respect to the vertical wavenumber, which is mainly controlled by dissipation due to viscosity

[Figure]

Figure 1: Background temperature $T_0$, horizontal velocity $u_0$ and ion number density $n_{i0}$ ($i = \mathrm{O}^+$) (upper panels), and their height derivatives (lower panels).

and thermal conduction in the neutral gas.

*General and simplified models.* The altitude profiles of the perturbed temperature, vertical velocity, and horizontal velocity computed using the general and simplified models are shown in Fig. 3. The plots show that, in the altitude range 150–300 km, the amplitudes obtained with the general model are larger than those obtained with the simplified model. If the imaginary parts of the vertical wavenumber for ascending gravity waves computed with the general and simplified models were plotted on the same graph (i.e. by merging the lower and middle panels of Fig. 2 ), it would be seen that the imaginary part of the vertical wavenumber corresponding to the general model is negative but larger (i.e., less negative) than that obtained with the simplified model. As a consequence, the exponential attenuation with altitude is weaker in the general model, leading to systematically larger wave amplitudes in this region. This difference reflects the modified balance between wave propagation and dissipation introduced by the inclusion of altitude-dependent background properties and by relaxing the assumption of constant kinematic viscosity in the general model, which reduces the effective vertical damping compared to the simplified, homogeneous approximation.

*Ion Drag.* The effect of ion drag on the perturbed temperature, vertical velocity, and horizontal velocity is illustrated in Fig. 4. A moderate attenuation is observed in the altitude range from 180 to 350 km, where the ion number density is relatively high. When $\mathbf{E} \times \mathbf{B}$ drifts are not included, ion drag does not exhibit the classical regime in which auroral convection strongly drives the neutral atmosphere. Instead, ion drag mainly arises from diffusion- and pressure-gradient-driven ion motion along the magnetic field, as well as from any relative ion–neutral motion induced by neutral winds. Consequently, ion drag does not constitute the dominant forcing mechanism for the neutral perturbations in this configuration.

*Horizontal wavelength and time period.* The influence of the horizontal wavelength $\lambda_\mathrm{x}$ and the wave period $\lambda_\mathrm{t}$ on the perturbed quantities is shown in Fig. 5. These plots indicate that the wave amplitude decreases with increasing $\lambda_\mathrm{x}$ and $\lambda_\mathrm{t}$. The underlying reasons are as follows:

1. Horizontal wavelength. As the horizontal wavelength increases, horizontal pressure gradients become weaker, which reduces the driving of the wave motion. This also weakens the coupling between horizontal and vertical motions, resulting in smaller gravity-wave amplitudes.

2. Wave period. As the wave period increases, the buoyancy restoring force acts more slowly, leading to weaker oscillations for a given forcing. In addition, dissipative processes such as viscosity and thermal diffusion act more effectively on low-frequency waves, further reducing their amplitudes.

*Computing ascending and descending wave modes.* The ascending and descending solution modes in layer $l$, denoted by $\mathbf{e}_l^+$ and $\mathbf{e}_l^-$, respectively, can be computed using the GMMA through Eq. (61) or using the GMMN via the recurrence relations (269) and (270). The total solution mode $\mathbf{e}_l$, obtained from

[Figure]

Figure 2: Upper panel: The imaginary part of the vertical wavenumber $k_z$ for ascending ($k_{z1}$) and descending ($k_{z4}$) gravity waves, with and without ion drag, computed using the general model. Middle panel: The imaginary part of the vertical wavenumber $k_z$ for ascending ($k_{z1}$) and descending ($k_{z4}$) gravity waves, computed using the simplified model. Lower panel: The imaginary part of the vertical wavenumber $k_z$ for all types of waves ($k_{zn}, n = 1, \ldots, 6$), computed using the general model. The results correspond to $\lambda_x = 400$ km and $\lambda_t = 40$ min

[Figure]

Figure 3: Altitude profiles of the perturbed temperature $\overline{T}$, vertical velocity $\overline{w}$, and horizontal velocity $\overline{u}$ for the general and simplified models. The results correspond to $\lambda_x = 400$ km and $\lambda_t = 40$ min

[Figure]

Figure 4: Altitude profiles of the perturbed temperature $\overline{T}$, vertical velocity $\overline{w}$, and horizontal velocity $\overline{u}$ for the general model, with and without ion drag. The results correspond to $\lambda_x = 400$ km and $\lambda_t = 40$ min

[Figure]

Figure 5: Altitude profiles of the perturbed temperature $\overline{T}$, vertical velocity $\overline{w}$, and horizontal velocity $\overline{u}$ for (i) $\lambda_x = 300$, 400, and 500 km with $\lambda_t = 40$ min (upper panels), and (ii) $\lambda_x = 400$ km with $\lambda_t = 40$, 60, and 80 min (lower panels). The results correspond to the general model with ion drag.

Eq. (59) in the GMMA formulation and by solving Eq. (266) in the GMMN formulation, should satisfy the relation $\mathbf{e}_l = \mathbf{e}_l^+ + \mathbf{e}_l^-$. In all our simulations, this identity is satisfied. Furthermore, the results shown in Fig. 6 indicate that the ascending mode is dominant, except in the altitude range between 120 km and 180 km in the case of the general model. This finding, which is consistent with the results presented by Knight et al. [16] (see their Fig. 6), suggests that in a simplified model one may assume the ascending modes to be dominant at altitudes above 200 km, that is, $\mathbf{e}_l \approx \mathbf{e}_l^+$ for $l = 1, \ldots, L$. Under this assumption, the state vector can be computed using the upward recurrence relation (269). In Ref. [16], this approach was referred to as the transmission-only approximation, whereas in Ref. [17] a related single-mode approximation was introduced.

*Time-dependent wave packet.* For the source function (82)–(83), Fig. 7 shows the perturbed temperature and vertical velocity as functions of time and altitude. Note the different time intervals used for each horizontal wavelength $\lambda_x$ in these plots. The maximum values of the perturbed temperature are 32.21 K, 31.15 K, and 32.73 K for the horizontal wavelengths 300 km, 500 km, and 700 km, respectively, whereas the corresponding maximum values of the vertical velocity are $21.40\,\mathrm{ms}^{-1}$, $16.21\,\mathrm{ms}^{-1}$, and $12.08\,\mathrm{ms}^{-1}$.

*Imaginary frequency shift.* For the time-dependent wave packet, we choose the minimum and maximum values of the imaginary frequency shift as $\delta_{\min} = 10^{-6}$ s$^{-1}$ and $\delta_{\max} = 10^{-4}$ s$^{-1}$, respectively, and set the discrete step to $\Delta\delta = \delta_{\min}$. Referring to Appendix D, the results obtained with the imaginary frequency shift approach for $\lambda_x = 500$ km and $\lambda_t = 40$ min are summarized as follows:

· In the first step, the input value $\delta_{\min} = 10^{-6}$ s$^{-1}$ passes the Source-Function Reconstruction (SFR) test.

· In the second step, the SFR test reduces the input value $\delta_{\max} = 10^{-4}$ s$^{-1}$ to $\delta_{\max} = 32.24 \times 10^{-6}$ s$^{-1}$.

· In the third step, it is found the the interval $[\delta_{\min}, \delta_{\max}]$ contains a sufficient number of internal grid points, spaced by $\Delta\delta$, to be used in the subsequent step.

· In the fourth step, the Layerwise Causality (LC) condition is evaluated at 10 altitude starting at 80 km with a spacing of 20 km. It is found to be satisfied for $\delta_{\mathrm{LC}} = \delta_{\min}$, and therefore for all $\delta \in [\delta_{\min}, \delta_{\max}]$, for which the SFR test also holds.

· In the fifth step, five equidistant frequency shifts in $[\delta_{\min}, \delta_{\max}]$ are considered; for each, the wave parameters and their maximum values are computed and the layerwise causality condition is verified over the full altitude range. All five frequency shifts satisfy this condition and yield very similar maximum amplitudes (Table 2). The final solution is selected as the one whose maximum-amplitude vector is closest to the center of mass in the space of perturbed horizontal velocity, vertical velocity, and temperature, corresponding to $\delta = 16.62 \times 10^{-6}$ s$^{-1}$. The maxima occur at

[Figure]

Figure 6: Altitude profiles of the perturbed temperature $\overline{T}$, vertical velocity $\overline{w}$, and horizontal velocity $\overline{u}$ for the total mode ($\mathbf{e}_l = \mathbf{e}_l^+ + \mathbf{e}_l^-$ in layer $l$), the ascending mode ($\mathbf{e}_l^+$), and the descending mode ($\mathbf{e}_l^-$). The upper panels correspond to the general model, and the lower panels to the simplified model. The horizontal wavelength is $\lambda_x = 400$ km and the wave period is $\lambda_t = 40$ min

[Figure]

Figure 7: Perturbed temperature (left panels) and vertical velocity (right panels) as functions of time and altitude. The upper panels correspond to $\lambda_x = 300$ km and $\lambda_t = 30$ min, the middle panels to $\lambda_x = 500$ km and $\lambda_t = 40$ min, and the lower panels to $\lambda_x = 700$ km and $\lambda_t = 60$ min

| $\delta\,[10^{-6}\mathrm{s}^{-1}]$ | $\overline{u}_{\max}\,[\mathrm{ms}^{-1}]$ | $\overline{w}_{\max}\,[\mathrm{ms}^{-1}]$ | $\overline{T}_{\max}\,[\mathrm{K}]$ |
|---|---|---|---|
| 32.24 | 38.084 | 16.210 | 31.153 |
| 24.43 | 38.115 | 16.224 | 31.132 |
| 16.62 | 38.079 | 16.210 | 31.158 |
| 8.81 | 38.110 | 16.225 | 31.186 |
| 1.00 | 38.017 | 16.186 | 31.113 |

Table 2: Maximum values of the perturbed horizontal velocity ($\overline{u}_{\max}$), vertical velocity ($\overline{w}_{\max}$), and temperature ($\overline{T}_{\max}$) for different values of the imaginary frequency shift $\delta$

11.06 hr and 248.00 km for the horizontal velocity, 10.98 hr and 303.64 km for the vertical velocity, and 10.90 hr and 257.45 km for the temperature.

**7 Conclusions**

We designed a numerical model for solving the linearized gravity-wave equations using a multilayer method, which is freely available as open-source code on GitHub. To decouple the hydrodynamic equations for the neutral atmosphere from the ionospheric equations, which are coupled through ion drag, we adopt a fast field-aligned diffusion approximation. This approximation may be viewed as a generalization of an approach originally proposed by Klostermeyer [9].

To solve the linearized equations, we employ (i) global matrix methods based on matrix exponentials and (ii) scattering matrix methods to determine either (a) the amplitudes of the characteristic solutions or (b) the grid-point values of the state vector. Ascending and descending wave modes are identified according to the criterion that the real parts of the eigenvalues of the characteristic equation for ascending modes are smaller than those for descending modes (or, equivalently, that the imaginary parts of the vertical wavenumbers are smaller). Global matrix methods using the scaling matrices (36) and (37) require the classification of ascending and descending modes only at the lower and upper boundaries, whereas scattering matrix methods require an explicit determination of the mode type at every altitude. The model is devoted to solving the linearized equations including viscosity, thermal conduction, and ion drag. A simplified model, corresponding to an isothermal, homogeneous atmosphere with constant kinematic viscosity, no background wind, and no ion drag, is also considered.

Depending on the form of the source function, either single-frequency waves or time-dependent wave packets can be analyzed. A heuristic approach for determining the imaginary frequency shift introduced by Knight et al. [16] is also considered. This approach is based on (i) a layerwise causality condition applied at selected altitude levels, and (ii) a source-function reconstruction test.

Numerical simulations demonstrate that both global matrix and scattering matrix methods achieve comparable accuracy. However, the former are significantly more efficient than the latter, particularly in simulations involving

time-dependent wave packets. Among the global matrix methods, the approach based on solving for the amplitudes of the characteristic solutions appears to provide the highest efficiency and accuracy.

The linearized equations on which the solution methods were tested correspond to ionospheric conditions. The ultimate goal of our research is to develop a comprehensive model for analyzing ionospheric gravity waves using satellite measurements. The approach presented in this paper represents only the first component of such a model. Two options are envisaged for extending it to a more complete formulation.

1. Fully coupled neutral–ion model. The linearized hydrodynamic equations would be solved together with the ion equations. In this case, the ion continuity equation would include perturbed production and loss terms, whereas ion inertia and ion–ion collisions would continue to be neglected in the ion momentum equation, and only transport parallel to the magnetic field lines would be retained. The state vector would then be augmented by two additional components, namely the perturbed ion number density and the ion diffusion velocity.

2. Two-step coupling strategy. In the first step, the neutral-atmosphere equations are solved using the fast field-aligned diffusion approximation. In the second step, the wave-induced perturbations obtained from the neutral solution are used as input to solve the ionospheric equations for the perturbed $O^+$ ion density. The ionospheric equations may be solved using the SAMI2 model [40] for low latitudes, where the $\mathbf{E} \times \mathbf{B}$ drift is neglected, or the SAMI3 model [47] at higher altitudes, where the $\mathbf{E} \times \mathbf{B}$ drift is included and the electric field is determined from the solution of a two-dimensional potential equation. In this strategy, priority is given to the ionospheric equations of the SAMI framework, while wave-induced perturbations are handled using the approximate approach developed in the present study. Along similar lines, Knight et al. [34] solved the neutral-atmosphere equations without ion drag in a first step, and subsequently addressed the ionospheric response using the Field-Line Interhemispheric Plasma (FLIP) model [48].

The development and application of these complete models will be addressed in future papers.

**Appendix A. Derivation of the linear system of ordinary differential equations**

In this appendix, we derive the explicit representation of the linear system of ordinary differential equations (5).

**Hydrodynamic equations**

The hydrodynamic equations for the neutral atmosphere consist in the continuity, momentum, heat, and ideal gas equations (e.g., Refs. [4, 5])

$$\frac{\mathrm{D}\rho}{\mathrm{D}t} = -\rho \nabla \cdot \mathbf{u}, \tag{117}$$

$$\rho \frac{\mathrm{D}\mathbf{u}}{\mathrm{D}t} = -\nabla p + \rho \mathbf{g} + \nabla \cdot \overline{\boldsymbol{\sigma}} - \mathbf{f}_{\mathrm{ID}}, \tag{118}$$

$$\rho c_{\mathrm{v}} \frac{\mathrm{D}T}{\mathrm{D}t} = -p \nabla \cdot \mathbf{u} + \overline{\boldsymbol{\sigma}} : \nabla \mathbf{u} + \nabla \cdot (\Lambda \nabla T) - q_{\mathrm{ID}}, \tag{119}$$

$$p = \rho R_{\mathrm{M}} T, \tag{120}$$

where $\rho$ is the density, $p$ the pressure, $T$ the temperature, $\mathbf{u}$ the velocity, $\mathrm{D}/\mathrm{D}t = \partial/\partial t + \mathbf{u} \cdot \nabla$ the material (substantial) derivative, $c_{\mathrm{v}}$ the specific heat at constant volume, $\Lambda$ the coefficient of thermal conductivity, $R_{\mathrm{M}}$ the specific gas constant, and $\overline{\boldsymbol{\sigma}}$ the viscous stress tensor. The quantities $\mathbf{f}_{\mathrm{ID}}$ and $q_{\mathrm{ID}}$ denote the ion-drag force exerted by neutrals on ions per unit volume, and the frictional heating rate per unit volume arising from ion–neutral collisions, respectively. In a Cartesian coordinate system $(x_1, x_2, x_3)$, the components of the viscous stress tensor are given by

$$\sigma_{ij} = \mu \left( \frac{\partial u_i}{\partial x_j} + \frac{\partial u_j}{\partial x_i} - \frac{2}{3} \delta_{ij} \nabla \cdot \mathbf{u} \right), \tag{121}$$

where $\mu$ is the dynamic viscosity and $\delta_{ij}$ the Kroneker delta. Accordingly, the double dot product of $\overline{\boldsymbol{\sigma}} = \sum_{ij} \sigma_{ij} \widehat{\mathbf{x}}_i \otimes \widehat{\mathbf{x}}_j$ with $\nabla \mathbf{u} = \sum_{ij} \partial u_i / \partial x_j \widehat{\mathbf{x}}_i \otimes \widehat{\mathbf{x}}_j$ is

$$\overline{\boldsymbol{\sigma}} : \nabla \mathbf{u} = \sum_{ij} \sigma_{ij} \frac{\partial u_i}{\partial x_j}. \tag{122}$$

Using the ideal-gas law $p = \rho R_{\mathrm{M}} T$ so that $\nabla p = \rho R_{\mathrm{M}} \nabla T + R_{\mathrm{M}} T \nabla \rho$, the momentum equation can be written as

$$\frac{\partial \mathbf{u}}{\partial t} = -\frac{R_{\mathrm{M}} T}{\rho} \nabla \rho - R_{\mathrm{M}} \nabla T - (\mathbf{u} \cdot \nabla) \mathbf{u} + \mathbf{g} + \frac{1}{\rho} \nabla \cdot \overline{\boldsymbol{\sigma}} - \frac{1}{\rho} \mathbf{f}_{\mathrm{ID}}. \tag{123}$$

Moreover, using

$$c_{\mathrm{p}} = c_{\mathrm{v}} + R_{\mathrm{M}}, \;\; \gamma = \frac{c_{\mathrm{p}}}{c_{\mathrm{v}}}, \;\; \Lambda = \frac{c_{\mathrm{p}} \mu}{\mathrm{Pr}}, \tag{124}$$

where $c_{\mathrm{p}}$ is the specific heat at constant pressure, $\gamma$ the ratio of specific heats, and Pr the Prandtl number, and assuming that $c_{\mathrm{p}}$ and Pr are constant, the heat equation becomes

$$\frac{\partial T}{\partial t} = -(\gamma - 1) T \nabla \cdot \mathbf{u} - \mathbf{u} \cdot \nabla T + \frac{1}{\rho c_{\mathrm{v}}} \overline{\boldsymbol{\sigma}} : \nabla \mathbf{u} + \frac{\gamma}{\rho \mathrm{Pr}} \nabla \cdot (\mu \nabla T) - \frac{1}{\rho c_{\mathrm{v}}} q_{\mathrm{ID}}. \tag{125}$$

In the momentum and heat equations, the ion-drag force and the corresponding heating per unit mass are given by

$$\frac{1}{\rho}\mathbf{f}_{\mathrm{ID}} = \nu_{ni}(\mathbf{u} - \mathbf{u}_i), \tag{126}$$

$$\frac{1}{\rho}q_{\mathrm{ID}} = \frac{1}{\rho}\mathbf{f}_{\mathrm{ID}} \cdot (\mathbf{u} - \mathbf{u}_i) = \nu_{ni}|\mathbf{u} - \mathbf{u}_i|^2, \tag{127}$$

where $\nu_{ni}$ is the neutral-ion collision frequency (the collision frequency between a neutral particle and all kind of ions).

We choose a rectangular coordinate system such that the $x$-axis is directed to the geographic south, the $y$-axis to the east and the z-axis upward. The wave propagates in the meridional plane, i.e., in the $(x, z)$ plane. The viscous terms per unit mass in the momentum equation are then given by

$$\frac{1}{\rho}(\nabla \cdot \overline{\boldsymbol{\sigma}})_{\mathrm{x}} = \mu_{\mathrm{k}}\left[\left(\frac{\partial^2 u}{\partial x^2} + \frac{\partial^2 u}{\partial z^2}\right) + \frac{1}{3}\left(\frac{\partial^2 u}{\partial x^2} + \frac{\partial^2 w}{\partial x \partial z}\right)\right] + \frac{1}{\rho}\frac{\partial \mu}{\partial z}\left(\frac{\partial u}{\partial z} + \frac{\partial w}{\partial x}\right), \tag{128}$$

$$\frac{1}{\rho}(\nabla \cdot \overline{\boldsymbol{\sigma}})_{\mathrm{z}} = \mu_{\mathrm{k}}\left[\left(\frac{\partial^2 w}{\partial x^2} + \frac{\partial^2 w}{\partial z^2}\right) + \frac{1}{3}\left(\frac{\partial^2 u}{\partial x \partial z} + \frac{\partial^2 w}{\partial z^2}\right)\right] + \frac{1}{\rho}\frac{\partial \mu}{\partial z}\left(\frac{4}{3}\frac{\partial w}{\partial z} - \frac{2}{3}\frac{\partial u}{\partial x}\right), \tag{129}$$

while the viscous dissipation term per unit mass appearing in the heat equation is

$$\frac{1}{\rho}\overline{\boldsymbol{\sigma}} : \nabla\mathbf{u} = \frac{4}{3}\mu_{\mathrm{k}}\left(\frac{\partial u}{\partial x}\right)^2 - \frac{4}{3}\mu_{\mathrm{k}}\frac{\partial u}{\partial x}\frac{\partial w}{\partial z} + \frac{4}{3}\mu_{\mathrm{k}}\left(\frac{\partial w}{\partial z}\right)^2$$
$$+ \mu_{\mathrm{k}}\left(\frac{\partial u}{\partial z}\right)^2 + 2\mu_{\mathrm{k}}\frac{\partial u}{\partial z}\frac{\partial w}{\partial x} + \mu_{\mathrm{k}}\left(\frac{\partial w}{\partial x}\right)^2. \tag{130}$$

Here, $\mu_{\mathrm{k}} = \mu/\rho$ is the kinematic viscosity, and we have assumed that the viscosity depends only on altitude, so that

$$\frac{\partial \mu}{\partial x} = 0. \tag{131}$$

Using Eqs. (128)–(130) together with assumption (131), we express the hydrodynamic equations as

$$\frac{\partial u}{\partial t} = -\frac{R_{\mathrm{M}}T}{\rho}\frac{\partial \rho}{\partial x} - R_{\mathrm{M}}\frac{\partial T}{\partial x} - \left(u\frac{\partial u}{\partial x} + w\frac{\partial u}{\partial z}\right)$$
$$+ \mu_{\mathrm{k}}\left[\left(\frac{\partial^2 u}{\partial x^2} + \frac{\partial^2 u}{\partial z^2}\right) + \frac{1}{3}\left(\frac{\partial^2 u}{\partial x^2} + \frac{\partial^2 w}{\partial x \partial z}\right)\right]$$
$$+ \frac{1}{\rho}\frac{\partial \mu}{\partial z}\left(\frac{\partial u}{\partial z} + \frac{\partial w}{\partial x}\right) - \frac{1}{\rho}f_{\mathrm{IDx}}, \tag{132}$$

$$\frac{\partial w}{\partial t} = -\frac{R_{\mathrm{M}}T}{\rho}\frac{\partial \rho}{\partial z} - R_{\mathrm{M}}\frac{\partial T}{\partial z} - \left( u\frac{\partial w}{\partial x} + w\frac{\partial w}{\partial z} \right) - g$$
$$+ \mu_{\mathrm{k}} \left[ \left( \frac{\partial^2 w}{\partial x^2} + \frac{\partial^2 w}{\partial z^2} \right) + \frac{1}{3}\left( \frac{\partial^2 u}{\partial x \partial z} + \frac{\partial^2 w}{\partial z^2} \right) \right]$$
$$+ \frac{1}{\rho}\frac{\partial \mu}{\partial z}\left( \frac{4}{3}\frac{\partial w}{\partial z} - \frac{2}{3}\frac{\partial u}{\partial x} \right) - \frac{1}{\rho}f_{\mathrm{IDz}}, \tag{133}$$

$$\frac{\partial T}{\partial t} = -(\gamma - 1)\,T\left( \frac{\partial u}{\partial x} + \frac{\partial w}{\partial z} \right) - \left( u\frac{\partial T}{\partial x} + w\frac{\partial T}{\partial z} \right)$$
$$+ \frac{1}{c_{\mathrm{v}}}\left[ \frac{4}{3}\mu_{\mathrm{k}}\left( \frac{\partial u}{\partial x} \right)^2 - \frac{4}{3}\mu_{\mathrm{k}}\frac{\partial u}{\partial x}\frac{\partial w}{\partial z} + \frac{4}{3}\mu_{\mathrm{k}}\left( \frac{\partial w}{\partial z} \right)^2 \right.$$
$$\left. + \mu_{\mathrm{k}}\left( \frac{\partial u}{\partial z} \right)^2 + 2\mu_{\mathrm{k}}\frac{\partial u}{\partial z}\frac{\partial w}{\partial x} + \mu_{\mathrm{k}}\left( \frac{\partial w}{\partial x} \right)^2 \right]$$
$$+ \frac{\mu_{\mathrm{k}}\gamma}{\mathrm{Pr}}\left( \frac{\partial^2 T}{\partial x^2} + \frac{\partial^2 T}{\partial z^2} \right) + \frac{\gamma}{\rho\mathrm{Pr}}\frac{\partial \mu}{\partial z}\frac{\partial T}{\partial z} - \frac{1}{c_{\mathrm{v}}\rho}q_{\mathrm{ID}}, \tag{134}$$

where $f_{\mathrm{IDx}}$ and $f_{\mathrm{IDz}}$ are the components of the ion drag force $\mathbf{f}_{\mathrm{ID}}$ on the $x$- and $z$-axis, respectively.

In the above hydrodynamic equations, the Coriolis force has been neglected. For a two-dimensional wave geometry in which both the background flow and the perturbation velocities are confined to the vertical $(x, z)$ plane and all variables are independent of the transverse horizontal coordinate $y$, the Coriolis acceleration associated with the Earth's rotation is directed entirely along the transverse direction and therefore does not enter the momentum equations considered here. More precisely, for $\mathbf{u} = (u, 0, w)$ and $\mathbf{\Omega} = (-\Omega\cos\phi, 0, \Omega\sin\phi)$, where $\phi$ is the geographic latitude and $\Omega = 7.29 \times 10^{-5}\,\mathrm{s}^{-1}$ the Earth's angular velocity, the Coriolis force per unit mass is $\mathbf{f}_{\mathrm{C}} = -2\mathbf{\Omega} \times \mathbf{u} = (0, 2\Omega(\cos\phi w + \sin\phi u), 0)$. Thus, the Coriolis acceleration is directed entirely along the transverse horizontal direction $y$ and does not affect the two-dimensional $(x, z)$ momentum equations.

**Linearized equations**

To linearize the hydrodynamic equations, we assume that all background (unperturbed) quantities vary only in the $z$-direction and write

$$f(x, z, t) = f_0(z) + f'(x, z, t),$$

where $f$ denotes any state variable. In particular, we assume

$$\mathbf{u}_0(z) = (u_0(z), 0, w_0(z) = 0), \quad \rho_0 = \rho_0(z), \quad T_0 = T_0(z). \tag{135}$$

Furthermore, we neglect the second derivative of the background horizontal wind and background temperature,

$$\frac{\mathrm{d}^2 u_0}{\mathrm{d}z^2} = 0, \quad \frac{\mathrm{d}^2 T_0}{\mathrm{d}z^2} = 0, \tag{136}$$

1. The linearized continuity equation is

$$\frac{\partial \rho'}{\partial t} = -u_0 \frac{\partial \rho'}{\partial x} + \frac{\rho_0}{H_\rho} w' - \rho_0 \left( \frac{\partial u'}{\partial x} + \frac{\partial w'}{\partial z} \right). \tag{137}$$

2. The linearized momentum equations are

$$\begin{aligned}
\frac{\partial u'}{\partial t} = &-\frac{\mathrm{d} u_0}{\mathrm{d} z} w' - u_0 \frac{\partial u'}{\partial x} - \frac{c_s^2}{\gamma \rho_0} \frac{\partial \rho'}{\partial x} - \frac{c_s^2}{\gamma T_0} \frac{\partial T'}{\partial x} \\
&+ \mu_{\mathrm{k}0} \left[ \left( \frac{\partial^2 u'}{\partial x^2} + \frac{\partial^2 u'}{\partial z^2} \right) + \frac{1}{3} \left( \frac{\partial^2 u'}{\partial x^2} + \frac{\partial^2 w'}{\partial x \partial z} \right) \right] \\
&+ \frac{1}{\rho_0} \frac{\mathrm{d} \mu_0}{\mathrm{d} z} \left( \frac{\partial u'}{\partial z} + \frac{\partial w'}{\partial x} \right) + \frac{1}{\rho_0} \frac{\mathrm{d} u_0}{\mathrm{d} z} \frac{\partial \mu'}{\partial z} - \frac{1}{\rho_0} \frac{\mathrm{d} \mu_0}{\mathrm{d} z} \frac{\mathrm{d} u_0}{\mathrm{d} z} \frac{\rho'}{\rho_0} - \left( \frac{1}{\rho} f_{\mathrm{IDx}} \right)'
\end{aligned} \tag{138}$$

and

$$\begin{aligned}
\frac{\partial w'}{\partial t} = &-\frac{c_s^2}{\gamma \rho_0} \frac{\partial \rho'}{\partial z} + \frac{c_s^2}{\gamma H_\rho} \left( \frac{T'}{T_0} - \frac{\rho'}{\rho_0} \right) - \frac{c_s^2}{\gamma T_0} \frac{\partial T'}{\partial z} - u_0 \frac{\partial w'}{\partial x} \\
&+ \mu_{\mathrm{k}0} \left[ \left( \frac{\partial^2 w'}{\partial x^2} + \frac{\partial^2 w'}{\partial z^2} \right) + \frac{1}{3} \left( \frac{\partial^2 u'}{\partial x \partial z} + \frac{\partial^2 w'}{\partial z^2} \right) \right] \\
&+ \frac{1}{\rho_0} \frac{\mathrm{d} \mu_0}{\mathrm{d} z} \left( \frac{4}{3} \frac{\partial w'}{\partial z} - \frac{2}{3} \frac{\partial u'}{\partial x} \right) - \left( \frac{1}{\rho} f_{\mathrm{IDz}} \right)',
\end{aligned} \tag{139}$$

where

$$c_{\mathrm{s}} = \sqrt{\gamma R_{\mathrm{M}} T_0} \tag{140}$$

is the speed of sound and $H_\rho$ is the density scale height defined by

$$\frac{\mathrm{d} \rho_0}{\mathrm{d} z} = -\frac{\rho_0}{H_\rho}. \tag{141}$$

Here, $(f_{\mathrm{IDx}}/\rho)'$ and $(f_{\mathrm{IDz}}/\rho)'$ denote the perturbations of the $x$- and $z$-components, respectively, of the ion-drag force per unit mass $f_{\mathrm{IDx}}/\rho$ and $f_{\mathrm{IDz}}/\rho$.

3. The linearized heat equation is

$$\begin{aligned}
\frac{\partial T'}{\partial t} = &-(\gamma - 1) T_0 \left( \frac{\partial u'}{\partial x} + \frac{\partial w'}{\partial z} \right) - \left( u_0 \frac{\partial T'}{\partial x} + \frac{\mathrm{d} T_0}{\mathrm{d} z} w' \right) \\
&+ \frac{2 \mu_{\mathrm{k}0}}{c_{\mathrm{v}}} \frac{\mathrm{d} u_0}{\mathrm{d} z} \left( \frac{\partial u'}{\partial z} + \frac{\partial w'}{\partial x} \right) + \frac{\mu_0}{c_{\mathrm{v}} \rho_0} \left( \frac{\mathrm{d} u_0}{\mathrm{d} z} \right)^2 \left( \frac{\mu'}{\mu_0} - \frac{\rho'}{\rho_0} \right) \\
&+ \frac{\mu_{\mathrm{k}0} \gamma}{\mathrm{Pr}} \left( \frac{\partial^2 T'}{\partial x^2} + \frac{\partial^2 T'}{\partial z^2} \right) + \frac{\gamma}{\rho_0 \mathrm{Pr}} \left( \frac{\mathrm{d} \mu_0}{\mathrm{d} z} \frac{\partial T'}{\partial z} + \frac{\mathrm{d} T_0}{\mathrm{d} z} \frac{\partial \mu'}{\partial z} \right) - \frac{1}{c_{\mathrm{v}}} \left( \frac{1}{\rho} q_{\mathrm{ID}} \right)',
\end{aligned} \tag{142}$$

where $(q_{\mathrm{ID}}/\rho)'$ denotes the perturbation of the ion-drag heating per unit mass.

The linearized equations (137), (138), (139), and (142) are structurally consistent with Eqs. (7), (4), (5), and (6) in Ref. [35], respectively, when only the terms in black and blue are considered (the terms indicated in red are not included). They are likewise related to Eqs. (3.5), (3.1), (3.3), and (3.4) in Ref. [11], when only the terms indicated in black are considered (the additional contributions indicated in red and blue are not included).

Using the following representation for the dynamic viscosity $\mu$ [49]

$$\mu = 3.34 \times 10^{-7} T^{0.71}, \tag{143}$$

we obtain

$$\mu' = 0.71 \mu_0 \frac{T'}{T_0}, \quad \frac{\partial \mu'}{\partial z} = 0.71 \frac{\mathrm{d}\mu_0}{\mathrm{d}z} \left( \frac{T'}{T_0} \right) + 0.71 \mu_0 \frac{\partial}{\partial z} \left( \frac{T'}{T_0} \right). \tag{144}$$

With these relations, the linearized equations can be written as follows:

1. $x$-momentum equation

$$
\begin{aligned}
\frac{\partial u'}{\partial t} = & -\frac{\mathrm{d}u_0}{\mathrm{d}z} w' - u_0 \frac{\partial u'}{\partial x} - \frac{c_s^2}{\gamma \rho_0} \frac{\partial \rho'}{\partial x} - \frac{c_s^2}{\gamma T_0} \frac{\partial T'}{\partial x} \\
& + \mu_{k0} \left[ \left( \frac{\partial^2 u'}{\partial x^2} + \frac{\partial^2 u'}{\partial z^2} \right) + \frac{1}{3} \left( \frac{\partial^2 u'}{\partial x^2} + \frac{\partial^2 w'}{\partial x \partial z} \right) \right] \\
& + \frac{1}{\rho_0} \frac{\mathrm{d}\mu_0}{\mathrm{d}z} \left( \frac{\partial u'}{\partial z} + \frac{\partial w'}{\partial x} \right) + \frac{1}{\rho_0} \frac{\mathrm{d}u_0}{\mathrm{d}z} \left[ 0.71 \frac{\mathrm{d}\mu_0}{\mathrm{d}z} \frac{T'}{T_0} + 0.71 \mu_0 \frac{\partial}{\partial z} \left( \frac{T'}{T_0} \right) \right] \\
& - \frac{1}{\rho_0} \frac{\mathrm{d}\mu_0}{\mathrm{d}z} \frac{\mathrm{d}u_0}{\mathrm{d}z} \frac{\rho'}{\rho_0} - \left( \frac{1}{\rho} f_{\mathrm{IDx}} \right)',
\end{aligned} \tag{145}
$$

2. $z$-momentum equation

$$
\begin{aligned}
\frac{\partial w'}{\partial t} = & -\frac{c_s^2}{\gamma \rho_0} \frac{\partial \rho'}{\partial z} + \frac{c_s^2}{\gamma H_\rho} \left( \frac{T'}{T_0} - \frac{\rho'}{\rho_0} \right) - \frac{c_s^2}{\gamma T_0} \frac{\partial T'}{\partial z} - u_0 \frac{\partial w'}{\partial x} \\
& + \mu_{k0} \left[ \left( \frac{\partial^2 w'}{\partial x^2} + \frac{\partial^2 w'}{\partial z^2} \right) + \frac{1}{3} \left( \frac{\partial^2 u'}{\partial x \partial z} + \frac{\partial^2 w'}{\partial z^2} \right) \right] \\
& + \frac{1}{\rho_0} \frac{\mathrm{d}\mu_0}{\mathrm{d}z} \left( \frac{4}{3} \frac{\partial w'}{\partial z} - \frac{2}{3} \frac{\partial u'}{\partial x} \right) - \left( \frac{1}{\rho} f_{\mathrm{IDz}} \right)',
\end{aligned} \tag{146}
$$

3. heat equation

$$
\begin{aligned}
\frac{\partial T'}{\partial t} = & -(\gamma - 1) T_0 \left( \frac{\partial u'}{\partial x} + \frac{\partial w'}{\partial z} \right) - \left( u_0 \frac{\partial T'}{\partial x} + \frac{\mathrm{d}T_0}{\mathrm{d}z} w' \right) \\
& + \frac{2\mu_{k0}}{c_v} \frac{\mathrm{d}u_0}{\mathrm{d}z} \left( \frac{\partial u'}{\partial z} + \frac{\partial w'}{\partial x} \right) + \frac{\mu_0}{c_v \rho_0} \left( \frac{\mathrm{d}u_0}{\mathrm{d}z} \right)^2 \left( 0.71 \frac{T'}{T_0} - \frac{\rho'}{\rho_0} \right) \\
& + \frac{\mu_{k0} \gamma}{\mathrm{Pr}} \left( \frac{\partial^2 T'}{\partial x^2} + \frac{\partial^2 T'}{\partial z^2} \right) + \frac{\gamma}{\rho_0 \mathrm{Pr}} \frac{\mathrm{d}\mu_0}{\mathrm{d}z} \frac{\partial T'}{\partial z} \\
& + \frac{\gamma}{\rho_0 \mathrm{Pr}} \frac{\mathrm{d}T_0}{\mathrm{d}z} \left[ 0.71 \frac{\mathrm{d}\mu_0}{\mathrm{d}z} \left( \frac{T'}{T_0} \right) + 0.71 \mu_0 \frac{\partial}{\partial z} \left( \frac{T'}{T_0} \right) \right] - \frac{1}{c_v} \left( \frac{1}{\rho} q_{\mathrm{ID}} \right)'.
\end{aligned} \tag{147}
$$

**Plane wave solution**

We assume that all perturbations vary harmonically in time and in the $x$-direction, that is,

$$f'(x, z, t) = \overline{f}(z)e^{j(\omega t - k_x x)}, \tag{148}$$

where $\omega$ is the angular frequency and $k_x$ the horizontal wavenumber. It then follows that

$$\frac{\partial f'}{\partial t} = j\omega f', \ \ \frac{\partial f'}{\partial x} = -jk_x f', \ \ \frac{\partial^2 f'}{\partial x^2} = -k_x^2 f'. \tag{149}$$

The linearized continuity equation becomes

$$\frac{\overline{\rho}}{\rho_0} = \frac{k_x}{\Omega}\overline{u} - \frac{j}{\Omega H_\rho}\overline{w} + \frac{j}{\Omega}\frac{d\overline{w}}{dz}, \tag{150}$$

where

$$\Omega = \omega - k_x u_0 \tag{151}$$

is the intrinsic frequency. Further, using

$$\frac{d}{dz}\left(\frac{\overline{\rho}}{\rho_0}\right) = \frac{1}{\rho_0}\frac{d\overline{\rho}}{dz} + \frac{1}{H_\rho}\frac{\overline{\rho}}{\rho_0} \tag{152}$$

together with

$$\frac{d\Omega}{dz} = -k_x\frac{du_0}{dz}, \ \ \frac{d\rho_0}{dz} = -\frac{\rho_0}{H_\rho}, \tag{153}$$

we obtain

$$\frac{1}{\rho_0}\frac{d\overline{\rho}}{dz} + \frac{1}{H_\rho}\frac{\overline{\rho}}{\rho_0} = \frac{k_x^2}{\Omega^2}\frac{du_0}{dz}\overline{u} - \frac{j}{\Omega H_\rho}\left(\frac{k_x}{\Omega}\frac{du_0}{dz} - \frac{1}{H_\rho}\frac{dH_\rho}{dz}\right)\overline{w}$$
$$+ \frac{k_x}{\Omega}\frac{d\overline{u}}{dz} + \frac{j}{\Omega}\left(\frac{k_x}{\Omega}\frac{du_0}{dz} - \frac{1}{H_\rho}\right)\frac{d\overline{w}}{dz} + \frac{j}{\Omega}\frac{d^2\overline{w}}{dz^2}. \tag{154}$$

In a first step, we use Eqs. (150) and (154), together with the linearized forms of the momentum and the heat equation given in Eqs. (145)–(147), to express the governing equations in terms of $\overline{u}$, $\overline{w}$, $\overline{T}/T_0$, and their vertical derivatives.

In a second step, we introduce the dimensionless state variables $\widehat{u}$, $\widehat{w}$, and $\widehat{T}$, defined by

$$\overline{u}(z) = \frac{\omega_0}{k_x}\widehat{u}(z), \ \ \overline{w}(z) = \frac{\omega_0}{k_x}\widehat{w}(z), \ \ \overline{T}(z) = T_0(z)\widehat{T}(z), \tag{155}$$

where $\omega_0$ is a reference frequency, together with their vertical derivatives

$$\widehat{\mathcal{U}} = \frac{d\widehat{u}}{dz}, \ \ \widehat{\mathcal{W}} = \frac{d\widehat{w}}{dz}, \ \ \widehat{\mathcal{T}} = \frac{d\widehat{T}}{dz}. \tag{156}$$

The state vector is organized as

$$\mathbf{e} = [\widehat{u}, \widehat{w}, \widehat{T}, \widehat{\mathcal{U}}, \widehat{\mathcal{W}}, \widehat{\mathcal{T}}]^{\mathrm{T}}$$

and the corresponding system of ordinary differential equations is given by

$$\frac{1}{k_{\mathrm{x}}} \frac{\mathrm{d}\widehat{u}}{\mathrm{d}z} = \frac{1}{k_{\mathrm{x}}} \widehat{\mathcal{U}}, \tag{157}$$

$$\frac{1}{k_{\mathrm{x}}} \frac{\mathrm{d}\widehat{w}}{\mathrm{d}z} = \frac{1}{k_{\mathrm{x}}} \widehat{\mathcal{W}}, \tag{158}$$

$$\frac{1}{k_{\mathrm{x}}} \frac{\mathrm{d}\widehat{T}}{\mathrm{d}z} = \frac{1}{k_{\mathrm{x}}} \widehat{\mathcal{T}}, \tag{159}$$

$$
\begin{aligned}
k_{\mathrm{x}} \mu_{\mathrm{k}0} \left( \frac{1}{k_{\mathrm{x}}} \frac{\mathrm{d}\widehat{\mathcal{U}}}{\mathrm{d}z} \right) =& \left[ \mathrm{j}\Omega + \frac{4}{3} k_{\mathrm{x}}^2 \mu_{\mathrm{k}0} + \frac{k_{\mathrm{x}}}{\Omega} \left( \frac{1}{\rho_0} \frac{\mathrm{d}\mu_0}{\mathrm{d}z} \frac{\mathrm{d}u_0}{\mathrm{d}z} - \mathrm{j}k_{\mathrm{x}} \frac{c_{\mathrm{s}}^2}{\gamma} \right) \right] \widehat{u} \\
&+ \left[ \frac{\mathrm{d}u_0}{\mathrm{d}z} + \mathrm{j}k_{\mathrm{x}} \frac{1}{\rho_0} \frac{\mathrm{d}\mu_0}{\mathrm{d}z} - \frac{\mathrm{j}}{\Omega H_\rho} \left( \frac{1}{\rho_0} \frac{\mathrm{d}\mu_0}{\mathrm{d}z} \frac{\mathrm{d}u_0}{\mathrm{d}z} - \mathrm{j}k_{\mathrm{x}} \frac{c_{\mathrm{s}}^2}{\gamma} \right) \right] \widehat{w} \\
&- \frac{k_{\mathrm{x}}}{\omega_0} \left( \mathrm{j}k_{\mathrm{x}} \frac{c_{\mathrm{s}}^2}{\gamma} + 0.71 \frac{1}{\rho_0} \frac{\mathrm{d}\mu_0}{\mathrm{d}z} \frac{\mathrm{d}u_0}{\mathrm{d}z} \right) \widehat{T} - \frac{1}{\rho_0} \frac{\mathrm{d}\mu_0}{\mathrm{d}z} \widehat{\mathcal{U}} \\
&+ \left[ \frac{1}{3} \mathrm{j}k_{\mathrm{x}} \mu_{\mathrm{k}0} + \frac{\mathrm{j}}{\Omega} \left( \frac{1}{\rho_0} \frac{\mathrm{d}\mu_0}{\mathrm{d}z} \frac{\mathrm{d}u_0}{\mathrm{d}z} - \mathrm{j}k_{\mathrm{x}} \frac{c_{\mathrm{s}}^2}{\gamma} \right) \right] \widehat{\mathcal{W}} \\
&- 0.71 \frac{\mu_0}{\rho_0} \frac{\mathrm{d}u_0}{\mathrm{d}z} \frac{k_{\mathrm{x}}}{\omega_0} \widehat{\mathcal{T}} + \frac{k_{\mathrm{x}}}{\omega_0} \overline{\left( \frac{1}{\rho} f_{\mathrm{IDx}} \right)},
\end{aligned} \tag{160}
$$

$$
\begin{aligned}
k_{\mathrm{x}} \left( \frac{4}{3} \mu_{\mathrm{k}0} - \frac{\mathrm{j}c_{\mathrm{s}}^2}{\gamma\Omega} \right) \left( \frac{1}{k_{\mathrm{x}}} \frac{\mathrm{d}\widehat{\mathcal{W}}}{\mathrm{d}z} \right) =& \left( -\frac{2}{3} \mathrm{j}k_{\mathrm{x}} \frac{1}{\rho_0} \frac{\mathrm{d}\mu_0}{\mathrm{d}z} + \frac{c_{\mathrm{s}}^2 k_{\mathrm{x}}^2}{\gamma\Omega^2} \frac{\mathrm{d}u_0}{\mathrm{d}z} \right) \widehat{u} \\
&+ \left[ \mathrm{j}\Omega + k_{\mathrm{x}}^2 \mu_{\mathrm{k}0} - \frac{\mathrm{j}c_{\mathrm{s}}^2}{\gamma\Omega H_\rho} \left( \frac{k_{\mathrm{x}}}{\Omega} \frac{\mathrm{d}u_0}{\mathrm{d}z} - \frac{1}{H_\rho} \frac{\mathrm{d}H_\rho}{\mathrm{d}z} \right) \right] \widehat{w} \\
&- \frac{c_{\mathrm{s}}^2}{\gamma} \frac{k_{\mathrm{x}}}{\omega_0} \left( \frac{1}{H_\rho} - \frac{1}{T_0} \frac{\mathrm{d}T_0}{\mathrm{d}z} \right) \widehat{T} + \left( \frac{1}{3} \mathrm{j}k_{\mathrm{x}} \mu_{\mathrm{k}0} + \frac{k_{\mathrm{x}} c_{\mathrm{s}}^2}{\gamma\Omega} \right) \widehat{\mathcal{U}} \\
&+ \left[ \frac{\mathrm{j}c_{\mathrm{s}}^2}{\gamma\Omega} \left( \frac{k_{\mathrm{x}}}{\Omega} \frac{\mathrm{d}u_0}{\mathrm{d}z} - \frac{1}{H_\rho} \right) - \frac{4}{3} \frac{1}{\rho_0} \frac{\mathrm{d}\mu_0}{\mathrm{d}z} \right] \widehat{\mathcal{W}} \\
&+ \frac{c_{\mathrm{s}}^2}{\gamma} \frac{k_{\mathrm{x}}}{\omega_0} \widehat{\mathcal{T}} + \frac{k_{\mathrm{x}}}{\omega_0} \overline{\left( \frac{1}{\rho} f_{\mathrm{IDz}} \right)},
\end{aligned} \tag{161}
$$

$$k_{\mathrm{x}} \frac{\mu_{\mathrm{k0}} \gamma}{\mathrm{Pr}} \left( \frac{1}{k_{\mathrm{x}}} \frac{\mathrm{d}\widehat{\mathcal{T}}}{\mathrm{d}z} \right) = \omega_0 \left[ -\mathrm{j}\left(\gamma - 1\right) + \frac{1}{\Omega} \frac{\mu_0}{c_{\mathrm{v}} T_0 \rho_0} \left( \frac{\mathrm{d}u_0}{\mathrm{d}z} \right)^2 \right] \widehat{u}$$

$$+ \frac{\omega_0}{k_{\mathrm{x}}} \left[ \frac{1}{T_0} \frac{\mathrm{d}T_0}{\mathrm{d}z} + \mathrm{j}\frac{2\mu_{\mathrm{k0}} k_{\mathrm{x}}}{c_{\mathrm{v}} T_0} \frac{\mathrm{d}u_0}{\mathrm{d}z} - \frac{\mathrm{j}}{\Omega H_\rho} \frac{\mu_0}{c_{\mathrm{v}} T_0 \rho_0} \left( \frac{\mathrm{d}u_0}{\mathrm{d}z} \right)^2 \right] \widehat{w}$$

$$+ \left[ \mathrm{j}\Omega + \frac{\mu_{\mathrm{k0}} \gamma}{\mathrm{Pr}} k_{\mathrm{x}}^2 - C_1 \frac{\gamma}{\rho_0 \mathrm{Pr}} \frac{\mathrm{d}\mu_0}{\mathrm{d}z} \frac{1}{T_0} \frac{\mathrm{d}T_0}{\mathrm{d}z} - 0.71 \frac{\mu_0}{c_{\mathrm{v}} T_0 \rho_0} \left( \frac{\mathrm{d}u_0}{\mathrm{d}z} \right)^2 \right] \widehat{T}$$

$$- \frac{2\mu_{\mathrm{k0}}}{c_{\mathrm{v}} T_0} \frac{\omega_0}{k_{\mathrm{x}}} \frac{\mathrm{d}u_0}{\mathrm{d}z} \widehat{\mathcal{U}} + \frac{\omega_0}{k_{\mathrm{x}}} \left[ \left(\gamma - 1\right) + \frac{\mathrm{j}}{\Omega} \frac{\mu_0}{c_{\mathrm{v}} T_0 \rho_0} \left( \frac{\mathrm{d}u_0}{\mathrm{d}z} \right)^2 \right] \widehat{\mathcal{W}}$$

$$- \frac{\gamma}{\mathrm{Pr}} \left( \frac{1}{\rho_0} \frac{\mathrm{d}\mu_0}{\mathrm{d}z} + C_2 \mu_{\mathrm{k0}} \frac{1}{T_0} \frac{\mathrm{d}T_0}{\mathrm{d}z} \right) \widehat{\mathcal{T}} + \frac{1}{c_{\mathrm{v}} T_0} \overline{\left( \frac{1}{\rho} q_{\mathrm{ID}} \right)}. \quad (162)$$

Equations (160), (161), and (162) with the constants $C_1 = 1.71$ and $C_2 = 2.71$ correspond to the general model, in which all altitude derivatives of the background parameters $u_0$, $T_0$, $H_\rho$, and $\mu_0$ are retained. The terms shown in black and blue in these equations, with the constants $C_1 = 1.0$ and $C_2 = 2.0$, correspond to the model of Vadas and Nicolls [35]. Moreover, Eqs. (160), (161), and (162), when only the black terms are retained and the constants are set to $C_1 = C_2 = 0$, are consistent in form with Eqs. (3.32), (3.31), and (3.33) in Ref. [11], respectively. Note that in Ref. [11], the wave propagates in three-dimensional space, the harmonic dependence of the perturbed quantities is taken as $\exp[-\mathrm{j}(\omega t - k_{\mathrm{x}}x - k_{\mathrm{y}}y)]$, rather than $\exp[\mathrm{j}(\omega t - k_{\mathrm{x}}x)]$, the characteristic solutions have an $\exp(\mathrm{j}mz)$ dependence, rather than $\exp(k_{\mathrm{x}}\lambda z)$, and the state vector is defined as $[\overline{u}, \overline{w}, \widehat{T}, \overline{\mathcal{U}}, \overline{\mathcal{W}}, \widehat{\mathcal{T}}]^T$ instead of $[\widehat{u}, \widehat{w}, \widehat{T}, \widehat{\mathcal{U}}, \widehat{\mathcal{W}}, \widehat{\mathcal{T}}]^T$, where $\overline{\mathcal{U}} = \mathrm{d}\overline{u}/\mathrm{d}z$ and $\overline{\mathcal{W}} = \mathrm{d}\overline{w}/\mathrm{d}z$. Essentially, the difference between the model of Vadas and Nicolls [35] and that of Knight et al. [11] is that, in the latter, the derivative of the dynamic viscosity $\mathrm{d}\mu_0/\mathrm{d}z$ is omitted. In the code, for testing purposes, we included a hard-coded logical flag that selects the linearized model to be used. Our numerical simulations show that there are no significant differences between the general model and that of Vadas and Nicolls [35], and that the effect of the assumption $\mathrm{d}\mu_0/\mathrm{d}z = 0$ is relatively small. This latter assumption was discussed in detail in Ref. [11]. In the general model, the derivative $\mathrm{d}\mu_0/\mathrm{d}z$ is computed as

$$\frac{\mathrm{d}\mu_0}{\mathrm{d}z} = 0.71 \mu_0 \frac{1}{T_0} \frac{\mathrm{d}T_0}{\mathrm{d}z},$$

whereas the derivatives of $u_0$, $T_0$, and $H_\rho$ are computed using central finite differences.

The model can be particularized as follows:

1. For an isothermal ($T_0 = \mathrm{constant}$), homogeneous ($\mu_{\mathrm{k0}} = \mu_0/\rho_0 = \mathrm{constant}$), and windless atmosphere ($u_0 = 0$), we set

$$\frac{\mathrm{d}u_0}{\mathrm{d}z} = 0, \ \ \frac{\mathrm{d}T_0}{\mathrm{d}z} = 0, \ \ \Omega = \omega_0, \ \ \frac{\mathrm{d}H_\rho}{\mathrm{d}z} = 0, \ \text{and} \ \frac{1}{\rho_0} \frac{\mathrm{d}\mu_0}{\mathrm{d}z} = -\frac{\mu_{\mathrm{k0}}}{H_\rho}. \quad (163)$$

In this case, the density scale height $H_\rho$ coincides with the atmospheric scale height $H_a$, which satisfies

$$\frac{1}{\rho_0}\frac{d\rho_0}{dz} = \frac{1}{p_0}\frac{dp_0}{dz} = -\frac{1}{H_a} \tag{164}$$

and is given by

$$H_a = \frac{p_0}{\rho_0 g} = \frac{R_M T_0}{g} = \text{constant.} \tag{165}$$

2. For an atmosphere without ion drag, we set

$$f_{IDx} = 0, \quad f_{IDz} = 0, \quad \text{and } q_{ID} = 0. \tag{166}$$

**Dispersion equation**

For $\widehat{f}(z) \propto \exp(k_x \lambda z)$, we obtain

$$\widehat{\mathcal{F}} = \frac{d\widehat{f}}{dz} = k_x \lambda \widehat{f}, \quad \frac{d\widehat{\mathcal{F}}}{dz} = k_x^2 \lambda^2 \widehat{f}, \tag{167}$$

where $f$ denotes $u$, $w$, and $T$, and $\mathcal{F}$ denotes $\mathcal{U}$, $\mathcal{W}$, and $\mathcal{T}$. Inserting Eq. (167) into Eqs. (160)–(162) yields a homogeneous system of equations. Requiring the determinant of this system to vanish, and neglecting the ion-drag terms, leads to the following dispersion equation:

$$-\frac{\Omega}{c_s^2}\left[\Omega - j\frac{\gamma\mu_{k0}}{Pr}k_x^2\left(1-\lambda^2\right)\right]\left[\Omega - j\mu_{k0}k_x^2\left(1-\lambda^2\right)\right]\left[\Omega - j\frac{4\mu_{k0}}{3}k_x^2\left(1-\lambda^2\right)\right]$$

$$+\left[\Omega - j\mu_{k0}k_x^2\left(1-\lambda^2\right)\right]\left[\Omega - j\frac{\mu_{k0}}{Pr}k_x^2\left(1-\lambda^2\right)\right]\left[k_x^2\left(1-\lambda^2\right)+\frac{k_x\lambda}{H_\rho}\right]-\frac{k_x^2 c_s^2}{\gamma^2 H_\rho^2}\left(\gamma - 1\right)$$

$$= 0. \tag{168}$$

The dispersion equation (168) is the counterpart of dispersion relation (3.35) in Knight et al. [11], under the aforementioned equivalences. Remarkably, it does not include derivatives of the background parameters $u_0$, $T_0$, $H_\rho$, and $\mu_0$. This result follows from the variable-change method discussed by Knight et al. [11] in their Section 2.2.

For an isothermal, homogeneous, and windless atmosphere without ion drag, the dispersion equation reduces to the cubic equation [4]

$$C_3 R^3 + C_2 R^2 + C_1 R + C_0 = 0, \tag{169}$$

for $R = -\lambda^2 + \alpha\lambda + 1$, where $\alpha = 1/k_x H_a$, or equivalently, $R = \kappa^2 - j\alpha\kappa + 1$, where

$$\kappa = j\lambda = \frac{1}{k_x}\left(k_z + j\frac{1}{2H_a}\right). \tag{170}$$

The coefficients of the cubic equations are given by

$$C_3 = -3\eta\nu(1 + 4\eta), \tag{171}$$

$$C_2 = \frac{3\eta(1 + 4\eta)}{\gamma - 1} + \nu\beta(1 + 7\eta) + 3\eta, \tag{172}$$

$$C_1 = -[\beta^2 - 2\eta\alpha^2(1 + 3\eta)]\nu - \frac{\beta(1 + 7\eta)}{\gamma - 1} - \beta, \tag{173}$$

$$C_0 = \frac{\beta^2 - 2\eta\alpha^2(1 + 3\eta)}{\gamma - 1} + \alpha^2(1 + 3\eta), \tag{174}$$

where

$$\eta = \mathrm{j}\frac{\omega_0\mu_0}{3p_0}, \quad \nu = \mathrm{j}\frac{k_\mathrm{x}^2\Lambda_0 T_0}{\omega_0 p_0}, \quad \Lambda_0 = \frac{\gamma c_\mathrm{v}\mu_0}{\mathrm{Pr}}, \quad \beta = \frac{\omega_0^2}{k_\mathrm{x}^2 g H_\mathrm{a}}. \tag{175}$$

If $R_m$, $m = 1, \ldots, 3$ are the solutions of the dispersion equation, the corresponding vertical wavenumbers $k_{zm}^\pm$ are given by

$$k_{zm}^\pm = \mp k_\mathrm{x}\sqrt{R_m - 1 - \frac{\alpha^2}{4}}. \tag{176}$$

The wavenumbers $k_{zm}^+$ with $\mathrm{Im}(k_{zm}^+) < 0$ are associated with ascending modes, whereas the wavenumbers $k_{zm}^-$ with $\mathrm{Im}(k_{zm}^-) > 0$ correspond to descending modes. Ordering the ascending wavenumbers as

$$\mathrm{Im}(k_{z3}^+) < \mathrm{Im}(k_{z2}^+) < \mathrm{Im}(k_{z1}^+) < 0,$$

we identify (i) $k_{z1}^+$ and $k_{z1}^-$ as ascending and descending gravity-wave modes, respectively, (ii) $k_{z2}^+$ and $k_{z2}^-$ as ascending and descending viscosity-wave modes, respectively, and (iii) $k_{z3}^+$ and $k_{z3}^-$ as ascending and descending thermal-conduction wave modes, respectively. A similar cubic equation was derived by Francis [7] under the assumption that the geomagnetic field is either in the horizontal or the vertical direction.

For an isothermal, homogeneous, and windless atmosphere, without viscosity and ion drag, the dispersion equation (169) reduces to the quadratic equation

$$C_2 R^2 + C_1 R + C_0 = 0, \tag{177}$$

with

$$C_2 = \nu\beta, \tag{178}$$

$$C_1 = -\beta^2\nu - \beta\frac{\gamma}{\gamma - 1}, \tag{179}$$

$$C_0 = \frac{\beta^2}{\gamma - 1} + \alpha^2. \tag{180}$$

For $\mathrm{Im}(k_{z2}^+) < \mathrm{Im}(k_{z1}^+) < 0$, the permissible modes are (i) the ascending and descending gravity-wave modes associated to the pair $(k_{z1}^+, k_{z1}^-)$, and (ii) the ascending and descending thermal conduction-wave modes associated to the pair $(k_{z2}^+, k_{z2}^-)$.

**Appendix B. Derivation of the ion-drag terms**

In this appendix we derive the expressions for the ion-drag terms that enter the hydrodynamic equations (160)–(162).

**Ion equations**

For each ion species $i$, the ion continuity equation is

$$\frac{\partial n_i}{\partial t} + \nabla \cdot (n_i \mathbf{u}_i) = P_i - n_i \mathcal{L}_i, \tag{181}$$

and the corresponding ion momentum equation, including pressure gradient, electric field, magnetic field, gravity, and collisions, is

$$\frac{\partial \mathbf{u}_i}{\partial t} + (\mathbf{u}_i \cdot \nabla)\mathbf{u}_i = -\frac{1}{m_i n_i}\nabla p_i + \frac{q_i}{m_i}\mathbf{E} + \frac{q_i}{m_i}\mathbf{u}_i \times \mathbf{B} + \mathbf{g}$$
$$- \nu_{in}(\mathbf{u}_i - \mathbf{u}) - \sum_j \nu_{ij}(\mathbf{u}_i - \mathbf{u}_j). \tag{182}$$

Here $n_i$, $\mathbf{u}_i$, $T_i$, $m_i$, and $p_i = n_i k_\mathrm{B} T_i$ are the number density, velocity, temperature, mass, and pressure of ion species $i$; $\mathbf{E}$ is the electric field, $\mathbf{B}$ the magnetic field, $q_i$ the ion charge, $\mathbf{g}$ the gravitational acceleration, and $k_\mathrm{B}$ the Boltzmann constant. The quantities $P_i$ and $\mathcal{L}_i$ denote the ionization production rate and the loss rate due to chemical processes of ion $i$, respectively. The neutral wind velocity is $\mathbf{u}$, and the collision frequencies $\nu_{in}$ and $\nu_{ij}$ describe ion–neutral and ion–ion collisions.

In addition to ion equations, we consider the electron momentum equation

$$0 = -\frac{1}{m_e n_e}\nabla p_e - \frac{e}{m_e}\mathbf{E} - \frac{e}{m_e}\mathbf{u}_e \times \mathbf{B}, \tag{183}$$

where $n_e$, $\mathbf{u}_e$, $T_e$, $m_e$, and $p_e = n_e k_\mathrm{B} T_e$ are the number density, velocity, temperature, mass, and pressure of electrons. In Eq. (183), electron inertia is neglected because of the small electron mass, while electron collisional terms are neglected because $\nu_e \ll \Omega_e$, where $\nu_e$ denotes the electron collision frequencies and $\Omega_e$ is the electron cyclotron frequency.

In the ion momentum equation we neglect the ion inertia and the ion–ion collisions, and introduce the drift velocity $\mathbf{u}_\mathrm{D}$ by writing $\mathbf{u}_i = \mathbf{u} + \mathbf{u}_\mathrm{D}$. This yields

$$m_i \nu_{in} \mathbf{u}_\mathrm{D} = q_i \left[\mathbf{E} + (\mathbf{u} + \mathbf{u}_\mathrm{D}) \times \mathbf{B}\right] - \frac{1}{n_i}\nabla p_i + m_i \mathbf{g}. \tag{184}$$

The momentum equation is projected along the direction of the magnetic field $\widehat{\mathbf{b}}$ and perpendicular to it. The parallel and perpendicular ·force-balance equations are

$$m_i \nu_{in} u_{\mathrm{D}\parallel} = q_i E_\parallel - \frac{1}{n_i}\left(\nabla p_i\right)_\parallel + m_i g_\parallel, \tag{185}$$

and

$$m_i \nu_{in} \mathbf{u}_{\mathrm{D}\perp} = q_i \left[ \mathbf{E}_\perp + (\mathbf{u}_\perp + \mathbf{u}_{\mathrm{D}\perp}) \times \mathbf{B} \right] - \frac{1}{n_i} (\nabla p_i)_\perp + m_i \mathbf{g}_\perp, \qquad (186)$$

respectively, where in general $\mathbf{a}_\| = (\mathbf{a} \cdot \widehat{\mathbf{b}}) \widehat{\mathbf{b}} = a_\| \widehat{\mathbf{b}}$ and $\mathbf{a}_\perp = \mathbf{a} - \mathbf{a}_\|$. The parallel transport equation for electrons reduces to

$$E_\| = -\frac{1}{e n_e} \frac{\partial p_e}{\partial b}. \qquad (187)$$

The parallel and perpendicular force-balance equations are solved as follows. We first consider the ambipolar diffusion velocity, and then the electromagnetic drift velocity.

1. **Ambipolar diffusion velocity**. For a single dominant ion species of charge $q_i = +e$, the parallel force balance equation for ions becomes

$$m_i \nu_{in} u_{\mathrm{D}\|} = e E_\| - \frac{1}{n_i} \frac{\partial p_i}{\partial b} + m_i g_\|, \qquad (188)$$

where $\partial p_i / \partial b = \nabla p_i \cdot \widehat{\mathbf{b}}$. Inserting Eq. (187) into Eq. (188), gives

$$m_i \nu_{in} u_{\mathrm{D}\|} = -\left( \frac{1}{n_e} \frac{\partial p_e}{\partial b} + \frac{1}{n_i} \frac{\partial p_i}{\partial b} \right) + m_i g_\|, \qquad (189)$$

where $g_\| = \mathbf{g} \cdot \widehat{\mathbf{b}}$. Using the ideal-gas relations $p_i = n_i k_{\mathrm{B}} T_i$ and $p_e = n_e k_{\mathrm{B}} T_e$, assuming quasi-neutrality $n_e = n_i$, and thermal equilibrium $T_i = T_e = T$, we obtain

$$u_{\mathrm{D}\|} = -D_{\mathrm{A}} \left( \frac{1}{n_i} \frac{\partial n_i}{\partial b} + \frac{1}{T} \frac{\partial T}{\partial b} \right) + \frac{g_\|}{\nu_{in}}, \qquad (190)$$

where

$$D_{\mathrm{A}} = \frac{2 k_{\mathrm{B}} T}{m_i \nu_{in}} \qquad (191)$$

is the ambipolar diffusion coefficient.

2. **Electromagnetic drift velocity**. Neglecting perpendicular pressure-gradient and gravity terms, which are often small compared to the electromagnetic drift terms, and assuming a collisionless or weakly collisional limit in which the ion–neutral drag term is negligible, the perpendicular momentum balance reduces to

$$\mathbf{E}_\perp + (\mathbf{u}_\perp + \mathbf{u}_{\mathrm{D}\perp}) \times \mathbf{B} = 0. \qquad (192)$$

With $\mathbf{u}_{i\perp} = \mathbf{u}_\perp + \mathbf{u}_{\mathrm{D}\perp}$, this gives $\mathbf{u}_{i\perp} \times \mathbf{B} = -\mathbf{E}_\perp$, whose solution is the electromagnetic drift velocity

$$\mathbf{u}_{\mathrm{E}} = \frac{\mathbf{E}_\perp \times \mathbf{B}}{B^2} = \frac{\mathbf{E} \times \mathbf{B}}{B^2}. \qquad (193)$$

Consequently,

$$\mathbf{u}_{\mathrm{D}\perp} = \mathbf{u}_{\mathrm{E}} - \mathbf{u}_\perp. \qquad (194)$$

Collecting all contributions, the ion velocity can be written as

$$\mathbf{u}_i = \mathbf{u}_\parallel + \mathbf{u}_{D\parallel} + \mathbf{u}_E, \tag{195}$$

where $\mathbf{u}_\parallel = (\mathbf{u} \cdot \widehat{\mathbf{b}})\widehat{\mathbf{b}}$ is the field-aligned neutral velocity, $\mathbf{u}_{D\parallel} = u_{D\parallel}\widehat{\mathbf{b}}$ is the ambipolar diffusion velocity given by Eq. (190), and $\mathbf{u}_E$ is the electromagnetic drift velocity given by Eq. (193). This expression follows from the decomposition $\mathbf{u}_i = \mathbf{u} + \mathbf{u}_D$, together with $\mathbf{u}_{D\perp} = \mathbf{u}_E - \mathbf{u}_\perp$, so that the perpendicular neutral velocity cancels.

In our model, the ion velocity is assumed to be aligned with the magnetic field lines. This assumption is introduced to decouple the hydrodynamic and ion equation systems. Accordingly, the ion velocity is approximated by

$$\mathbf{u}_i \approx (\mathbf{u} \cdot \widehat{\mathbf{b}})\widehat{\mathbf{b}} + u_{D\parallel}\widehat{\mathbf{b}}. \tag{196}$$

This approximation is justified when perpendicular ion transport is small compared to the dominant field-aligned diffusion. The resulting formulation captures the leading-order effects of ambipolar diffusion along the magnetic field lines, while deliberately neglecting perpendicular electrodynamic coupling, such as cross-field advection and $\mathbf{E} \times \mathbf{B}$ drifts. Consequently, the model is applicable to regimes in which field-aligned transport dominates and perpendicular electrodynamic effects play a secondary role. We note, however, that the neglect of the electromagnetic drift velocity is not appropriate for all geophysical regimes. At high latitudes, ion convection is largely controlled by magnetospheric forcing [50], and realistic modeling generally requires externally imposed convection electric fields, for example from empirical models such as Weimer [51]. At mid-latitudes, perpendicular ion motion may be influenced by inter-hemispheric coupling and neutral-wind differences between conjugate hemispheres [52, 53]. A fully self-consistent electrodynamic formulation, in which the electric field is obtained from an electrostatic potential $\Phi$ via $\mathbf{E} = -\nabla\Phi$, with $\Phi$ determined from quasi-neutral current continuity and the conductivity-tensor relation [51], is therefore beyond the scope of the present study but constitutes an important extension for future work.

In the following, for simplicity, the subscript $\parallel$ is omitted, and we write $\mathbf{u}_D$ and $u_D$ instead of $\mathbf{u}_{D\parallel}$ and $u_{D\parallel}$, respectively. Let

$$\mathbf{g} = (0, 0, -g), \ \ \widehat{\mathbf{b}} = (-\cos I, 0, -\sin I), \ \ \mathbf{u} = (u, 0, w), \tag{197}$$

where $I$ is the geomagnetic inclination. The field-aligned derivative is

$$\frac{\partial}{\partial b} = \nabla \cdot \widehat{\mathbf{b}} = -\left(\cos I \frac{\partial}{\partial x} + \sin I \frac{\partial}{\partial z}\right),$$

and the scalar field-aligned diffusion velocity becomes (cf. Eq. (190))

$$u_D = D_A \left(\cos I \frac{1}{n_i}\frac{\partial n_i}{\partial x} + \sin I \frac{1}{n_i}\frac{\partial n_i}{\partial z} + \cos I \frac{1}{T}\frac{\partial T}{\partial x} + \sin I \frac{1}{T}\frac{\partial T}{\partial z}\right) + \frac{g}{\nu_{in}}\sin I. \tag{198}$$

**Linearized equations**

The linearized continuity equation, together with the linearized expressions for the diffusion velocity, ion-drag force, and ion-drag heating, are as follows.

1. **Ion continuity equation**. Neglecting the perturbed production and loss terms, the perturbed ion continuity equation is

$$\frac{\partial n_i'}{\partial t} + n_i'\nabla \cdot \mathbf{u}_{i0} + \mathbf{u}_{i0} \cdot \nabla n_i' + n_{i0}\nabla \cdot \mathbf{u}_i' + \mathbf{u}_i' \cdot \nabla n_{i0} = 0. \tag{199}$$

Using Eq. (197) and the standard assumptions $n_{i0} = n_{i0}(z)$, $\mathbf{u}_0(z) = (u_0(z), 0, 0)$, and $u_{\mathrm{D}0} = u_{\mathrm{D}0}(z)$, we obtain

$$\frac{\partial}{\partial t}\left(\frac{n_i'}{n_{i0}}\right) = -u_{\mathrm{D}0}\left[\frac{\partial}{\partial b}\left(\frac{n_i'}{n_{i0}}\right) - \frac{1}{n_{i0}}\frac{\mathrm{d}n_{i0}}{\mathrm{d}z}\sin I\left(\frac{n_i'}{n_{i0}}\right)\right] + \frac{\mathrm{d}u_{\mathrm{D}0}}{\mathrm{d}z}\sin I\left(\frac{n_i'}{n_{i0}}\right)$$

$$+ \left(\frac{\partial u'}{\partial b} - \frac{1}{n_{i0}}\frac{\mathrm{d}n_{i0}}{\mathrm{d}z}\sin I u'\right)\cos I + \left(\frac{\partial w'}{\partial b} - \frac{1}{n_{i0}}\frac{\mathrm{d}n_{i0}}{\mathrm{d}z}\sin I w'\right)\sin I$$

$$- \left(\frac{\partial u_{\mathrm{D}}'}{\partial b} - \frac{1}{n_{i0}}\frac{\mathrm{d}n_{i0}}{\mathrm{d}z}\sin I u_{\mathrm{D}}'\right) + u_0\cos I\frac{\partial}{\partial b}\left(\frac{n_i'}{n_{i0}}\right)$$

$$- \cos I \sin I\left(\frac{\mathrm{d}u_0}{\mathrm{d}z} + u_0\frac{1}{n_{i0}}\frac{\mathrm{d}n_{i0}}{\mathrm{d}z}\right)\left(\frac{n_i'}{n_{i0}}\right). \tag{200}$$

2. **Diffusion velocity**. For the perturbed diffusion velocity $u_{\mathrm{D}}'$, we have the representation

$$u_{\mathrm{D}}' = -D_{\mathrm{A}0}\frac{\partial}{\partial b}\left(\frac{T'}{T_0}\right) - D_{\mathrm{A}0}\frac{\partial}{\partial b}\left(\frac{n_i'}{n_{i0}}\right)$$

$$+ \left(\frac{1}{n_{i0}}\frac{\mathrm{d}n_{i0}}{\mathrm{d}z} + \frac{1}{T_0}\frac{\mathrm{d}T_0}{\mathrm{d}z}\right)\sin I D_{\mathrm{A}}' - \frac{g\sin I}{\nu_{in0}}\left(\frac{\nu_{in}'}{\nu_{in0}}\right). \tag{201}$$

3. **Ion-drag force and ion-drag heating**. For the ion-drag force per unit mass (cf. Eq. (126) of Appendix A),

$$\frac{1}{\rho}\mathbf{f}_{\mathrm{ID}} = \nu_{ni}(\mathbf{u} - \mathbf{u}_i),$$

we obtain

$$\left(\frac{1}{\rho}f_{\mathrm{IDx}}\right)' = \nu_{ni0}\left[\sin^2 I u' - \sin I\cos I w' + \cos I u_{\mathrm{D}}'\right.$$

$$\left. + \left(u_{\mathrm{D}0}\cos I + u_0\sin^2 I\right)\left(\frac{\nu_{ni}'}{\nu_{ni0}}\right)\right], \tag{202}$$

$$\left(\frac{1}{\rho}f_{\mathrm{IDz}}\right)' = \nu_{ni0}\left[-\sin I\cos I u' + \cos^2 I w' + \sin I u_{\mathrm{D}}'\right.$$

$$\left. + \left(u_{\mathrm{D}0}\sin I - u_0\cos I\sin I\right)\left(\frac{\nu_{ni}'}{\nu_{ni0}}\right)\right]. \tag{203}$$

For the ion-drag heating per unit mass (cf. Eq. (127) of Appendix A)

$$\frac{1}{\rho} q_{\text{ID}} = \nu_{ni} |\mathbf{u} - \mathbf{u}_i|^2,$$

we find

$$
\left(\frac{1}{\rho} q_{\text{ID}}\right)' = \nu_{ni0} \left[ 2 \left( \sin^2 I u_0 u' - \cos I \sin I u_0 w' + u_{\text{D}0} u'_{\text{D}} \right) \right.
$$
$$
\left. + \left( u_0^2 \sin^2 I + u_{\text{D}0}^2 \right) \left( \frac{\nu'_{ni}}{\nu_{ni0}} \right) \right]. \tag{204}
$$

Under the assumption that, in the ionosphere, the atomic oxygen O and $O^+$-ions are the main neutral and ionic constituents, we compute the background neutral-ion and ion-neutral collision frequencies as [46, 54]

$$\nu_{ni0} = 7.22 \times 10^{-17} T_0^{0.37} n_{i0}, \ \ \nu_{in0} = \frac{\nu_{ni0}}{n_{i0}} n_n, \ \ n = \text{O}, \ i = \text{O}^+$$

and their perturbed values as

$$D'_{\text{A}} = D_{\text{A}0} \left( \frac{T'}{T_0} - \frac{\nu'_{in}}{\nu_{in0}} \right), \tag{205}$$

$$\frac{\nu'_{in}}{\nu_{in0}} = 0.37 \frac{T'}{T_0} + \frac{\rho'}{\rho_0}, \tag{206}$$

$$\frac{\nu'_{ni}}{\nu_{ni0}} = 0.37 \frac{T'}{T_0} + \frac{n'_i}{n_{i0}}. \tag{207}$$

**Decoupled system of equations**

From Eqs. (200)–(204) we deduce that the hydrodynamic equations should be solved together with the ion continuity and momentum equations by introducing two additional state variables, namely $n'_i/n_{i0}$ and $u'_{\text{D}}$. The resulting system then consists of eight equations, obtained by augmenting the hydrodynamic system with the ion continuity and momentum equations. This fully coupled approach was used by Shibata [46].

In our analysis we instead employ a simplified, approximate model that decouples the hydrodynamic and ion equations. We have several options.

1. **Klostermeyer Approximation**. Klostermeyer solved the ion-continuity equation by neglecting the diffuse velocity and neutral winds, i.e.,

$$
\frac{\partial}{\partial t} \left( \frac{n'_i}{n_{i0}} \right) = - \left( \cos I \frac{\partial u'}{\partial x} + \sin I \frac{\partial u'}{\partial z} + \frac{1}{n_{i0}} \frac{\mathrm{d} n_{i0}}{\mathrm{d} z} \sin I u' \right) \cos I
$$
$$
- \left( \cos I \frac{\partial w'}{\partial x} + \sin I \frac{\partial w'}{\partial z} + \frac{1}{n_{i0}} \frac{\mathrm{d} n_{i0}}{\mathrm{d} z} \sin I w' \right) \sin I, \tag{208}
$$

and then computed the ion-drag force and ion-drag heating by including the background neutral wind, i.e.,

$$\left(\frac{1}{\rho}f_{\text{IDx}}\right)' = \nu_{ni0}\left[\sin^2 I u' - \sin I \cos I w' + u_0 \sin^2 I \left(\frac{\nu'_{ni}}{\nu_{ni0}}\right)\right], \quad (209)$$

$$\left(\frac{1}{\rho}f_{\text{IDz}}\right)' = \nu_{ni0}\left[-\sin I \cos I u' + \cos^2 I w' - u_0 \cos I \sin I \left(\frac{\nu'_{ni}}{\nu_{ni0}}\right)\right], \quad (210)$$

and

$$\left(\frac{1}{\rho}q_{\text{ID}}\right)' = \nu_{ni0}\left[2\left(\sin^2 I u_0 u' - \cos I \sin I u_0 w'\right) + u_0^2 \sin^2 I \left(\frac{\nu'_{ni}}{\nu_{ni0}}\right)\right]. \quad (211)$$

Klostermeyer's method is best viewed as a semi–diagnostic approximation: one solves a simplified ion continuity equation driven only by the wave-induced divergence, and then evaluates ion-drag force and heating, including $u_0$ and $\nu'_{ni}$, but assuming no diffusion velocity. In other words, field-aligned diffusion and background neutral wind are neglected in the ion continuity equation when computing $n'_i$, and the ion drag is treated as a diagnostic based on the neutral wave field and background wind.

2. **Fast Field-Aligned Diffusion**. In the second method, we assume that field-aligned diffusion is sufficiently strong that the relative ion perturbation and the perturbed diffusion velocity are nearly constant along the magnetic field line (but can vary across it):

$$\frac{\partial}{\partial b}\left(\frac{n'_i}{n_{i0}}\right) = -\left(\cos I \frac{\partial}{\partial x} + \sin I \frac{\partial}{\partial z}\right)\left(\frac{n'_i}{n_{i0}}\right) \approx 0 \quad (212)$$

and

$$\frac{\partial u'_{\text{D}}}{\partial b} = \cos I \frac{\partial u'_{\text{D}}}{\partial x} + \sin I \frac{\partial u'_{\text{D}}}{\partial z} \approx 0. \quad (213)$$

We then obtain

$$\begin{aligned}
\frac{\partial}{\partial t}\left(\frac{n'_i}{n_{i0}}\right) = &\left(u_{\text{D}0}\frac{1}{n_{i0}}\frac{\mathrm{d}n_{i0}}{\mathrm{d}z} - \cos I \frac{\mathrm{d}u_0}{\mathrm{d}z} - \cos I u_0 \frac{1}{n_{i0}}\frac{\mathrm{d}n_{i0}}{\mathrm{d}z}\right. \\
&+ \left.\frac{\mathrm{d}u_{\text{D}0}}{\mathrm{d}z}\right)\sin I \left(\frac{n'_i}{n_{i0}}\right) + \frac{1}{n_{i0}}\frac{\mathrm{d}n_{i0}}{\mathrm{d}z}\sin I u'_{\text{D}} \\
&- \left(\cos I \frac{\partial u'}{\partial x} + \sin I \frac{\partial u'}{\partial z} + \frac{1}{n_{i0}}\frac{\mathrm{d}n_{i0}}{\mathrm{d}z}\sin I u'\right)\cos I \\
&- \left(\cos I \frac{\partial w'}{\partial x} + \sin I \frac{\partial w'}{\partial z} + \frac{1}{n_{i0}}\frac{\mathrm{d}n_{i0}}{\mathrm{d}z}\sin I w'\right)\sin I \quad (214)
\end{aligned}$$

and

$$u'_{\mathrm{D}} = D_{\mathrm{A0}} \left[ \cos I \, \frac{\partial}{\partial x} \left( \frac{T'}{T_0} \right) + \sin I \, \frac{\partial}{\partial z} \left( \frac{T'}{T_0} \right) \right]$$
$$+ \left( \frac{1}{n_{i0}} \frac{\mathrm{d}n_{i0}}{\mathrm{d}z} + \frac{1}{T_0} \frac{\mathrm{d}T_0}{\mathrm{d}z} \right) \sin I \, D'_{\mathrm{A}} - \frac{g \sin I}{\nu_{in0}} \left( \frac{\nu'_{in}}{\nu_{in0}} \right). \tag{215}$$

The ion-drag force and ion-drag heating are still computed from Eqs. (202)–(204).

**Plane wave solutions**

Let

$$n'_i(x, z, t) = \overline{n}_i(z) \, \mathrm{e}^{\mathrm{j}(\omega t - k_{\mathrm{x}} x)}, \quad \overline{n}_i(z) = n_{i0}(z) \, \widehat{n}_i(z), \tag{216}$$

and (cf. Eq. (148) of Appendix A)

$$f'(x, z, t) = \overline{f}(z) \, \mathrm{e}^{\mathrm{j}(\omega t - k_{\mathrm{x}} x)}.$$

As in Appendix A, we introduce the dimensionless state variables $\widehat{u}$, $\widehat{w}$, and $\widehat{T}$, defined by

$$\overline{u} = \frac{\omega_0}{k_{\mathrm{x}}} \widehat{u}, \quad \overline{w} = \frac{\omega_0}{k_{\mathrm{x}}} \widehat{w}, \quad \frac{\overline{T}}{T_0} = \widehat{T},$$

together with their derivatives

$$\frac{\mathrm{d}\overline{u}}{\mathrm{d}z} = \frac{\omega_0}{k_{\mathrm{x}}} \widehat{\mathcal{U}}, \quad \frac{\mathrm{d}\overline{w}}{\mathrm{d}z} = \frac{\omega_0}{k_{\mathrm{x}}} \widehat{\mathcal{W}}, \quad \frac{\mathrm{d}}{\mathrm{d}z} \left( \frac{\overline{T}}{T_0} \right) = \frac{\mathrm{d}\widehat{T}}{\mathrm{d}z} = \widehat{\mathcal{T}}$$

1. **Representation of $\overline{u}_{\mathrm{D}}$.** From Eq. (215) we obtain

$$\overline{u}_{\mathrm{D}} = -\mathrm{j}k_{\mathrm{x}} D_{\mathrm{A0}} \cos I \, \widehat{T} + D_{\mathrm{A0}} \sin I \, \widehat{\mathcal{T}}$$
$$+ \left( \frac{1}{n_{i0}} \frac{\mathrm{d}n_{i0}}{\mathrm{d}z} + \frac{1}{T_0} \frac{\mathrm{d}T_0}{\mathrm{d}z} \right) \sin I \, \overline{D}_{\mathrm{A}} - \frac{g \sin I}{\nu_{in0}} \left( \frac{\overline{\nu}_{in}}{\nu_{in0}} \right). \tag{217}$$

Using

$$\frac{\overline{D}_{\mathrm{A}}}{D_{\mathrm{A0}}} = \frac{\overline{T}}{T_0} - \frac{\overline{\nu}_{in}}{\nu_{in0}}, \tag{218}$$

$$\frac{\overline{\nu}_{in}}{\nu_{in0}} = 0.37 \frac{\overline{T}}{T_0} + \frac{\overline{\rho}}{\rho_0}, \tag{219}$$

together with (cf. Eq. (150) of Appendix A)

$$\frac{\overline{\rho}}{\rho_0} = \frac{k_{\mathrm{x}}}{\Omega} \overline{u} - \frac{\mathrm{j}}{\Omega H_\rho} \overline{w} + \frac{\mathrm{j}}{\Omega} \frac{\mathrm{d}\overline{w}}{\mathrm{d}z}$$

we express $\overline{u}_{\mathrm{D}}$ as

$$\overline{u}_{\mathrm{D}} = U_{\mathrm{u}} \widehat{u} + U_{\mathrm{w}} \widehat{w} + U_{\mathrm{T}} \widehat{T} + U_{\mathcal{W}} \widehat{\mathcal{W}} + U_{\mathcal{T}} \widehat{\mathcal{T}}, \tag{220}$$

where the $U$ coefficients are given by

$$U_{\mathrm{u}} = -\frac{\omega_0}{\Omega}\left(D_{\mathrm{A}0}E + G\right), \tag{221}$$

$$U_{\mathrm{w}} = \mathrm{j}\frac{\omega_0}{\Omega}\frac{1}{k_{\mathrm{x}}H_\rho}\left(D_{\mathrm{A}0}E + G\right), \tag{222}$$

$$U_{\mathrm{T}} = -\mathrm{j}k_{\mathrm{x}}D_{\mathrm{A}0}\cos I + 0.63\,D_{\mathrm{A}0}E - 0.37\,G, \tag{223}$$

$$U_{\mathcal{W}} = -\mathrm{j}\frac{\omega_0}{\Omega}\frac{1}{k_{\mathrm{x}}}\left(D_{\mathrm{A}0}E + G\right), \tag{224}$$

$$U_{\mathcal{T}} = D_{\mathrm{A}0}\sin I, \tag{225}$$

with

$$E = \sin I\left(\frac{1}{n_{i0}}\frac{\mathrm{d}n_{i0}}{\mathrm{d}z} + \frac{1}{T_0}\frac{\mathrm{d}T_0}{\mathrm{d}z}\right), \quad G = \frac{g\sin I}{\nu_{in0}}. \tag{226}$$

2. **Representation of $\widehat{n}_i$.** From Eq. (214) we obtain

$$\left(\mathrm{j}\omega - u_{\mathrm{D}0}\frac{1}{n_{i0}}\frac{\mathrm{d}n_{i0}}{\mathrm{d}z}\sin I + \frac{\mathrm{d}u_0}{\mathrm{d}z}\cos I\sin I\right.$$
$$\left. + u_0\frac{1}{n_{i0}}\frac{\mathrm{d}n_{i0}}{\mathrm{d}z}\cos I\sin I - \frac{\mathrm{d}u_{\mathrm{D}0}}{\mathrm{d}z}\sin I\right)\widehat{n}_i$$
$$= \frac{1}{n_{i0}}\frac{\mathrm{d}n_{i0}}{\mathrm{d}z}\sin I\,\bar{u}_{\mathrm{D}} + \left(\mathrm{j}k_{\mathrm{x}}\cos I - \frac{1}{n_{i0}}\frac{\mathrm{d}n_{i0}}{\mathrm{d}z}\sin I\right)\cos I\frac{\omega_0}{k_{\mathrm{x}}}\widehat{u}$$
$$+ \left(\mathrm{j}k_{\mathrm{x}}\cos I - \frac{1}{n_{i0}}\frac{\mathrm{d}n_{i0}}{\mathrm{d}z}\sin I\right)\sin I\frac{\omega_0}{k_{\mathrm{x}}}\widehat{w} - \cos I\sin I\frac{\omega_0}{k_{\mathrm{x}}}\widehat{\mathcal{U}} - \sin^2 I\frac{\omega_0}{k_{\mathrm{x}}}\widehat{\mathcal{W}}. \tag{227}$$

After some manipulations, the expression for $\widehat{n}_i$ reads

$$\widehat{n}_i = N_{\mathrm{u}}\widehat{u} + N_{\mathrm{w}}\widehat{w} + N_{\mathrm{T}}\widehat{T} + N_{\mathcal{U}}\widehat{\mathcal{U}} + N_{\mathcal{W}}\widehat{\mathcal{W}} + N_{\mathcal{T}}\widehat{\mathcal{T}}, \tag{228}$$

where the $N$ coefficients are given by

$$N_{\mathrm{u}} = \frac{1}{N_0}\left[\left(\mathrm{j}k_{\mathrm{x}}\cos I - \frac{1}{n_{i0}}\frac{\mathrm{d}n_{i0}}{\mathrm{d}z}\sin I\right)\cos I\frac{\omega_0}{k_{\mathrm{x}}} + \frac{1}{n_{i0}}\frac{\mathrm{d}n_{i0}}{\mathrm{d}z}\sin I U_{\mathrm{u}}\right],$$
(229)

$$N_{\mathrm{w}} = \frac{1}{N_0}\left[\left(\mathrm{j}k_{\mathrm{x}}\cos I - \frac{1}{n_{i0}}\frac{\mathrm{d}n_{i0}}{\mathrm{d}z}\sin I\right)\sin I\frac{\omega_0}{k_{\mathrm{x}}} + \frac{1}{n_{i0}}\frac{\mathrm{d}n_{i0}}{\mathrm{d}z}\sin I U_{\mathrm{w}}\right],$$
(230)

$$N_{\mathrm{T}} = \frac{1}{N_0}\frac{1}{n_{i0}}\frac{\mathrm{d}n_{i0}}{\mathrm{d}z}\sin I U_{\mathrm{T}},$$
(231)

$$N_{\mathcal{U}} = -\frac{1}{N_0}\cos I\sin I\frac{\omega_0}{k_{\mathrm{x}}},$$
(232)

$$N_{\mathcal{W}} = -\frac{1}{N_0}\left(\sin^2 I\frac{\omega_0}{k_{\mathrm{x}}} - \frac{1}{n_{i0}}\frac{\mathrm{d}n_{i0}}{\mathrm{d}z}\sin I U_{\mathcal{W}}\right),$$
(233)

$$N_{\mathcal{T}} = \frac{1}{N_0}\frac{1}{n_{i0}}\frac{\mathrm{d}n_{i0}}{\mathrm{d}z}\sin I U_{\mathcal{T}},$$
(234)

and

$$N_0 = \mathrm{j}\omega - u_{\mathrm{D}0}\frac{1}{n_{i0}}\frac{\mathrm{d}n_{i0}}{\mathrm{d}z}\sin I + \frac{\mathrm{d}u_0}{\mathrm{d}z}\cos I\sin I$$
$$+ u_0\frac{1}{n_{i0}}\frac{\mathrm{d}n_{i0}}{\mathrm{d}z}\cos I\sin I - \frac{\mathrm{d}u_{\mathrm{D}0}}{\mathrm{d}z}\sin I.$$
(235)

3. **Ion-drag force and ion-drag heating**. Using Eqs. (220) and (228), together with

$$\frac{\overline{\nu}_{ni}}{\nu_{ni0}} = 0.37\frac{\overline{T}}{T_0} + \frac{\overline{n}_i}{n_{i0}} = 0.37\widehat{T} + \widehat{n}_i,$$
(236)

we obtain

$$\frac{1}{\nu_{ni0}}\overline{\left(\frac{1}{\rho}f_{\mathrm{IDx}}\right)} = F_{\mathrm{xu}}\widehat{u} + F_{\mathrm{xw}}\widehat{w} + F_{\mathrm{xT}}\widehat{T} + F_{\mathrm{x}\mathcal{U}}\widehat{\mathcal{U}} + F_{\mathrm{x}\mathcal{W}}\widehat{\mathcal{W}} + F_{\mathrm{x}\mathcal{T}}\widehat{\mathcal{T}},$$
(237)

$$\frac{1}{\nu_{ni0}}\overline{\left(\frac{1}{\rho}f_{\mathrm{IDz}}\right)} = F_{\mathrm{zu}}\widehat{u} + F_{\mathrm{zw}}\widehat{w} + F_{\mathrm{zT}}\widehat{T} + F_{\mathrm{z}\mathcal{U}}\widehat{\mathcal{U}} + F_{\mathrm{z}\mathcal{W}}\widehat{\mathcal{W}} + F_{\mathrm{z}\mathcal{T}}\widehat{\mathcal{T}},$$
(238)

$$\frac{1}{\nu_{ni0}}\overline{\left(\frac{1}{\rho}q_{\mathrm{ID}}\right)} = P_{\mathrm{u}}\widehat{u} + P_{\mathrm{w}}\widehat{w} + P_{\mathrm{T}}\widehat{T} + P_{\mathcal{U}}\widehat{\mathcal{U}} + P_{\mathcal{W}}\widehat{\mathcal{W}} + P_{\mathcal{T}}\widehat{\mathcal{T}}.$$
(239)

With

$$A_{\mathrm{x}} = u_{\mathrm{D}0}\cos I + u_0\sin^2 I, \quad A_{\mathrm{z}} = u_{\mathrm{D}0}\sin I - u_0\cos I\sin I, \quad B = u_{\mathrm{D}0}^2 + u_0^2\sin^2 I,$$
(240)

the coefficients corresponding to $f_{\mathrm{IDx}}$ are

$$F_{\mathrm{xu}} = \sin^2 I \frac{\omega_0}{k_{\mathrm{x}}} + A_{\mathrm{x}} N_{\mathrm{u}} + \cos I \, U_{\mathrm{u}}, \tag{241}$$

$$F_{\mathrm{xw}} = -\sin I \cos I \frac{\omega_0}{k_{\mathrm{x}}} + A_{\mathrm{x}} N_{\mathrm{w}} + \cos I \, U_{\mathrm{w}}, \tag{242}$$

$$F_{\mathrm{xT}} = 0.37 \, A_{\mathrm{x}} + A_{\mathrm{x}} N_{\mathrm{T}} + \cos I \, U_{\mathrm{T}}, \tag{243}$$

$$F_{\mathrm{x}\mathcal{U}} = A_{\mathrm{x}} N_{\mathcal{U}}, \tag{244}$$

$$F_{\mathrm{x}\mathcal{W}} = A_{\mathrm{x}} N_{\mathcal{W}} + \cos I \, U_{\mathcal{W}}, \tag{245}$$

$$F_{\mathrm{x}\mathcal{T}} = A_{\mathrm{x}} N_{\mathcal{T}} + \cos I \, U_{\mathcal{T}}, \tag{246}$$

the coefficients corresponding to $f_{\mathrm{IDz}}$ are

$$F_{\mathrm{zu}} = -\sin I \cos I \frac{\omega_0}{k_{\mathrm{x}}} + A_{\mathrm{z}} N_{\mathrm{u}} + \sin I \, U_{\mathrm{u}}, \tag{247}$$

$$F_{\mathrm{zw}} = \cos^2 I \frac{\omega_0}{k_{\mathrm{x}}} + A_{\mathrm{z}} N_{\mathrm{w}} + \sin I \, U_{\mathrm{w}}, \tag{248}$$

$$F_{\mathrm{zT}} = 0.37 \, A_{\mathrm{z}} + A_{\mathrm{z}} N_{\mathrm{T}} + \sin I \, U_{\mathrm{T}}, \tag{249}$$

$$F_{\mathrm{z}\mathcal{U}} = A_{\mathrm{z}} N_{\mathcal{U}}, \tag{250}$$

$$F_{\mathrm{z}\mathcal{W}} = A_{\mathrm{z}} N_{\mathcal{W}} + \sin I \, U_{\mathcal{W}}, \tag{251}$$

$$F_{\mathrm{z}\mathcal{T}} = A_{\mathrm{z}} N_{\mathcal{T}} + \sin I \, U_{\mathcal{T}}, \tag{252}$$

and the coefficients corresponding to $q_{\mathrm{ID}}$ are

$$P_{\mathrm{u}} = 2 \sin^2 I \frac{\omega_0}{k_{\mathrm{x}}} u_0 + B N_{\mathrm{u}} + 2 u_{\mathrm{D0}} U_{\mathrm{u}}, \tag{253}$$

[remaining 27,819 characters of this post omitted]

---

## Author Comment (AC2)

**Response to reviewer comments**

Based on the comments provided by the reviewers, we undertook a major revision and completely reformulated the manuscript. The major changes to the manuscript are listed below.

1. The revised manuscript is organized as follows. In Section 2, we present the derivation of the matrix exponential solution of the linearized equations, while Section 3 describes stable numerical methods for computing the amplitudes of the characteristic solution in a stratified atmosphere. Here, we focus only on the global matrix method based on matrix exponentials and the scattering matrix method for computing the amplitudes of the characteristic solutions. Section 4 addresses the computation of the perturbed quantities for both harmonic and non-harmonic source functions, that is, for single-frequency waves and time-dependent wave packets. The concepts of causality and the imaginary frequency shift are also briefly discussed. Aspects of the numerical implementation are addressed in Section 5, and representative simulation results are presented in Section 6. Additional theoretical issues are discussed in the appendices. Some of the theoretical aspects discussed, especially in Appendices A and B, may perhaps be unnecessary; however, our intention was to provide a self-consistent and complete description of the models.

2. The linearization model is described in Appendix A. We reformulate the linearized equations for the state vector $\mathbf{e} = [\widehat{u}, \widehat{w}, \widehat{T}, \widehat{\mathcal{U}}, \widehat{\mathcal{W}}, \widehat{\mathcal{T}}]^{\mathrm{T}}$, following the formulations of Vadas and Nicolls (2012) and Knight et al. (2024), instead of using $\mathbf{e} = [\widehat{u}, \widehat{w}, \widehat{p}, \widehat{T}, \widehat{\mathcal{U}}, \widehat{\mathcal{T}}]^{T}$, as adopted in earlier studies by Midgley and Liemohn (1966), Volland (1969), Francis (1973), and Yeh and Liu (1974). Appendix A provides a general model that accounts for the altitude derivatives of the background velocity $u_0$, temperature $T_0$, density scale height $H_\rho$, and dynamic viscosity $\mu_0$. In addition, it includes a simplified model for an isothermal ($T_0 = \text{constant}$), homogeneous ($\mu_{\mathrm{k}0} = \mu_0/\rho_0 = \text{constant}$), and windless atmosphere without ion drag. Appendix A is organized into the following sections: *Hydrodynamic equations, Linearized equations, Plane wave solution*, and *Dispersion equation*.

3. The computation of the ion-drag force and ion-drag heating is presented in Appendix B. In the revised version, ion-drag effects are incorporated in an approximate manner with the explicit aim of decoupling the hydrodynamic and ion equation systems, while explicitly accounting for the plasma diffusion velocity. This is achieved by neglecting perturbed production and loss terms in the ion continuity equation, reducing the ion momentum equation to ambipolar diffusion by neglecting ion inertia and ion–ion collisions and retaining only field-aligned transport, and assuming fast field-aligned diffusion so that ion perturbations and the plasma diffusion velocity remain nearly constant along magnetic field lines. Appendix

B is organized into the following sections: *Ion equations, Linearized equations, Decoupled system of equations,* and *Plane wave solutions.*

4. The application the global matrix method based on matrix exponentials and the scattering matrix method to compute the grid-point values of the state vector is described in Appendix C. This appendix is organized into the following sections: *Global matrix method with matrix exponential* and *Scattering matrix method.*

5. In Appendix D, we discuss several implementation issues related to the computation of lower and upper bounds for the wave period, the choice of frequency and time discretization for the Fourier transform, and the determination of the imaginary frequency shift using a practical, albeit heuristic, approach.

6. We plan to provide a freely available open-source code for solving the linearized gravity-wave equations on GitHub. Accordingly, only representative simulation results are presented in Section 6. The simulation results are new for two reasons: (i) we employ a new linearization model, and (ii) the previous implementation contained an error in the coefficient of thermal conductivity, which was set to $\lambda_0 = 6.71 \times 10^{-7} T_0^{0.71}$ instead of the correct value $\lambda_0 = 6.71 \times 10^{-4} T_0^{0.71}$, thereby substantially reducing the effect of thermal conduction.

For a better understanding of the revised manuscript, we have included the new version in our response (not in the final form required by the journal). Please find below our detailed replies (in black font) to the reviewer comments (in blue font).

**Reviewer 2**

We are grateful to Stephan Buchert for his insightful and pertinent comments, especially those concerning ion drag.

My review focuses on the issue of the ion drag, how it is introduced and discussed.

Comment 1)

Lines 144-165: This discussion is about taking into account ion drag in the neutral dynamics. It is based on equation (13) which seems to have appeared 1st in Klostermeyer (1972). The implications of (13) are not entirely clear to me, so I try to rephrase my understanding of the issue: In the direction parallel to the magnetic field B the ions are dragged with the neutrals, and the unperturbed velocities u|| and ui,|| are approximately the same (in lines 172-177 an approach by Shibata (1983) which includes also diffusion and gravity is mentioned, but then not applied). But the interesting yet in the manuscript not elaborated aspect is what happens in the directions perpendicular to B?

Equations (14) and (15) suggest that the background ion velocity ui,⊥ is assumed to be zero in the reference frame where the neutral wind u is given

(presumably a co-rotating frame). The drag force fID and frictional heat PID can only depend on the difference between ion and neutral velocity. So obviously $u_{i,\perp} = 0$ is assumed. At high latitudes this assumption is quite unrealistic. The aurora zone ion convection is dominated by coupling to the magnetosphere as described by Dungey (1961). The Weimer (2005) model, among others, could be used in combination with the HWM for the background difference $u_\perp$ - $u_{ui,\perp}$. At mid-latitudes there is inter-hemispheric coupling which determines the ion velocity depending on inter-hemispheric wind differences, for example according to the HWM (Laundal et al., 2025; Buchert, 2020). The SAMI2 model, as far as I understand, includes inter-hemispheric coupling by particle transport, but not electrodynamically, i.e. without a mid-latitude Sq current system.

This study is about gravity waves and linearized perturbations. According to equations (16) and (17) the background state has an effect also on the perturbed fID' and PID'. My guess is that quantitatively it is quite negligible. Also reviewer 1 remarked on the very small difference that seems to arise from the ion drag. An update to a more realistic background ion velocity model could be done, and at high latitudes the effect should then become larger than observed in the present draft. Finally I would like to remark that the perturbations by gravity waves themselves should also affect the ion velocities, which would then affect the perturbed ion drag. Equation (16) and (17) are incomplete, as it is assumed that $u_{i,\perp}$ faithfully remains unaffected by the perturbed $u_\perp$', or that $u_{i,\perp}' = 0$ as well. To abandon this assumption and solve the ion momentum equation (7) would be complicated and is understandably avoided in this work. However, ion drag forcing and dissipation by gravity wave perturbations might well be important and have comparable or even larger effect than a realistic background state via the perturbed densities in the 2nd term of equations (16) and (17). Even if a complete, linearized solution were available for both neutrals and ions, the very high mobility of electrons along magnetic field lines should have the effect that these are electric equipotential also within gravity waves. This seems not guaranteed with a complete neutral and ion solution unless additional constraints are introduced.

Alternatively, a possible assumption is that the ions are completely dragged by the neutrals, $u_i = u$ also in the perpendicular directions. This is equivalent to ignoring the ions for neutral dynamics. Compared to the effective assumption $u_{i,\perp} = 0$, which is chosen in the draft, this seems to me slightly preferable. The above mentioned condition that magnetic field-lines are electric equipotential also under perturbations by gravity waves would then generally be violated.

In summary, the authors have followed the approach in previous works when taking into account the ion-neutral coupling in atmospheric dynamics which is fine. My complaint is that the implications of the chosen approach had not been made clear up to now. Therefore I recommend to explicitly state that the Klostermeyer (1972) equation (13) implies that the ion velocities perpendicular to B, both background and perturbed ones are assumed to be zero, and this might be unrealistic in some situations. If possible, give quantitative estimates about an expected inaccuracy related to this assumption.

We fully agree with these comments. In the revised manuscript, we derive

in Appendix B the expressions for the ion-drag terms that enter the hydro-dynamic equations. In this appendix, we treat the ion equations, derive the linearized system, discuss the assumptions that lead to a decoupled set of equations, and present the corresponding plane-wave solutions. In particular, we now state: *"This approximation is justified when perpendicular ion transport is small compared to the dominant field-aligned diffusion. The resulting formulation captures the leading-order effects of ambipolar diffusion along the magnetic field lines, while deliberately neglecting perpendicular electrodynamic coupling, such as cross-field advection and $\mathbf{E} \times \mathbf{B}$ drifts. Consequently, the model is applicable to regimes in which field-aligned transport dominates and perpendicular electrodynamic effects play a secondary role. We note, however, that the neglect of the electromagnetic drift velocity is not appropriate for all geophysical regimes. At high latitudes, ion convection is largely controlled by magnetospheric forcing [47], and realistic modeling generally requires externally imposed convection electric fields, for example from empirical models such as Weimer [48]. At mid-latitudes, perpendicular ion motion may be influenced by inter-hemispheric coupling and neutral-wind differences between conjugate hemispheres [49,50]. A fully self-consistent electrodynamic formulation, in which the electric field is obtained from an electrostatic potential $\Phi$ via $\mathbf{E} = -\nabla\Phi$, with $\Phi$ determined from quasi-neutral current continuity and the conductivity-tensor relation [?], is therefore beyond the scope of the present study but constitutes an important extension for future work."*

In Section 6 (Numerical Simulations), we now state: *"Ion Drag. The effect of ion drag on the perturbed temperature, vertical velocity, and horizontal velocity is illustrated in Fig. 4. A moderate attenuation is observed in the altitude range from 180 to 350 km, where the ion number density is relatively high. When $\mathbf{E} \times \mathbf{B}$ drifts are not included, ion drag does not exhibit the classical regime in which auroral convection strongly drives the neutral atmosphere. Instead, ion drag mainly arises from diffusion- and pressure-gradient-driven ion motion along the magnetic field, as well as from any relative ion–neutral motion induced by neutral winds. Consequently, ion drag does not constitute the dominant forcing mechanism for the neutral perturbations in this configuration."*

In the Conclusions, we now add the following perspective on future extensions of the model: *"The approach presented in this paper represents only the first component of such a model. Two options are envisaged for extending it to a more complete formulation.*

1. *Fully coupled neutral–ion model. The linearized hydrodynamic equations would be solved together with the ion equations. In this case, the ion continuity equation would include perturbed production and loss terms, whereas ion inertia and ion–ion collisions would continue to be neglected in the ion momentum equation, and only transport parallel to the magnetic field lines would be retained. The state vector would then be augmented by two additional components, namely the perturbed ion number density and the ion diffusion velocity.*

2. *Two-step coupling strategy. In the first step, the neutral-atmosphere equa-*

*tions are solved using the fast field-aligned diffusion approximation. In the second step, the wave-induced perturbations obtained from the neutral solution are used as input to solve the ionospheric equations for the perturbed $O^+$ ion density. The ionospheric equations may be solved using the SAMI2 model for low latitudes, where the $\mathbf{E} \times \mathbf{B}$ drift is neglected, or the SAMI3 model at higher altitudes, where the $\mathbf{E} \times \mathbf{B}$ drift is included. In this strategy, priority is given to the ionospheric equations of the SAMI framework, while wave-induced perturbations are handled using the approximate approach developed in the present study. Along similar lines, Knight et al. [34] solved the neutral-atmosphere equations without ion drag in a first step, and subsequently addressed the ionospheric response using the Field-Line Interhemispheric Plasma (FLIP) model [44]."*

Comment 2)

Lines 181-182: "This topic will be discussed in more detail in the Conclusions." I cannot find a more detailed discussion of the topic in the Conclusions?

In the previous version of the manuscript, this topic was discussed in the Appendix rather than in the Conclusions. In the revised manuscript, we have clarified this point and now discuss the topic in detail in Appendix B.

Comment 3)

Lines 129-130: "... neglected the Coriolis force ..." The statement is not very specific. According to Klostermeyer (1972) the Earth rotation should be taken into account for wave periods t>1 hour. So I guess for long period/large scale gravity waves there could be effects from the Coriolis force. How is a limit "... gravity waves with an angular frequency w > 2W, where W = 7.3 × 10−5 s−1 is the Earth's angular velocity" justified, what does it mean in terms of small, medium, and large scales?

In Appendix A of the new version of the paper, we use the following argument: *"In the above hydrodynamic equations, the Coriolis force has been neglected. For a two-dimensional wave geometry in which both the background flow and the perturbation velocities are confined to the vertical $(x, z)$ plane and all variables are independent of the transverse horizontal coordinate y, the Coriolis acceleration associated with the Earth's rotation is directed entirely along the transverse direction and therefore does not enter the momentum equations considered here. More precisely, for $\mathbf{u} = (u, 0, w)$ and $\mathbf{\Omega} = (-\Omega \cos \phi, 0, \Omega \sin \phi)$, where $\phi$ is the geographic latitude and $\Omega = 7.29 \times 10^{-5} \ s^{-1}$ the Earth's angular velocity, the Coriolis force per unit mass is $\mathbf{f}_C = -2\mathbf{\Omega} \times \mathbf{u} = (0, 2\Omega(\cos \phi w + \sin \phi u), 0)$. Thus, the Coriolis acceleration is directed entirely along the transverse horizontal direction y and does not affect the two-dimensional $(x, z)$ momentum equations."* On the other hand, an alternative explanation is the following: *"In the above hydrodynamic equations, the Coriolis force has been neglected because we are interested in gravity waves with frequencies $\omega > 2\Omega$, corresponding to wave periods shorter than $\pi/\Omega \approx 12$ h, where $\Omega = 7.3 \times 10^{-5} \ s^{-1}$ is the Earth's angular rotation rate."*

Comment 4)

Lines 620-624: The references to the SAMI2, MSIS and HWM models are a

bit old, updates and newer versions of these models exist. I think that using not the newest of these models is fine and does not significantly affect the results and conclusions of this work. Updating the models is not necessary. But a brief statement about which models exactly were used is recommended. According to my research the latest models would be:

Huba, J. D. (2023). On the development of the SAMI2 ionosphere model. Perspectives of Earth and Space Scientists, 4, e2022CN000195. https://doi.org/10.1029/2022CN000195

with a link to the SAMI2 code on Github, https://github.com/NRL-Plasma-Physics-Division/SAMI2

Emmert, J. T., Drob, D. P., Picone, J. M., Siskind, D. E., Jones, M. Jr., Mlynczak, M. G., et al. (2021). NRLMSIS 2.0: A whole-atmosphere empirical model of temperature and neutral species densities. Earth and Space Science, 8, e2020EA001321. https://doi.org/10.1029/2020EA001321 with link to the public code.

Drob, D. P., J. T. Emmert, J. W. Meriwether, J. J. Makela, E. Doornbos, M. Conde, G. Hernandez, J. Noto, K. A. Zawdie, S. E. McDonald, et al. (2015), An update to the Horizontal Wind Model (HWM): The quiet time thermosphere, Earth and Space Science, 2, 301–319, doi:10.1002/2014EA000089.

We added the references indicated by the reviewer and now state: *"The IRI data are subsequently used in a manner analogous to that in the SAMI2 model of the Naval Research Laboratory (https://github.com/NRL-Plasma-Physics-Division/SAMI2). In the present implementation, the ionospheric equations follow the SAMI2 framework originally developed by Huba et al. [40] and described in detail by Huba [41]. In SAMI2, the neutral atmospheric parameters–namely the neutral number density, total mass density, and temperature–are specified using the MSIS family of models. In this study, these parameters are based on the MSIS formulation of Hedin [42], while we note that more recent updates are provided by the NRLMSIS 2.0 model of Emmert et al. [43]. The meridional and zonal winds are specified using the Horizontal Wind Model. In the present implementation, we follow the formulation of Hedin et al. [44], while more recent updates are described by Drob et al. [45]."*

References

J. W. Dungey (1961), Interplanetary Magnetic Field and the Auroral Zones, Phys. Rev. Lett. 6, 47, DOI:https://doi.org/10.1103/PhysRevLett.6.47

Weimer, D. R. (2005), Improved ionospheric electrodynamic models and application to calculating Joule heating rates, J. Geophys. Res., 110, A05306, doi:10.1029/2004JA010884

Laundal, K. M., Skeidsvoll, A. S., Popescu Braileanu, B., Hatch, S. M., Olsen, N., and Vanhamäki, H.: Global inductive magnetosphere-ionosphere-thermosphere coupling, Ann. Geophys., 43, 803–833, https://doi.org/10.5194/angeo-43-803-2025, 2025.

Buchert, S. C.: Entangled dynamos and Joule heating in the Earth's ionosphere, Ann. Geophys., 38, 1019–1030, https://doi.org/10.5194/angeo-38-1019-2020, 2020. Citation: https://doi.org/10.5194/egusphere-2025-3406-RC2

We have included these references in the revised list of references.

**A numerical model for solving the linearized gravity-wave equations by a multilayer method**

Alexandru Doicu[a], Dmitry S. Efremenko[b], Thomas Trautmann[b], Adrian Doicu[b]

January 22, 2026

[a]Independent Researcher, 82110 Germering, Germany

[b]Deutsches Zentrum für Luft- und Raumfahrt (DLR), Institut für Methodik der Fernerkundung (IMF), 82234 Oberpfaffenhofen, Germany

**Abstract**

We developed a numerical model for solving the linearized gravity-wave equations using a multilayer approach that explicitly accounts for viscosity, thermal conduction, and ion drag. The solution strategy is based on a matrix-exponential formalism and comprises two classes of methods: global matrix methods and scattering matrix methods. The model supports both single-frequency waves and time-dependent wave packets. Particular emphasis is placed on the global matrix method, which exploits the structured form of the multilayer system to achieve high computational efficiency while maintaining numerical accuracy. Numerical experiments demonstrate that all methods yield identical accuracy, although the global matrix method is significantly more efficient than the scattering matrix method, especially for time-dependent wave packets. The impact of ion drag on wave characteristics is quantified within this framework. The implementation is freely available as open-source code on GitHub.

**1 Introduction**

Time-step methods [1, 2, 3] are commonly used to solve fully nonlinear sets of governing equations for upper-atmospheric gravity waves, thereby allowing the modeling of wave breaking, secondary wave generation, and weakly nonlinear effects. However, as compared to linear methods for gravity waves [4, 5, 6, 7, 8, 9, 10, 11] they are computationally expensive. In Ref. [12] it was found that a time-step model took several hours to run, while a linear method only took several seconds. In this regard, linear methods are more suitable for analyzing measured data.

[revised manuscript text omitted]

The main purpose of this article is to apply radiative transfer techniques to solve the linearized gravity-wave equations. As a prototype, we will consider the equations that describe gravity waves in the ionosphere, and that include viscosity, thermal conduction, and ion drag. In principle, a full wave model for the ionosphere comprises the hydrodynamic equations for the neutral atmosphere and the ionospheric equations. These two sets of equations are coupled through the ion drag, and should be solved together. However, to simplify the analysis, we decouple the two sets of equations by adopting a fast field-aligned diffusion approximation, which may be viewed as a generalization of an approximation originally proposed by Klostermeyer [9].

Our paper is organized as follows. In Section 2, we present the derivation of the matrix exponential solution of the linearized equations, while Section 3 describes stable numerical methods for computing the amplitudes of the characteristic solution in a stratified atmosphere. Section 4, which is largely inspired by the works of Knight et al. [11, 15, 16, 17], addresses the computation of the perturbed quantities for both harmonic and non-harmonic source functions, that is, for single-frequency waves and time-dependent wave packets. The concepts of causality and the imaginary frequency shift, which are rigorously treated in Refs. [15–17], are also briefly discussed. Aspects of the numerical implementation are addressed in Section 5, and representative simulation results are presented in Section 6. Additional theoretical issues are discussed in the appendices. Appendix A contains the linearized hydrodynamic equations for the neutral atmosphere and the derivation of the underlying system of differential equations. Appendix B outlines the linearized ionospheric equations and discusses the assumptions employed to decouple the hydrodynamic and ionospheric systems. Appendix C describes methods for computing grid-point values of the state vector in a stratified atmosphere. Appendix D addresses several implementation issues, including a practical, albeit heuristic, approach for determining the imaginary frequency shift.

**2 Matrix exponential solution of the linearized equations**

To design a full wave model for the ionosphere, we use the hydrodynamic equations for the neutral atmosphere and the ionospheric equations. In a linearized (perturbation) method, a quantity $f$ is expressed as

$$f = f_0 + f', \tag{1}$$

where $f_0$ and $f'$ are the unperturbed (background) and the perturbed quantity, respectively. The perturbations are assumed to be small so that it is justified to neglect all terms of higher than the first order.

Concretely, we solve the linearized hydrodynamic equations for the neutral atmosphere together with the linearized ion continuity and momentum equations. The linearized neutral-atmosphere equations are solved under the following assumptions:

**A1.** The geographic and geomagnetic coordinates are identical.

**A2.** The wave propagates in the meridional plane (the $x$-coordinate is positive southwards while the $z$-coordinate is positive upwards), i.e.,

$$f = f(x, z, t). \tag{2}$$

**A3.** All background (unperturbed) quantities vary only in the $z$-direction, i.e.,

$$f_0 = f_0(z), \tag{3}$$

while all perturbations vary harmonically in time and the $x$-direction, i.e.,

$$f' = f'(x, z, t) = \overline{f}(z)e^{j(\omega t - k_x x)}, \tag{4}$$

where $\omega$ is the angular frequency and $k_x$ the horizontal wavenumber. Note that in some gravity-wave studies, the opposite sign convention for frequency and horizontal wavenumber is used (e.g. Ref. [34]).

The linearization model is described in Appendix A. It provides a general framework that accounts for the altitude derivatives of the background velocity $u_0$, temperature $T_0$, density scale Height $H_\rho$, and dynamic viscosity $\mu_0$. Apart from the ion-drag terms, the formulation follows a structure similar to those employed by Vadas and Nicolls [35] and Knight et al. [11].

The computation of the ion-drag force and ion-drag heating is presented in Appendix B. The ion-drag terms are introduced in an approximate manner, with the explicit aim of decoupling the hydrodynamic and ion equation systems. To this end, we adopt the following assumptions:

**B1.** In the ion continuity equation, the perturbed production and loss terms are neglected.

**B2.** In the ion momentum equation, ion inertia and ion–ion collisions are neglected, and only transport parallel to the magnetic field lines is retained. Under these assumptions, the ion momentum equation reduces to the ambipolar diffusion equation.

**B3.** To decouple the ion continuity equation from the diffusion equation, fast field-aligned diffusion is assumed, meaning that the field-aligned diffusion is sufficiently strong for the relative ion perturbation and the perturbed diffusion velocity to remain nearly constant along a magnetic field line.

The linearized equations lead to a linear system of ordinary differential equations, written in matrix form as

$$\frac{1}{k_{\mathrm{x}}}\frac{\mathrm{d}\mathbf{e}}{\mathrm{d}z} = \mathbf{A}\mathbf{e}, \tag{5}$$

where

$$\mathbf{e} = [\widehat{u}, \widehat{w}, \widehat{T}, \widehat{\mathcal{U}}, \widehat{\mathcal{W}}, \widehat{\mathcal{T}}]^{\mathrm{T}} \tag{6}$$

is the state vector, and A is the propagation matrix with altitude independent elements (whose expressions follow from Eqs. (157)–(162) of Appendix A). In general, the unknowns (the hat quantities in Eq. (6)) are defined through the relation

$$\overline{f}(z) = C(z)\widehat{f}(z), \tag{7}$$

where $\overline{f}$ is defined by Eq. (4), and $C$ is a known quantity that ensures that $\widehat{f}$ is dimensionless and that may or may not depend on altitude (here, we indicate that $C$ depend on $z$). Specifically, for the background velocity $\mathbf{u}_0 = (u_0, 0, 0)$, and the perturbed velocity $\mathbf{u}' = (u', 0, w')$, we have (cf. Eqs. (155) and (156) of Appendix A)

$$\overline{u}(z) = \frac{\omega_0}{k_{\mathrm{x}}}\widehat{u}(z), \tag{8}$$

$$\overline{w}(z) = \frac{\omega_0}{k_{\mathrm{x}}}\widehat{w}(z), \tag{9}$$

$$\overline{T}(z) = T_0(z)\widehat{T}(z), \tag{10}$$

and

$$\widehat{\mathcal{U}} = \frac{\mathrm{d}\widehat{u}}{\mathrm{d}z}, \quad \widehat{\mathcal{W}} = \frac{\mathrm{d}\widehat{w}}{\mathrm{d}z}, \quad \widehat{\mathcal{T}} = \frac{\mathrm{d}\widehat{T}}{\mathrm{d}z}.$$

where $\omega_0$ is a reference frequency.

If $(\lambda_n, \mathbf{v}_n)$ is an eigenpair of the matrix A, i.e., $\mathbf{A}\mathbf{v}_n = \lambda_n\mathbf{v}_n$ for $n = 1, \ldots, N$, where $N = \dim(\mathbf{e})$, the general solution of Eq. (5) is a linear combination of the characteristic solutions $\exp(k_{\mathrm{x}}\lambda_n z)\mathbf{v}_n$, that is,

$$\mathbf{e}(z) = \sum_{n=1}^{N} a_n e^{k_{\mathrm{x}}\lambda_n z}\mathbf{v}_n$$

$$= [\mathbf{v}_1, \ldots \mathbf{v}_N]\begin{bmatrix} e^{k_{\mathrm{x}}\lambda_1 z} & \cdots & 0 \\ \vdots & \ddots & \vdots \\ 0 & \cdots & e^{k_{\mathrm{x}}\lambda_N z} \end{bmatrix}\begin{bmatrix} a_1 \\ \vdots \\ a_N \end{bmatrix}$$

$$= \mathrm{V}\mathrm{diag}[e^{k_{\mathrm{x}}\lambda_n z}]\mathbf{a}, \tag{11}$$

where

$$\mathrm{V} = [\mathbf{v}_1, \ldots, \mathbf{v}_N], \ \mathrm{diag}[e^{k_{\mathrm{x}}\lambda_n z}] = \begin{bmatrix} e^{k_{\mathrm{x}}\lambda_1 z} & \cdots & 0 \\ \vdots & \ddots & \vdots \\ 0 & \cdots & e^{k_{\mathrm{x}}\lambda_N z} \end{bmatrix}, \tag{12}$$

and $\mathbf{a} = [a_1, \ldots, a_N]^T$. At $z = 0$, we have $\mathbf{e}(0) = \mathrm{V}\mathbf{a}$; thus,

$$\mathbf{a} = \mathrm{V}^{-1}\mathbf{e}(0), \tag{13}$$

implying (cf. Eq. (11)),

$$\mathbf{e}(z) = \mathrm{Vdiag}[\mathrm{e}^{k_\mathrm{x}\lambda_n z}]\mathrm{V}^{-1}\mathbf{e}(0) = \mathrm{e}^{k_\mathrm{x}\mathrm{A}z}\mathbf{e}(0), \tag{14}$$

and conversely,

$$\mathbf{e}(0) = \mathrm{Vdiag}[\mathrm{e}^{-k_\mathrm{x}\lambda_n z}]\mathrm{V}^{-1}\mathbf{e}(z) = \mathrm{e}^{-k_\mathrm{x}\mathrm{A}z}\mathbf{e}(z). \tag{15}$$

From the theory of gravity waves within an isothermal, nondissipative atmosphere, it is generally known that the amplitude of an ascending modes increases like $\exp[z/(2H_\mathrm{a})]$, where $H_\mathrm{a}$ is the atmospheric scale height [36]. This is necessary to keep the wave energy constant in an atmosphere where the pressure decreases exponentially with height. In this regard, we define the vertical wavenumber $k_{zn}$ through the relation

$$\mathrm{diag}[\mathrm{e}^{k_\mathrm{x}\lambda_n z}] = \mathrm{diag}[\mathrm{e}^{z/(2H_\mathrm{a})}\mathrm{e}^{-\mathrm{j}k_{zn} z}], \tag{16}$$

yielding

$$\lambda_n = -\frac{\mathrm{j}}{k_\mathrm{x}}k_{zn} + \frac{1}{2}\alpha, \tag{17}$$

and conversely,

$$k_{zn} = \mathrm{j}k_\mathrm{x}\left(\lambda_n - \frac{1}{2}\alpha\right), \tag{18}$$

where $\alpha = 1/(k_\mathrm{x}H_\mathrm{a})$. The characteristic equation $\det(\mathrm{A} - \lambda \mathrm{I}_N) = 0$ has $N = 6$ solutions. As shown in Appendix A, for a constant kinematic viscosity the solutions occur in pairs and correspond to (i) ascending and descending gravity-wave modes, (ii) ascending and descending viscosity-wave modes, and (iii) ascending and descending thermal-conduction wave modes [6, 7]. In that appendix, this pairing is explicitly demonstrated by deriving the dispersion relation for the special case of an isothermal (constant background temperature), homogeneous (constant kinematic viscosity), and windless atmosphere without ion drag. This solution classification is made according to the imaginary part of the vertical wavenumber $k_{zn}$. In the more realistic case of a constant background dynamic viscosity, it is generally not possible to define ascending and descending modes as corresponding pairs (see Eq. (168) in Appendix A). However, in our model we will use the same rule as in the case of a homogeneous atmosphere, even though the traditional concept of classifying waves in pairs is no longer applicable. Specifically, we compute $k_{zn}$ for $n = 1, \ldots, N$ by means of Eq. (18), and order the set $\{k_{zn}\}_{n=1}^N$, and accordingly, $\{\lambda_n\}_{n=1}^N$, such that

$$\mathrm{Im}(k_{z3}) < \mathrm{Im}(k_{z2}) < \mathrm{Im}(k_{z1}) < \mathrm{Im}(k_{z4}) < \mathrm{Im}(k_{z5}) < \mathrm{Im}(k_{z6}). \tag{19}$$

By convention, (i) the pairs $(k_{z1} = k_{z1}^+, \lambda_1 = \lambda_1^+)$ and $(k_{z4} = k_{z1}^-, \lambda_4 = \lambda_1^-)$ will correspond to ascending and descending gravity-wave modes, respectively,

(ii) the pairs $(k_{z2} = k_{z2}^+, \lambda_2 = \lambda_2^+)$ and $(k_{z5} = k_{z2}^-, \lambda_5 = \lambda_2^-)$ to ascending and descending viscosity-wave modes, respectively, and (iii) the pairs $(k_{z3} = k_{z3}^+, \lambda_3 = \lambda_3^+)$ and $(k_{z6} = k_{z3}^-, \lambda_6 = \lambda_3^-)$ to ascending and descending thermal conduction-wave modes, respectively. Thus, the vertical wavenumber is an auxiliary quantity that is used only to identify the different modes. According to the notation introduced above, $\{\lambda_m^+\}_{m=1}^M$, where $M = N/2$ is the number of modes, is the set of eigenvalues defining ascending modes, and $\{\lambda_m^-\}_{m=1}^M$ is the set of eigenvalues defining descending modes. Because $\mathrm{Re}(\lambda_n) = \mathrm{Im}(k_{zn})/k_x + \alpha/2$, it is obvious that we can put aside the concept of vertical wavenumber when identifying the different wave modes. We can simply order the set $\{\lambda_n\}_{n=1}^N$, such that

$$\mathrm{Re}(\lambda_3) < \mathrm{Re}(\lambda_2) < \mathrm{Re}(\lambda_1) < \mathrm{Re}(\lambda_4) < \mathrm{Re}(\lambda_5) < \mathrm{Re}(\lambda_6), \qquad (20)$$

and use the same classification rule as above. A commonly cited interpretation of condition (20) is that, for increasing $z$, the exponential term $\exp(k_x \lambda_n z)$ will tend to be damped more for ascending modes than for descending modes; conversely, for decreasing $z$, the roles of ascending and descending modes are reversed. However, such a classification of upgoing and downgoing roots (e.g., Ref. [6] and related works) was primarily heuristic and lacked a rigorous theoretical justification. By contrast, the approach of Knight et al. [16], which is discussed in Section 4, introduces additional constraints beyond condition (20) that are explicitly related to causality and is therefore grounded in theoretical considerations rather than heuristic arguments.

To highlight the different wave modes, we organize the state vector $\mathbf{e}(z)$ as

$$
\begin{aligned}
\mathbf{e}(z) &= \mathbf{e}_+(z) + \mathbf{e}_-(z) \\
&= \left( \sum_{m=1}^M a_m^+ e^{k_x \lambda_m^+ z} \mathbf{v}_m^+ \right) + \left( \sum_{m=1}^M a_m^- e^{k_x \lambda_m^- z} \mathbf{v}_m^- \right) \\
&= [\mathrm{V}_+, \mathrm{V}_-] \begin{bmatrix} \mathrm{diag}[e^{k_x \lambda_m^+ z}] & 0_M \\ 0_M & \mathrm{diag}[e^{k_x \lambda_m^- z}] \end{bmatrix} \begin{bmatrix} \mathbf{a}_+ \\ \mathbf{a}_- \end{bmatrix},
\end{aligned}
\qquad (21)
$$

where the eigenvector $\mathbf{v}_m^\pm$ corresponds to the eigenvalue $\lambda_m^\pm$,

$$\mathrm{V} = [\mathrm{V}_+, \mathrm{V}_-], \quad \mathrm{V}_\pm = [\mathbf{v}_1^\pm, \dots, \mathbf{v}_M^\pm], \qquad (22)$$

$$\mathbf{a} = \begin{bmatrix} \mathbf{a}_+ \\ \mathbf{a}_- \end{bmatrix}, \quad \mathbf{a}_\pm = \begin{bmatrix} a_1^\pm \\ \vdots \\ a_M^\pm \end{bmatrix}, \qquad (23)$$

and $0_M$ is the zero matrix of dimension $M \times M$. Some useful relations are listed below

1. From Eq. (13), we find

$$\mathbf{a}_+ = [\mathrm{I}_M, 0_M]\mathbf{a} = [\mathrm{I}_M, 0_M]\mathrm{V}^{-1}\mathbf{e}(0), \qquad (24)$$

$$\mathbf{a}_- = [0_M, \mathrm{I}_M]\mathbf{a} = [0_M, \mathrm{I}_M]\mathrm{V}^{-1}\mathbf{e}(0), \qquad (25)$$

where $I_M$ is the identity matrix of dimension $M \times M$.

2. From Eq. (21), that is,

$$\mathbf{e}_\pm(z) = \sum_{m=1}^{M} a_m^\pm e^{k_x \lambda_m^\pm z} \mathbf{v}_m^\pm = V_\pm \mathrm{diag}[e^{k_x \lambda_m^\pm z}] \mathbf{a}_\pm, \qquad (26)$$

we deduce that

$$\mathbf{e}_\pm(0) = V_\pm \mathbf{a}_\pm. \qquad (27)$$

3. From Eq. (14), we obtain

$$\mathbf{e}_+(z) = T_+ \mathbf{e}(0), \qquad (28)$$

where

$$T_+ = V \begin{bmatrix} \mathrm{diag}[e^{k_x \lambda_m^+ z}] & 0_M \\ 0_M & 0_M \end{bmatrix} V^{-1}, \qquad (29)$$

while from Eq. (15), we find

$$\mathbf{e}_-(0) = T_- \mathbf{e}(z), \qquad (30)$$

where

$$T_- = V \begin{bmatrix} 0_M & 0_M \\ 0_M & \mathrm{diag}[e^{-k_x \lambda_m^- z}] \end{bmatrix} V^{-1}. \qquad (31)$$

**3 Solution of the linearized equations for a stratified atmosphere**

Consider an equidistant discretization of the atmosphere, i.e., $\widehat{z}_i = z_{\min} + (i - 1)\Delta\widehat{z}$ for $i = 1, ..., 2L + 1$. A layer $l$, where $l = 1, ..., L$ and $L$ is the number of layers, is bounded from below and from above by the grid points $z_l = \widehat{z}_{2l-1}$ and $z_{l+1} = \widehat{z}_{2l+1}$, respectively, and its center is located at the grid point $\overline{z}_l = \widehat{z}_{2l}$. The atmosphere extends from $z_{\min} = z_1 = \widehat{z}_1$ to $z_{\max} = z_{L+1} = \widehat{z}_{2L+1} = z_{\min} + L(2\Delta\widehat{z})$. We adopt a numerical multilayer method [9, 15, 16], and approximate the altitude dependent matrix A in each layer $l$ by its value at the layer center, i.e., $A_l = A(\overline{z}_l)$. The eigenpairs of the propagation matrix $A_l$ are denote by $(\lambda_{nl}, \mathbf{v}_{nl})$ for $n = 1, ..., N$. The matrix differential equation (5) can be solved either (i) in terms of the amplitudes $\mathbf{a}_l$, $l = 1, ..., L$ of the characteristic solutions, or (ii) in terms of the grid-point values $\mathbf{e}_l = \mathbf{e}(z_l)$, $l = 1, ..., L$ of the state vector $\mathbf{e}(z)$. In the following we present the method based on the amplitude of the characteristic solutions, whereas the second method is described in Appendix C.

In the layers $l$ and $l + 1$, the solutions are given by (cf. Eq. (11))

$$\mathbf{e}_l(z) = V_l \mathrm{diag}[e^{k_x \lambda_{nl}(z-z_l)}] \mathbf{a}_l, \ \ z_l \leq z \leq z_{l+1}, \qquad (32)$$

and
$$\mathbf{e}_{l+1}(z) = \mathrm{V}_{l+1}\mathrm{diag}[\mathrm{e}^{k_x\lambda_{n,l+1}(z-z_{l+1})}]\mathbf{a}_{l+1}, \ \ z_{l+1} \leq z \leq z_l, \tag{33}$$

respectively. The continuity condition at the interface $z = z_{l+1}$,

$$\mathbf{e}_l(z_{l+1}) = \mathbf{e}_{l+1}(z_{l+1}), \tag{34}$$

gives

$$\mathrm{V}_l^{-1}\mathrm{V}_{l+1}\mathbf{a}_{l+1} = \mathrm{diag}[\mathrm{e}^{k_x\lambda_{n,l}\Delta_l}]\mathbf{a}_l, \tag{35}$$

where $\Delta_l = z_{l+1} - z_l$. To obtain a stable system of equations, we define a scaling matrix $\mathrm{K}_l^1$ with entries

$$[\mathrm{K}_l^1]_{nn} = \begin{cases} \mathrm{e}^{-k_x\lambda_{nl}\Delta_l}, & \mathrm{Re}(\lambda_{nl}) > 0 \\ \\ 1, & \mathrm{Re}(\lambda_{nl}) \leq 0 \end{cases}, \tag{36}$$

and a second scaling matrix $\mathrm{K}_l^0$ by

$$\mathrm{K}_l^0 = \mathrm{K}_l^1\mathrm{diag}[\mathrm{e}^{k_x\lambda_{nl}\Delta_l}], \ \mathrm{i.e.}, \ [\mathrm{K}_l^0]_{nn} = \begin{cases} 1, & \mathrm{Re}(\lambda_{nl}) > 0 \\ \\ \mathrm{e}^{k_x\lambda_{nl}\Delta_l}, & \mathrm{Re}(\lambda_{nl}) \leq 0 \end{cases}. \tag{37}$$

Multiplying Eq. (35) from the left with $\mathrm{K}_l^1$ yields the continuity equation

$$\mathbb{A}_{l,l+1}^1\mathbf{a}_{l+1} - \mathbb{A}_{l,l+1}^0\mathbf{a}_l = \mathbf{0}_{2M}, \ \ l = 1, ..., L-1, \tag{38}$$

where $\mathbf{0}_{2M}$ is the $2M$-dimensional zero vector, and

$$\mathbb{A}_{l,l+1}^1 = \mathrm{K}_l^1(\mathrm{V}_l^{-1}\mathrm{V}_{l+1}), \tag{39}$$

$$\mathbb{A}_{l,l+1}^0 = \mathrm{K}_l^0. \tag{40}$$

The scaling matrices $\mathrm{K}_l^1$ and $\mathrm{K}_l^0$ prevent a possible blow-up of the exponential terms for $\mathrm{Re}(\lambda_{nl}) > 0$ and $\mathrm{Re}(\lambda_{nl}) \leq 0$, respectively. Such scaling techniques are standard in radiative transfer theory and are commonly used to obtain stable numerical algorithms for computing the radiance field in multilayered atmospheres [29, 30].

Actually, we have $L-1$ continuity equations imposed at the levels $z_2, \ldots, z_L$ for the $L$ unknowns $\mathbf{a}_1, \ldots, \mathbf{a}_L$. The two missing equations are obtained from the lower and upper boundary conditions.

1. At the lower boundary, i.e., at $z = z_1(= z_{\min})$, we assume that only the ascending wave modes transport energy upward. In this regard, we impose that in the layer $l = 1$, we have $a_{1,l=1}^+ = s = $ finite, and that the rest of $a_{m,l=1}^+$ are zero, that is, $a_{m,l=1}^+ = 0$ for $m \neq 1$ [9]. Note that $a_{1,l=1}^+$ is the amplitude of the ascending gravity-wave modes, while the condition

$a_{m,l=1}^{+} = 0$ for $m \neq 1$ means that the amplitudes of the ascending viscosity-wave and thermal conduction-wave modes are assumed to be zero. In this case, the boundary condition for ascending modes is

$$\mathbf{e}_{l=1}^{+}(z_1) = \begin{bmatrix} \widehat{u}_{l=1}^{+}(z_1) \\ \widehat{w}_{l=1}^{+}(z_1) \\ \widehat{T}_{l=1}^{+}(z_1) \\ \widehat{\mathcal{U}}_{l=1}^{+}(z_1) \\ \widehat{\mathcal{W}}_{l=1}^{+}(z_1) \\ \widehat{\mathcal{T}}_{l=1}^{+}(z_1) \end{bmatrix} = \sum_{m=1}^{M} a_{m,l=1}^{+} \mathbf{v}_{m,l=1}^{+} = s\mathbf{v}_{1,l=1}^{+}. \qquad (41)$$

Excluding for the moment the scale factor $s$, we express the boundary condition for amplitudes,

$$\mathbf{a}_{l=1}^{+} = \begin{bmatrix} a_{1,l=1}^{+} \\ a_{2,l=1}^{+} \\ \vdots \\ a_{M,l=1}^{+} \end{bmatrix} = \mathbf{i}_1 \text{ with } \mathbf{i}_1 = \begin{bmatrix} 1 \\ 0 \\ \vdots \\ 0 \end{bmatrix}, \qquad (42)$$

in matrix form as

$$[\mathbf{I}_M, \mathbf{0}_M]\mathbf{a}_1 = [\mathbf{I}_M, \mathbf{0}_M] \begin{bmatrix} \mathbf{a}_1^{+} \\ \mathbf{a}_1^{-} \end{bmatrix} = \mathbf{i}_1, \qquad (43)$$

where in general, $\mathbf{a}_{l_0}^{\pm} = \mathbf{a}_{l=l_0}^{\pm}$, for $l_0 = 1, \ldots, L$. The boundary condition (41) is a modal (eigenvector-based) boundary condition, which imposes that the state at $z_1$ is exactly aligned with a chosen eigenmode. In this way, a pure normal mode is injected into the system.

2. A reasonable upper boundary condition is that there is no downgoing energy at great altitudes, so that the amplitudes of all descending wave modes must be zero at the upper boundary [9]. In this regard, we impose $a_{m,l=L}^{-} = 0$ for all $m = 1, \ldots, M$, in which case, in the layer $L$, the boundary condition for descending modes is

$$\mathbf{e}_{l=L}^{-}(z) = \sum_{m=1}^{M} a_{m,l=L}^{-} \mathbf{e}^{k_{\mathbf{x}} \lambda_{m,l=L}^{-} z} \mathbf{v}_{m,l=L}^{-} = \mathbf{0}_{2M} \qquad (44)$$

for all $z_L \leq z \leq z_{L+1}$. In matrix form, the boundary condition for amplitudes

$$\mathbf{a}_{l=L}^{-} = \begin{bmatrix} a_{1,l=L}^{-} \\ a_{2,l=L}^{-} \\ \vdots \\ a_{M,l=L}^{-} \end{bmatrix} = \mathbf{0}_M \qquad (45)$$

is written as

$$[\mathbf{0}_M, \mathbf{I}_M]\mathbf{a}_L = [\mathbf{0}_M, \mathbf{I}_M] \begin{bmatrix} \mathbf{a}_L^{+} \\ \mathbf{a}_L^{-} \end{bmatrix} = \mathbf{0}_M. \qquad (46)$$

Comments.

1. The scaling matrices defined by Eqs. (36) and (37) do not take into account a classification of the wave modes as ascending and descending (as defined by Eq. (20)). Consequently, the continuity equations (38) do not account for this classification, and the only equations in which it is necessary to distinguish between ascending and descending modes are the boundary condition equations (43) and (46). From this point of view, the method is similar to finite-difference methods [37, 38, 39].

2. An alternative type of lower boundary condition was proposed by Knight et al. [15, 16]. In this approach, the lower boundary condition for ascending modes is prescribed in terms of $M$ values $b_{1,k}$, $k = 1, \ldots, M$, according to (compare with Eq. (41))

$$\left[ \frac{\mathrm{d}^{k-1} \mathbf{e}^+_{l=1}}{\mathrm{d}z^{k-1}}(z_1) \right]_q = b_{1,k}, \quad k = 1, ..., M, \tag{47}$$

where the notation $[\mathbf{x}]_q$ denotes the $q$th component of the vector $\mathbf{x}$. In the present context, this refers to the first $M$ components, corresponding to $\widehat{u}$ ($q = 1$), $\widehat{w}$ ($q = 2$), and $\widehat{T}$ ($q = 3$). In Eq. 47, $k$ denotes the derivative order, and in the case $M = 3$, we have explicitly,

$$\left[ \mathbf{e}^+_{l=1}(z_1) \right]_q = b_{1,1}, \quad \left[ \frac{\mathrm{d}\mathbf{e}^+_{l=1}}{\mathrm{d}z}(z_1) \right]_q = b_{1,2}, \quad \left[ \frac{\mathrm{d}^2\mathbf{e}^+_{l=1}}{\mathrm{d}z^2}(z_1) \right]_q = b_{1,3}. \tag{48}$$

Note that Eq. 47 generalizes Eq. (2.19) in Ref. [15], which is formulated for the first state variable rather than for an arbitrary state variable. Using the relations

$$\frac{\mathrm{d}^{k-1} \mathbf{e}^+_{l=1}}{\mathrm{d}z^{k-1}}(z_1) = \sum_{m=1}^{M} a^+_{m,l=1} (k_{\mathrm{x}} \lambda^+_{m,l=1})^{k-1} \mathbf{v}^+_{m,l=1}, \quad k = 1, \ldots, M, \tag{49}$$

and

$$\left[ \frac{\mathrm{d}^{k-1} \mathbf{e}^+_{l=1}}{\mathrm{d}z^{k-1}}(z_1) \right]_q = \widehat{\mathbf{i}}^T_q \frac{\mathrm{d}^{k-1} \mathbf{e}^+_{l=1}}{\mathrm{d}z^{k-1}}(z_1) = \sum_{m=1}^{M} a^+_{m,l=1} (k_{\mathrm{x}} \lambda^+_{m,l=1})^{k-1} \widehat{\mathbf{i}}^T_q \mathbf{v}^+_{m,l=1}, \tag{50}$$

where $\widehat{\mathbf{i}}_q$ is a $2M$-dimensional vector with components (compare with Eq. (42))

$$[\widehat{\mathbf{i}}_q]_k = \begin{cases} 1, & k = q \\ 0, & k \neq q \end{cases}, \quad k = 1, \ldots, 2M, \tag{51}$$

we find

$$\sum_{m=1}^{M} [\mathsf{B}]_{mk} a^+_{m,l=1} = b_{1,k}, \quad k = 1, ...M, \tag{52}$$

where B is a matrix with entries

$$[\mathsf{B}]_{mk} = (k_\mathrm{x}\lambda^+_{m,l=1})^{k-1}\widehat{\mathbf{i}}^T_q\mathbf{v}^+_{m,l=1}, \ \ m,k = 1,\ldots,M. \tag{53}$$

Setting $\mathbf{b}_1 = [b_{1,1},\ldots,b_{1,M}]^\mathrm{T}$, we consider the boundary condition for amplitudes

$$\mathbf{a}^+_1 = \mathsf{B}^{-1}\mathbf{b}_1, \tag{54}$$

that is (compare with Eq. (43))

$$[\mathsf{I}_M, 0_M]\mathbf{a}_1 = \mathsf{B}^{-1}\mathbf{b}_1. \tag{55}$$

For the choice $b_{1,k} = 0$ with $k \geq 2$, the first component of the boundary-value vector $\mathbf{b}_1$ can be identified with the scale factor $s$, that is, $s = b_{1,1}$. Consequently, for a unit scale factor and $M = 3$, we have $\mathbf{b}_1 = [1,0,0]^\mathrm{T}$. The boundary condition (48) is a localized condition that prescribes the value of a single state variable while enforcing vanishing slope and curvature at the boundary. It effectively acts as an external driver applied to one variable and is appropriate for non-harmonic source functions. Note that this form of the lower boundary condition is used in the statement of Theorem 1 in Ref. [16]. For causality considerations, boundary conditions must be expressed in terms of state variables rather than modal amplitudes, since modes are defined in the frequency domain.

3. The eigenvectors are not uniquely defined and may be scaled by an arbitrary nonzero complex factor. When the LAPACK routine ZGEEV is used, the eigenvectors are returned with a built-in normalization, namely unit Euclidean norm together with a fixed phase convention. In the present work, we follow Knight et al. [15, 16] and apply a component-wise normalization, i.e.,

$$[\mathbf{v}_n]_j = \frac{1}{|[\mathbf{v}_n]_q|}[\mathbf{v}_n]_j, \ \ j = 1,\ldots,N,$$

in which each eigenvector is rescaled such that a selected reference component has unit magnitude ($|[\mathbf{v}_n]_q| = 1$). This reference component is chosen to correspond to a boundary value, thereby fixing the overall amplitude of the eigenmode in a manner consistent with the imposed boundary conditions.

Starting from the continuity equation (38), we will determine the amplitudes $\mathbf{a}_l$ by using two solution methods, namely, (i) the so-called global matrix method with matrix exponential and (ii) the scattering matrix method.

**3.1   Global matrix method with matrix exponential**

The continuity equations (38), and the boundary conditions (43) and (46) for a unit scale factor, are assembled into a system of equations for the stratified atmosphere, i.e.,

$$\mathbb{A}\mathbf{a} = \mathbf{b}, \tag{56}$$

where

$$
\mathbb{A} = \begin{bmatrix}
[0_M, I_M] & 0 & \dots & 0 & 0 \\
\mathbb{A}^1_{L-1,L} & -\mathbb{A}^0_{L-1,L} & \dots & 0 & 0 \\
\vdots & \vdots & \ddots & \vdots & \vdots \\
0 & 0 & \dots & \mathbb{A}^1_{12} & -\mathbb{A}^0_{12} \\
0 & 0 & \dots & 0 & [I_M, 0_M]
\end{bmatrix}, \tag{57}
$$

$$
\mathbf{a} = \begin{bmatrix}
\mathbf{a}_L \\
\mathbf{a}_{L-1} \\
\vdots \\
\mathbf{a}_2 \\
\mathbf{a}_1
\end{bmatrix}, \text{ and } \mathbf{b} = \begin{bmatrix}
\mathbf{0}_M \\
\mathbf{0}_{2M} \\
\vdots \\
\mathbf{0}_{2M} \\
\mathbf{i}_1
\end{bmatrix}. \tag{58}
$$

For the lower boundary condition (55), $\mathbf{i}_1$ in Eq. (58) should be replaced by $\mathrm{B}^{-1}\mathbf{b}_1$, where, for a unit scale factor, $\mathbf{b}_1 = [1,0,0]^{\mathrm{T}}$. The matrix $\mathbb{A}$ has $3M-1$ sub- and superdiagonals (excluding the main diagonal) and can therefore be stored in banded form and treated using standard band-matrix techniques. To solve the resulting banded system of linear equations, we employed the LA-PACK routines ZGBTRF and ZGBTRS. The routine ZGBTRF performs an LU factorization with partial pivoting of the complex band matrix, and ZGBTRS subsequently uses this factorization to solve the linear system for the prescribed right-hand side. In this approach, the inverse of the full system matrix is not computed explicitly, which improves the computational efficiency.

After solving Eq. (56), we compute the state vector as

$$
\mathbf{e}_l = \mathbf{e}(z_l) = \begin{bmatrix}
\widehat{u}(z_l) \\
\widehat{w}(z_l) \\
\widehat{T}(z_l) \\
\widehat{\mathcal{U}}(z_l) \\
\widehat{\mathcal{W}}(z_l) \\
\widehat{\mathcal{T}}(z_l)
\end{bmatrix} = \mathrm{V}_l \mathbf{a}_l, \ \ l = 1, \dots, L, \tag{59}
$$

and the wave amplitudes by means of the relation

$$
\overline{f}(z) = C(z)\widehat{f}(z), \tag{60}
$$

where $f$ stands for $u$, $w$, and $T$. The ascending and descending solution modes are computed by using Eq. (27), that is,

$$
\mathbf{e}_l^\pm = \mathrm{V}_l^\pm \mathbf{a}_l^\pm, \ l = 1, \dots, L. \tag{61}
$$

**3.2   Scattering matrix method**

We consider the continuity equation (38) and partition the matrices $\mathbb{A}^i_{l,l+1}$, with $i = 0, 1$, as

$$
\mathbb{A}^i_{l,l+1} = \begin{bmatrix}
[\mathbb{A}^i_{l,l+1}]_{11} & [\mathbb{A}^i_{l,l+1}]_{12} \\
[\mathbb{A}^i_{l,l+1}]_{21} & [\mathbb{A}^i_{l,l+1}]_{22}
\end{bmatrix}. \tag{62}
$$

Further, we define the scattering matrix at the interface between the layers $l$ and $l+1$ (in fact, at the layer grid point $z_{l+1}$), $S_{l,l+1}$ through the relation

$$\begin{bmatrix} \mathbf{a}_l^- \\ \mathbf{a}_{l+1}^+ \end{bmatrix} = S_{l,l+1} \begin{bmatrix} \mathbf{a}_l^+ \\ \mathbf{a}_{l+1}^- \end{bmatrix},$$  (63)

where

$$S_{l,l+1} = \begin{bmatrix} R_{l,l+1}^+ & T_{l,l+1}^- \\ T_{l,l+1}^+ & R_{l,l+1}^- \end{bmatrix},$$  (64)

and $R_{l,l+1}^\pm$ and $T_{l,l+1}^\pm$ with $\dim(R_{l,l+1}^\pm) = \dim(T_{l,l+1}^\pm) = M \times M$, are the reflection and transmission matrices, respectively. In analogy with radiative transfer theory (e.g., Refs. [31, 32]), Eq. (63) is referred to as the interaction principle equation at the interface $(l, l+1)$. It shows that the scattering matrix $S_{l,l+1}$ relates the amplitudes $\mathbf{a}_l^-$ and $\mathbf{a}_{l+1}^+$ of the waves leaving the interface with the amplitudes $\mathbf{a}_l^+$ and $\mathbf{a}_{l+1}^-$ of the waves entering the interface. From Eqs. (38) and (63), we find

$$\begin{bmatrix} R_{l,l+1}^+ & T_{l,l+1}^- \\ T_{l,l+1}^+ & R_{l,l+1}^- \end{bmatrix} = \begin{bmatrix} [\mathbb{A}_{l,l+1}^0]_{12} & -[\mathbb{A}_{l,l+1}^1]_{11} \\ [\mathbb{A}_{l,l+1}^0]_{22} & -[\mathbb{A}_{l,l+1}^1]_{21} \end{bmatrix}^{-1} \begin{bmatrix} -[\mathbb{A}_{l,l+1}^0]_{11} & [\mathbb{A}_{l,l+1}^1]_{12} \\ -[\mathbb{A}_{l,l+1}^0]_{21} & [\mathbb{A}_{l,l+1}^1]_{22} \end{bmatrix}.$$  (65)

We organize the computational process as an upward recurrence using the concept of a "stack". The stack $\mathcal{S}_{l_0 l}$ with $l_0 < l$, is a group of interfaces characterized by the interaction principle equation

$$\begin{bmatrix} \mathbf{a}_{l_0}^- \\ \mathbf{a}_l^+ \end{bmatrix} = \begin{bmatrix} \mathcal{R}_{l_0 l}^+ & \mathcal{T}_{l_0 l}^- \\ \mathcal{T}_{l_0 l}^+ & \mathcal{R}_{l_0 l}^- \end{bmatrix} \begin{bmatrix} \mathbf{a}_{l_0}^+ \\ \mathbf{a}_l^- \end{bmatrix},$$  (66)

where the matrices $\mathcal{R}_{l_0 l}^\pm$ and $\mathcal{T}_{l_0 l}^\pm$ are obtained through a successive application of the interaction principle equation at the interfaces $(l_0, l_0 + 1)$, $(l_0 + 1, l_0 + 2)$,...,$(l-1, l)$. Adding a new layer $l+1$, and taking into account that at the interface $(l, l+1)$, the reflection and transmission matrices are $R_{l,l+1}^\pm$ and $T_{l,l+1}^\pm$, respectively, we find that the interaction principle equation for the stack $\mathcal{S}_{l_0,l+1}$, is

$$\begin{bmatrix} \mathbf{a}_{l_0}^- \\ \mathbf{a}_{l+1}^+ \end{bmatrix} = \begin{bmatrix} \mathcal{R}_{l_0,l+1}^+ & \mathcal{T}_{l_0,l+1}^- \\ \mathcal{T}_{l_0.l+1}^+ & \mathcal{R}_{l_0.l+1}^- \end{bmatrix} \begin{bmatrix} \mathbf{a}_{l_0}^+ \\ \mathbf{a}_{l+1}^- \end{bmatrix},$$  (67)

where $\mathcal{R}_{l_0,l+1}^\pm$ and $\mathcal{T}_{l_0,l+1}^\pm$ are computed recursively by using of the "adding formulas"

$$\mathcal{R}_{l_0,l+1}^+ = \mathcal{R}_{l_0 l}^+ + \mathcal{T}_{l_0 l}^- (I - R_{l,l+1}^+ \mathcal{R}_{l_0 l}^-)^{-1} R_{l,l+1}^+ \mathcal{T}_{l_0 l}^+,$$  (68)

$$\mathcal{T}_{l_0,l+1}^- = \mathcal{T}_{l_0 l}^- (I - R_{l,l+1}^+ \mathcal{R}_{l_0 l}^-)^{-1} T_{l,l+1}^-,$$  (69)

$$\mathcal{T}_{l_0,l+1}^+ = T_{l,l+1}^+ (I - \mathcal{R}_{l_0 l}^- R_{l,l+1}^+)^{-1} \mathcal{T}_{l_0 l}^+,$$  (70)

$$\mathcal{R}_{l_0,l+1}^- = R_{l,l+1}^- + T_{l,l+1}^+ (I - \mathcal{R}_{l_0 l}^- R_{l,l+1}^+)^{-1} \mathcal{R}_{l_0 l}^- T_{l,l+1}^-,$$  (71)

for $l = l_0 + 1, ..., L - 1$. Note that Eqs. (68)–(71) are mathematically equivalent to Eqs. (4.30)–(4.33) in Ref. [16]. The procedure is initialized with $\mathcal{R}_{l_0,l_0+1}^\pm =$

$R_{l_0,l_0+1}^{\pm}$ and $\mathcal{T}_{l_0,l_0+1}^{\pm} = T_{l_0,l_0+1}^{\pm}$, and is repeated until the last interface is added to the stack. For the stack $\mathcal{S}_{1L}$, the interaction principle equation is

$$\begin{bmatrix} \mathbf{a}_1^- \\ \mathbf{a}_L^+ \end{bmatrix} = \begin{bmatrix} \mathcal{R}_{1L}^+ & \mathcal{T}_{1L}^- \\ \mathcal{T}_{1L}^+ & \mathcal{R}_{1L}^- \end{bmatrix} \begin{bmatrix} \mathbf{a}_1^+ \\ \mathbf{a}_L^- \end{bmatrix}, \tag{72}$$

and from the boundary conditions for amplitudes (42) and (45), that is, from the relations $\mathbf{a}_1^+ = \mathbf{i}_1$ and $\mathbf{a}_L^- = \mathbf{0}_M$, respectively, we find

$$\mathbf{a}_1^- = \mathcal{R}_{1L}^+\mathbf{a}_1^+ \text{ and } \mathbf{a}_L^+ = \mathcal{T}_{1L}^+\mathbf{a}_1^+. \tag{73}$$

For the lower boundary condition (55), $\mathbf{a}_1^+$ in Eq. (73) is given by $\mathbf{a}_1^+ = \mathbf{B}^{-1}\mathbf{b}_1$, where, for a unit scale factor, $\mathbf{b}_1 = [1,0,0]^{\mathrm{T}}$. To restore the entire set of amplitude vectors $\mathbf{a}_l$, we consider the interaction principle equations for the stacks $\mathcal{S}_{1l}$ and $\mathcal{S}_{lL}$, yielding

$$\mathbf{a}_l^+ = (I - \mathcal{R}_{1l}^-\mathcal{R}_{lL}^+)^{-1}\mathcal{T}_{1l}^+\mathbf{a}_1^+, \tag{74}$$

$$\mathbf{a}_l^- = \mathcal{R}_{lL}^+\mathbf{a}_l^+, \tag{75}$$

for $l = L-1, ..., 1$. The state vector and the wave amplitudes are then computed by using Eqs. (59) and (60), respectively. In contrast to the previous method, this approach requires a clear differentiation between ascending and descending modes as defined by Eq. (20).

**4 Source function**

In the derivation so far, the amplitude vector is uniquely defined up to a multiplicative factor, namely the scale factor $s$. Accordingly, the general solution can be written as $\mathbf{a}_{\mathrm{s}} = s\mathbf{a}$, where, here and it what follows, the subscript s indicates the dependence on $s$. Since $\mathbf{a}_{\mathrm{s}}$ satisfies the equation $\mathbb{A}\mathbf{a}_{\mathrm{s}} = s\mathbf{b}$ (cf. Eq. (56)), the scale factor can be interpreted as a source factor. The source factor is constant in the case of a harmonic (monochromatic) source function, corresponding to a single-frequency wave, but is time dependent for a non-harmonic source function, corresponding to a time-dependent wave packet. In this section, we describe the computation of the perturbed quantities for both harmonic and non-harmonic source functions. We also present a brief overview of the causality condition and the imaginary frequency shift introduced by Knight et al. [16], and latter extended and applied in Refs. [11, 15, 17, 34]. Although it would be sufficient to simply refer to these works, we include a short discussion here because the underlying mathematical structure provides valuable insight into the method.

**4.1 Non-harmonic source (time-dependent wave packet)**

If the source term is not purely harmonic in time (i.e., it cannot be written as a single factor $\exp(\mathrm{j}\omega t)$), the perturbed quantity $f'(x,z,t)$ is not a single-frequency wave with a specified angular frequency $\omega$. In this case, the equations

are treated in the frequency domain by considering the Fourier transform in time [15, 16, 17]. This is defined by

$$F'(x, z, \omega) = \int_{-\infty}^{\infty} f'(x, z, t) e^{-j\omega t} dt = \mathcal{F}[f'(x, z, t)](x, z, \omega) \tag{76}$$

and its inverse by

$$f'(x, z, t) = \frac{1}{2\pi} \int_{-\infty}^{\infty} F'(x, z, \omega) e^{j\omega t} d\omega = \mathcal{F}^{-1}[F'(x, z, \omega)](x, z, t). \tag{77}$$

Applying the Fourier transform to the linearized equations (145)–(147) of Appendix A, using the result

$$\mathcal{F}\left[\frac{\partial f'}{\partial t}(x, z, t)\right](x, z, \omega) = j\omega F'(x, z, \omega), \tag{78}$$

and setting

$$F'(x, z, \omega) = \overline{F}(z, \omega) e^{-jk_x x} \tag{79}$$

as the counterpart of Eq. (148) (in which the exponential term $\exp(j\omega t)$ is absorbed into $\overline{f}(z)$), together with

$$\overline{F}(z, \omega) = C(z)\widehat{F}(z, \omega) \tag{80}$$

as the counterpart of Eq. (7), we are led to the system of differential equations (157)–(162) of Appendix A (or equivalently, to the matrix differential equation (5)), but with $\widehat{F}(z, \omega)$ replacing $\widehat{f}(z)$.

At the lower boundary $z_1$, we consider the localized boundary conditions

$$f'_{sq_0}(x, z_1, t) = C_{q_0}(z_1) s(x, t), \quad \frac{\partial f'_{sq_0}}{\partial z}(x, z_1, t) = 0, \quad \frac{\partial^2 f'_{sq_0}}{\partial z^2}(x, z_1, t) = 0, \tag{81}$$

where $q_0$ takes the values 1, 2, and 3 for the horizontal velocity, vertical velocity, and temperature, respectively. In Eq. (81), the source function is given by

$$s(x, t) = As(t) e^{-jk_x x}, \tag{82}$$

with $A$ denoting the scalar source amplitude and $s(t)$ its prescribed time dependence. Here, and in what follows, the index s s is used to indicate that a quantity depends on the source function. In our implementation, the time-dependent part of the source function is chosen as

$$s(t) = e^{j\omega_0(t-t_0)} e^{-\frac{(t-t_0)^2}{2\sigma_t^2}} \tag{83}$$

with the Fourier transform

$$S(\omega) = \frac{\sqrt{2\pi}}{\sigma_\omega} e^{-j\omega t_0} e^{-\frac{(\omega-\omega_0)^2}{2\sigma_\omega^2}}, \tag{84}$$

where $\omega_0$ is the reference frequency (the central frequency in the Fourier spectrum), $t_0$ is the time at which the source function is maximum, and $\sigma_\mathrm{t}$ and $\sigma_\omega = 1/\sigma_\mathrm{t}$ are the standard deviations in the time and frequency domains, respectively. The amplitude of the source function $A$ is specified by imposing the normalization condition:

$$|\mathrm{Re}\{f'_{sq_0}(x=0,z_1,t_0)\}| = f_{\mathrm{b}q_0}, \tag{85}$$

for some prescribed boundary value $f_{\mathrm{b}q_0} > 0$. For example, in the case $q_0 = 1$, $f_{\mathrm{b}q_0}$ may be chosen as a fraction of the maximum horizontal velocity of neutrals in the south direction over the altitude range, whereas in the case $q_0 = 3$, $f_{\mathrm{b}q_0}$ may be chosen as a fraction of the maximum temperature of neutrals over the altitude range.

Applying the Fourier transform to Eq. (81) and using Eqs. (79) and (80), we obtain the following boundary conditions in the frequency domain (note that $AS(\omega)$ is the Fourier transform of $As(t)$):

$$\widehat{F}_{sq_0}(z_1,\omega) = AS(\omega), \quad \frac{\partial \widehat{F}_{sq_0}}{\partial z}(z_1,\omega) = 0, \quad \frac{\partial^2 \widehat{F}_{sq_0}}{\partial z^2}(z_1,\omega) = 0. \tag{86}$$

Comparing Eqs. (86) and (48), we see that the latter corresponds to the choice $b_{1,2} = b_{1,3} = 0$. In this case, the source factor $s = b_{1,1}$ can be identified with $AS(\omega)$. Therefore, as in Section 3, we define $\widehat{F}_q(z,\omega) = [\mathbf{e}(z,\omega)]_q$, $q = 1,2,3$, where $\mathbf{e}(z,\omega)$ denotes the solution of the differential equation (5) for a unit source factor in the frequency domain (i.e., for $\mathbf{b}_1 = [1,0,0]^\mathrm{T}$). The perturbed quantity $f'_{sq}(x,z,t)$ is then obtained by applying the inverse transform (77) to

$$F'_{sq}(x,z,\omega) = AS(\omega)\mathrm{e}^{-\mathrm{j}k_\mathrm{x}x}C_q(z)\widehat{F}_q(z,\omega), \tag{87}$$

that is,

$$f'_{sq}(x,z,t) = \frac{1}{2\pi}\int_{-\infty}^{\infty} F'_{sq}(x,z,\omega)\mathrm{e}^{\mathrm{j}\omega t}\mathrm{d}\omega = A\overline{f}_q(z,t)\mathrm{e}^{-\mathrm{j}k_\mathrm{x}x}, \tag{88}$$

where

$$\overline{f}_q(z,t) = \frac{C_q(z)}{2\pi}\int_{-\infty}^{\infty} S(\omega)\widehat{F}_q(z,\omega)\mathrm{e}^{\mathrm{j}\omega t}\mathrm{d}\omega. \tag{89}$$

For $S(\omega)$ as above, $\overline{f}_q(z,t)$ can be written as

$$\overline{f}_q(z,t) = \frac{C_q(z)}{2\pi}\int_{-\infty}^{\infty} \mathscr{S}(\omega)\widehat{F}_q(z,\omega)\mathrm{e}^{\mathrm{j}\omega(t-t_0)}\mathrm{d}\omega, \tag{90}$$

with

$$\mathscr{S}(\omega) = \frac{\sqrt{2\pi}}{\sigma_\omega}\mathrm{e}^{-\frac{(\omega-\omega_0)^2}{2\sigma_\omega^2}}. \tag{91}$$

The computation of $\widehat{F}_q(z,\omega)$ can be performed using any of the methods presented in Section 3. The computed quantity is $\overline{f}_q(z,t)$, the amplitude $A > 0$, is determined from the normalization condition (85) as

$$A = \frac{f_{bq_0}}{|\text{Re}\{\overline{f}_{q_0}(z_1, t_0)\}|},\tag{92}$$

and the perturbed quantity $f'_{sq}(x, z, t)$ is computed from Eq. (88).

**4.2   Monochromatic source (single-frequency wave)**

The case of a monochromatic source is obtained as a special case of the above approach by choosing

$$s(t) = e^{j\omega_0 t},\tag{93}$$

whose Fourier transform is

$$S(\omega) = \int_{-\infty}^{\infty} s(t)\, e^{-j\omega t}\, \mathrm{d}t = \int_{-\infty}^{\infty} e^{-j(\omega - \omega_0)t}\, \mathrm{d}t = 2\pi\delta(\omega - \omega_0),\tag{94}$$

where the equality is understood in the sense of distributions.

The lower boundary conditions in the time domain are specified as

$$f'_{sq_0}(x, z_1, t) = \frac{1}{2\pi} C_{q_0}(z_1)\, s(x, t), \quad \frac{\partial f'_{sq_0}}{\partial z}(x, z_1, t) = 0, \quad \frac{\partial^2 f'_{sq_0}}{\partial z^2}(x, z_1, t) = 0,\tag{95}$$

with

$$s(x, t) = As(t)e^{-jk_x x} = Ae^{j\omega_0 t}\, e^{-jk_x x}.\tag{96}$$

Applying the Fourier transform with respect to time yields

$$\widehat{F}_{sq_0}(z_1, \omega) = A\,\delta(\omega - \omega_0).\tag{97}$$

Since the forcing is monochromatic, the frequency-domain problem is solved only at the excitation frequency $\omega = \omega_0$. Accordingly, the localized boundary conditions are imposed directly on the harmonic amplitudes at $\omega_0$, namely

$$\widehat{F}_{sq_0}(z_1, \omega_0) = A, \quad \frac{\partial \widehat{F}_{sq_0}}{\partial z}(z_1, \omega_0) = 0, \quad \frac{\partial^2 \widehat{F}_{sq_0}}{\partial z^2}(z_1, \omega_0) = 0.\tag{98}$$

Again, by comparing Eqs. (98) and (48), we see that the source factor $s = b_{1,1}$ can be identified with $A$. In this regard, let $\widehat{F}_q(z, \omega_0) = \left[\mathbf{e}(z, \omega_0)\right]_q$, $q = 1, 2, 3$, where $\mathbf{e}(z, \omega_0)$ denotes the solution of the differential equation (5) for a unit source factor in the frequency domain. The perturbed quantity $f'_{sq}(x, z, t)$ is then obtained by the inverse Fourier transform (77) of $F'_{sq}(x, z, \omega)$ (cf. Eq. (87)), and the result is

$$f'_{sq}(x, z, t) = A\,\overline{f}_{sq}(z)e^{j(\omega_0 t - k_x x)},\tag{99}$$

where
$$\overline{f}_{sq}(z) = C_q(z)\,\widehat{F}_q(z, \omega_0). \tag{100}$$

Thus, the computed quantity is $\overline{f}_{sq}(z)$, and the amplitude $A$ is determined from the normalization condition

$$|\mathrm{Re}\{f'_{sq_0}(x=0, z_1, t=0)\}| = f_{bq_0}, \tag{101}$$

which yields

$$A = \frac{f_{bq_0}}{|\mathrm{Re}\{\overline{f}_{sq}(z_1)\}|}. \tag{102}$$

Note that for a monochromatic source, the modal boundary condition (41) can be used instead of the localized boundary condition (98).

**4.3   Causality and imaginary frequency shifting**

Causality means that the wave field in response to any source function cannot be nonzero prior to the earliest time at which the source function is nonzero. According to the classification rule (20), we have

$$\mathrm{Re}[\lambda_{1l}^+(\omega)] < \mathrm{Re}[\lambda_{1l}^-(\omega)], \tag{103}$$

for any layer $l = 1, \ldots, L$ and any real frequency $\omega$. To preserve causality in solutions of two-point boundary value problems, a stronger condition is required, namely

$$\max_{l=1,\ldots,L} \mathrm{Re}[\lambda_{1l}^+(\omega)] < \min_{l=1,\ldots,L} \mathrm{Re}[\lambda_{1l}^-(\omega)] \tag{104}$$

for all $\omega \in \mathbb{R}$. Equivalently, this condition requires that there exists a single real constant $\sigma$, such that

$$\mathrm{Re}[\lambda_{1l}^+(\omega)] < \sigma < \mathrm{Re}[\lambda_{1l}^-(\omega)], \tag{105}$$

for all $l$ and all $\omega \in \mathbb{R}$.

In some situations, condition (105) is not satisfied on the real frequency axis but can be enforced by introducing an imaginary frequency shift $\omega \to \omega - \mathrm{j}\delta$. Following Ref. [16], we impose a causality requirement, which we refer to as the Global Causality (GC) condition. This condition demands that there exists a single real constant $\sigma$, such that

$$\mathrm{Re}[\lambda_{1l}^+(\omega - \mathrm{j}\delta)] < \sigma < \mathrm{Re}[\lambda_{1l}^-(\omega - \mathrm{j}\delta)], \tag{106}$$

for all $l$ and all $\omega \in \mathbb{R}$.

Knight et al. [11] subsequently relaxed the requirement that (106) hold for all layers $l$ for a fixed $\sigma$. The new condition, which we refer to as the Layerwise Causality (LC) condition, requires that at each layer $l$ there is a $\sigma_l$ such that

$$\mathrm{Re}[\lambda_{1l}^+(\omega - \mathrm{j}\delta)] < \sigma_l < \mathrm{Re}[\lambda_{1l}^-(\omega - \mathrm{j}\delta)], \tag{107}$$

for all $\omega \in \mathbb{R}$. Equivalently, if at each layer $l$,

$$d_l(\omega) = \mathrm{Re}[\lambda_{1l}^-(\omega - \mathrm{j}\delta)] - \mathrm{Re}[\lambda_{1l}^+(\omega - \mathrm{j}\delta)] > 0, \tag{108}$$

for all $\omega \in \mathbb{R}$, then the multilayer algorithm will still preserve causality. The LC condition requires a strict separation between the two eigenvalue families within each layer, but it does not require that the same separator works for all layers. Thus, each layer may have its own separating value $\sigma_l$. Equivalently, $d_l(\omega) > 0$ means that, in layer $l$, the real parts of the eigenvalues associated with the ascending and descending gravity waves remain separated (and therefore do not cross) as functions of the real frequency $\omega$ after the shift.

To summarize the approach for computing $f'_{Sq}(x, z, t)$ in the case of imaginary frequency shifting, we introduce the shifted spectrum

$$S_\delta(\omega) = S(\omega - \mathrm{j}\delta) = \int_{-\infty}^{\infty} s(t)\mathrm{e}^{-\mathrm{j}(\omega - \mathrm{j}\delta)t}\mathrm{d}t = \int_{-\infty}^{\infty} \left[s(t)\mathrm{e}^{-\delta t}\right]\mathrm{e}^{-\mathrm{j}\omega t}\mathrm{d}t, \tag{109}$$

which can be viewed as the analytic continuation of $S(\omega)$ to complex frequencies. Note that the shift $\omega \to \omega - \mathrm{j}\delta$ corresponds in the time domain to multiplication by $\exp(-\delta t)$. Let

$$F'_{\delta sq}(x, z, \omega) = AS(\omega - \mathrm{j}\delta)C_q(z)\widehat{F}_q(z, \omega - \mathrm{j}\delta)\mathrm{e}^{-\mathrm{j}k_x x} \tag{110}$$

be the Fourier transform (in time) of the perturbed quantity with frequency shifting $f'_{\delta sq}(x, z, t)$, where as usual, $\widehat{F}_q(z, \omega - \mathrm{j}\delta) = [\mathbf{e}(z, \omega - \mathrm{j}\delta)]_q$ is solution of the differential equation (5) for a unit source factor in the frequency domain. Under the usual analyticity and decay assumptions (so that contour shifting is permitted), Cauchy's theorem yields the shift relation

$$f'_{\delta sq}(x, z, t) = \frac{1}{2\pi} \int_{-\infty}^{\infty} F'_{\delta sq}(x, z, \omega)\mathrm{e}^{\mathrm{j}\omega t}\mathrm{d}\omega = \mathrm{e}^{-\delta t} f'_{sq}(x, z, t), \tag{111}$$

where $f'_{sq}$ is the perturbed quantity without frequency shifting given by Eq. (88). Equivalently, this implies the shift-invariance property

$$f'_{sq}(x, z, t) = \mathrm{e}^{\delta t} f'_{\delta sq}(x, z, t). \tag{112}$$

Summarizing, the computational steps for the frequency-shifting approach are as follows:

1. Compute $\mathbf{e}(z, \omega - \mathrm{j}\delta)$ as the solution of the differential equation (5) for a unit source factor, and set $\widehat{F}_q(z, \omega - \mathrm{j}\delta) = [\mathbf{e}(z, \omega - \mathrm{j}\delta)]_q$.

2. Calculate $F'_{\delta sq}$ by means of Eq. (110) with $S(\omega)$ replaced by $S(\omega - \mathrm{j}\delta)$ and $\widehat{F}_q(z, \omega)$ replaced by $\widehat{F}_q(z, \omega - \mathrm{j}\delta)$.

3. Compute $f'_{\delta sq}$ by inverse Fourier transform

$$f'_{\delta sq}(x, z, t) = A\mathrm{e}^{-\mathrm{j}k_x x}\frac{C_q(z)}{2\pi} \int_{-\infty}^{\infty} S(\omega - \mathrm{j}\delta)\widehat{F}_q(\omega - \mathrm{j}\delta)e^{\mathrm{j}\omega t}\,\mathrm{d}\omega. \tag{113}$$

4. Recover $f'_{\mathrm{s}q}$ from the shift-invariance property (112).

For $S(\omega)$ as in Eq. (84), it is convenient to write the recovered solution in the form

$$f'_{sq}(x, z, t) = A\overline{f}_{\delta q}(z, t)e^{-jk_x x}, \tag{114}$$

where (compare with Eq. (90))

$$\overline{f}_{\delta q}(z, t) = e^{\delta(t-t_0)} \frac{C_q(z)}{2\pi} \int_{-\infty}^{\infty} \mathscr{S}(\omega - j\delta)\widehat{F}_q(z, \omega - j\delta)e^{j\omega(t-t_0)} d\omega, \tag{115}$$

and $\mathscr{S}$ is given by Eq. (91). The computed quantity is $\overline{f}_{\delta q}(z, t)$, the amplitude $A > 0$ is determined from the normalization condition (85) as

$$A = \frac{f_{\mathrm{b}q_0}}{|\mathrm{Re}\{\overline{f}_{\delta q}(z_1, t_0)\}|}, \tag{116}$$

and the perturbed quantity $f'_{\mathrm{s}q}(x, z, t)$ is computed from Eq. (114). The Fourier integral in Eq. (115) is evaluated using a direct discrete Fourier transform (FT) rather than a fast Fourier transform (FFT). The frequency and time discretization used in the Fourier transform are discussed in Appendix D.

A potential numerical issue with this approach is that a large frequency shift $\delta$, while ensuring the causality condition, may amplify rounding errors when recovering $f'_{\mathrm{s}q}$ from $f'_{\delta \mathrm{s}q}$ through the exponential term $\exp[\delta(t - t_0)]$. For large $t$, this may lead to an uncontrolled growth of the right-hand side of Eq. (115). Therefore, care is required in selecting $\delta$: it must be large enough to ensure the layerwise causality condition , but not significantly larger than that. Rigorous methods for determining the minimum sufficient $\delta$ were described by Knight et al. in Refs. [15, 16, 17], while the numerical blow-up associated with the exponential growth term was discussed in Appendix B of Ref. [17]. In our implementation we employ a heuristic approach that combines (i) the Layerwise Causality (LC) condition applied at selected altitude levels, and (ii) a Source-Function Reconstruction (SFR) test. First, an admissible interval $[\delta_{\min}, \delta_{\max}]$ is constructed by enforcing the SFR criterion, and within this interval, the LC condition is applied at selected altitude levels to obtain a refined lower bound $\delta_{\min}$. In the final selection step, a discrete set of candidate shifts is evaluated, and for each candidate the LC condition is checked over the entire altitude range. Among all shifts that satisfy causality at all altitudes, the algorithm selects the one whose maximum-amplitude vector is closest to the center of mass of the admissible solutions. A detailed description of this approach is provided in Appendix D.

**5  Numerical implementation**

An implementation of the method is freely available as an open-source code on GitHub. The code uses as input the data file produced by the International Reference Ionosphere (IRI) code available at https://ccmc.gsfc.nasa.gov/models/IRI~2016/ is used. From these date, we read

| Location | Latitude [deg] | Longitude [deg] | Height [km] |
|---|---|---|---|
| Jicamarca (Peru) | -12 | 283 | 300 |
| Arecibo (Puerto Rico) | +18 | 293 | 300 |
| Millstone Hill (USA) | +42 | 288 | 300 |
| Saint-Santin (France) | +44 | 2 | 300 |
| EISCAT Tromsø (Auroral) | +70 | 19 | 300 |
| Svalbard archipelago (Norway) | +80 | 15 | 300 |

Table 1: Geographic locations and heights of the IRI data sets

1. the date (year, month, and day) and the time,

2. the geographic latitude and longitude,

3. the magnetic dip angle,

4. the solar radio flux f10.7 and its 81-day average,

5. the number density of $O^+$ ions as a function of altitude.

In its present implementation, the IRI data files correspond to the locations summarized in Table 1. The altitude grid extends from $z_{min} = 80$ km to $z_{max} = 500$ km with a step size of $dz = 1.0$ km. Users may generate custom data files by running the IRI code and specifying the corresponding file names in the input namelist.

The IRI data are subsequently used in a manner analogous to that in the SAMI2 model of the Naval Research Laboratory (https://github.com/NRL-Plasma-Physics-Division/SAMI2). In the present implementation, the ionospheric equations follow the SAMI2 framework originally developed by Huba et al. [40] and described in detail by Huba [41]. In SAMI2, the neutral atmospheric parameters–namely the neutral number density, total mass density, and temperature–are specified using the MSIS family of models. In this study, these parameters are based on the MSIS formulation of Hedin [42], while we note that more recent updates are provided by the NRLMSIS 2.0 model of Emmert et al. [43]. The meridional and zonal winds are specified using the Horizontal Wind Model. In the present implementation, we follow the formulation of Hedin et al. [44], while more recent updates are described by Drob et al. [45].

The derivatives of the background parameters are computed using central finite differences. Prior to applying the finite-difference calculations, the background parameters are smoothed by means of cubic spline interpolation with regularization.

Other features of the model are summarized as follows:

1. Two linearization models are included in the code:

   (a) a general model that accounts for the altitude derivatives of the background velocity, temperature, density scale height, and dynamic viscosity; and

(b) a simplified model for an isothermal, homogeneous, and windless atmosphere without ion drag.

2. The methods for solving the linearized equations based on the matrix exponential formalism comprise:

   (a) the Global Matrix Method for the Amplitudes (GMMA) of the characteristic solutions;

   (b) the Scattering Matrix Method for the Amplitudes (SMMA) of the characteristic solutions; and

   (c) the Global Matrix Method for the Nodal (grid-point) values (GMMN) of the state vector.

3. At the lower boundary, we impose that a selected component of the state vector is finite and that its first and second derivatives with respect to height vanish. At the upper boundary, we assume that there is no downward energy propagation, i.e., the amplitudes of all descending wave modes are set to zero.

4. The code first computes the wave parameters for a single-frequency wave and then for a time-dependent wave packet.

5. Typical values of the horizontal wavelength lie in the range 300–700 km.

6. The algorithm computes lower and upper bounds for the wave period by solving the inviscid dispersion equation for two prescribed minimum and maximum values of the vertical wavelength. The computational procedure is described in Appendix D. The user then selects an appropriate value within this range.

7. For a single-frequency wave, the output quantity of interest is $A\overline{f}_{sq}(z)$, where $\overline{f}_{sq}(z)$ and $A$ are given by Eqs. (100) and (102), respectively, whereas for a time-dependent wave packet the corresponding output quantity is $A\overline{f}_{\delta q}(z,t)$, where $\overline{f}_{\delta q}(z,t)$ and $A$ are given by Eqs. (115) and (116), respectively.

**6 Numerical simulations**

The simulations are performed using, as input, an IRI data file corresponding to the EISCAT Tromsø (auroral) location on 11 February 2012 at 10:00. The solar zenith angle is 84.2°, the magnetic inclination angle is 78.28°, the daily solar radio flux F10.7 is 109.4 sfu, and the 81-day averaged solar radio flux is 116.9 sfu, where $1\,\mathrm{sfu} = 10^{-22}\ \mathrm{Wm^{-2}Hz^{-1}}$. In the simulations, the Prandtl number is 0.66, the magnetic index is 7.0, and the horizontal wind model 14 implemented in SAMI2 is used. The altitude grid extends from 80 km to 500 km and contains 801 grid points. The lower boundary can, in principle, be set to smaller altitudes (e.g., 50 km), but we choose 80 km because this level

is typically adopted as the lower boundary for ionospheric equations. Unless stated otherwise, the horizontal wavelength is $\lambda_x = 400$ km, the wave period is $\lambda_t = 40$ min, and the imaginary frequency shift for a single-frequency wave is $10^{-6}$ s$^{-1}$. The lower boundary conditions are imposed on the vertical velocity with $f_{b2} = 5 \times 10^{-2}$ ms$^{-1}$ in Eqs. (85) and (101).

*Accuracy and efficiency of the solution methods.* Taking the global matrix method for amplitudes as a reference, we find that the relative root-mean-square errors in the perturbed temperature, vertical velocity, and horizontal velocity obtained with the other two solution methods are smaller than $10^{-6}$. Thus, all methods exhibit comparable accuracy. On the other hand, we find that the scattering matrix method is more time-consuming than the global matrix methods, particularly for time-dependent wave packets. This is because the scattering matrix approach requires numerous matrix operations in each layer, whereas solving a system of equations compressed into band storage is computationally less expensive.

*Background atmospheric parameters.* The code provides altitude-dependent profiles of the input parameters used in the numerical model. These include the temperature, mass density, pressure, southward horizontal velocity, atmospheric scale height, density scale height, specific heat capacity, ratio of specific heats, sound speed, number density of O$^+$ ions, neutral–ion collision frequency, ion–neutral collision frequency, and the diffusion velocity. In addition, altitude derivatives of the temperature, horizontal velocity, mass density, pressure, density scale height, and ion number density are also provided. As an illustrative example, Fig. 1 shows the background temperature $T_0$, horizontal velocity $u_0$ and the ion number density $n_{i0}$, together with their corresponding altitude derivatives.

*Pairwise classification of ascending and descending* modes. In the upper and middle panels of Fig. 2, we plot the imaginary part of the vertical wavenumber for ascending ($k_{z1}$) and descending ($k_{z4}$) gravity waves, computed using the general and the simplified models, respectively. The plots demonstrate that only in the latter case do the vertical wavenumbers appear in pairs. In the former case, a problematic altitude range for gravity waves is observed between 80 km and 120 km, where the imaginary parts of the vertical wavenumbers for the ascending and descending modes are nearly identical, being either both positive or both negative. In the lower panel of Fig. 2, we show the imaginary part of the vertical wavenumber $k_z$ for all wave types ($k_{zn}, n = 1, \ldots, 6$), computed using the general model. The plots reveal a clear distinction between gravity waves and viscosity- and thermal-conduction waves. However, the viscosity and thermal-conduction waves are very close to each other. In the upper panel of Fig. 2, we also compare the imaginary part of the vertical wavenumber computed with and without ion drag. No pronounced effect of ion drag on the vertical wavenumber is observed. A small effect appears in the altitude range from 180 to 300 km, where the ion number density is relatively high. This finding is consistent with the results of Shibata [46], who showed that, for gravity waves, plasma diffusion is of minor importance with respect to the vertical wavenumber, which is mainly controlled by dissipation due to viscosity

[Figure]

Figure 1: Background temperature $T_0$, horizontal velocity $u_0$ and ion number density $n_{i0}$ ($i = O^+$) (upper panels), and their height derivatives (lower panels).

and thermal conduction in the neutral gas.

*General and simplified models.* The altitude profiles of the perturbed temperature, vertical velocity, and horizontal velocity computed using the general and simplified models are shown in Fig. 3. The plots show that, in the altitude range 150–300 km, the amplitudes obtained with the general model are larger than those obtained with the simplified model. If the imaginary parts of the vertical wavenumber for ascending gravity waves computed with the general and simplified models were plotted on the same graph (i.e. by merging the lower and middle panels of Fig. 2 ), it would be seen that the imaginary part of the vertical wavenumber corresponding to the general model is negative but larger (i.e., less negative) than that obtained with the simplified model. As a consequence, the exponential attenuation with altitude is weaker in the general model, leading to systematically larger wave amplitudes in this region. This difference reflects the modified balance between wave propagation and dissipation introduced by the inclusion of altitude-dependent background properties and by relaxing the assumption of constant kinematic viscosity in the general model, which reduces the effective vertical damping compared to the simplified, homogeneous approximation.

*Ion Drag.* The effect of ion drag on the perturbed temperature, vertical velocity, and horizontal velocity is illustrated in Fig. 4. A moderate attenuation is observed in the altitude range from 180 to 350 km, where the ion number density is relatively high. When $\mathbf{E} \times \mathbf{B}$ drifts are not included, ion drag does not exhibit the classical regime in which auroral convection strongly drives the neutral atmosphere. Instead, ion drag mainly arises from diffusion- and pressure-gradient-driven ion motion along the magnetic field, as well as from any relative ion–neutral motion induced by neutral winds. Consequently, ion drag does not constitute the dominant forcing mechanism for the neutral perturbations in this configuration.

*Horizontal wavelength and time period.* The influence of the horizontal wavelength $\lambda_{\mathrm{x}}$ and the wave period $\lambda_{\mathrm{t}}$ on the perturbed quantities is shown in Fig. 5. These plots indicate that the wave amplitude decreases with increasing $\lambda_{\mathrm{x}}$ and $\lambda_{\mathrm{t}}$. The underlying reasons are as follows:

1. Horizontal wavelength. As the horizontal wavelength increases, horizontal pressure gradients become weaker, which reduces the driving of the wave motion. This also weakens the coupling between horizontal and vertical motions, resulting in smaller gravity-wave amplitudes.

2. Wave period. As the wave period increases, the buoyancy restoring force acts more slowly, leading to weaker oscillations for a given forcing. In addition, dissipative processes such as viscosity and thermal diffusion act more effectively on low-frequency waves, further reducing their amplitudes.

*Computing ascending and descending wave modes.* The ascending and descending solution modes in layer $l$, denoted by $\mathbf{e}_l^+$ and $\mathbf{e}_l^-$, respectively, can be computed using the GMMA through Eq. (61) or using the GMMN via the recurrence relations (269) and (270). The total solution mode $\mathbf{e}_l$, obtained from

[Figure]

Figure 2: Upper panel: The imaginary part of the vertical wavenumber $k_z$ for ascending ($k_{z1}$) and descending ($k_{z4}$) gravity waves, with and without ion drag, computed using the general model. Middle panel: The imaginary part of the vertical wavenumber $k_z$ for ascending ($k_{z1}$) and descending ($k_{z4}$) gravity waves, computed using the simplified model. Lower panel: The imaginary part of the vertical wavenumber $k_z$ for all types of waves ($k_{zn}, n = 1, \ldots, 6$), computed using the general model. The results correspond to $\lambda_x = 400$ km and $\lambda_t = 40$ min

[Figure]

Figure 3: Altitude profiles of the perturbed temperature $\overline{T}$, vertical velocity $\overline{w}$, and horizontal velocity $\overline{u}$ for the general and simplified models. The results correspond to $\lambda_\mathrm{x} = 400$ km and $\lambda_\mathrm{t} = 40$ min

[Figure]

Figure 4: Altitude profiles of the perturbed temperature $\overline{T}$, vertical velocity $\overline{w}$, and horizontal velocity $\overline{u}$ for the general model, with and without ion drag. The results correspond to $\lambda_x = 400$ km and $\lambda_t = 40$ min

[Figure]

Figure 5: Altitude profiles of the perturbed temperature $\overline{T}$, vertical velocity $\overline{w}$, and horizontal velocity $\overline{u}$ for (i) $\lambda_x = 300$, 400, and 500 km with $\lambda_t = 40$ min (upper panels), and (ii) $\lambda_x = 400$ km with $\lambda_t = 40$, 60, and 80 min (lower panels). The results correspond to the general model with ion drag.

Eq. (59) in the GMMA formulation and by solving Eq. (266) in the GMMN formulation, should satisfy the relation $\mathbf{e}_l = \mathbf{e}_l^+ + \mathbf{e}_l^-$. In all our simulations, this identity is satisfied. Furthermore, the results shown in Fig. 6 indicate that the ascending mode is dominant, except in the altitude range between 120 km and 180 km in the case of the general model. This finding, which is consistent with the results presented by Knight et al. [16] (see their Fig. 6), suggests that in a simplified model one may assume the ascending modes to be dominant at altitudes above 200 km, that is, $\mathbf{e}_l \approx \mathbf{e}_l^+$ for $l = 1, \ldots, L$. Under this assumption, the state vector can be computed using the upward recurrence relation (269). In Ref. [16], this approach was referred to as the transmission-only approximation, whereas in Ref. [17] a related single-mode approximation was introduced.

*Time-dependent wave packet.* For the source function (82)–(83), Fig. 7 shows the perturbed temperature and vertical velocity as functions of time and altitude. Note the different time intervals used for each horizontal wavelength $\lambda_x$ in these plots. The maximum values of the perturbed temperature are 32.21 K, 31.15 K, and 32.73 K for the horizontal wavelengths 300 km, 500 km, and 700 km, respectively, whereas the corresponding maximum values of the vertical velocity are $21.40\,\mathrm{ms}^{-1}$, $16.21\,\mathrm{ms}^{-1}$, and $12.08\,\mathrm{ms}^{-1}$.

*Imaginary frequency shift.* For the time-dependent wave packet, we choose the minimum and maximum values of the imaginary frequency shift as $\delta_{\min} = 10^{-6}$ s$^{-1}$ and $\delta_{\max} = 10^{-4}$ s$^{-1}$, respectively, and set the discrete step to $\Delta\delta = \delta_{\min}$. Referring to Appendix D, the results obtained with the imaginary frequency shift approach for $\lambda_x = 500$ km and $\lambda_t = 40$ min are summarized as follows:

· In the first step, the input value $\delta_{\min} = 10^{-6}$ s$^{-1}$ passes the Source-Function Reconstruction (SFR) test.

· In the second step, the SFR test reduces the input value $\delta_{\max} = 10^{-4}$ s$^{-1}$ to $\delta_{\max} = 32.24 \times 10^{-6}$ s$^{-1}$.

· In the third step, it is found the the interval $[\delta_{\min}, \delta_{\max}]$ contains a sufficient number of internal grid points, spaced by $\Delta\delta$, to be used in the subsequent step.

· In the fourth step, the Layerwise Causality (LC) condition is evaluated at 10 altitude starting at 80 km with a spacing of 20 km. It is found to be satisfied for $\delta_{LC} = \delta_{\min}$, and therefore for all $\delta \in [\delta_{\min}, \delta_{\max}]$, for which the SFR test also holds.

· In the fifth step, five equidistant frequency shifts in $[\delta_{\min}, \delta_{\max}]$ are considered; for each, the wave parameters and their maximum values are computed and the layerwise causality condition is verified over the full altitude range. All five frequency shifts satisfy this condition and yield very similar maximum amplitudes (Table 2). The final solution is selected as the one whose maximum-amplitude vector is closest to the center of mass in the space of perturbed horizontal velocity, vertical velocity, and temperature, corresponding to $\delta = 16.62 \times 10^{-6}$ s$^{-1}$. The maxima occur at

[Figure]

Figure 6: Altitude profiles of the perturbed temperature $\overline{T}$, vertical velocity $\overline{w}$, and horizontal velocity $\overline{u}$ for the total mode ($\mathbf{e}_l = \mathbf{e}_l^+ + \mathbf{e}_l^-$ in layer $l$), the ascending mode ($\mathbf{e}_l^+$), and the descending mode ($\mathbf{e}_l^-$). The upper panels correspond to the general model, and the lower panels to the simplified model. The horizontal wavelength is $\lambda_\mathrm{x} = 400$ km and the wave period is $\lambda_\mathrm{t} = 40$ min

[Figure]

Figure 7: Perturbed temperature (left panels) and vertical velocity (right panels) as functions of time and altitude. The upper panels correspond to $\lambda_x = 300$ km and $\lambda_t = 30$ min, the middle panels to $\lambda_x = 500$ km and $\lambda_t = 40$ min, and the lower panels to $\lambda_x = 700$ km and $\lambda_t = 60$ min

| $\delta\,[10^{-6}\mathrm{s}^{-1}]$ | $\overline{u}_{\max}\,[\mathrm{ms}^{-1}]$ | $\overline{w}_{\max}\,[\mathrm{ms}^{-1}]$ | $\overline{T}_{\max}\,[\mathrm{K}]$ |
|---|---|---|---|
| 32.24 | 38.084 | 16.210 | 31.153 |
| 24.43 | 38.115 | 16.224 | 31.132 |
| 16.62 | 38.079 | 16.210 | 31.158 |
| 8.81 | 38.110 | 16.225 | 31.186 |
| 1.00 | 38.017 | 16.186 | 31.113 |

Table 2: Maximum values of the perturbed horizontal velocity ($\overline{u}_{\max}$), vertical velocity ($\overline{w}_{\max}$), and temperature ($\overline{T}_{\max}$) for different values of the imaginary frequency shift $\delta$

11.06 hr and 248.00 km for the horizontal velocity, 10.98 hr and 303.64 km for the vertical velocity, and 10.90 hr and 257.45 km for the temperature.

**7 Conclusions**

We designed a numerical model for solving the linearized gravity-wave equations using a multilayer method, which is freely available as open-source code on GitHub. To decouple the hydrodynamic equations for the neutral atmosphere from the ionospheric equations, which are coupled through ion drag, we adopt a fast field-aligned diffusion approximation. This approximation may be viewed as a generalization of an approach originally proposed by Klostermeyer [9].

To solve the linearized equations, we employ (i) global matrix methods based on matrix exponentials and (ii) scattering matrix methods to determine either (a) the amplitudes of the characteristic solutions or (b) the grid-point values of the state vector. Ascending and descending wave modes are identified according to the criterion that the real parts of the eigenvalues of the characteristic equation for ascending modes are smaller than those for descending modes (or, equivalently, that the imaginary parts of the vertical wavenumbers are smaller). Global matrix methods using the scaling matrices (36) and (37) require the classification of ascending and descending modes only at the lower and upper boundaries, whereas scattering matrix methods require an explicit determination of the mode type at every altitude. The model is devoted to solving the linearized equations including viscosity, thermal conduction, and ion drag. A simplified model, corresponding to an isothermal, homogeneous atmosphere with constant kinematic viscosity, no background wind, and no ion drag, is also considered.

Depending on the form of the source function, either single-frequency waves or time-dependent wave packets can be analyzed. A heuristic approach for determining the imaginary frequency shift introduced by Knight et al. [16] is also considered. This approach is based on (i) a layerwise causality condition applied at selected altitude levels, and (ii) a source-function reconstruction test.

Numerical simulations demonstrate that both global matrix and scattering matrix methods achieve comparable accuracy. However, the former are significantly more efficient than the latter, particularly in simulations involving

time-dependent wave packets. Among the global matrix methods, the approach based on solving for the amplitudes of the characteristic solutions appears to provide the highest efficiency and accuracy.

The linearized equations on which the solution methods were tested correspond to ionospheric conditions. The ultimate goal of our research is to develop a comprehensive model for analyzing ionospheric gravity waves using satellite measurements. The approach presented in this paper represents only the first component of such a model. Two options are envisaged for extending it to a more complete formulation.

1. Fully coupled neutral–ion model. The linearized hydrodynamic equations would be solved together with the ion equations. In this case, the ion continuity equation would include perturbed production and loss terms, whereas ion inertia and ion–ion collisions would continue to be neglected in the ion momentum equation, and only transport parallel to the magnetic field lines would be retained. The state vector would then be augmented by two additional components, namely the perturbed ion number density and the ion diffusion velocity.

2. Two-step coupling strategy. In the first step, the neutral-atmosphere equations are solved using the fast field-aligned diffusion approximation. In the second step, the wave-induced perturbations obtained from the neutral solution are used as input to solve the ionospheric equations for the perturbed $O^+$ ion density. The ionospheric equations may be solved using the SAMI2 model [40] for low latitudes, where the $\mathbf{E} \times \mathbf{B}$ drift is neglected, or the SAMI3 model [47] at higher altitudes, where the $\mathbf{E} \times \mathbf{B}$ drift is included and the electric field is determined from the solution of a two-dimensional potential equation. In this strategy, priority is given to the ionospheric equations of the SAMI framework, while wave-induced perturbations are handled using the approximate approach developed in the present study. Along similar lines, Knight et al. [34] solved the neutral-atmosphere equations without ion drag in a first step, and subsequently addressed the ionospheric response using the Field-Line Interhemispheric Plasma (FLIP) model [48].

The development and application of these complete models will be addressed in future papers.

**Appendix A. Derivation of the linear system of ordinary differential equations**

In this appendix, we derive the explicit representation of the linear system of ordinary differential equations (5).

**Hydrodynamic equations**

The hydrodynamic equations for the neutral atmosphere consist in the continuity, momentum, heat, and ideal gas equations (e.g., Refs. [4, 5])

$$\frac{\mathrm{D}\rho}{\mathrm{D}t} = -\rho \nabla \cdot \mathbf{u}, \tag{117}$$

$$\rho \frac{\mathrm{D}\mathbf{u}}{\mathrm{D}t} = -\nabla p + \rho \mathbf{g} + \nabla \cdot \boldsymbol{\sigma} - \mathbf{f}_{\mathrm{ID}}, \tag{118}$$

$$\rho c_{\mathrm{v}} \frac{\mathrm{D}T}{\mathrm{D}t} = -p \nabla \cdot \mathbf{u} + \boldsymbol{\sigma} : \nabla \mathbf{u} + \nabla \cdot (\varLambda \nabla T) - q_{\mathrm{ID}}, \tag{119}$$

$$p = \rho R_{\mathrm{M}} T, \tag{120}$$

where $\rho$ is the density, $p$ the pressure, $T$ the temperature, $\mathbf{u}$ the velocity, $\mathrm{D}/\mathrm{D}t = \partial/\partial t + \mathbf{u} \cdot \nabla$ the material (substantial) derivative, $c_{\mathrm{v}}$ the specific heat at constant volume, $\varLambda$ the coefficient of thermal conductivity, $R_{\mathrm{M}}$ the specific gas constant, and $\boldsymbol{\sigma}$ the viscous stress tensor. The quantities $\mathbf{f}_{\mathrm{ID}}$ and $q_{\mathrm{ID}}$ denote the ion-drag force exerted by neutrals on ions per unit volume, and the frictional heating rate per unit volume arising from ion–neutral collisions, respectively. In a Cartesian coordinate system $(x_1, x_2, x_3)$, the components of the viscous stress tensor are given by

$$\sigma_{ij} = \mu \left( \frac{\partial u_i}{\partial x_j} + \frac{\partial u_j}{\partial x_i} - \frac{2}{3} \delta_{ij} \nabla \cdot \mathbf{u} \right), \tag{121}$$

where $\mu$ is the dynamic viscosity and $\delta_{ij}$ the Kroneker delta. Accordingly, the double dot product of $\boldsymbol{\sigma} = \sum_{ij} \sigma_{ij} \widehat{\mathbf{x}}_i \otimes \widehat{\mathbf{x}}_j$ with $\nabla \mathbf{u} = \sum_{ij} \partial u_i / \partial x_j \widehat{\mathbf{x}}_i \otimes \widehat{\mathbf{x}}_j$ is

$$\boldsymbol{\sigma} : \nabla \mathbf{u} = \sum_{ij} \sigma_{ij} \frac{\partial u_i}{\partial x_j}. \tag{122}$$

Using the ideal-gas law $p = \rho R_{\mathrm{M}} T$ so that $\nabla p = \rho R_{\mathrm{M}} \nabla T + R_{\mathrm{M}} T \nabla \rho$, the momentum equation can be written as

$$\frac{\partial \mathbf{u}}{\partial t} = -\frac{R_{\mathrm{M}} T}{\rho} \nabla \rho - R_{\mathrm{M}} \nabla T - (\mathbf{u} \cdot \nabla) \mathbf{u} + \mathbf{g} + \frac{1}{\rho} \nabla \cdot \boldsymbol{\sigma} - \frac{1}{\rho} \mathbf{f}_{\mathrm{ID}}. \tag{123}$$

Moreover, using

$$c_{\mathrm{p}} = c_{\mathrm{v}} + R_{\mathrm{M}}, \quad \gamma = \frac{c_{\mathrm{p}}}{c_{\mathrm{v}}}, \quad \varLambda = \frac{c_{\mathrm{p}} \mu}{\mathrm{Pr}}, \tag{124}$$

where $c_{\mathrm{p}}$ is the specific heat at constant pressure, $\gamma$ the ratio of specific heats, and $\mathrm{Pr}$ the Prandtl number, and assuming that $c_{\mathrm{p}}$ and $\mathrm{Pr}$ are constant, the heat equation becomes

$$\frac{\partial T}{\partial t} = -(\gamma - 1) T \nabla \cdot \mathbf{u} - \mathbf{u} \cdot \nabla T + \frac{1}{\rho c_{\mathrm{v}}} \boldsymbol{\sigma} : \nabla \mathbf{u} + \frac{\gamma}{\rho \mathrm{Pr}} \nabla \cdot (\mu \nabla T) - \frac{1}{\rho c_{\mathrm{v}}} q_{\mathrm{ID}}. \tag{125}$$

In the momentum and heat equations, the ion-drag force and the corresponding heating per unit mass are given by

$$\frac{1}{\rho}\mathbf{f}_{\mathrm{ID}} = \nu_{ni}(\mathbf{u} - \mathbf{u}_i), \tag{126}$$

$$\frac{1}{\rho}q_{\mathrm{ID}} = \frac{1}{\rho}\mathbf{f}_{\mathrm{ID}} \cdot (\mathbf{u} - \mathbf{u}_i) = \nu_{ni}|\mathbf{u} - \mathbf{u}_i|^2, \tag{127}$$

where $\nu_{ni}$ is the neutral-ion collision frequency (the collision frequency between a neutral particle and all kind of ions).

We choose a rectangular coordinate system such that the $x$-axis is directed to the geographic south, the $y$-axis to the east and the z-axis upward. The wave propagates in the meridional plane, i.e., in the $(x, z)$ plane. The viscous terms per unit mass in the momentum equation are then given by

$$\frac{1}{\rho}(\nabla \cdot \overline{\boldsymbol{\sigma}})_{\mathrm{x}} = \mu_{\mathrm{k}}\left[\left(\frac{\partial^2 u}{\partial x^2} + \frac{\partial^2 u}{\partial z^2}\right) + \frac{1}{3}\left(\frac{\partial^2 u}{\partial x^2} + \frac{\partial^2 w}{\partial x\partial z}\right)\right] + \frac{1}{\rho}\frac{\partial \mu}{\partial z}\left(\frac{\partial u}{\partial z} + \frac{\partial w}{\partial x}\right), \tag{128}$$

$$\frac{1}{\rho}(\nabla \cdot \overline{\boldsymbol{\sigma}})_{\mathrm{z}} = \mu_{\mathrm{k}}\left[\left(\frac{\partial^2 w}{\partial x^2} + \frac{\partial^2 w}{\partial z^2}\right) + \frac{1}{3}\left(\frac{\partial^2 u}{\partial x\partial z} + \frac{\partial^2 w}{\partial z^2}\right)\right] + \frac{1}{\rho}\frac{\partial \mu}{\partial z}\left(\frac{4}{3}\frac{\partial w}{\partial z} - \frac{2}{3}\frac{\partial u}{\partial x}\right), \tag{129}$$

while the viscous dissipation term per unit mass appearing in the heat equation is

$$\frac{1}{\rho}\overline{\boldsymbol{\sigma}} : \nabla\mathbf{u} = \frac{4}{3}\mu_{\mathrm{k}}\left(\frac{\partial u}{\partial x}\right)^2 - \frac{4}{3}\mu_{\mathrm{k}}\frac{\partial u}{\partial x}\frac{\partial w}{\partial z} + \frac{4}{3}\mu_{\mathrm{k}}\left(\frac{\partial w}{\partial z}\right)^2$$
$$+ \mu_{\mathrm{k}}\left(\frac{\partial u}{\partial z}\right)^2 + 2\mu_{\mathrm{k}}\frac{\partial u}{\partial z}\frac{\partial w}{\partial x} + \mu_{\mathrm{k}}\left(\frac{\partial w}{\partial x}\right)^2. \tag{130}$$

Here, $\mu_{\mathrm{k}} = \mu/\rho$ is the kinematic viscosity, and we have assumed that the viscosity depends only on altitude, so that

$$\frac{\partial \mu}{\partial x} = 0. \tag{131}$$

Using Eqs. (128)–(130) together with assumption (131), we express the hydrodynamic equations as

$$\frac{\partial u}{\partial t} = -\frac{R_{\mathrm{M}}T}{\rho}\frac{\partial \rho}{\partial x} - R_{\mathrm{M}}\frac{\partial T}{\partial x} - \left(u\frac{\partial u}{\partial x} + w\frac{\partial u}{\partial z}\right)$$
$$+ \mu_{\mathrm{k}}\left[\left(\frac{\partial^2 u}{\partial x^2} + \frac{\partial^2 u}{\partial z^2}\right) + \frac{1}{3}\left(\frac{\partial^2 u}{\partial x^2} + \frac{\partial^2 w}{\partial x\partial z}\right)\right]$$
$$+ \frac{1}{\rho}\frac{\partial \mu}{\partial z}\left(\frac{\partial u}{\partial z} + \frac{\partial w}{\partial x}\right) - \frac{1}{\rho}f_{\mathrm{IDx}}, \tag{132}$$

$$\frac{\partial w}{\partial t} = -\frac{R_\mathrm{M}T}{\rho}\frac{\partial \rho}{\partial z} - R_\mathrm{M}\frac{\partial T}{\partial z} - \left(u\frac{\partial w}{\partial x} + w\frac{\partial w}{\partial z}\right) - g$$
$$+ \mu_\mathrm{k}\left[\left(\frac{\partial^2 w}{\partial x^2} + \frac{\partial^2 w}{\partial z^2}\right) + \frac{1}{3}\left(\frac{\partial^2 u}{\partial x\partial z} + \frac{\partial^2 w}{\partial z^2}\right)\right]$$
$$+ \frac{1}{\rho}\frac{\partial \mu}{\partial z}\left(\frac{4}{3}\frac{\partial w}{\partial z} - \frac{2}{3}\frac{\partial u}{\partial x}\right) - \frac{1}{\rho}f_\mathrm{IDz}, \tag{133}$$

$$\frac{\partial T}{\partial t} = -(\gamma - 1)\,T\left(\frac{\partial u}{\partial x} + \frac{\partial w}{\partial z}\right) - \left(u\frac{\partial T}{\partial x} + w\frac{\partial T}{\partial z}\right)$$
$$+ \frac{1}{c_\mathrm{v}}\left[\frac{4}{3}\mu_\mathrm{k}\left(\frac{\partial u}{\partial x}\right)^2 - \frac{4}{3}\mu_\mathrm{k}\frac{\partial u}{\partial x}\frac{\partial w}{\partial z} + \frac{4}{3}\mu_\mathrm{k}\left(\frac{\partial w}{\partial z}\right)^2\right.$$
$$\left.+ \mu_\mathrm{k}\left(\frac{\partial u}{\partial z}\right)^2 + 2\mu_\mathrm{k}\frac{\partial u}{\partial z}\frac{\partial w}{\partial x} + \mu_\mathrm{k}\left(\frac{\partial w}{\partial x}\right)^2\right]$$
$$+ \frac{\mu_\mathrm{k}\gamma}{\mathrm{Pr}}\left(\frac{\partial^2 T}{\partial x^2} + \frac{\partial^2 T}{\partial z^2}\right) + \frac{\gamma}{\rho\mathrm{Pr}}\frac{\partial \mu}{\partial z}\frac{\partial T}{\partial z} - \frac{1}{c_\mathrm{v}\rho}q_\mathrm{ID}, \tag{134}$$

where $f_\mathrm{IDx}$ and $f_\mathrm{IDz}$ are the components of the ion drag force $\mathbf{f}_\mathrm{ID}$ on the $x$- and $z$-axis, respectively.

In the above hydrodynamic equations, the Coriolis force has been neglected. For a two-dimensional wave geometry in which both the background flow and the perturbation velocities are confined to the vertical $(x, z)$ plane and all variables are independent of the transverse horizontal coordinate $y$, the Coriolis acceleration associated with the Earth's rotation is directed entirely along the transverse direction and therefore does not enter the momentum equations considered here. More precisely, for $\mathbf{u} = (u, 0, w)$ and $\mathbf{\Omega} = (-\Omega\cos\phi, 0, \Omega\sin\phi)$, where $\phi$ is the geographic latitude and $\Omega = 7.29 \times 10^{-5}\,\mathrm{s}^{-1}$ the Earth's angular velocity, the Coriolis force per unit mass is $\mathbf{f}_\mathrm{C} = -2\mathbf{\Omega} \times \mathbf{u} = (0, 2\Omega(\cos\phi w + \sin\phi u), 0)$. Thus, the Coriolis acceleration is directed entirely along the transverse horizontal direction $y$ and does not affect the two-dimensional $(x, z)$ momentum equations.

**Linearized equations**

To linearize the hydrodynamic equations, we assume that all background (unperturbed) quantities vary only in the $z$-direction and write

$$f(x, z, t) = f_0(z) + f'(x, z, t),$$

where $f$ denotes any state variable. In particular, we assume

$$\mathbf{u}_0(z) = (u_0(z), 0, w_0(z) = 0), \ \ \rho_0 = \rho_0(z), \ \ T_0 = T_0(z). \tag{135}$$

Furthermore, we neglect the second derivative of the background horizontal wind and background temperature,

$$\frac{\mathrm{d}^2 u_0}{\mathrm{d}z^2} = 0, \ \ \frac{\mathrm{d}^2 T_0}{\mathrm{d}z^2} = 0, \tag{136}$$

1. The linearized continuity equation is

$$\frac{\partial \rho'}{\partial t} = -u_0 \frac{\partial \rho'}{\partial x} + \frac{\rho_0}{H_\rho} w' - \rho_0 \left( \frac{\partial u'}{\partial x} + \frac{\partial w'}{\partial z} \right). \tag{137}$$

2. The linearized momentum equations are

$$\begin{aligned}
\frac{\partial u'}{\partial t} = &-\frac{\mathrm{d}u_0}{\mathrm{d}z} w' - u_0 \frac{\partial u'}{\partial x} - \frac{c_\mathrm{s}^2}{\gamma \rho_0} \frac{\partial \rho'}{\partial x} - \frac{c_\mathrm{s}^2}{\gamma T_0} \frac{\partial T'}{\partial x} \\
&+ \mu_\mathrm{k0} \left[ \left( \frac{\partial^2 u'}{\partial x^2} + \frac{\partial^2 u'}{\partial z^2} \right) + \frac{1}{3} \left( \frac{\partial^2 u'}{\partial x^2} + \frac{\partial^2 w'}{\partial x \partial z} \right) \right] \\
&+ \frac{1}{\rho_0} \frac{\mathrm{d}\mu_0}{\mathrm{d}z} \left( \frac{\partial u'}{\partial z} + \frac{\partial w'}{\partial x} \right) + \frac{1}{\rho_0} \frac{\mathrm{d}u_0}{\mathrm{d}z} \frac{\partial \mu'}{\partial z} - \frac{1}{\rho_0} \frac{\mathrm{d}\mu_0}{\mathrm{d}z} \frac{\mathrm{d}u_0}{\mathrm{d}z} \frac{\rho'}{\rho_0} - \left( \frac{1}{\rho} f_\mathrm{IDx} \right)'
\end{aligned} \tag{138}$$

and

$$\begin{aligned}
\frac{\partial w'}{\partial t} = &-\frac{c_\mathrm{s}^2}{\gamma \rho_0} \frac{\partial \rho'}{\partial z} + \frac{c_\mathrm{s}^2}{\gamma H_\rho} \left( \frac{T'}{T_0} - \frac{\rho'}{\rho_0} \right) - \frac{c_\mathrm{s}^2}{\gamma T_0} \frac{\partial T'}{\partial z} - u_0 \frac{\partial w'}{\partial x} \\
&+ \mu_\mathrm{k0} \left[ \left( \frac{\partial^2 w'}{\partial x^2} + \frac{\partial^2 w'}{\partial z^2} \right) + \frac{1}{3} \left( \frac{\partial^2 u'}{\partial x \partial z} + \frac{\partial^2 w'}{\partial z^2} \right) \right] \\
&+ \frac{1}{\rho_0} \frac{\mathrm{d}\mu_0}{\mathrm{d}z} \left( \frac{4}{3} \frac{\partial w'}{\partial z} - \frac{2}{3} \frac{\partial u'}{\partial x} \right) - \left( \frac{1}{\rho} f_\mathrm{IDz} \right)',
\end{aligned} \tag{139}$$

where

$$c_\mathrm{s} = \sqrt{\gamma R_\mathrm{M} T_0} \tag{140}$$

is the speed of sound and $H_\rho$ is the density scale height defined by

$$\frac{\mathrm{d}\rho_0}{\mathrm{d}z} = -\frac{\rho_0}{H_\rho}. \tag{141}$$

Here, $(f_\mathrm{IDx}/\rho)'$ and $(f_\mathrm{IDz}/\rho)'$ denote the perturbations of the $x$- and $z$-components, respectively, of the ion-drag force per unit mass $f_\mathrm{IDx}/\rho$ and $f_\mathrm{IDz}/\rho$.

3. The linearized heat equation is

$$\begin{aligned}
\frac{\partial T'}{\partial t} = &-(\gamma - 1) T_0 \left( \frac{\partial u'}{\partial x} + \frac{\partial w'}{\partial z} \right) - \left( u_0 \frac{\partial T'}{\partial x} + \frac{\mathrm{d}T_0}{\mathrm{d}z} w' \right) \\
&+ \frac{2\mu_\mathrm{k0}}{c_\mathrm{v}} \frac{\mathrm{d}u_0}{\mathrm{d}z} \left( \frac{\partial u'}{\partial z} + \frac{\partial w'}{\partial x} \right) + \frac{\mu_0}{c_\mathrm{v}\rho_0} \left( \frac{\mathrm{d}u_0}{\mathrm{d}z} \right)^2 \left( \frac{\mu'}{\mu_0} - \frac{\rho'}{\rho_0} \right) \\
&+ \frac{\mu_\mathrm{k0}\gamma}{\mathrm{Pr}} \left( \frac{\partial^2 T'}{\partial x^2} + \frac{\partial^2 T'}{\partial z^2} \right) + \frac{\gamma}{\rho_0 \mathrm{Pr}} \left( \frac{\mathrm{d}\mu_0}{\mathrm{d}z} \frac{\partial T'}{\partial z} + \frac{\mathrm{d}T_0}{\mathrm{d}z} \frac{\partial \mu'}{\partial z} \right) - \frac{1}{c_\mathrm{v}} \left( \frac{1}{\rho} q_\mathrm{ID} \right)',
\end{aligned} \tag{142}$$

where $(q_\mathrm{ID}/\rho)'$ denotes the perturbation of the ion-drag heating per unit mass.

The linearized equations (137), (138), (139), and (142) are structurally consistent with Eqs. (7), (4), (5), and (6) in Ref. [35], respectively, when only the terms in black and blue are considered (the terms indicated in red are not included). They are likewise related to Eqs. (3.5), (3.1), (3.3), and (3.4) in Ref. [11], when only the terms indicated in black are considered (the additional contributions indicated in red and blue are not included).

Using the following representation for the dynamic viscosity $\mu$ [49]

$$\mu = 3.34 \times 10^{-7} T^{0.71}, \tag{143}$$

we obtain

$$\mu' = 0.71\mu_0 \frac{T'}{T_0}, \quad \frac{\partial \mu'}{\partial z} = 0.71 \frac{\mathrm{d}\mu_0}{\mathrm{d}z}\left(\frac{T'}{T_0}\right) + 0.71\mu_0 \frac{\partial}{\partial z}\left(\frac{T'}{T_0}\right). \tag{144}$$

With these relations, the linearized equations can be written as follows:

1. $x$-momentum equation

$$
\begin{aligned}
\frac{\partial u'}{\partial t} = & -\frac{\mathrm{d}u_0}{\mathrm{d}z}w' - u_0\frac{\partial u'}{\partial x} - \frac{c_{\mathrm{s}}^2}{\gamma \rho_0}\frac{\partial \rho'}{\partial x} - \frac{c_{\mathrm{s}}^2}{\gamma T_0}\frac{\partial T'}{\partial x} \\
& + \mu_{\mathrm{k}0}\left[\left(\frac{\partial^2 u'}{\partial x^2} + \frac{\partial^2 u'}{\partial z^2}\right) + \frac{1}{3}\left(\frac{\partial^2 u'}{\partial x^2} + \frac{\partial^2 w'}{\partial x \partial z}\right)\right] \\
& + \frac{1}{\rho_0}\frac{\mathrm{d}\mu_0}{\mathrm{d}z}\left(\frac{\partial u'}{\partial z} + \frac{\partial w'}{\partial x}\right) + \frac{1}{\rho_0}\frac{\mathrm{d}u_0}{\mathrm{d}z}\left[0.71\frac{\mathrm{d}\mu_0}{\mathrm{d}z}\frac{T'}{T_0} + 0.71\mu_0\frac{\partial}{\partial z}\left(\frac{T'}{T_0}\right)\right] \\
& - \frac{1}{\rho_0}\frac{\mathrm{d}\mu_0}{\mathrm{d}z}\frac{\mathrm{d}u_0}{\mathrm{d}z}\frac{\rho'}{\rho_0} - \left(\frac{1}{\rho}f_{\mathrm{IDx}}\right)',
\end{aligned}
\tag{145}
$$

2. $z$-momentum equation

$$
\begin{aligned}
\frac{\partial w'}{\partial t} = & -\frac{c_{\mathrm{s}}^2}{\gamma \rho_0}\frac{\partial \rho'}{\partial z} + \frac{c_{\mathrm{s}}^2}{\gamma H_\rho}\left(\frac{T'}{T_0} - \frac{\rho'}{\rho_0}\right) - \frac{c_{\mathrm{s}}^2}{\gamma T_0}\frac{\partial T'}{\partial z} - u_0\frac{\partial w'}{\partial x} \\
& + \mu_{\mathrm{k}0}\left[\left(\frac{\partial^2 w'}{\partial x^2} + \frac{\partial^2 w'}{\partial z^2}\right) + \frac{1}{3}\left(\frac{\partial^2 u'}{\partial x \partial z} + \frac{\partial^2 w'}{\partial z^2}\right)\right] \\
& + \frac{1}{\rho_0}\frac{\mathrm{d}\mu_0}{\mathrm{d}z}\left(\frac{4}{3}\frac{\partial w'}{\partial z} - \frac{2}{3}\frac{\partial u'}{\partial x}\right) - \left(\frac{1}{\rho}f_{\mathrm{IDz}}\right)',
\end{aligned}
\tag{146}
$$

3. heat equation

$$
\begin{aligned}
\frac{\partial T'}{\partial t} = & -(\gamma - 1)T_0\left(\frac{\partial u'}{\partial x} + \frac{\partial w'}{\partial z}\right) - \left(u_0\frac{\partial T'}{\partial x} + \frac{\mathrm{d}T_0}{\mathrm{d}z}w'\right) \\
& + \frac{2\mu_{\mathrm{k}0}}{c_{\mathrm{v}}}\frac{\mathrm{d}u_0}{\mathrm{d}z}\left(\frac{\partial u'}{\partial z} + \frac{\partial w'}{\partial x}\right) + \frac{\mu_0}{c_{\mathrm{v}}\rho_0}\left(\frac{\mathrm{d}u_0}{\mathrm{d}z}\right)^2\left(0.71\frac{T'}{T_0} - \frac{\rho'}{\rho_0}\right) \\
& + \frac{\mu_{\mathrm{k}0}\gamma}{\mathrm{Pr}}\left(\frac{\partial^2 T'}{\partial x^2} + \frac{\partial^2 T'}{\partial z^2}\right) + \frac{\gamma}{\rho_0 \mathrm{Pr}}\frac{\mathrm{d}\mu_0}{\mathrm{d}z}\frac{\partial T'}{\partial z} \\
& + \frac{\gamma}{\rho_0 \mathrm{Pr}}\frac{\mathrm{d}T_0}{\mathrm{d}z}\left[0.71\frac{\mathrm{d}\mu_0}{\mathrm{d}z}\left(\frac{T'}{T_0}\right) + 0.71\mu_0\frac{\partial}{\partial z}\left(\frac{T'}{T_0}\right)\right] - \frac{1}{c_{\mathrm{v}}}\left(\frac{1}{\rho}q_{\mathrm{ID}}\right)'.
\end{aligned}
\tag{147}
$$

**Plane wave solution**

We assume that all perturbations vary harmonically in time and in the $x$-direction, that is,

$$f'(x, z, t) = \overline{f}(z) e^{j(\omega t - k_x x)}, \tag{148}$$

where $\omega$ is the angular frequency and $k_x$ the horizontal wavenumber. It then follows that

$$\frac{\partial f'}{\partial t} = j\omega f', \quad \frac{\partial f'}{\partial x} = -jk_x f', \quad \frac{\partial^2 f'}{\partial x^2} = -k_x^2 f'. \tag{149}$$

The linearized continuity equation becomes

$$\frac{\overline{\rho}}{\rho_0} = \frac{k_x}{\Omega} \overline{u} - \frac{j}{\Omega H_\rho} \overline{w} + \frac{j}{\Omega} \frac{d\overline{w}}{dz}, \tag{150}$$

where

$$\Omega = \omega - k_x u_0 \tag{151}$$

is the intrinsic frequency. Further, using

$$\frac{d}{dz} \left( \frac{\overline{\rho}}{\rho_0} \right) = \frac{1}{\rho_0} \frac{d\overline{\rho}}{dz} + \frac{1}{H_\rho} \frac{\overline{\rho}}{\rho_0} \tag{152}$$

together with

$$\frac{d\Omega}{dz} = -k_x \frac{du_0}{dz}, \quad \frac{d\rho_0}{dz} = -\frac{\rho_0}{H_\rho}, \tag{153}$$

we obtain

$$\frac{1}{\rho_0} \frac{d\overline{\rho}}{dz} + \frac{1}{H_\rho} \frac{\overline{\rho}}{\rho_0} = \frac{k_x^2}{\Omega^2} \frac{du_0}{dz} \overline{u} - \frac{j}{\Omega H_\rho} \left( \frac{k_x}{\Omega} \frac{du_0}{dz} - \frac{1}{H_\rho} \frac{dH_\rho}{dz} \right) \overline{w}$$
$$+ \frac{k_x}{\Omega} \frac{d\overline{u}}{dz} + \frac{j}{\Omega} \left( \frac{k_x}{\Omega} \frac{du_0}{dz} - \frac{1}{H_\rho} \right) \frac{d\overline{w}}{dz} + \frac{j}{\Omega} \frac{d^2\overline{w}}{dz^2}. \tag{154}$$

In a first step, we use Eqs. (150) and (154), together with the linearized forms of the momentum and the heat equation given in Eqs. (145)–(147), to express the governing equations in terms of $\overline{u}$, $\overline{w}$, $\overline{T}/T_0$, and their vertical derivatives.

In a second step, we introduce the dimensionless state variables $\widehat{u}$, $\widehat{w}$, and $\widehat{T}$, defined by

$$\overline{u}(z) = \frac{\omega_0}{k_x} \widehat{u}(z), \quad \overline{w}(z) = \frac{\omega_0}{k_x} \widehat{w}(z), \quad \overline{T}(z) = T_0(z) \widehat{T}(z), \tag{155}$$

where $\omega_0$ is a reference frequency, together with their vertical derivatives

$$\widehat{\mathcal{U}} = \frac{d\widehat{u}}{dz}, \quad \widehat{\mathcal{W}} = \frac{d\widehat{w}}{dz}, \quad \widehat{\mathcal{T}} = \frac{d\widehat{T}}{dz}. \tag{156}$$

The state vector is organized as

$$\mathbf{e} = [\widehat{u}, \widehat{w}, \widehat{T}, \widehat{\mathcal{U}}, \widehat{\mathcal{W}}, \widehat{\mathcal{T}}]^{\mathrm{T}}$$

and the corresponding system of ordinary differential equations is given by

$$\frac{1}{k_{\mathrm{x}}} \frac{\mathrm{d}\widehat{u}}{\mathrm{d}z} = \frac{1}{k_{\mathrm{x}}} \widehat{\mathcal{U}}, \tag{157}$$

$$\frac{1}{k_{\mathrm{x}}} \frac{\mathrm{d}\widehat{w}}{\mathrm{d}z} = \frac{1}{k_{\mathrm{x}}} \widehat{\mathcal{W}}, \tag{158}$$

$$\frac{1}{k_{\mathrm{x}}} \frac{\mathrm{d}\widehat{T}}{\mathrm{d}z} = \frac{1}{k_{\mathrm{x}}} \widehat{\mathcal{T}}, \tag{159}$$

$$
\begin{aligned}
k_{\mathrm{x}}\mu_{\mathrm{k0}} \left( \frac{1}{k_{\mathrm{x}}} \frac{\mathrm{d}\widehat{\mathcal{U}}}{\mathrm{d}z} \right) &= \left[ \mathrm{j}\Omega + \frac{4}{3} k_{\mathrm{x}}^2 \mu_{\mathrm{k0}} + \frac{k_{\mathrm{x}}}{\Omega} \left( \frac{1}{\rho_0} \frac{\mathrm{d}\mu_0}{\mathrm{d}z} \frac{\mathrm{d}u_0}{\mathrm{d}z} - \mathrm{j}k_{\mathrm{x}} \frac{c_{\mathrm{s}}^2}{\gamma} \right) \right] \widehat{u} \\
&\quad + \left[ \frac{\mathrm{d}u_0}{\mathrm{d}z} + \mathrm{j}k_{\mathrm{x}} \frac{1}{\rho_0} \frac{\mathrm{d}\mu_0}{\mathrm{d}z} - \frac{\mathrm{j}}{\Omega H_\rho} \left( \frac{1}{\rho_0} \frac{\mathrm{d}\mu_0}{\mathrm{d}z} \frac{\mathrm{d}u_0}{\mathrm{d}z} - \mathrm{j}k_{\mathrm{x}} \frac{c_{\mathrm{s}}^2}{\gamma} \right) \right] \widehat{w} \\
&\quad - \frac{k_{\mathrm{x}}}{\omega_0} \left( \mathrm{j}k_{\mathrm{x}} \frac{c_{\mathrm{s}}^2}{\gamma} + 0.71 \frac{1}{\rho_0} \frac{\mathrm{d}\mu_0}{\mathrm{d}z} \frac{\mathrm{d}u_0}{\mathrm{d}z} \right) \widehat{T} - \frac{1}{\rho_0} \frac{\mathrm{d}\mu_0}{\mathrm{d}z} \widehat{\mathcal{U}} \\
&\quad + \left[ \frac{1}{3} \mathrm{j}k_{\mathrm{x}}\mu_{\mathrm{k0}} + \frac{\mathrm{j}}{\Omega} \left( \frac{1}{\rho_0} \frac{\mathrm{d}\mu_0}{\mathrm{d}z} \frac{\mathrm{d}u_0}{\mathrm{d}z} - \mathrm{j}k_{\mathrm{x}} \frac{c_{\mathrm{s}}^2}{\gamma} \right) \right] \widehat{\mathcal{W}} \\
&\quad - 0.71 \frac{\mu_0}{\rho_0} \frac{\mathrm{d}u_0}{\mathrm{d}z} \frac{k_{\mathrm{x}}}{\omega_0} \widehat{\mathcal{T}} + \frac{k_{\mathrm{x}}}{\omega_0} \overline{\left( \frac{1}{\rho} f_{\mathrm{IDx}} \right)},
\end{aligned}
\tag{160}
$$

$$
\begin{aligned}
k_{\mathrm{x}} \left( \frac{4}{3}\mu_{\mathrm{k0}} - \frac{\mathrm{j}c_{\mathrm{s}}^2}{\gamma\Omega} \right) \left( \frac{1}{k_{\mathrm{x}}} \frac{\mathrm{d}\widehat{\mathcal{W}}}{\mathrm{d}z} \right) &= \left( -\frac{2}{3} \mathrm{j}k_{\mathrm{x}} \frac{1}{\rho_0} \frac{\mathrm{d}\mu_0}{\mathrm{d}z} + \frac{c_{\mathrm{s}}^2 k_{\mathrm{x}}^2}{\gamma\Omega^2} \frac{\mathrm{d}u_0}{\mathrm{d}z} \right) \widehat{u} \\
&\quad + \left[ \mathrm{j}\Omega + k_{\mathrm{x}}^2 \mu_{\mathrm{k0}} - \frac{\mathrm{j}c_{\mathrm{s}}^2}{\gamma\Omega H_\rho} \left( \frac{k_{\mathrm{x}}}{\Omega} \frac{\mathrm{d}u_0}{\mathrm{d}z} - \frac{1}{H_\rho} \frac{\mathrm{d}H_\rho}{\mathrm{d}z} \right) \right] \widehat{w} \\
&\quad - \frac{c_{\mathrm{s}}^2}{\gamma} \frac{k_{\mathrm{x}}}{\omega_0} \left( \frac{1}{H_\rho} - \frac{1}{T_0} \frac{\mathrm{d}T_0}{\mathrm{d}z} \right) \widehat{T} + \left( \frac{1}{3} \mathrm{j}k_{\mathrm{x}}\mu_{\mathrm{k0}} + \frac{k_{\mathrm{x}} c_{\mathrm{s}}^2}{\gamma\Omega} \right) \widehat{\mathcal{U}} \\
&\quad + \left[ \frac{\mathrm{j}c_{\mathrm{s}}^2}{\gamma\Omega} \left( \frac{k_{\mathrm{x}}}{\Omega} \frac{\mathrm{d}u_0}{\mathrm{d}z} - \frac{1}{H_\rho} \right) - \frac{4}{3} \frac{1}{\rho_0} \frac{\mathrm{d}\mu_0}{\mathrm{d}z} \right] \widehat{\mathcal{W}} \\
&\quad + \frac{c_{\mathrm{s}}^2}{\gamma} \frac{k_{\mathrm{x}}}{\omega_0} \widehat{\mathcal{T}} + \frac{k_{\mathrm{x}}}{\omega_0} \overline{\left( \frac{1}{\rho} f_{\mathrm{IDz}} \right)},
\end{aligned}
\tag{161}
$$

$$k_{\mathrm{x}}\frac{\mu_{\mathrm{k0}}\gamma}{\mathrm{Pr}}\left(\frac{1}{k_{\mathrm{x}}}\frac{\mathrm{d}\widehat{\mathcal{T}}}{\mathrm{d}z}\right) = \omega_0\left[-\mathrm{j}\left(\gamma-1\right)+\frac{1}{\Omega}\frac{\mu_0}{c_{\mathrm{v}}T_0\rho_0}\left(\frac{\mathrm{d}u_0}{\mathrm{d}z}\right)^2\right]\widehat{u}$$

$$+\frac{\omega_0}{k_{\mathrm{x}}}\left[\frac{1}{T_0}\frac{\mathrm{d}T_0}{\mathrm{d}z}+\mathrm{j}\frac{2\mu_{\mathrm{k0}}k_{\mathrm{x}}}{c_{\mathrm{v}}T_0}\frac{\mathrm{d}u_0}{\mathrm{d}z}-\frac{\mathrm{j}}{\Omega H_\rho}\frac{\mu_0}{c_{\mathrm{v}}T_0\rho_0}\left(\frac{\mathrm{d}u_0}{\mathrm{d}z}\right)^2\right]\widehat{w}$$

$$+\left[\mathrm{j}\Omega+\frac{\mu_{\mathrm{k0}}\gamma}{\mathrm{Pr}}k_{\mathrm{x}}^2-C_1\frac{\gamma}{\rho_0\mathrm{Pr}}\frac{\mathrm{d}\mu_0}{\mathrm{d}z}\frac{1}{T_0}\frac{\mathrm{d}T_0}{\mathrm{d}z}-0.71\frac{\mu_0}{c_{\mathrm{v}}T_0\rho_0}\left(\frac{\mathrm{d}u_0}{\mathrm{d}z}\right)^2\right]\widehat{T}$$

$$-\frac{2\mu_{\mathrm{k0}}}{c_{\mathrm{v}}T_0}\frac{\omega_0}{k_{\mathrm{x}}}\frac{\mathrm{d}u_0}{\mathrm{d}z}\widehat{\mathcal{U}}+\frac{\omega_0}{k_{\mathrm{x}}}\left[\left(\gamma-1\right)+\frac{\mathrm{j}}{\Omega}\frac{\mu_0}{c_{\mathrm{v}}T_0\rho_0}\left(\frac{\mathrm{d}u_0}{\mathrm{d}z}\right)^2\right]\widehat{\mathcal{W}}$$

$$-\frac{\gamma}{\mathrm{Pr}}\left(\frac{1}{\rho_0}\frac{\mathrm{d}\mu_0}{\mathrm{d}z}+C_2\mu_{\mathrm{k0}}\frac{1}{T_0}\frac{\mathrm{d}T_0}{\mathrm{d}z}\right)\widehat{\mathcal{T}}+\frac{1}{c_{\mathrm{v}}T_0}\overline{\left(\frac{1}{\rho}q_{\mathrm{ID}}\right)}. \quad (162)$$

Equations (160), (161), and (162) with the constants $C_1 = 1.71$ and $C_2 = 2.71$ correspond to the general model, in which all altitude derivatives of the background parameters $u_0$, $T_0$, $H_\rho$, and $\mu_0$ are retained. The terms shown in black and blue in these equations, with the constants $C_1 = 1.0$ and $C_2 = 2.0$, correspond to the model of Vadas and Nicolls [35]. Moreover, Eqs. (160), (161), and (162), when only the black terms are retained and the constants are set to $C_1 = C_2 = 0$, are consistent in form with Eqs. (3.32), (3.31), and (3.33) in Ref. [11], respectively. Note that in Ref. [11], the wave propagates in three-dimensional space, the harmonic dependence of the perturbed quantities is taken as $\exp[-\mathrm{j}(\omega t - k_{\mathrm{x}}x - k_{\mathrm{y}}y)]$, rather than $\exp[\mathrm{j}(\omega t - k_{\mathrm{x}}x)]$, the characteristic solutions have an $\exp(\mathrm{j}mz)$ dependence, rather than $\exp(k_{\mathrm{x}}\lambda z)$, and the state vector is defined as $[\overline{u},\overline{w},\widehat{T},\overline{\mathcal{U}},\overline{\mathcal{W}},\widehat{\mathcal{T}}]^T$ instead of $[\widehat{u},\widehat{w},\widehat{T},\widehat{\mathcal{U}},\widehat{\mathcal{W}},\widehat{\mathcal{T}}]^T$, where $\overline{\mathcal{U}} = \mathrm{d}\overline{u}/\mathrm{d}z$ and $\overline{\mathcal{W}} = \mathrm{d}\overline{w}/\mathrm{d}z$. Essentially, the difference between the model of Vadas and Nicolls [35] and that of Knight et al. [11] is that, in the latter, the derivative of the dynamic viscosity $\mathrm{d}\mu_0/\mathrm{d}z$ is omitted. In the code, for testing purposes, we included a hard-coded logical flag that selects the linearized model to be used. Our numerical simulations show that there are no significant differences between the general model and that of Vadas and Nicolls [35], and that the effect of the assumption $\mathrm{d}\mu_0/\mathrm{d}z = 0$ is relatively small. This latter assumption was discussed in detail in Ref. [11]. In the general model, the derivative $\mathrm{d}\mu_0/\mathrm{d}z$ is computed as

$$\frac{\mathrm{d}\mu_0}{\mathrm{d}z} = 0.71\mu_0\frac{1}{T_0}\frac{\mathrm{d}T_0}{\mathrm{d}z},$$

whereas the derivatives of $u_0$, $T_0$, and $H_\rho$ are computed using central finite differences.

The model can be particularized as follows:

1. For an isothermal ($T_0 = $ constant), homogeneous ($\mu_{\mathrm{k0}} = \mu_0/\rho_0 = $ constant), and windless atmosphere ($u_0 = 0$), we set

$$\frac{\mathrm{d}u_0}{\mathrm{d}z} = 0, \ \ \frac{\mathrm{d}T_0}{\mathrm{d}z} = 0, \ \ \Omega = \omega_0, \ \ \frac{\mathrm{d}H_\rho}{\mathrm{d}z} = 0, \ \text{and} \ \frac{1}{\rho_0}\frac{\mathrm{d}\mu_0}{\mathrm{d}z} = -\frac{\mu_{\mathrm{k0}}}{H_\rho}. \quad (163)$$

In this case, the density scale height $H_\rho$ coincides with the atmospheric scale height $H_a$, which satisfies

$$\frac{1}{\rho_0}\frac{\mathrm{d}\rho_0}{\mathrm{d}z} = \frac{1}{p_0}\frac{\mathrm{d}p_0}{\mathrm{d}z} = -\frac{1}{H_a} \tag{164}$$

and is given by

$$H_a = \frac{p_0}{\rho_0 g} = \frac{R_M T_0}{g} = \text{constant}. \tag{165}$$

2. For an atmosphere without ion drag, we set

$$f_{\mathrm{IDx}} = 0, \quad f_{\mathrm{IDz}} = 0, \quad \text{and } q_{\mathrm{ID}} = 0. \tag{166}$$

**Dispersion equation**

For $\widehat{f}(z) \propto \exp(k_x \lambda z)$, we obtain

$$\widehat{\mathcal{F}} = \frac{\mathrm{d}\widehat{f}}{\mathrm{d}z} = k_x \lambda \widehat{f}, \quad \frac{\mathrm{d}\widehat{\mathcal{F}}}{\mathrm{d}z} = k_x^2 \lambda^2 \widehat{f}, \tag{167}$$

where $f$ denotes $u$, $w$, and $T$, and $\mathcal{F}$ denotes $\mathcal{U}$, $\mathcal{W}$, and $\mathcal{T}$. Inserting Eq. (167) into Eqs. (160)–(162) yields a homogeneous system of equations. Requiring the determinant of this system to vanish, and neglecting the ion-drag terms, leads to the following dispersion equation:

$$-\frac{\Omega}{c_s^2}\left[\Omega - \mathrm{j}\frac{\gamma\mu_{k0}}{\mathrm{Pr}}k_x^2\left(1-\lambda^2\right)\right]\left[\Omega - \mathrm{j}\mu_{k0}k_x^2\left(1-\lambda^2\right)\right]\left[\Omega - \mathrm{j}\frac{4\mu_{k0}}{3}k_x^2\left(1-\lambda^2\right)\right]$$

$$+\left[\Omega - \mathrm{j}\mu_{k0}k_x^2\left(1-\lambda^2\right)\right]\left[\Omega - \mathrm{j}\frac{\mu_{k0}}{\mathrm{Pr}}k_x^2\left(1-\lambda^2\right)\right]\left[k_x^2\left(1-\lambda^2\right)+\frac{k_x\lambda}{H_\rho}\right] - \frac{k_x^2 c_s^2}{\gamma^2 H_\rho^2}\left(\gamma - 1\right)$$

$$= 0. \tag{168}$$

The dispersion equation (168) is the counterpart of dispersion relation (3.35) in Knight et al. [11], under the aforementioned equivalences. Remarkably, it does not include derivatives of the background parameters $u_0$, $T_0$, $H_\rho$, and $\mu_0$. This result follows from the variable-change method discussed by Knight et al. [11] in their Section 2.2.

For an isothermal, homogeneous, and windless atmosphere without ion drag, the dispersion equation reduces to the cubic equation [4]

$$C_3 R^3 + C_2 R^2 + C_1 R + C_0 = 0, \tag{169}$$

for $R = -\lambda^2 + \alpha\lambda + 1$, where $\alpha = 1/k_x H_a$, or equivalently, $R = \kappa^2 - \mathrm{j}\alpha\kappa + 1$, where

$$\kappa = \mathrm{j}\lambda = \frac{1}{k_x}\left(k_z + \mathrm{j}\frac{1}{2H_a}\right). \tag{170}$$

The coefficients of the cubic equations are given by

$$C_3 = -3\eta\nu(1 + 4\eta), \tag{171}$$

$$C_2 = \frac{3\eta(1 + 4\eta)}{\gamma - 1} + \nu\beta(1 + 7\eta) + 3\eta, \tag{172}$$

$$C_1 = -[\beta^2 - 2\eta\alpha^2(1 + 3\eta)]\nu - \frac{\beta(1 + 7\eta)}{\gamma - 1} - \beta, \tag{173}$$

$$C_0 = \frac{\beta^2 - 2\eta\alpha^2(1 + 3\eta)}{\gamma - 1} + \alpha^2(1 + 3\eta), \tag{174}$$

where

$$\eta = j\frac{\omega_0\mu_0}{3p_0}, \quad \nu = j\frac{k_x^2\Lambda_0 T_0}{\omega_0 p_0}, \quad \Lambda_0 = \frac{\gamma c_v \mu_0}{\mathrm{Pr}}, \quad \beta = \frac{\omega_0^2}{k_x^2 g H_a}. \tag{175}$$

If $R_m$, $m = 1, \ldots, 3$ are the solutions of the dispersion equation, the corresponding vertical wavenumbers $k_{zm}^\pm$ are given by

$$k_{zm}^\pm = \mp k_x\sqrt{R_m - 1 - \frac{\alpha^2}{4}}. \tag{176}$$

The wavenumbers $k_{zm}^+$ with $\mathrm{Im}(k_{zm}^+) < 0$ are associated with ascending modes, whereas the wavenumbers $k_{zm}^-$ with $\mathrm{Im}(k_{zm}^-) > 0$ correspond to descending modes. Ordering the ascending wavenumbers as

$$\mathrm{Im}(k_{z3}^+) < \mathrm{Im}(k_{z2}^+) < \mathrm{Im}(k_{z1}^+) < 0,$$

we identify (i) $k_{z1}^+$ and $k_{z1}^-$ as ascending and descending gravity-wave modes, respectively, (ii) $k_{z2}^+$ and $k_{z2}^-$ as ascending and descending viscosity-wave modes, respectively, and (iii) $k_{z3}^+$ and $k_{z3}^-$ as ascending and descending thermal-conduction wave modes, respectively. A similar cubic equation was derived by Francis [7] under the assumption that the geomagnetic field is either in the horizontal or the vertical direction.

For an isothermal, homogeneous, and windless atmosphere, without viscosity and ion drag, the dispersion equation (169) reduces to the quadratic equation

$$C_2 R^2 + C_1 R + C_0 = 0, \tag{177}$$

with

$$C_2 = \nu\beta, \tag{178}$$

$$C_1 = -\beta^2\nu - \beta\frac{\gamma}{\gamma - 1}, \tag{179}$$

$$C_0 = \frac{\beta^2}{\gamma - 1} + \alpha^2. \tag{180}$$

For $\mathrm{Im}(k_{z2}^+) < \mathrm{Im}(k_{z1}^+) < 0$, the permissible modes are (i) the ascending and descending gravity-wave modes associated to the pair $(k_{z1}^+, k_{z1}^-)$, and (ii) the ascending and descending thermal conduction-wave modes associated to the pair $(k_{z2}^+, k_{z2}^-)$.

**Appendix B. Derivation of the ion-drag terms**

In this appendix we derive the expressions for the ion-drag terms that enter the hydrodynamic equations (160)–(162).

**Ion equations**

For each ion species $i$, the ion continuity equation is

$$\frac{\partial n_i}{\partial t} + \nabla \cdot (n_i \mathbf{u}_i) = P_i - n_i \mathcal{L}_i, \tag{181}$$

and the corresponding ion momentum equation, including pressure gradient, electric field, magnetic field, gravity, and collisions, is

$$\frac{\partial \mathbf{u}_i}{\partial t} + (\mathbf{u}_i \cdot \nabla)\mathbf{u}_i = -\frac{1}{m_i n_i}\nabla p_i + \frac{q_i}{m_i}\mathbf{E} + \frac{q_i}{m_i}\mathbf{u}_i \times \mathbf{B} + \mathbf{g}$$
$$- \nu_{in}(\mathbf{u}_i - \mathbf{u}) - \sum_j \nu_{ij}(\mathbf{u}_i - \mathbf{u}_j). \tag{182}$$

Here $n_i$, $\mathbf{u}_i$, $T_i$, $m_i$, and $p_i = n_i k_{\mathrm{B}} T_i$ are the number density, velocity, temperature, mass, and pressure of ion species $i$; $\mathbf{E}$ is the electric field, $\mathbf{B}$ the magnetic field, $q_i$ the ion charge, $\mathbf{g}$ the gravitational acceleration, and $k_{\mathrm{B}}$ the Boltzmann constant. The quantities $P_i$ and $\mathcal{L}_i$ denote the ionization production rate and the loss rate due to chemical processes of ion $i$, respectively. The neutral wind velocity is $\mathbf{u}$, and the collision frequencies $\nu_{in}$ and $\nu_{ij}$ describe ion–neutral and ion–ion collisions.

In addition to ion equations, we consider the electron momentum equation

$$0 = -\frac{1}{m_e n_e}\nabla p_e - \frac{e}{m_e}\mathbf{E} - \frac{e}{m_e}\mathbf{u}_e \times \mathbf{B}, \tag{183}$$

where $n_e$, $\mathbf{u}_e$, $T_e$, $m_e$, and $p_e = n_e k_{\mathrm{B}} T_e$ are the number density, velocity, temperature, mass, and pressure of electrons. In Eq. (183), electron inertia is neglected because of the small electron mass, while electron collisional terms are neglected because $\nu_e \ll \Omega_e$, where $\nu_e$ denotes the electron collision frequencies and $\Omega_e$ is the electron cyclotron frequency.

In the ion momentum equation we neglect the ion inertia and the ion–ion collisions, and introduce the drift velocity $\mathbf{u}_{\mathrm{D}}$ by writing $\mathbf{u}_i = \mathbf{u} + \mathbf{u}_{\mathrm{D}}$. This yields

$$m_i \nu_{in} \mathbf{u}_{\mathrm{D}} = q_i \left[ \mathbf{E} + (\mathbf{u} + \mathbf{u}_{\mathrm{D}}) \times \mathbf{B} \right] - \frac{1}{n_i}\nabla p_i + m_i \mathbf{g}. \tag{184}$$

The momentum equation is projected along the direction of the magnetic field $\widehat{\mathbf{b}}$ and perpendicular to it. The parallel and perpendicular ·force-balance equations are

$$m_i \nu_{in} u_{\mathrm{D}\parallel} = q_i E_\parallel - \frac{1}{n_i}\left(\nabla p_i\right)_\parallel + m_i g_\parallel, \tag{185}$$

and

$$m_i \nu_{in} \mathbf{u}_{\mathrm{D}\perp} = q_i \left[ \mathbf{E}_\perp + (\mathbf{u}_\perp + \mathbf{u}_{\mathrm{D}\perp}) \times \mathbf{B} \right] - \frac{1}{n_i} \left( \nabla p_i \right)_\perp + m_i \mathbf{g}_\perp, \qquad (186)$$

respectively, where in general $\mathbf{a}_\parallel = (\mathbf{a} \cdot \widehat{\mathbf{b}}) \widehat{\mathbf{b}} = a_\parallel \widehat{\mathbf{b}}$ and $\mathbf{a}_\perp = \mathbf{a} - \mathbf{a}_\parallel$. The parallel transport equation for electrons reduces to

$$E_\parallel = -\frac{1}{e n_e} \frac{\partial p_e}{\partial b}. \qquad (187)$$

The parallel and perpendicular force-balance equations are solved as follows. We first consider the ambipolar diffusion velocity, and then the electromagnetic drift velocity.

1. **Ambipolar diffusion velocity**. For a single dominant ion species of charge $q_i = +e$, the parallel force balance equation for ions becomes

$$m_i \nu_{in} u_{\mathrm{D}\parallel} = eE_\parallel - \frac{1}{n_i} \frac{\partial p_i}{\partial b} + m_i g_\parallel, \qquad (188)$$

where $\partial p_i / \partial b = \nabla p_i \cdot \widehat{\mathbf{b}}$. Inserting Eq. (187) into Eq. (188), gives

$$m_i \nu_{in} u_{\mathrm{D}\parallel} = -\left( \frac{1}{n_e} \frac{\partial p_e}{\partial b} + \frac{1}{n_i} \frac{\partial p_i}{\partial b} \right) + m_i g_\parallel, \qquad (189)$$

where $g_\parallel = \mathbf{g} \cdot \widehat{\mathbf{b}}$. Using the ideal-gas relations $p_i = n_i k_{\mathrm{B}} T_i$ and $p_e = n_e k_{\mathrm{B}} T_e$, assuming quasi-neutrality $n_e = n_i$, and thermal equilibrium $T_i = T_e = T$, we obtain

$$u_{\mathrm{D}\parallel} = -D_{\mathrm{A}} \left( \frac{1}{n_i} \frac{\partial n_i}{\partial b} + \frac{1}{T} \frac{\partial T}{\partial b} \right) + \frac{g_\parallel}{\nu_{in}}, \qquad (190)$$

where

$$D_{\mathrm{A}} = \frac{2 k_{\mathrm{B}} T}{m_i \nu_{in}} \qquad (191)$$

is the ambipolar diffusion coefficient.

2. **Electromagnetic drift velocity**. Neglecting perpendicular pressure-gradient and gravity terms, which are often small compared to the electromagnetic drift terms, and assuming a collisionless or weakly collisional limit in which the ion–neutral drag term is negligible, the perpendicular momentum balance reduces to

$$\mathbf{E}_\perp + (\mathbf{u}_\perp + \mathbf{u}_{\mathrm{D}\perp}) \times \mathbf{B} = 0. \qquad (192)$$

With $\mathbf{u}_{i\perp} = \mathbf{u}_\perp + \mathbf{u}_{\mathrm{D}\perp}$, this gives $\mathbf{u}_{i\perp} \times \mathbf{B} = -\mathbf{E}_\perp$, whose solution is the electromagnetic drift velocity

$$\mathbf{u}_{\mathrm{E}} = \frac{\mathbf{E}_\perp \times \mathbf{B}}{B^2} = \frac{\mathbf{E} \times \mathbf{B}}{B^2}. \qquad (193)$$

Consequently,

$$\mathbf{u}_{\mathrm{D}\perp} = \mathbf{u}_{\mathrm{E}} - \mathbf{u}_\perp. \qquad (194)$$

Collecting all contributions, the ion velocity can be written as

$$\mathbf{u}_i = \mathbf{u}_\| + \mathbf{u}_{\mathrm{D}\|} + \mathbf{u}_\mathrm{E}, \qquad (195)$$

where $\mathbf{u}_\| = (\mathbf{u} \cdot \widehat{\mathbf{b}})\widehat{\mathbf{b}}$ is the field-aligned neutral velocity, $\mathbf{u}_{\mathrm{D}\|} = u_{\mathrm{D}\|}\widehat{\mathbf{b}}$ is the ambipolar diffusion velocity given by Eq. (190), and $\mathbf{u}_\mathrm{E}$ is the electromagnetic drift velocity given by Eq. (193). This expression follows from the decomposition $\mathbf{u}_i = \mathbf{u} + \mathbf{u}_\mathrm{D}$, together with $\mathbf{u}_{\mathrm{D}\perp} = \mathbf{u}_\mathrm{E} - \mathbf{u}_\perp$, so that the perpendicular neutral velocity cancels.

In our model, the ion velocity is assumed to be aligned with the magnetic field lines. This assumption is introduced to decouple the hydrodynamic and ion equation systems. Accordingly, the ion velocity is approximated by

$$\mathbf{u}_i \approx (\mathbf{u} \cdot \widehat{\mathbf{b}})\widehat{\mathbf{b}} + u_{\mathrm{D}\|}\widehat{\mathbf{b}}. \qquad (196)$$

This approximation is justified when perpendicular ion transport is small compared to the dominant field-aligned diffusion. The resulting formulation captures the leading-order effects of ambipolar diffusion along the magnetic field lines, while deliberately neglecting perpendicular electrodynamic coupling, such as cross-field advection and $\mathbf{E} \times \mathbf{B}$ drifts. Consequently, the model is applicable to regimes in which field-aligned transport dominates and perpendicular electrodynamic effects play a secondary role. We note, however, that the neglect of the electromagnetic drift velocity is not appropriate for all geophysical regimes. At high latitudes, ion convection is largely controlled by magnetospheric forcing [50], and realistic modeling generally requires externally imposed convection electric fields, for example from empirical models such as Weimer [51]. At mid-latitudes, perpendicular ion motion may be influenced by inter-hemispheric coupling and neutral-wind differences between conjugate hemispheres [52, 53]. A fully self-consistent electrodynamic formulation, in which the electric field is obtained from an electrostatic potential $\Phi$ via $\mathbf{E} = -\nabla\Phi$, with $\Phi$ determined from quasi-neutral current continuity and the conductivity-tensor relation [51], is therefore beyond the scope of the present study but constitutes an important extension for future work.

In the following, for simplicity, the subscript $\|$ is omitted, and we write $\mathbf{u}_\mathrm{D}$ and $u_\mathrm{D}$ instead of $\mathbf{u}_{\mathrm{D}\|}$ and $u_{\mathrm{D}\|}$, respectively. Let

$$\mathbf{g} = (0, 0, -g), \ \ \widehat{\mathbf{b}} = (-\cos I, 0, -\sin I), \ \ \mathbf{u} = (u, 0, w), \qquad (197)$$

where $I$ is the geomagnetic inclination. The field-aligned derivative is

$$\frac{\partial}{\partial b} = \nabla \cdot \widehat{\mathbf{b}} = -\left(\cos I \frac{\partial}{\partial x} + \sin I \frac{\partial}{\partial z}\right),$$

and the scalar field-aligned diffusion velocity becomes (cf. Eq. (190))

$$u_\mathrm{D} = D_\mathrm{A}\left(\cos I \frac{1}{n_i}\frac{\partial n_i}{\partial x} + \sin I \frac{1}{n_i}\frac{\partial n_i}{\partial z} + \cos I \frac{1}{T}\frac{\partial T}{\partial x} + \sin I \frac{1}{T}\frac{\partial T}{\partial z}\right) + \frac{g}{\nu_{in}}\sin I.$$
$$(198)$$

**Linearized equations**

The linearized continuity equation, together with the linearized expressions for the diffusion velocity, ion-drag force, and ion-drag heating, are as follows.

1. **Ion continuity equation**. Neglecting the perturbed production and loss terms, the perturbed ion continuity equation is

$$\frac{\partial n_i'}{\partial t} + n_i'\nabla \cdot \mathbf{u}_{i0} + \mathbf{u}_{i0} \cdot \nabla n_i' + n_{i0}\nabla \cdot \mathbf{u}_i' + \mathbf{u}_i' \cdot \nabla n_{i0} = 0. \tag{199}$$

Using Eq. (197) and the standard assumptions $n_{i0} = n_{i0}(z)$, $\mathbf{u}_0(z) = (u_0(z), 0, 0)$, and $u_{D0} = u_{D0}(z)$, we obtain

$$\frac{\partial}{\partial t}\left(\frac{n_i'}{n_{i0}}\right) = -u_{D0}\left[\frac{\partial}{\partial b}\left(\frac{n_i'}{n_{i0}}\right) - \frac{1}{n_{i0}}\frac{\mathrm{d}n_{i0}}{\mathrm{d}z}\sin I\left(\frac{n_i'}{n_{i0}}\right)\right] + \frac{\mathrm{d}u_{D0}}{\mathrm{d}z}\sin I\left(\frac{n_i'}{n_{i0}}\right)$$
$$+ \left(\frac{\partial u'}{\partial b} - \frac{1}{n_{i0}}\frac{\mathrm{d}n_{i0}}{\mathrm{d}z}\sin I u'\right)\cos I + \left(\frac{\partial w'}{\partial b} - \frac{1}{n_{i0}}\frac{\mathrm{d}n_{i0}}{\mathrm{d}z}\sin I w'\right)\sin I$$
$$- \left(\frac{\partial u_D'}{\partial b} - \frac{1}{n_{i0}}\frac{\mathrm{d}n_{i0}}{\mathrm{d}z}\sin I u_D'\right) + u_0\cos I\frac{\partial}{\partial b}\left(\frac{n_i'}{n_{i0}}\right)$$
$$- \cos I\sin I\left(\frac{\mathrm{d}u_0}{\mathrm{d}z} + u_0\frac{1}{n_{i0}}\frac{\mathrm{d}n_{i0}}{\mathrm{d}z}\right)\left(\frac{n_i'}{n_{i0}}\right). \tag{200}$$

2. **Diffusion velocity**. For the perturbed diffusion velocity $u_D'$, we have the representation

$$u_D' = -D_{A0}\frac{\partial}{\partial b}\left(\frac{T'}{T_0}\right) - D_{A0}\frac{\partial}{\partial b}\left(\frac{n_i'}{n_{i0}}\right)$$
$$+ \left(\frac{1}{n_{i0}}\frac{\mathrm{d}n_{i0}}{\mathrm{d}z} + \frac{1}{T_0}\frac{\mathrm{d}T_0}{\mathrm{d}z}\right)\sin I D_A' - \frac{g\sin I}{\nu_{in0}}\left(\frac{\nu_{in}'}{\nu_{in0}}\right). \tag{201}$$

3. **Ion-drag force and ion-drag heating**. For the ion-drag force per unit mass (cf. Eq. (126) of Appendix A),

$$\frac{1}{\rho}\mathbf{f}_{ID} = \nu_{ni}(\mathbf{u} - \mathbf{u}_i),$$

we obtain

$$\left(\frac{1}{\rho}f_{IDx}\right)' = \nu_{ni0}\left[\sin^2 I u' - \sin I\cos I w' + \cos I u_D'\right.$$
$$\left. + \left(u_{D0}\cos I + u_0\sin^2 I\right)\left(\frac{\nu_{ni}'}{\nu_{ni0}}\right)\right], \tag{202}$$

$$\left(\frac{1}{\rho}f_{IDz}\right)' = \nu_{ni0}\left[-\sin I\cos I u' + \cos^2 I w' + \sin I u_D'\right.$$
$$\left. + \left(u_{D0}\sin I - u_0\cos I\sin I\right)\left(\frac{\nu_{ni}'}{\nu_{ni0}}\right)\right]. \tag{203}$$

For the ion-drag heating per unit mass (cf. Eq. (127) of Appendix A)

$$\frac{1}{\rho}q_{\mathrm{ID}} = \nu_{ni}|\mathbf{u} - \mathbf{u}_i|^2,$$

we find

$$\left(\frac{1}{\rho}q_{\mathrm{ID}}\right)' = \nu_{ni0}\left[2\left(\sin^2 I u_0 u' - \cos I \sin I u_0 w' + u_{\mathrm{D}0}u'_{\mathrm{D}}\right)\right.$$
$$\left. + \left(u_0^2 \sin^2 I + u_{\mathrm{D}0}^2\right)\left(\frac{\nu'_{ni}}{\nu_{ni0}}\right)\right]. \tag{204}$$

Under the assumption that, in the ionosphere, the atomic oxygen O and $O^+$-ions are the main neutral and ionic constituents, we compute the background neutral-ion and ion-neutral collision frequencies as [46, 54]

$$\nu_{ni0} = 7.22 \times 10^{-17}T_0^{0.37}n_{i0}, \ \ \nu_{in0} = \frac{\nu_{ni0}}{n_{i0}}n_n, \ \ n = \mathrm{O}, \ i = \mathrm{O}^+$$

and their perturbed values as

$$D'_{\mathrm{A}} = D_{\mathrm{A}0}\left(\frac{T'}{T_0} - \frac{\nu'_{in}}{\nu_{in0}}\right), \tag{205}$$

$$\frac{\nu'_{in}}{\nu_{in0}} = 0.37\frac{T'}{T_0} + \frac{\rho'}{\rho_0}, \tag{206}$$

$$\frac{\nu'_{ni}}{\nu_{ni0}} = 0.37\frac{T'}{T_0} + \frac{n'_i}{n_{i0}}. \tag{207}$$

**Decoupled system of equations**

From Eqs. (200)–(204) we deduce that the hydrodynamic equations should be solved together with the ion continuity and momentum equations by introducing two additional state variables, namely $n'_i/n_{i0}$ and $u'_{\mathrm{D}}$. The resulting system then consists of eight equations, obtained by augmenting the hydrodynamic system with the ion continuity and momentum equations. This fully coupled approach was used by Shibata [46].

In our analysis we instead employ a simplified, approximate model that decouples the hydrodynamic and ion equations. We have several options.

1. **Klostermeyer Approximation**. Klostermeyer solved the ion-continuity equation by neglecting the diffuse velocity and neutral winds, i.e.,

$$\frac{\partial}{\partial t}\left(\frac{n'_i}{n_{i0}}\right) = -\left(\cos I\frac{\partial u'}{\partial x} + \sin I\frac{\partial u'}{\partial z} + \frac{1}{n_{i0}}\frac{\mathrm{d}n_{i0}}{\mathrm{d}z}\sin I u'\right)\cos I$$
$$-\left(\cos I\frac{\partial w'}{\partial x} + \sin I\frac{\partial w'}{\partial z} + \frac{1}{n_{i0}}\frac{\mathrm{d}n_{i0}}{\mathrm{d}z}\sin I w'\right)\sin I, \tag{208}$$

and then computed the ion-drag force and ion-drag heating by including the background neutral wind, i.e.,

$$\left(\frac{1}{\rho}f_{\text{IDx}}\right)' = \nu_{ni0}\left[\sin^2 I u' - \sin I \cos I w' + u_0 \sin^2 I \left(\frac{\nu'_{ni}}{\nu_{ni0}}\right)\right], \quad (209)$$

$$\left(\frac{1}{\rho}f_{\text{IDz}}\right)' = \nu_{ni0}\left[-\sin I \cos I u' + \cos^2 I w' - u_0 \cos I \sin I \left(\frac{\nu'_{ni}}{\nu_{ni0}}\right)\right], \quad (210)$$

and

$$\left(\frac{1}{\rho}q_{\text{ID}}\right)' = \nu_{ni0}\left[2\left(\sin^2 I u_0 u' - \cos I \sin I u_0 w'\right) + u_0^2 \sin^2 I \left(\frac{\nu'_{ni}}{\nu_{ni0}}\right)\right]. \quad (211)$$

Klostermeyer's method is best viewed as a semi–diagnostic approximation: one solves a simplified ion continuity equation driven only by the wave-induced divergence, and then evaluates ion-drag force and heating, including $u_0$ and $\nu'_{ni}$, but assuming no diffusion velocity. In other words, field-aligned diffusion and background neutral wind are neglected in the ion continuity equation when computing $n'_i$, and the ion drag is treated as a diagnostic based on the neutral wave field and background wind.

2. **Fast Field-Aligned Diffusion**. In the second method, we assume that field-aligned diffusion is sufficiently strong that the relative ion perturbation and the perturbed diffusion velocity are nearly constant along the magnetic field line (but can vary across it):

$$\frac{\partial}{\partial b}\left(\frac{n'_i}{n_{i0}}\right) = -\left(\cos I \frac{\partial}{\partial x} + \sin I \frac{\partial}{\partial z}\right)\left(\frac{n'_i}{n_{i0}}\right) \approx 0 \quad (212)$$

and

$$\frac{\partial u'_{\text{D}}}{\partial b} = \cos I \frac{\partial u'_{\text{D}}}{\partial x} + \sin I \frac{\partial u'_{\text{D}}}{\partial z} \approx 0. \quad (213)$$

We then obtain

$$\begin{aligned}
\frac{\partial}{\partial t}\left(\frac{n'_i}{n_{i0}}\right) = &\left(u_{\text{D}0}\frac{1}{n_{i0}}\frac{\mathrm{d}n_{i0}}{\mathrm{d}z} - \cos I \frac{\mathrm{d}u_0}{\mathrm{d}z} - \cos I u_0 \frac{1}{n_{i0}}\frac{\mathrm{d}n_{i0}}{\mathrm{d}z}\right. \\
&\left. + \frac{\mathrm{d}u_{\text{D}0}}{\mathrm{d}z}\right)\sin I \left(\frac{n'_i}{n_{i0}}\right) + \frac{1}{n_{i0}}\frac{\mathrm{d}n_{i0}}{\mathrm{d}z}\sin I u'_{\text{D}} \\
&- \left(\cos I \frac{\partial u'}{\partial x} + \sin I \frac{\partial u'}{\partial z} + \frac{1}{n_{i0}}\frac{\mathrm{d}n_{i0}}{\mathrm{d}z}\sin I u'\right)\cos I \\
&- \left(\cos I \frac{\partial w'}{\partial x} + \sin I \frac{\partial w'}{\partial z} + \frac{1}{n_{i0}}\frac{\mathrm{d}n_{i0}}{\mathrm{d}z}\sin I w'\right)\sin I \quad (214)
\end{aligned}$$

and

$$u'_D = D_{A0} \left[ \cos I \, \frac{\partial}{\partial x} \left( \frac{T'}{T_0} \right) + \sin I \, \frac{\partial}{\partial z} \left( \frac{T'}{T_0} \right) \right]$$
$$+ \left( \frac{1}{n_{i0}} \frac{dn_{i0}}{dz} + \frac{1}{T_0} \frac{dT_0}{dz} \right) \sin I \, D'_A - \frac{g \sin I}{\nu_{in0}} \left( \frac{\nu'_{in}}{\nu_{in0}} \right). \tag{215}$$

The ion-drag force and ion-drag heating are still computed from Eqs. (202)–(204).

**Plane wave solutions**

Let

$$n'_i(x, z, t) = \overline{n}_i(z) \, e^{j(\omega t - k_x x)}, \quad \overline{n}_i(z) = n_{i0}(z) \, \widehat{n}_i(z), \tag{216}$$

and (cf. Eq. (148) of Appendix A)

$$f'(x, z, t) = \overline{f}(z) \, e^{j(\omega t - k_x x)}.$$

As in Appendix A, we introduce the dimensionless state variables $\widehat{u}$, $\widehat{w}$, and $\widehat{T}$, defined by

$$\overline{u} = \frac{\omega_0}{k_x} \widehat{u}, \quad \overline{w} = \frac{\omega_0}{k_x} \widehat{w}, \quad \frac{\overline{T}}{T_0} = \widehat{T},$$

together with their derivatives

$$\frac{d\overline{u}}{dz} = \frac{\omega_0}{k_x} \widehat{\mathcal{U}}, \quad \frac{d\overline{w}}{dz} = \frac{\omega_0}{k_x} \widehat{\mathcal{W}}, \quad \frac{d}{dz} \left( \frac{\overline{T}}{T_0} \right) = \frac{d\widehat{T}}{dz} = \widehat{\mathcal{T}}$$

1. **Representation of $\overline{u}_D$.** From Eq. (215) we obtain

$$\overline{u}_D = -j k_x D_{A0} \cos I \, \widehat{T} + D_{A0} \sin I \, \widehat{\mathcal{T}}$$
$$+ \left( \frac{1}{n_{i0}} \frac{dn_{i0}}{dz} + \frac{1}{T_0} \frac{dT_0}{dz} \right) \sin I \, \overline{D}_A - \frac{g \sin I}{\nu_{in0}} \left( \frac{\overline{\nu}_{in}}{\nu_{in0}} \right). \tag{217}$$

Using

$$\frac{\overline{D}_A}{D_{A0}} = \frac{\overline{T}}{T_0} - \frac{\overline{\nu}_{in}}{\nu_{in0}}, \tag{218}$$

$$\frac{\overline{\nu}_{in}}{\nu_{in0}} = 0.37 \frac{\overline{T}}{T_0} + \frac{\overline{\rho}}{\rho_0}, \tag{219}$$

together with (cf. Eq. (150) of Appendix A)

$$\frac{\overline{\rho}}{\rho_0} = \frac{k_x}{\Omega} \overline{u} - \frac{j}{\Omega H_\rho} \overline{w} + \frac{j}{\Omega} \frac{d\overline{w}}{dz}$$

we express $\overline{u}_D$ as

$$\overline{u}_D = U_u \widehat{u} + U_w \widehat{w} + U_T \widehat{T} + U_{\mathcal{W}} \widehat{\mathcal{W}} + U_{\mathcal{T}} \widehat{\mathcal{T}}, \tag{220}$$

where the $U$ coefficients are given by

$$U_{\mathrm{u}} = -\frac{\omega_0}{\Omega}\left(D_{\mathrm{A}0}E + G\right), \tag{221}$$

$$U_{\mathrm{w}} = \mathrm{j}\,\frac{\omega_0}{\Omega}\,\frac{1}{k_{\mathrm{x}}H_\rho}\left(D_{\mathrm{A}0}E + G\right), \tag{222}$$

$$U_{\mathrm{T}} = -\mathrm{j}k_{\mathrm{x}}D_{\mathrm{A}0}\cos I + 0.63\,D_{\mathrm{A}0}E - 0.37\,G, \tag{223}$$

$$U_{\mathcal{U}} = -\mathrm{j}\,\frac{\omega_0}{\Omega}\,\frac{1}{k_{\mathrm{x}}}\left(D_{\mathrm{A}0}E + G\right), \tag{224}$$

$$U_{\mathcal{T}} = D_{\mathrm{A}0}\sin I, \tag{225}$$

with

$$E = \sin I\left(\frac{1}{n_{i0}}\frac{\mathrm{d}n_{i0}}{\mathrm{d}z} + \frac{1}{T_0}\frac{\mathrm{d}T_0}{\mathrm{d}z}\right), \quad G = \frac{g\sin I}{\nu_{in0}}. \tag{226}$$

2. **Representation of $\widehat{n}_i$.** From Eq. (214) we obtain

$$\left(\mathrm{j}\omega - u_{\mathrm{D}0}\frac{1}{n_{i0}}\frac{\mathrm{d}n_{i0}}{\mathrm{d}z}\sin I + \frac{\mathrm{d}u_0}{\mathrm{d}z}\cos I\sin I\right.$$
$$\left. + u_0\frac{1}{n_{i0}}\frac{\mathrm{d}n_{i0}}{\mathrm{d}z}\cos I\sin I - \frac{\mathrm{d}u_{\mathrm{D}0}}{\mathrm{d}z}\sin I\right)\widehat{n}_i$$
$$= \frac{1}{n_{i0}}\frac{\mathrm{d}n_{i0}}{\mathrm{d}z}\sin I\,\overline{u}_{\mathrm{D}} + \left(\mathrm{j}k_{\mathrm{x}}\cos I - \frac{1}{n_{i0}}\frac{\mathrm{d}n_{i0}}{\mathrm{d}z}\sin I\right)\cos I\frac{\omega_0}{k_{\mathrm{x}}}\widehat{u}$$
$$+ \left(\mathrm{j}k_{\mathrm{x}}\cos I - \frac{1}{n_{i0}}\frac{\mathrm{d}n_{i0}}{\mathrm{d}z}\sin I\right)\sin I\frac{\omega_0}{k_{\mathrm{x}}}\widehat{w} - \cos I\sin I\frac{\omega_0}{k_{\mathrm{x}}}\widehat{\mathcal{U}} - \sin^2 I\frac{\omega_0}{k_{\mathrm{x}}}\widehat{\mathcal{W}}. \tag{227}$$

After some manipulations, the expression for $\widehat{n}_i$ reads

$$\widehat{n}_i = N_{\mathrm{u}}\widehat{u} + N_{\mathrm{w}}\widehat{w} + N_{\mathrm{T}}\widehat{T} + N_{\mathcal{U}}\widehat{\mathcal{U}} + N_{\mathcal{W}}\widehat{\mathcal{W}} + N_{\mathcal{T}}\widehat{\mathcal{T}}, \tag{228}$$

where the $N$ coefficients are given by

$$N_{\text{u}} = \frac{1}{N_0} \left[ \left( \text{j}k_{\text{x}} \cos I - \frac{1}{n_{i0}} \frac{\text{d}n_{i0}}{\text{d}z} \sin I \right) \cos I \frac{\omega_0}{k_{\text{x}}} + \frac{1}{n_{i0}} \frac{\text{d}n_{i0}}{\text{d}z} \sin I U_{\text{u}} \right],$$
(229)

$$N_{\text{w}} = \frac{1}{N_0} \left[ \left( \text{j}k_{\text{x}} \cos I - \frac{1}{n_{i0}} \frac{\text{d}n_{i0}}{\text{d}z} \sin I \right) \sin I \frac{\omega_0}{k_{\text{x}}} + \frac{1}{n_{i0}} \frac{\text{d}n_{i0}}{\text{d}z} \sin I U_{\text{w}} \right],$$
(230)

$$N_{\text{T}} = \frac{1}{N_0} \frac{1}{n_{i0}} \frac{\text{d}n_{i0}}{\text{d}z} \sin I U_{\text{T}},$$
(231)

$$N_{\mathcal{U}} = -\frac{1}{N_0} \cos I \sin I \frac{\omega_0}{k_{\text{x}}},$$
(232)

$$N_{\mathcal{W}} = -\frac{1}{N_0} \left( \sin^2 I \frac{\omega_0}{k_{\text{x}}} - \frac{1}{n_{i0}} \frac{\text{d}n_{i0}}{\text{d}z} \sin I U_{\mathcal{W}} \right),$$
(233)

$$N_{\mathcal{T}} = \frac{1}{N_0} \frac{1}{n_{i0}} \frac{\text{d}n_{i0}}{\text{d}z} \sin I U_{\mathcal{T}},$$
(234)

and

$$N_0 = \text{j}\omega - u_{\text{D}0} \frac{1}{n_{i0}} \frac{\text{d}n_{i0}}{\text{d}z} \sin I + \frac{\text{d}u_0}{\text{d}z} \cos I \sin I$$
$$+ u_0 \frac{1}{n_{i0}} \frac{\text{d}n_{i0}}{\text{d}z} \cos I \sin I - \frac{\text{d}u_{\text{D}0}}{\text{d}z} \sin I.$$
(235)

3. **Ion-drag force and ion-drag heating**. Using Eqs. (220) and (228), together with

$$\frac{\overline{\nu}_{ni}}{\nu_{ni0}} = 0.37 \frac{\overline{T}}{T_0} + \frac{\overline{n}_i}{n_{i0}} = 0.37 \widehat{T} + \widehat{n}_i,$$
(236)

we obtain

$$\frac{1}{\nu_{ni0}} \overline{\left( \frac{1}{\rho} f_{\text{IDx}} \right)} = F_{\text{xu}} \widehat{u} + F_{\text{xw}} \widehat{w} + F_{\text{xT}} \widehat{T} + F_{\text{x}\mathcal{U}} \widehat{\mathcal{U}} + F_{\text{x}\mathcal{W}} \widehat{\mathcal{W}} + F_{\text{x}\mathcal{T}} \widehat{\mathcal{T}},$$
(237)

$$\frac{1}{\nu_{ni0}} \overline{\left( \frac{1}{\rho} f_{\text{IDz}} \right)} = F_{\text{zu}} \widehat{u} + F_{\text{zw}} \widehat{w} + F_{\text{zT}} \widehat{T} + F_{\text{z}\mathcal{U}} \widehat{\mathcal{U}} + F_{\text{z}\mathcal{W}} \widehat{\mathcal{W}} + F_{\text{z}\mathcal{T}} \widehat{\mathcal{T}},$$
(238)

$$\frac{1}{\nu_{ni0}} \overline{\left( \frac{1}{\rho} q_{\text{ID}} \right)} = P_{\text{u}} \widehat{u} + P_{\text{w}} \widehat{w} + P_{\text{T}} \widehat{T} + P_{\mathcal{U}} \widehat{\mathcal{U}} + P_{\mathcal{W}} \widehat{\mathcal{W}} + P_{\mathcal{T}} \widehat{\mathcal{T}}.$$
(239)

With

$$A_{\text{x}} = u_{\text{D}0} \cos I + u_0 \sin^2 I, \quad A_{\text{z}} = u_{\text{D}0} \sin I - u_0 \cos I \sin I, \quad B = u_{\text{D}0}^2 + u_0^2 \sin^2 I,$$
(240)

the coefficients corresponding to $f_{\mathrm{IDx}}$ are

$$F_{\mathrm{xu}} = \sin^2 I \, \frac{\omega_0}{k_{\mathrm{x}}} + A_{\mathrm{x}} N_{\mathrm{u}} + \cos I \, U_{\mathrm{u}}, \tag{241}$$

$$F_{\mathrm{xw}} = -\sin I \cos I \, \frac{\omega_0}{k_{\mathrm{x}}} + A_{\mathrm{x}} N_{\mathrm{w}} + \cos I \, U_{\mathrm{w}}, \tag{242}$$

$$F_{\mathrm{xT}} = 0.37 \, A_{\mathrm{x}} + A_{\mathrm{x}} N_{\mathrm{T}} + \cos I \, U_{\mathrm{T}}, \tag{243}$$

$$F_{\mathrm{x}\mathcal{U}} = A_{\mathrm{x}} N_{\mathcal{U}}, \tag{244}$$

$$F_{\mathrm{x}\mathcal{W}} = A_{\mathrm{x}} N_{\mathcal{W}} + \cos I \, U_{\mathcal{W}}, \tag{245}$$

$$F_{\mathrm{x}\mathcal{T}} = A_{\mathrm{x}} N_{\mathcal{T}} + \cos I \, U_{\mathcal{T}}, \tag{246}$$

the coefficients corresponding to $f_{\mathrm{IDz}}$ are

$$F_{\mathrm{zu}} = -\sin I \cos I \, \frac{\omega_0}{k_{\mathrm{x}}} + A_{\mathrm{z}} N_{\mathrm{u}} + \sin I \, U_{\mathrm{u}}, \tag{247}$$

$$F_{\mathrm{zw}} = \cos^2 I \, \frac{\omega_0}{k_{\mathrm{x}}} + A_{\mathrm{z}} N_{\mathrm{w}} + \sin I \, U_{\mathrm{w}}, \tag{248}$$

$$F_{\mathrm{zT}} = 0.37 \, A_{\mathrm{z}} + A_{\mathrm{z}} N_{\mathrm{T}} + \sin I \, U_{\mathrm{T}}, \tag{249}$$

$$F_{\mathrm{z}\mathcal{U}} = A_{\mathrm{z}} N_{\mathcal{U}}, \tag{250}$$

$$F_{\mathrm{z}\mathcal{W}} = A_{\mathrm{z}} N_{\mathcal{W}} + \sin I \, U_{\mathcal{W}}, \tag{251}$$

$$F_{\mathrm{z}\mathcal{T}} = A_{\mathrm{z}} N_{\mathcal{T}} + \sin I \, U_{\mathcal{T}}, \tag{252}$$

and the coefficients corresponding to $q_{\mathrm{ID}}$ are

$$P_{\mathrm{u}} = 2 \sin^2 I \, \frac{\omega_0}{k_{\mathrm{x}}} u_0 + B N_{\mathrm{u}} + 2 u_{\mathrm{D0}} U_{\mathrm{u}}, \tag{253}$$

$$P_{\mathrm{w}} = -2 \cos I \sin I \, \frac{\omega_0}{k_{\mathrm{x}}} u_0 + B N_{\mathrm{w}} + 2 u_{\mathrm{D0}} U_{\mathrm{w}}, \tag{254}$$

$$P_{\mathrm{T}} = 0.37 \, B + B N_{\mathrm{T}} + 2 u_{\mathrm{D0}} U_{\mathrm{T}}, \tag{255}$$

$$P_{\mathcal{U}} = B N_{\mathcal{U}}, \tag{256}$$

$$P_{\mathcal{W}} = B N_{\mathcal{W}} + 2 u_{\mathrm{D0}} U_{\mathcal{W}}, \tag{257}$$

$$P_{\mathcal{T}} = B N_{\mathcal{T}} + 2 u_{\mathrm{D0}} U_{\mathcal{T}}. \tag{258}$$

Note that, in the algorithm implementation, the derivatives $\mathrm{d}n_{i0}/\mathrm{d}z$ and $\mathrm{d}u_{\mathrm{D0}}/\mathrm{d}z$ are computed using central finite differences.

**Appendix C. Solution methods for the grid-point values of the state vector**

In this appendix, the global matrix method with matrix exponential and the scattering matrix method will be formulated for the grid-point values of the

state vector.

**Global matrix method with matrix exponential**

In the layer $l$, with boundaries $z_l$ and $z_{l+1}$, the discrete values $\mathbf{e}_{l+1} = \mathbf{e}(z_{l+1})$ and $\mathbf{e}_l = \mathbf{e}(z_l)$ are related through the relation (cf. Eq. (14))

$$\mathbf{e}_{l+1} = V_l \operatorname{diag}[e^{k_x \lambda_{nl} \Delta_l}] V_l^{-1} \mathbf{e}_l, \tag{259}$$

or equivalently,

$$V_l^{-1} \mathbf{e}_{l+1} = \operatorname{diag}[e^{k_x \lambda_{nl} \Delta_l}] V_l^{-1} \mathbf{e}_l. \tag{260}$$

Taking into account that by Eqs. (13) and (14), we have $\mathbf{e}_l = V_l \mathbf{a}_l$ and $\mathbf{e}_{l+1} = V_{l+1} \mathbf{a}_{l+1}$, we see that Eq. (35) and (260) are completely equivalent. Multiplying Eq. (260) with the scaling matrix $K_{l,}^1$, we obtain the layer equation

$$\mathbb{A}_l^1 \mathbf{e}_{l+1} - \mathbb{A}_l^0 \mathbf{e}_l = \mathbf{0}_{2M}, \quad l = 1, ..., L - 1, \tag{261}$$

where

$$\mathbb{A}_l^1 = K_{l,}^1 V_l^{-1}, \tag{262}$$

$$\mathbb{A}_l^0 = K_{l,}^0 V_l^{-1}, \tag{263}$$

and $K_l^1$ and $K_l^0$, are given by Eqs. (36) and (37), respectively. Essentially, we have $L - 1$ equations imposed on layers $1, \ldots, L - 1$ for the $L$ unknowns $\mathbf{e}_1, \ldots, \mathbf{e}_L$. On the layer $l = 1$, the boundary condition (cf. Eq. (42)) $\mathbf{a}_1^+ = \mathbf{i}_1$, translates into (cf. Eq. (24))

$$[I_M, 0_M] V_1^{-1} \mathbf{e}_1 = \mathbf{i}_1, \tag{264}$$

while on the layer $l = L$, the boundary condition (cf. Eq. (45)) $\mathbf{a}_L^- = \mathbf{0}_M$ translates into (cf. Eq. (25))

$$[0_M, I_M] V_L^{-1} \mathbf{e}_L = \mathbf{0}_M. \tag{265}$$

As in Section 3, the layer equations (261) together with the boundary conditions (264) and (265) for a unit scale factor, are assembled into a system of equations for the stratified atmosphere, i.e.,

$$\mathbb{A} \mathbf{e} = \mathbf{b}, \tag{266}$$

where

$$
\mathbb{A} = \begin{bmatrix}
[0_M, I_M]V_L^{-1} & 0 & \dots & 0 & 0 \\
\mathbb{A}_{L-1}^1 & -\mathbb{A}_{L-1}^0 & \dots & 0 & 0 \\
\vdots & \vdots & \ddots & \vdots & \vdots \\
0 & 0 & \dots & \mathbb{A}_1^1 & -\mathbb{A}_1^0 \\
0 & 0 & \dots & 0 & [I_M, 0_M]V_1^{-1}
\end{bmatrix} , \qquad (267)
$$

$$
\mathbf{e} = \begin{bmatrix}
\mathbf{e}_L \\
\mathbf{e}_{L-1} \\
\vdots \\
\mathbf{e}_2 \\
\mathbf{e}_1
\end{bmatrix} \quad \text{and } \mathbf{b} = \begin{bmatrix}
\mathbf{0}_M \\
\mathbf{0}_{2M} \\
\vdots \\
\mathbf{0}_{2M} \\
\mathbf{i}_1
\end{bmatrix} . \qquad (268)
$$

When applying the lower boundary condition (55), $\mathbf{i}_1$ in Eq. (268) should be replaced by $\mathrm{B}^{-1}\mathbf{b}_1$, where, for a unit scale factor, $\mathbf{b}_1 = [1, 0, 0]^{\mathrm{T}}$. After solving Eq. (266), we compute the wave amplitudes by using Eq. (60).

Comments.

1. The ascending and descending solution modes can be derived by using the upward and downward recurrence relations (cf. Eqs. (28)–(31), (41) and (44))

$$
\mathbf{e}_{l+1}^+ = \mathrm{T}_l^+ \mathbf{e}_l, \text{ for } l = 1, \dots, L-1, \text{ with } \mathbf{e}_1^+ = \mathbf{v}_1^+, \text{ and} \qquad (269)
$$

$$
\mathbf{e}_l^- = \mathrm{T}_l^- \mathbf{e}_{l+1}, \text{ for } l = L-1, \dots, 1, \text{ with } \mathbf{e}_L^- = \mathbf{0}_{2M}, \qquad (270)
$$

respectively, where

$$
\mathrm{T}_l^+ = \mathrm{V}_l \begin{bmatrix}
\mathrm{diag}[e^{k_\mathrm{x}\lambda_{ml}^+ \Delta_l}] & 0_M \\
0_M & 0_M
\end{bmatrix} \mathrm{V}_l^{-1}, \qquad (271)
$$

$$
\mathrm{T}_l^- = \mathrm{V}_l \begin{bmatrix}
0_M & 0_M \\
0_M & \mathrm{diag}[e^{-k_\mathrm{x}\lambda_{ml}^- \Delta_l}]
\end{bmatrix} \mathrm{V}_l^{-1}. \qquad (272)
$$

Obviously, the relation $\mathbf{e}_l = \mathbf{e}_l^+ + \mathbf{e}_l^-$, $l = 1, \dots, L$, can be used to verify the numerical algorithm.

2. If we assume that the ascending modes are the dominant modes, i.e., $\mathbf{e}_l \approx \mathbf{e}_l^+$ for $l = 1, \dots, L$, we may compute the state vector by means of the upward recurrence relation (cf. Eq. (269))

$$
\mathbf{e}_{l+1} = \mathrm{T}_l^+ \mathbf{e}_l, \text{ for } l = 1, \dots, L-1, \text{ with } \mathbf{e}_1 = \mathbf{v}_1^+. \qquad (273)
$$

3. The layer equation (261) was derived from the solution representation (259). In fact, this equation is simply the matrix-exponential representation of the solution, i.e., $\mathbf{e}_{l+1} = \exp(\mathrm{A}_l k_\mathrm{x} \Delta_l)\mathbf{e}_l$, where the matrix exponential is calculated using an eigendecomposition of the propagation

matrix $A_l$, i.e., $A_l = V_l \text{diag}[\lambda_{nl}] V_l^{-1}$. However, instead of an eigende-composition method, we can use the Padé approximation to compute the matrix exponential [29, 30]. The $n$th diagonal Padé approximation to the matrix exponential is $\exp(Ax) = [D(Ax)]^{-1}N(Ax)$, where $D(Ax)$ and $N(Ax)$ are polynomials in $Ax$ of degree $n$ given respectively, by $D(Ax) = \sum_{k=0}^{n}(-1)^k c_k x^k A^k$ and $N(Ax) = D(-Ax) = \sum_{k=0}^{n} c_k x^k A^k$. The coefficients $c_k$ can be computed recursively by means of the relation

$$c_k = \frac{n-k+1}{k(2n-k+1)} c_{k-1}, \ \ k \geq 1 \tag{274}$$

with the initial value $c_0 = 1$. The layer equation then becomes

$$\mathbb{A}_l^1 \mathbf{e}_{l+1} - \mathbb{A}_l^0 \mathbf{e}_l = \mathbf{0}_{2M}, \ \ l = 1, ..., L-1, \tag{275}$$

where $\mathbb{A}_l^1 = D(A_l k_x \Delta_l)$ and $\mathbb{A}_l^0 = N(A_l k_x \Delta_l)$. The resulting system of layer equations, together with the boundary conditions, is assembled into a banded linear system. Eigendecomposition is required only in the lower and upper layers to apply the boundary conditions, whereas the Padé approximation is used in all interior layers. This approach is presum-ably more efficient than a full eigendecomposition. However, since the first-order Padé approximation corresponds to a centered finite-difference scheme, whereas the first-order Taylor expansion yields a forward finite-difference scheme, the method is less accurate. For this reason, the Padé approximation of the matrix exponential is not implemented in our com-puter code and is included here solely for theoretical completeness.

**Scattering matrix method**

In principle, the scattering matrix method can also be formulated in terms of the discrete values of the state vector. Starting from the interaction principle equation (67), using Eq. (27), i.e., $\mathbf{e}_l^\pm = V_l^\pm \mathbf{a}_l^\pm$, and Eqs. (24)–(25), i.e.,

$$\mathbf{a}_l^+ = [I_M, 0_M]\mathbf{a}_l = [I_M, 0_M]V_l^{-1}\mathbf{e}_l, \tag{276}$$

$$\mathbf{a}_l^- = [0_M, I_M]\mathbf{a}_l = [0_M, I_M]V_l^{-1}\mathbf{e}_l, \tag{277}$$

we find that for the stack $\mathcal{S}_{l_0 l}$, the interaction principle equation involving the discrete values of the state vector is

$$\left[ \begin{array}{c} \mathbf{e}_{l_0}^- \\ \mathbf{e}_{l+1}^+ \end{array} \right] = \left[ \begin{array}{cc} \mathscr{R}_{l_0 l}^+ & \mathscr{T}_{l_0 l}^- \\ \mathscr{T}_{l_0 l}^+ & \mathscr{R}_{l_0 l}^- \end{array} \right] \left[ \begin{array}{c} \mathbf{e}_{l_0}^+ \\ \mathbf{e}_{l+1}^- \end{array} \right], \tag{278}$$

where $\dim(\mathscr{R}_{l_0 l}^\pm) = \dim(\mathscr{T}_{l_0 l}^\pm) = 2M \times 2M$, and

$$\left[ \begin{array}{cc} \mathscr{R}_{l_0 l}^+ & \mathscr{T}_{l_0 l}^- \\ \mathscr{T}_{l_0 l}^+ & \mathscr{R}_{l_0 l}^- \end{array} \right] = (I_{4M} - A)^{-1}A, \tag{279}$$

with

$$A = \begin{bmatrix} V_{l_0}^- \mathcal{R}_{l_0 l}^+ [V_{l_0}^{-1}]_1 & V_{l_0}^- \mathcal{T}_{l_0 l}^- [V_l^{-1}]_2 \\ V_l^+ \mathcal{T}_{l_0 l}^+ [V_{l_0}^{-1}]_1 & V_l^+ \mathcal{R}_{l_0 l}^- [V_l^{-1}]_2 \end{bmatrix}, \tag{280}$$

$$V_l = [V_l^+, V_l^-], \text{ and } V_l^{-1} = \begin{bmatrix} [V_l^{-1}]_1 \\ [V_l^{-1}]_2 \end{bmatrix}. \tag{281}$$

Since the matrices $\mathscr{R}_{l_0 l}^\pm$ and $\mathscr{T}_{l_0 l}^\pm$ are twice the size of the matrices $\mathcal{R}_{l_0 l}^\pm$ and $\mathcal{T}_{l_0 l}^\pm$, the associated computational cost is significantly higher. Owing to its low numerical efficiency, this method is therefore of purely theoretical interest and is not implemented in our computer code.

**Appendix D. Implementation issues**

In this appendix, we discuss several implementation issues related to the computation of lower and upper bounds for the wave period, the choice of frequency and time discretization for the Fourier transform, and the determination of the imaginary frequency shift.

**Wave period**

To define practical bounds for the wave period $\lambda_t$, we solve the inviscid dispersion equation (cf. Eq. (177) with $\nu = 0$ and with the intrinsic frequency $\Omega = \omega - k_x u_0$ replacing $\omega$)

$$\Omega^4 - \left[ \omega_a^2 + c_s^2 \left( k_x^2 + k_z^2 \right) \right] \Omega^2 + c_s^2 k_x^2 \omega_g^2 = 0, \tag{282}$$

for an assumed value of the vertical wavenumber $k_z$, where $\omega_g = \sqrt{\gamma - 1} g / c_s$ is the buoyancy (gravity-wave) frequency and $\omega_a = \gamma g / (2 c_s)$ is the acoustic cutoff frequency. Thus, for a stratified atmosphere, the solution $\Omega_{inv}$ depends on the altitude $z$ and the wavenumber $k_z$, and satisfies $\Omega_{inv}(z, k_z) < \omega_g(z)$. In this context we define $\omega_{inv}(k_z) = \min_z [\Omega_{inv}(z, k_z) + k_x u_0(z)]$. The equation is solved for two assumed minimum and maximum values of the vertical wavelength $\lambda_z$, namely $\lambda_{zmin} = 125$ km and $\lambda_{zmax} = 250$ km, implying $k_{zmin} = 2\pi / \lambda_{zmax}$ and $k_{zmax} = 2\pi / \lambda_{zmin}$. The corresponding solutions are denoted by $\omega_{min} = \omega_{inv}(k_{zmax})$ and $\omega_{max} = \omega_{inv}(k_{zmin})$, and the resulting time periods are $\lambda_{t,min} = 2\pi / \omega_{max}$ and $\lambda_{t,max} = 2\pi / \omega_{min}$. These values are likely underestimates, since dissipative effects ($\nu \neq 0$) tend to reduce the oscillation frequencies and thus increase the wave periods. For this reason, we round the bounds upward to the nearest multiple of 10 min, and set $\lambda_{t,min} = 10 \left[ \lambda_{t,min} / 10 \right]$ and $\lambda_{t,max} = 10 \left[ \lambda_{t,max} / 10 \right]$, where $[x]$ denotes the upward rounding of $x$ to the next integer.

To provide a physical interpretation of this approach, we note that Eq. (282) yields

$$k_z^2 = \frac{\Omega^2 - \omega_a^2}{c_s^2} + k_x^2 \left( \frac{\omega_g^2}{\Omega^2} - 1 \right), \tag{283}$$

so that, under the assumption $\Omega \ll \omega_{\mathrm{a}}$ , we obtain

$$\frac{k_{\mathrm{z}}}{k_x} = \pm\sqrt{\frac{\omega_{\mathrm{g}}^2}{\Omega^2} - 1 - \frac{1}{4k_{\mathrm{x}}^2 H_{\mathrm{a}}^2}}. \tag{284}$$

If

$$\frac{\omega_{\mathrm{g}}^2}{\Omega^2} - 1 - \frac{1}{4k_{\mathrm{x}}^2 H_{\mathrm{a}}^2} > 0 \text{ for all } z, \tag{285}$$

then $k_{\mathrm{z}}$ is real and $\exp(-\mathrm{j}k_{\mathrm{z}}z) \in \mathbb{C}$; thus, the wave is propagating. When this condition is not satisfied, $k_{\mathrm{z}}$ is purely imaginary and $\exp(-\mathrm{j}k_{\mathrm{z}}z) \in \mathbb{R}$; thus, for $\mathrm{Im}(k_{\mathrm{z}}) < 0$, the wave is evanescent. This condition yields

$$\omega < \Omega_{\mathrm{inv}}(z, k_{\mathrm{z}} = 0) + k_{\mathrm{x}}u_0(z) \text{ for all } z, \tag{286}$$

where

$$\Omega_{\mathrm{inv}}(z, k_{\mathrm{z}} = 0) = \frac{\omega_{\mathrm{g}}}{\sqrt{1 + \frac{1}{4k_{\mathrm{x}}^2 H_{\mathrm{a}}^2}}} \tag{287}$$

is the solution of the inviscid dispersion equation in the case $k_{\mathrm{z}} = 0$ ($\lambda_{\mathrm{z}} \to \infty$). We conclude that the condition $\omega_{\mathrm{inv}}(k_{\mathrm{z}}) = \min_z[\Omega_{\mathrm{inv}}(z, k_{\mathrm{z}}) + k_{\mathrm{x}}u_0(z)]$ implies that the time period is chosen such that only propagating waves are considered. Condition (286), written $\Omega(z) < \Omega_{\mathrm{inv}}(z, k_{\mathrm{z}} = 0)$ was used by Knight et al. [11] to determine the intrinsic evanescent frequencies that lead to the so-called "anomalous results". In practice, evanescent frequencies occur at altitudes up to at least 100 km; consequently, these anomalous results are of limited relevance for upper-thermospheric gravity-wave studies, since evanescence below about 100 km and critical layers in the lower thermosphere strongly attenuate such waves before they can reach higher altitudes [11].

**Frequency and time discretization for the Fourier transform**

In the code, we do not use a Fast Fourier Transform (FFT). Instead, we perform a direct (discrete) Fourier Transform (FT) by explicitly discretizing the Fourier integral. The discretization parameters are chosen to resolve a source that is localized both in frequency and in time.

1. **Frequency band centered on** $\omega_0$. The Fourier transform is performed over a frequency band centered on the reference frequency $\omega_0$. We introduce a frequency standard deviation $\sigma_\omega$ defined as a fraction of $\omega_0$, i.e., $\sigma_\omega = \omega_0/\kappa_\omega$, where $\kappa_\omega \geq 6$ is an input parameter. The effective frequency band of interest is chosen to cover approximately plus/minus three standard deviations of the source spectrum: $\omega_{\mathrm{min}} = \omega_0 - 3\sigma_\omega$ and $\omega_{\mathrm{max}} = \omega_0 + 3\sigma_\omega$. The total frequency interval and the frequency step are, respectively, $L_\omega = \omega_{\mathrm{max}} - \omega_{\mathrm{min}} = 6\omega_0/\kappa_\omega$ and $\Delta\omega = L_\omega/(N_{\mathrm{FT}} - 1)$, where $N_{\mathrm{FT}}$ is the number of discrete points. The discrete frequency grid $\omega_k$ is then constructed as $\omega_k = \omega_{\mathrm{min}} + (k - 1)\Delta\omega$, $k = 1, 2, \ldots, N_{\mathrm{FT}}$.

2. **Time interval and time step**. The time grid is chosen to be consistent with the assumed temporal localization of the source and with the chosen frequency bandwidth. First, we define the time standard deviation $\sigma_t$ as $\sigma_t = 1/\sigma_\omega$ . The time interval $L_t$ is chosen to contain $N_p$ periods of the reference frequency. The time period is given by $\lambda_t = 2\pi/\omega_0$, and therefore, $L_t = N_p \lambda_t$. The time step $\Delta t$ is then computed as $\Delta t = L_t/(N_{FT}-1)$. The time grid $t_k$ is defined on the interval $[t_{\min}, t_{\max}]$, where $t_{\min}$ is an input parameter and $t_{\max} = t_{\min} + L_t$, by $t_k = t_{\min} + (k-1)\Delta t$, $k = 1, 2, \ldots, N_{FT}$. We also define a time shift by $t_0 = (t_{\max} - t_{\min})/2$ which places the reference time close to the center of the time window.

In the code, $N_{FT}$, $\kappa_\omega$, $N_p$ and $t_{\min}$ are input parameters. Specifically, $\kappa_\omega$ determines the length of the frequency interval $L_\omega$, whereas $N_p$ determines the length of the time interval $L_t$. The discretization is considered adequate if the following conditions are satisfied:

1. The time window is long enough for the chosen frequency bandwidth. This requires that the number of time standard deviations contained in the interval satisfies $L_t/\sigma_t > 6$, which ensures that the main part of the source is well contained within the time window and that truncation effects are negligible.

2. The Nyquist condition $\Delta t < \pi/\omega_{\max}$, and the dual Nyquist condition $\Delta \omega < 2\pi/L_t$ are satisfied. These conditions ensure that the temporal signal is properly sampled in time and frequency, prevent aliasing effects, and guarantee a consistent discrete Fourier transform between the time and frequency domains

Since the Fourier transform is computed by direct discretization of the Fourier integral, no FFT-specific constraint is imposed between the frequency step $\Delta\omega$ and the time step $\Delta t$. In particular, the grid relation $\Delta t \, \Delta\omega = 2\pi/N_{FT}$, which is characteristic of discrete Fourier transforms based on periodic sampling and exact discrete orthogonality, is not required here. Instead, the frequency and time grids are chosen independently, based on the desired frequency band and time window needed to resolve a source that is localized in both domains. This provides greater flexibility in selecting the bandwidth, resolution, and window length.

**Imaginary frequency shift**

The imaginary frequency shift is determined using a heuristic approach that combines two criteria: application of the Layerwise Causality (LC) condition at selected altitude levels, and a Source-Function Reconstruction (SFR) test.

1. **LC condition at selected altitude levels**. Choose a subset $L_0 \subset \{1, 2, \ldots, L\}$ of altitude indices, and let $N_{L_0}$ be the number of elements of the set $L_0$, i.e., $N_{L_0} = |L_0|$. At each altitude level $l_0 \in L_0$, start with

$\delta_{l_0} = \delta_{\text{start}}$ and increases $\delta_{l_0}$ in steps of $\Delta\delta$, i.e., $\delta_{l_0} \leftarrow \min(\delta_{l_0} + \Delta\delta, \delta_{\text{stop}})$, until the LC condition (c.f. Eq. (108))

$$d_{l_0}(\omega_k) = \text{Re}[\lambda_{1l_0}^-(\omega_k - j\delta_{l_0})] - \text{Re}[\lambda_{1l_0}^+(\omega_k - j\delta_{l_0})] > \epsilon_{\text{LC}} \qquad (288)$$

is satisfied for all $\omega_k = \omega_{\min} + (k-1)\Delta\omega$, $k = 1, 2, \ldots, N_{\text{FT}}$, and some prescribed tolerance $\epsilon_{\text{LC}}$. If there exists an altitude level for which it is not possible to satisfy the LC condition for all frequencies $\omega_k$ by increasing the imaginary frequency shift within the interval from the start value $\delta_{\text{start}}$ to the stop value $\delta_{\text{stop}}$, then the subset-based LC criterion is said to fail. Otherwise, the LC frequency shift is computed as

$$\delta_{\text{LC}} = \max_{l_0 \in L_0} \delta_{l_0}. \qquad (289)$$

2. **SFR**. A frequency shift $\delta$ is considered valid if the relative RMS error between the source function $s(t)$ and the inverse Fourier transform applied to $S(\omega - j\delta)$ is below a prescribed tolerance $\epsilon_{\text{SFR}} = 5 \times 10^{-3}$. In practice, we compare the normalized functions

$$\widehat{s}(t) = \frac{s(t)}{\max_t \text{Re}\{s(t)\}}, \quad \widehat{s}_\delta(t) = \frac{s_\delta(t)}{\max_t \text{Re}\{s_\delta(t)\}},$$

where (compare with Eq. (115))

$$s_\delta(t) = A e^{\delta(t-t_0)} \frac{1}{2\pi} \int_{-\infty}^{\infty} \mathscr{S}(\omega - j\delta) e^{j\omega(t-t_0)} d\omega, \qquad (290)$$

and $s(t)$ and $\mathscr{S}(\omega)$ are given by Eqs. (83) and (91), respectively. The relative RMS error is computed as

$$\varepsilon_{s\delta} = \sqrt{\frac{\sum_{k=1}^{N_{\text{FT}}} [\widehat{s}(t_k) - \widehat{s}_\delta(t_k)]^2}{\sum_{k=1}^{N_{\text{FT}}} \widehat{s}(t_k)^2}} \qquad (291)$$

with $t_k = t_{\min} + (k-1)\Delta t$, $k = 1, 2, \ldots, N_{\text{FT}}$. In the algorithm,

(a) if $\varepsilon_{s\delta} > \epsilon_{\text{SFR}}$, the frequency shift is relaxed according to $\delta \leftarrow \max(\delta - \Delta\delta, \delta_{\text{stop}})$, and

(b) if $\varepsilon_{s\delta} > \epsilon_{\text{SFR}}$ also holds for $\delta = \delta_{\text{stop}}$, the SFR test fails.

The algorithm proceeds as follows.

**Step 1: Initialization and SFR test at $\delta_{\min}$** We choose $\delta_{\min} = \Delta\delta$ and $\delta_{\max} = k_\delta \Delta\delta$, where $\Delta\delta$ (e.g., $\Delta\delta = 10^{-6}$ s$^{-1}$) is the initial discrete step of the imaginary frequency shift and $k_\delta$ is an integer. We first apply the SFR test to the input $\delta_{\min}$, using $\delta_{\text{stop}} = \Delta\delta$ and the step $\Delta\delta$ as a *decreasing* update of $\delta$. With this choice, the algorithm effectively checks whether $\delta_{\min}$ passes the SFR test, that is, without invoking a frequency-shift relaxation. If the SFR test fails, we stop the algorithm. Otherwise, the value $\delta_{\min}$ is accepted as SFR-admissible. Based on the monotonic behavior of the SFR error in $\delta$, we assume that any $\delta \leq \delta_{\min}$ would also satisfy the SFR test.

**Step 2: SFR test at $\delta_{\max}$ and refinement of $\delta_{\max}$**   We apply the SFR test to the input $\delta_{\max}$, using $\delta_{\text{stop}} = \delta_{\min}$ and the step $\Delta\delta$ again as a *decreasing* update of $\delta$. Since the SFR test has already been successful at $\delta_{\min}$, this second SFR call is guaranteed to succeed: if necessary, the procedure can always relax $\delta$ down to $\delta_{\min}$. Let $\delta_{\text{SFR}} = k_{\text{SFR}}\,\Delta\delta$ denote the value returned by the SFR routine. We then reset $\delta_{\max} \leftarrow \delta_{\text{SFR}}$. By construction, every $\delta \in [\delta_{\min}, \delta_{\max}]$ satisfies the SFR test.

**Step 3: Construction and adjustment of an internal $\delta$-grid**   The next steps require an internal grid of discrete values of $\delta$ between $\delta_{\min}$ and $\delta_{\max}$, with a sufficiently large number of points (e.g., at least five) to reliably apply the LC procedure. This internal grid uses the spacing $\Delta\delta$, but is *independent* of the number $N_\delta$ used later in the final selection step. To avoid a degenerate interval and to obtain a reasonably fine internal spacing, we adjust $\delta_{\min}$ and $\Delta\delta$ according to the following rules: if $\delta_{\max} - \delta_{\min} < \Delta\delta/2$ (i.e., $\delta_{\max} \approx \delta_{\min}$), then set $\delta_{\min} \leftarrow \delta_{\max} - \Delta\delta/2$ and $\Delta\delta \leftarrow \Delta\delta/8$; else if $\Delta\delta/2 < \delta_{\max} - \delta_{\min} < 3\Delta\delta/2$ (i.e., $\delta_{\max} - \delta_{\min} \approx \Delta\delta$), then set $\Delta\delta \leftarrow \Delta\delta/4$; else if $3\Delta\delta/2 < \delta_{\max} - \delta_{\min} < 5\Delta\delta/2$ (i.e., $\delta_{\max} - \delta_{\min} \approx 2\Delta\delta$), then set $\Delta\delta \leftarrow 2\Delta\delta/3$. After this adjustment, the interval $[\delta_{\min}, \delta_{\max}]$ contains a suitable number of internal grid points spaced by $\Delta\delta$. These internal points are used only in the LC procedure described in the next step.

**Step 4: LC procedure and update of $\delta_{\min}$**   We apply the subset-based LC procedure to $\delta$, starting from the current value $\delta_{\min}$ and moving towards larger values, with an upper bound $\delta_{\text{stop}} = \delta_{\max} - \Delta\delta$ and an *increasing* step $\Delta\delta$. If there exists an altitude level for which the LC test fails, we stop the algorithm. Otherwise, let $\delta_{\text{LC}} = k_{\text{LC}}\,\Delta\delta$ denote the LC-based estimate returned by the procedure. We then update $\delta_{\min} \leftarrow \delta_{\text{LC}}$. By construction, for every $\delta \in [\delta_{\min}, \delta_{\max}]$, both the SFR test and the LC condition at the selected altitude levels are satisfied.

**Step 5: Final selection of the frequency shift**   The final selection step is based on a separate set of $N_\delta$ equidistant values of $\delta$ in the interval $\delta_{\min} \leq \delta \leq \delta_{\max}$. Here, $N_\delta$ is chosen independently of the internal grid used in Steps 3 and 4, and we define $\delta_j = \delta_{\min} + (j-1)\,\Delta\delta_{\text{final}}$ , $j = 1, 2, \ldots, N_\delta$, with $\Delta\delta_{\text{final}} = (\delta_{\max} - \delta_{\min})/(N_\delta - 1)$. For each $\delta_j$, we perform the following tasks:

1. Compute the wave parameters corresponding to $\delta_j$.

2. Evaluate the maximum perturbed horizontal velocity, the maximum perturbed vertical velocity, and the maximum perturbed temperature.

3. Check whether the layerwise causality condition is satisfied over the entire altitude range.

Finally, among all frequency shifts for which the causality condition is satisfied over the full altitude range, the algorithm determines the center of mass in

the space spanned by the maximum values of the perturbed horizontal velocity, vertical velocity, and temperature, and selects the solution whose maximum-amplitude vector is closest to this center of mass.

Comments:

1. The algorithm determines not a single value but an interval of admissible imaginary frequency shifts. This is necessary for two reasons.

   (a) First, the LC procedure evaluates the layerwise causality condition only at a selected subset of altitude levels. As a consequence, the value returned by the LC test guarantees causality only at these selected levels, but not necessarily across the entire altitude range. By retaining an interval rather than a single value, we ensure that the subsequent analysis can identify those values of the frequency shift that satisfy the LC condition everywhere, not just at the sampled altitudes.

   (b) Second, even when the LC condition is applied across the full altitude range, it is often the case that more than one frequency shift satisfies the LC condition over all altitudes. Since these admissible shifts may lead to solutions with different numerical behaviors, a mechanism is needed to select an optimal value. For this purpose, the algorithm evaluates the wave parameters for a discrete set of shifts within the admissible interval and selects the optimal value based on the distance to the center of mass.

2. In principle, a Global Causality (CG) test can be applied at selected frequencies. Such a test proceeds as follows. Choose a set of $N_{K_0}$ equidistant frequencies $\omega_{k_0} \in \{\omega_k\}_{k=1}^{N_{FT}}$ with $k_0 \in K_0 = \{1, 2, \ldots, N_{K_0}\}$. For each test frequency $\omega_{k_0}$, start with $\delta_{k_0} = \delta_{\mathrm{start}}$ and increases $\delta_{k_0}$ in steps of $\Delta\delta$, i.e., $\delta_{k_0} \leftarrow \min(\delta_{k_0} + \Delta\delta, \delta_{\mathrm{stop}})$, until the GC condition (c.f. Eq. (104))

$$\max_{l=1,\ldots,L} \mathrm{Re}[\lambda_{1l}^+(\omega_{k_0} - \mathrm{j}\delta_{k_0})] < \min_{l=1,\ldots,L} \mathrm{Re}[\lambda_{1l}^-(\omega_{k_0} - \mathrm{j}\delta_{k_0})]$$

is satisfied. If there exists a test frequency for which the GC condition is not satisfied, we say that the subset-based GC criterion fails for $\delta_{\mathrm{stop}}$. Otherwise, the GC frequency shift is computed as

$$\delta_{\mathrm{GC}} = \max_{k_0 \in K_0} \delta_{k_0}.$$

This test can be applied after the first step to determine $\delta_{\mathrm{max}} = \delta_{\mathrm{GC}}$. However, our numerical experiments indicate that this test is computationally very expensive and typically yields an excessively large value of $\delta_{\mathrm{GC}}$. In the subsequent step, this large value would be reduced by the SFR test. Importantly, if one instead prescribes a moderate but still sufficiently large input value for $\delta_{\mathrm{max}}$, the source-function reconstruction test reduces $\delta_{\mathrm{max}}$ to the same final value as when starting from $\delta_{\mathrm{GC}}$. Since the LC test alone is sufficient to ensure causality, we therefore do not employ the GC test in practice and use a prescribed moderate value of $\delta_{\mathrm{max}}$.

.